# Mapping the human genetic architecture of COVID-19

COVID-19 Host Genetics Initiative*

The genetic make-up of an individual contributes to the susceptibility and response to viral infection. Although environmental, clinical and social factors have a role in the chance of exposure to SARS-CoV-2 and the severity of COVID-19[1,2], host genetics may also be important. Identifying host-specific genetic factors may reveal biological mechanisms of therapeutic relevance and clarify causal relationships of modifiable environmental risk factors for SARS-CoV-2 infection and outcomes. We formed a global network of researchers to investigate the role of human genetics in SARS-CoV-2 infection and COVID-19 severity. Here we describe the results of three genome-wide association meta-analyses that consist of up to 49,562 patients with COVID-19 from 46 studies across 19 countries. We report 13 genome-wide significant loci that are associated with SARS-CoV-2 infection or severe manifestations of COVID-19. Several of these loci correspond to previously documented associations to lung or autoimmune and inflammatory diseases[3–7]. They also represent potentially actionable mechanisms in response to infection. Mendelian randomization analyses support a causal role for smoking and body-mass index for severe COVID-19 although not for type II diabetes. The identification of novel host genetic factors associated with COVID-19 was made possible by the community of human genetics researchers coming together to prioritize the sharing of data, results, resources and analytical frameworks. This working model of international collaboration underscores what is possible for future genetic discoveries in emerging pandemics, or indeed for any complex human disease.

The COVID-19 pandemic, caused by infection with SARS-CoV-2, has resulted in an enormous health and economic burden worldwide. One of the most remarkable features of SARS-CoV-2 infection is the variation in consequences, which range from asymptomatic to life-threatening, viral pneumonia and acute respiratory distress syndrome[8]. Although established host factors correlate with disease severity (for example, increasing age, being a man and higher body-mass index[1]), these risk factors alone do not explain all of the variability in disease severity observed across individuals.

Genetic factors contributing to COVID-19 susceptibility and severity may provide new biological insights into disease pathogenesis and identify mechanistic targets for therapeutic development or drug repurposing, as treating the disease remains a highly important goal despite the recent development of vaccines. Further supporting this line of inquiry, rare loss-of-function variants in genes involved in the type I interferon response may be involved in severe forms of COVID-19[9–11]. At the same time, several genome-wide association studies that investigate the contribution of common genetic variation[12–15] to COVID-19 have provided robust support for the involvement of several genomic loci associated with COVID-19 severity and susceptibility, with the strongest and most robust finding for severity being at the 3p21.31 locus[12–16]. However, much remains unknown about the genetic basis of susceptibility to SARS-CoV-2 and severity of COVID-19.

The COVID-19 Host Genetics Initiative (COVID-19 HGI) (https://www.covid19hg.org/)[17] is an international, open-science collaboration to share scientific methods and resources with research groups across the world with the goal to robustly map the host genetic determinants of SARS-CoV-2 infection and the severity of the resulting COVID-19 disease. Here, we report the latest results of meta-analyses of 46 studies from 19 countries (Fig. 1) for COVID-19 host genetic effects.

## Meta-analyses of COVID-19

Overall, the COVID-19 HGI combined genetic data from 49,562 cases and 2 million controls across 46 distinct studies (Fig. 1). The data included studies from populations of different genetic ancestries, including European, admixed American, African, Middle Eastern, South Asian and East Asian individuals (Supplementary Table 1). An overview of the study design is provided in Extended Data Fig. 1. We performed case–control meta-analyses in three main categories of COVID-19 disease according to predefined and partially overlapping phenotypic criteria. These included (1) critically ill cases of COVID-19 defined as those individuals who required respiratory support in hospital or who died due to the disease; (2) cases of moderate or severe COVID-19 defined as those participants who were hospitalized due to symptoms associated with the infection; and (3) all cases with reported SARS-CoV-2 infection

*Lists of authors and their affiliations appear in the online version of the paper.

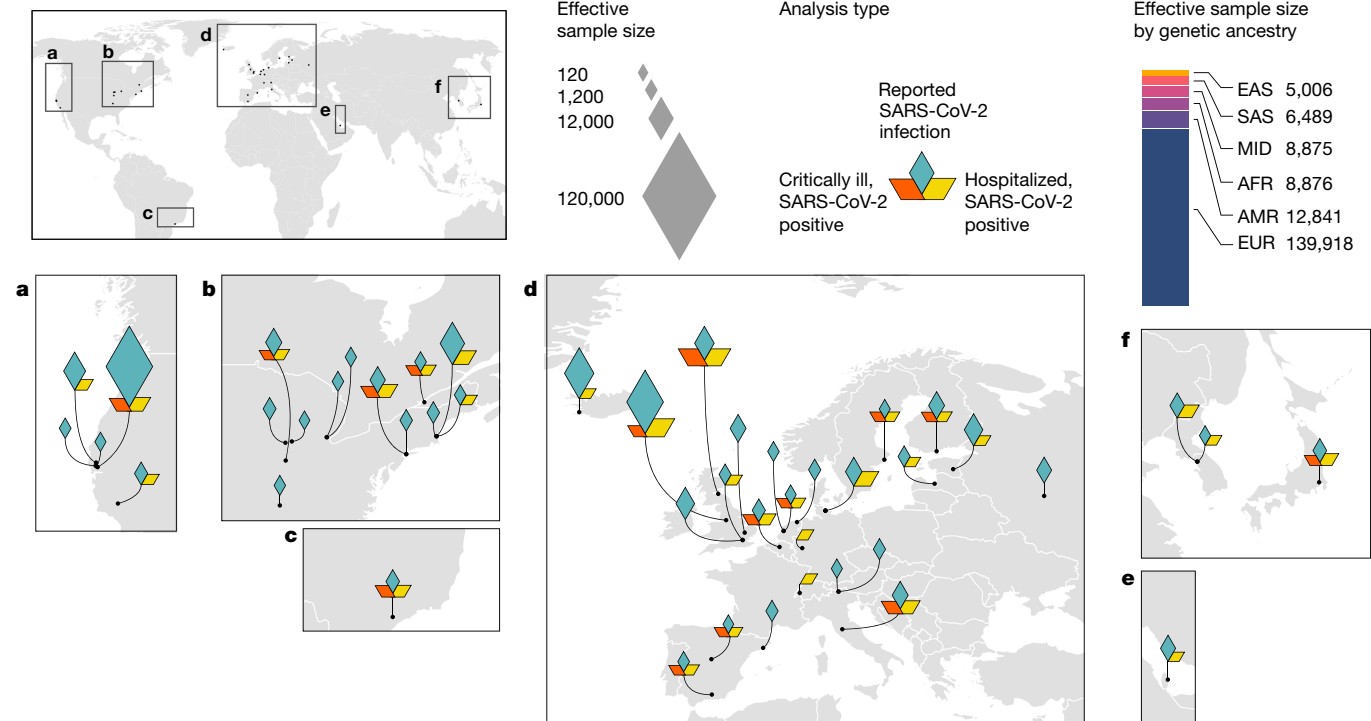

**Fig. 1 | Geographical overview of the contributing studies to the COVID-19 HGI and composition by major ancestry groups.** Populations are defined as African (AFR), admixed American (AMR), East Asian (EAS), European (EUR), Middle Eastern (MID) and South Asian (SAS).

regardless of symptoms (Methods). Controls for all three analyses were selected as genetically ancestry-matched samples without known SARS-CoV-2 infection, if that information was available (Methods). The average age of the participants with COVID-19 across studies was 55 years (Supplementary Table 1). We report quantile–quantile plots in Supplementary Fig. 1 and ancestry principal component plots for contributing studies in Extended Data Fig. 2.

Across our three analyses, we reported a total of 13 independent genome-wide significant loci associated with COVID-19 (the threshold of $P < 1.67 \times 10^{-8}$ is adjusted for multiple trait testing) (Supplementary Table 2), most of which were shared between two or more COVID-19 phenotypes. Two of these loci are in very close proximity within the 3p21.31 region, which was previously reported as a single locus associated with COVID-19 severity[12–16] (Extended Data Fig. 3). Overall, we find six genome-wide significant associations for critical illness due to COVID-19, using data from 6,179 cases and 1,483,780 controls from 16 studies (Extended Data Fig. 4). Nine genome-wide significant loci were detected for moderate to severe hospitalized COVID-19 (including five of the six critical illness loci) from an analysis of 13,641 cases of COVID-19 and 2,070,709 controls across 29 studies (Fig. 2a, top). Finally, seven loci reached genome-wide significance in the analysis using data for all available 49,562 reported cases of SARS-CoV-2 infection and 1,770,206 controls, using data from a total of 44 studies (Fig. 2a, bottom). The proportion of cases with non-European genetic ancestry for each of the three analyses was 23%, 29% and 22%. We report the results for the lead variants at the 13 loci in different ancestry-group meta-analyses in Supplementary Table 3. We note that two loci, tagged by lead variants rs1886814 and rs72711165, had higher allele frequencies in southeast Asian (rs1886814; 15%) and East Asian genetic ancestry (rs72711165; 8%) whereas the minor allele frequencies in European populations were less than 3%. This highlights the value of including data from diverse populations for genetic discovery. We discuss the replication of previous findings and the new discoveries from these three analyses in the Supplementary Note.

## Variant effects on severity and susceptibility

We found no genome-wide significant sex-specific effects at the 13 loci. However, we did identify significant heterogeneous effects ($P < 0.004$) across studies for 3 out of the 13 loci (Methods), which probably reflects the differential ascertainment of cases (Supplementary Table 2). There was a small number of overlapping samples ($n = 8,380$ European ancestry; $n = 745$ East Asian ancestry) between controls from the genOMICC and the UK Biobank studies, but leave-one-out sensitivity analyses did not reveal any bias in the corresponding effect sizes or $P$ values (Extended Data Fig. 5 and Supplementary Information).

We next wanted to better understand whether the 13 significant loci were acting through mechanisms that increased the susceptibility to infection or that affected the progression of symptoms towards more severe disease. For all 13 loci, we compared the lead variant (strongest association $P$ value) odds ratios (ORs) for the risk-increasing allele across our different COVID-19 phenotype definitions.

Focusing on the two better powered analyses: all cases with a reported SARS-CoV-2 infection and all cases hospitalized due to COVID-19, we find that four of the loci have similar odds ratios between these two analyses (Methods and Supplementary Table 2). Such consistency suggests a stronger link to susceptibility to SARS-CoV-2 infection rather than to the development of severe COVID-19. The strongest susceptibility signal was the previously reported *ABO* locus (rs912805253)[12,13,15,16]. Notably, and in agreement with a previously reported study[15], we also report a locus within the 3p21.31 region that was more strongly associated with susceptibility to SARS-CoV-2 than progression to more severe COVID-19 phenotypes. rs2271616 showed a stronger association with a reported SARS-CoV-2 infection ($P = 1.79 \times 10^{-34}$; OR (95% confidence interval (CI)) = 1.15 (1.13–1.18)) than hospitalization ($P = 1.05 \times 10^{-5}$; OR (95% CI) = 1.12 (1.06–1.19)). For this locus—which contains additional independent signals—the linkage-disequilibrium (LD) pattern is discordant with the $P$-value expectation (Extended Data Fig. 6 and Supplementary Note), pointing to a key missing causal variant or to a potentially undiscovered multi-allelic or structural variant in this locus.

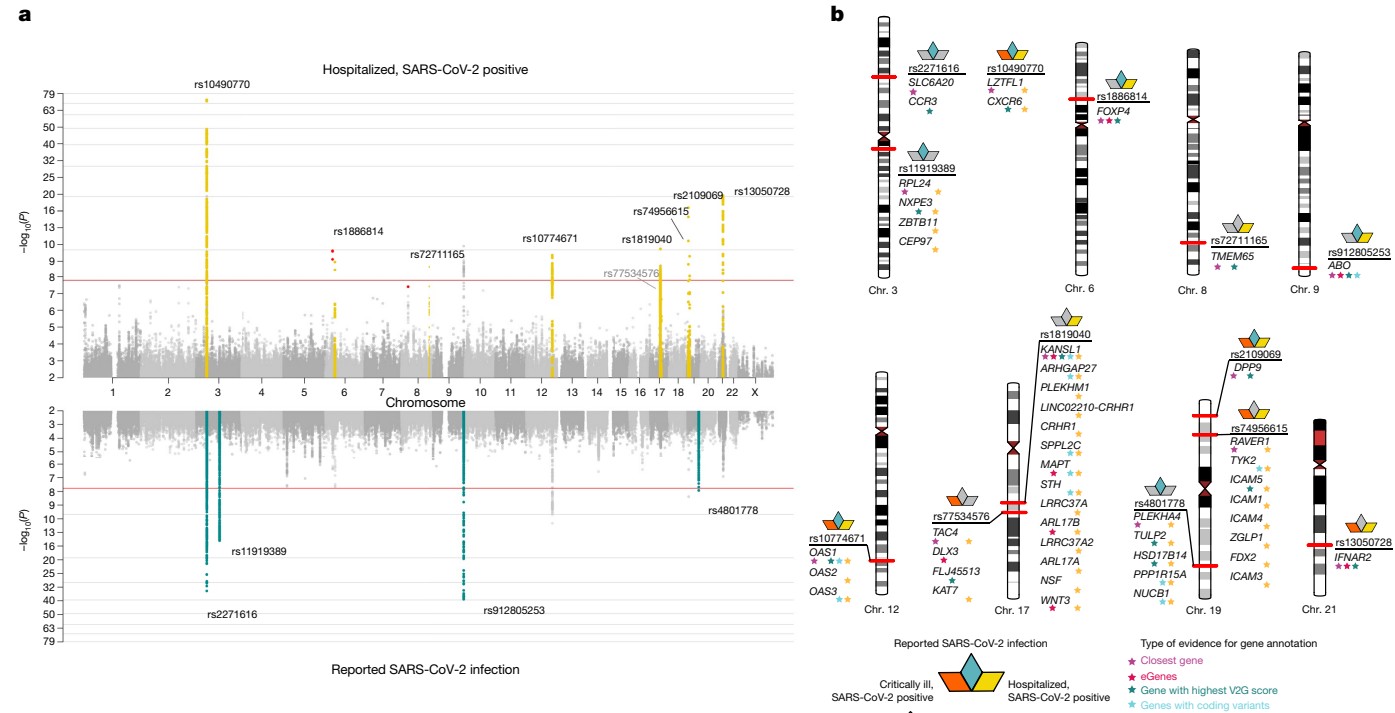

**Fig. 2 | Genome-wide association results for COVID-19. a**, Top, results of a genome-wide association study of hospitalized cases of COVID-19 ($n$ = 13,641 cases and $n$ = 2,070,709 controls). Bottom, the results of reported SARS-CoV-2 infections ($n$ = 49,562 cases and $n$ = 1,770,206 controls). Loci highlighted in yellow (top) represent regions associated with the severity of the COVID-19 manifestation—that is, increased odds of more severe COVID-19 phenotypes. Loci highlighted in green (bottom) are regions associated with susceptibility to a SARS-CoV-2 infection—that is, the effect is the same across mild and severe COVID-19 phenotypes. We highlight in red genome-wide significant variants that had high heterogeneity across contributing studies and that were therefore excluded from the list of loci found. **b**, Results of gene prioritization using different evidence measures of gene annotation. Genes in the LD region, genes with coding variants and eGenes (fine-mapped *cis*-eQTL variant PIP > 0.1 in GTEx Lung) are annotated if in LD with a COVID-19 lead variant ($r^2$ > 0.6). V2G, highest gene prioritized by the V2G score of Open Target Genetics.

By contrast, 9 out of the 13 loci were associated with increased risk of severe symptoms with significantly larger odds ratios for hospitalized COVID-19 compared with the mildest phenotype of reported SARS-CoV-2 infection (eight loci were below the threshold of $P$ < 0.004 (test for effect size difference) and, in addition, the lead variant rs10774671 had a clear increase in odds ratios despite not passing this threshold) (Supplementary Table 2). We further compared the odds ratios for these nine loci for critical illness due to COVID-19 versus hospitalized due to COVID-19, and found that these loci exhibited a general increase in effect risk for critical illness (Methods, Extended Data Fig. 7a and Supplementary Table 4), but the lower power for association analysis of critically ill COVID-19 means that these results should be considered as suggestive. Overall, these results indicated that these nine loci were more likely to be associated with progression of the disease and worse outcome from SARS-CoV-2 infection compared to being associated with susceptibility to SARS-CoV-2 infection.

For some of these analyses, the controls were simply existing population controls without knowledge of SARS-CoV-2 infection or COVID-19 status, which may bias effect size estimates as some of these individuals may have either become infected with SARS-CoV-2 or developed COVID-19. We perform several sensitivity analyses (Extended Data Fig. 7b, Supplementary Note and Supplementary Table 4) in which we show that using population controls can be a valid and powerful strategy for host genetic discovery of infectious disease, and particularly those that are widespread and with rare severe outcomes.

## Gene prioritization and association with other traits

To better understand the potential biological mechanism of each locus, we applied several approaches to prioritize candidate causal genes and explore additional associations with other diseases and traits. Of the 13 genome-wide significant loci, we found that nine loci implicated biologically plausible genes (Supplementary Tables 2, 5). Protein-altering variants in LD with lead variants implicated genes at six loci, including *TYK2* (chromosome and cytogenetic band (chr.) 19p13.2) and *PPP1R15A* (chr. 19q13.33). The COVID-19 lead variant rs74956615T>A in *TYK2*, which confers risk for critical illness (OR (95% CI) = 1.43 (1.29–1.59), $P$ = 9.71 × 10⁻¹²) and hospitalization due to COVID-19 (OR (95% CI) = 1.27 (1.18–1.36), $P$ = 5.05 × 10⁻¹⁰) is correlated with the missense variant rs34536443:G>C (p.Pro1104Ala; $r^2$ = 0.82) . This is consistent with the primary immunodeficiency described with complete *TYK2* loss of function[3] as this variant is known to reduce function[18,19]. By contrast, this missense variant was previously reported to be protective against autoimmune diseases (Extended Data Fig. 8 and Supplementary Table 6), including rheumatoid arthritis (OR = 0.74, $P$ = 3.0 × 10⁻⁸; UK Biobank SAIGE) and hypothyroidism (OR = 0.84, $P$ = 1.8 × 10⁻¹⁰; UK Biobank). At the 19q13.33 locus, the lead variant rs4801778, which was significantly associated with a reported SARS-CoV-2 infection (OR (95% CI) = 0.95 (0.93–0.96), $P$ = 2.1 × 10⁻⁸), is in LD ($r^2$ = 0.93) with a missense variant rs11541192:G>A (p.Gly312Ser) in *PPP1R15A*.

A lung-specific *cis*-expression quantitative trait loci (*cis*-eQTLs) from GTEx v.8[20] ($n$ = 515) and the Lung eQTL Consortium[21] ($n$ = 1,103) provided further support for a subset of loci (Supplementary Table 7), including *FOXP4* (chr. 6p21.1) and *ABO* (chr. 9q34.2), *OAS1/OAS3/OAS2* (chr.12q24.13) and *IFNAR2/IL10RB* (21q22.11), where the COVID-19-associated variants modify gene expression in lung. Furthermore, our phenome-wide association study (PheWAS) analysis (Supplementary Table 6) implicated three additional loci related to lung function, with modest lung eQTL evidence—that is, the lead variant was not fine-mapped but significantly associated. An intronic variant

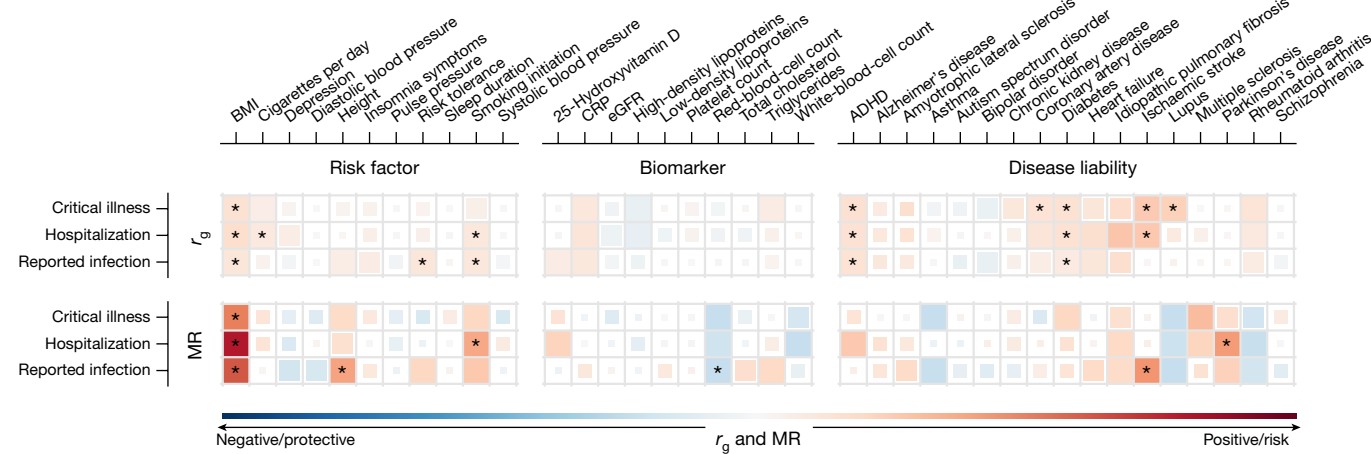

**Fig. 3 | Genetic correlations and Mendelian randomization causal estimates between 38 traits and COVID-19 critical illness, hospitalization and reported SARS-CoV-2 infection.** Larger squares correspond to *P* values with higher significance, with genetic correlations ($r_g$) or Mendelian randomization (MR) causal estimates significantly different from zero. The size of each coloured square indicates the magnitude of the *P* value, with *P* < 0.05 shown as a full-sized square, *P* = 0.05–0.1 as a large square, *P* = 0.1–0.5 as a medium square and *P* > 0.5 as a small square. Genetic correlations or causal estimates that are significantly different from zero at an FDR of 5% are marked with an asterisk. Two-sided *P* values were calculated using LDSC for genetic correlations and inverse-variance-weighted analysis for Mendelian randomization. ADHD, attention-deficit hyperactivity disorder; BMI, body mass index; CRP, C-reactive protein; eGFR, estimated glomerular filtration rate.

rs2109069:G>A in *DPP9* (chr. 19p13.3), which is positively associated with critical illness, was previously reported to be risk-increasing for interstitial lung disease (tag lead variant rs12610495:A>G (p.Leu8Pro); OR = 1.29, $P = 2.0 \times 10^{-12}$)[5]. The COVID-19 lead variant rs1886814:A>C in the *FOXP4* locus is correlated ($r^2 = 0.64$) with a lead variant of lung adenocarcinoma (tag variant is rs7741164; OR = 1.2, $P = 6.0 \times 10^{-13}$)[6,22] and similarly with a lead variant reported for subclinical interstitial lung disease[23]. In severe COVID-19, lung cancer and interstitial lung disease, the minor, expression-increasing allele is associated with increased risk. We also found that intronic variants (chr. 1q22) and rs1819040:T>A in *KANSL1* (chr. 17q21.31), associated with protection against hospitalization due to COVID-19, were previously reported for reduced lung function (for example, tag lead variant rs141942982:G>T; OR (95% CI) = 0.96 (0.95–0.97), $P = 1.00 \times 10^{-20}$)[7]. Notably, the 17q21.31 locus is a well-known locus for structural variants containing a megabase inversion polymorphism (H1 and inverted H2 forms) and complex copy-number variations, in which the inverted H2 forms were shown to be positively selected in European individuals[24,25].

Lastly, there are two loci in the 3p21.31 region with varying genes prioritized by different methods for different independent signals. For the severity lead variant rs10490770:T>C, we prioritized *CXCR6* with the Variant2Gene (V2G) algorithm[26], although *LZTFL1* is the closest gene. The *CXCR6* has a role in chemokine signalling[27] and *LZTFL1* has been implicated in lung cancer[28]. rs2271616:G>T, which is associated with susceptibility, tags a complex region including several independent signals (Supplementary Note) that are all located within the gene body of *SLC6A20*, which encodes a protein that is known to functionally interact with the SARS-CoV-2 receptor ACE2[29]. However, none of the lead variants in the 3p21.31 region has been previously associated with other traits or diseases in our PheWAS analysis. Although these results provide supporting in silico evidence for candidate causal gene prioritization, further functional characterization is needed. Detailed locus descriptions and LocusZoom plots are provided in Supplementary Fig. 2.

## Polygenic architecture of COVID-19

To further investigate the genetic architecture of COVID-19, we used results from meta-analyses including samples from European ancestries (sample sizes are described in the Methods and Supplementary Table 1) to estimate the heritability explained by common single-nucleotide polymorphisms—that is, the proportion of variation in the two phenotypes that was attributable to common genetic variants—and to determine whether heritability of COVID-19 phenotypes was enriched in genes that were specifically expressed in certain tissues[30] from the GTEx dataset[31]. We detected low, but significant, heritability across all three analyses (<1% on observed scale, all *P* values were *P* < 0.0001) (Supplementary Table 8). The values are low compared to previously published studies[14], but may be explained by differences in the reported estimate scale (observed versus liability), the specific method used, disease-prevalence estimates, phenotypic differences between patient cohorts or ascertainment of controls. Despite the low reported values, we found that heritability of a reported SARS-CoV-2 infection was significantly enriched in genes that were specifically expressed in the lung ($P = 5.0 \times 10^{-4}$) (Supplementary Table 9). These findings, together with the genome-wide significant loci identified in the meta-analyses, suggest that there is a significant polygenic architecture that can be better leveraged with future, larger, sample sizes.

## Genetic correlation and Mendelian randomization

Genetic correlations ($r_g$) between the three COVID-19 phenotypes was high, although lower correlations were observed between hospitalized COVID-19 and reported SARS-CoV-2 infection (critical illness versus hospitalized: $r_g$ (95% CI) = 1.37 (1.08–1.65), $P = 2.9 \times 10^{-21}$; critical illness versus reported SARS-CoV-2 infection, $r_g$ (95% CI) = 0.96 (0.71–1.20), $P = 1.1 \times 10^{-14}$; hospitalized versus reported SARS-CoV-2 infection: $r_g$ (95% CI) = 0.85 (0.68–1.02), $P = 1.1 \times 10^{-22}$). To better understand which traits are genetically correlated and/or potentially causally associated with COVID-19 hospitalization, critical illness and reported SARS-CoV-2 infection, we chose a set of 38 disease, health and neuropsychiatric phenotypes as potential COVID-19 risk factors based on their clinical correlation with disease susceptibility, severity or mortality (Supplementary Table 10).

We found evidence (false-discovery rate (FDR) < 0.05) of significant genetic correlations between nine traits and hospitalized COVID-19 and reported SARS-CoV-2 infection (Fig. 3, Extended Data Fig. 9 and Supplementary Table 11). Notably, genetic liability to ischaemic

stroke was only significantly positively correlated with critical illness or hospitalization due to COVID-19, but not with a higher likelihood of reported SARS-CoV-2 infection (infection $r_g$ = 0.019 versus hospitalization $r_g$ = 0.41, $z$ = 2.7, $P$ = 0.006; infection $r_g$ = 0.019 versus critical illness $r_g$ = 0.40, $z$ = 2.49, $P$ = 0.013).

We next used two-sample Mendelian randomization to infer potentially causal relationships between these traits. After correcting for multiple testing (FDR < 0.05), eight exposure–COVID-19 trait pairs showed suggestive evidence of a causal association (Fig. 3, Extended Data Fig. 10, Supplementary Table 12 and Supplementary Fig. 3). Five of these associations were robust to potential violations of the underlying assumptions of Mendelian randomization. Corroborating our genetic correlation results and evidence from epidemiological studies, genetically predicted higher body-mass index (OR (95% CI) = 1.4 (1.3–1.6), $P$ = 8.5 × 10⁻¹¹) and smoking (OR (95% CI) = 1.9 (1.3–2.8), $P$ = 0.0012) were associated with increased risk of COVID-19 hospitalization, with body-mass index also being associated with increased risk of SARS-CoV-2 infection (OR (95% CI) = 1.1 (1.1–1.2), $P$ = 4.8 × 10⁻⁷). Genetically predicted increased height (OR (95% CI) = 1.1 (1–1.1)), $P$ = 8.9 × 10⁻⁴) was associated with an increased risk of reported SARS-CoV-2 infection, whereas a genetically predicted higher red-blood-cell count (OR (95% CI) = 0.93 (0.89–0.96), $P$ = 5.7 × 10⁻⁵) was associated with a reduced risk of reported SARS-CoV-2 infection. Despite evidence of a genetic correlation between type II diabetes and COVID-19 outcomes, there was no evidence of a causal association in the Mendelian randomization analyses, which suggests that the observed genetic correlations are due to pleiotropic effects between body-mass index and type 2 diabetes. Further sensitivity analyses relating to sample overlap are discussed in the Supplementary Information.

## Discussion

The COVID-19 HGI has brought together investigators from across the world to advance genetic discovery for SARS-CoV-2 infection and severe COVID-19 disease. We report 13 genome-wide significant loci associated with some aspect of SARS-CoV-2 infection or COVID-19. Many of these loci overlap with previously reported associations with lung-related phenotypes or autoimmune or inflammatory diseases, but some loci have no obvious candidate gene.

Four out of the thirteen genome-wide significant loci showed similar effects in the reported SARS-CoV-2 infection analysis (a proxy for disease susceptibility) and all-hospitalized COVID-19 (a proxy for disease severity). Of these, one locus was in close proximity to, yet independent of, the major genetic signal for COVID-19 severity at the 3p21.31 locus. Notably, this locus was associated with COVID-19 susceptibility rather than severity. The locus overlaps *SLC6A20*, which encodes an amino acid transporter that interacts with ACE2. Nonetheless, we caution that more data are needed to resolve the nature of the relationship between genetic variation and COVID-19 at this locus, particularly as the physical proximity, LD structure and patterns of association suggest that untagged genetic variation could drive the association signal in the region. Our findings support the notion that some genetic variants, most notably at the *ABO* and *PPP1R15A* loci, in addition to *SLC6A20*, can indeed affect susceptibility to infection rather than progression to severe COVID-19 once infected.

Several of the loci reported here—as noted in previous publications[12,14]—intersect with well-known genetic variants that have established genetic associations. Examples of these include variants at *DPP9* and *FOXP4*, which show previous evidence of increasing risk for interstitial lung disease[5], and missense variants within *TYK2* that show a protective effect on several autoimmune-related diseases[32–35]. Together with the heritability enrichment observed in genes expressed in lung tissues, these results highlight the involvement of lung-related biological pathways in the development of severe COVID-19. Several other loci show no previously documented genome-wide significant

associations, despite the high significance and attractive candidate genes for COVID-19 (for example, *CXCR6*, *LZTFL1*, *IFNAR2* and *OAS1/OAS2/OAS3* loci). The previously reported associations for the strongest association for COVID-19 severity at the 3p21.31 locus and monocytes count are likely to be due to proximity and not a true co-localization.

Increasing the global representation in genetic studies enhances the ability to detect novel associations. Two of the loci that affect disease severity were only discovered by including the four studies of individuals with East Asian ancestry. One of these loci—close to *FOXP4*—is common particularly in East Asian participants (32%) as well as admixed American participants in the Americas (20%) and Middle Eastern participants (7%), but has a low frequency in most European ancestries (2–3%) in our data. Although we cannot be certain of the mechanism of action, the *FOXP4* association is an attractive biological target, as it is expressed in the proximal and distal airway epithelium[36] and has been shown to have a role in controlling epithelial cell fate during lung development[37]. The COVID-19 HGI continues to pursue expansion of the datasets included in the analyses of the consortium to populations from underrepresented populations in upcoming data releases. We plan to release ancestry-specific results in full once the sample sizes allow for a well-powered meta-analysis.

Care should be taken when interpreting the results from a meta-analysis because of challenges with case and control ascertainment and collider bias (see Supplementary Note for a more detailed discussion on study limitations). Drawing a comprehensive and reproducible map of the host genetics factors associated with COVID-19 severity and SARS-CoV-2 requires a sustained international effort to include diverse ancestries and study designs. To accelerate downstream research and therapeutic discovery, the COVID-19 HGI regularly publishes meta-analysis results from periodic data freezes on the website https://www.covid19hg.org/ and provides an interactive explorer through which researchers can browse the results and the genomic loci in more detail. Future work will be required to better understand the biological and clinical value of these findings. Continued efforts to collect more samples and detailed phenotypic data should be endorsed globally, allowing for more thorough investigation of variable, heritable symptoms, particularly in light of the newly emerging strains of SARS-CoV-2, which may provoke different host responses that lead to disease.

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

**COVID-19 Host Genetics Initiative**
**Leadership**
Mari E. K. Niemi[1,1280], Juha Karjalainen[1,1280], Rachel G. Liao[2], Benjamin M. Neale[4,1280✉], Mark Daly[1,2,3,1280✉] & Andrea Ganna[1,2,3✉]

**Writing group**
**Writing group leaders**
Mari E. K. Niemi[1,1280], Gita A. Pathak[6], Shea J. Andrews[7] & Masahiro Kanai[2]

**Writing group members**
Kumar Veerapen[2], Israel Fernandez-Cadenas[8], Eva C. Schulte[9,10,11], Pasquale Striano[12,13], Minttu Marttila[75], Camelia Minica[2], Eirini Marouli[14], Mohd Anisul Karim[15,38], Frank R. Wendt[6], Jeanne Savage[16], Laura Sloofman[7], Guillaume Butler-Laporte[17,53], Han-Na Kim[18], Stavroula Kanoni[14], Yukinori Okada[19], Jinyoung Byun[20], Younghun Han[20], Mohammed Jashim Uddin[21], George Davey Smith[22], Cristen J. Willer[23,24,25] & Joseph D. Buxbaum[26]

**Analysis group**
**Manuscript analyses team leader**
Juha Karjalainen[1,1280]

**Manuscript analyses team member: meta-analysis**
Juha Mehtonen[1]

**Manuscript analyses team member: heritability, methods and supplements**
Mari E. K. Niemi[1,1280]

**Manuscript analyses team member: PHEWAS**
Gita A. Pathak[6]

**Manuscript analyses team member: Mendelian randomization**
Shea J. Andrews[7]

**Manuscript analyses team member: PC projection and gene prioritization**
Masahiro Kanai[2]

**Manuscript analyses team member: gene prioritization**
Hilary Finucane[3,29]

**Manuscript analyses team member: sensitivity analysis**
Mattia Cordioli[1]

**Manuscript analyses team members: PC projection**
Alicia R. Martin[3,29] & Wei Zhou[3,29]

**In silico analysis team members**
Mattia Cordioli[1], Bogdan Pasaniuc[32], Hanna Julienne[33], Hugues Aschard[33], Huwenbo Shi[34], Loic Yengo[35], Renato Polimanti[6], Maya Ghoussaini[15,38], Jeremy Schwartzentruber[15,38], Ian Dunham[15,40] & Joseph D. Buxbaum[26]

**Project management group**
**Project management leader**
Rachel G. Liao[2]

**Project management support**
Karolina Chwialkowska[42], Margherita Francescatto[43], Amy Trankiem[2] & Mary K. Balaconis[2]

**Phenotype steering group**
Lea Davis[44], Andrea Ganna[1,2,3], Sulggi Lee[45], James Priest[46], Alessandra Renieri[47,65], Vijay G. Sankaran[49], David van Heel[50], Patrick Deelen[51,52], J. Brent Richards[17,53,55,56], Tomoko Nakanishi[17,56,57], Les Biesecker[59], V. Eric Kerchberger[44] & J. Kenneth Baillie[60,61,62]

**Data dictionary**
Alessandra Renieri[47,65], Francesca Mari[63,64,65], Anna Bernasconi[66], Stefano Ceri[67] & Arif Canakoglu[67]

**Scientific communication group**
**Scientific communication leaders**
Kumar Veerapen[2] & Brooke Wolford[68]

**Scientific communication members**
Amy Trankiem[2], Annika Faucon[69], Atanu Kumar Dutta[70], Claudia Schurmann[71], Emi Harry[72], Ewan Birney[73], Gita A. Pathak[6], Huy Nguyen[2], Jamal Nasir[74], James Priest[46], Mari Kaunisto[1], Minttu Marttila[75], Matthew Solomonson[2], Nicole Dueker[76], Nirmal Vadgama[46], Shea J. Andrews[7], Sophie Limou[78], Rachel G. Liao[2] & Andrea Ganna[1,2,3]

**Translators**
Souad Rahmouni[79], Hamdi Mbarek[80], Dima Darwish[80], Atanu Kumar Dutta[70], Md Mesbah Uddin[82,83], Israel Fernandez-Cadenas[8], Raquel Albertos[84], Jordi Pérez-Tur[85,86,87], Ruolin Li[88], Lasse Folkersen[89], Ida Moltke[90], Nils Koelling[91], Eva C. Schulte[9,10,11], Claudia Schurmann[71], Alexander Teumer[93,94], Athanasios Kousathanas[95], Eirini Marouli[14],

Alicia Utrilla[96], Ricardo A. Verdugo[97], Ruth Zárate[98], Carolina Medina-Gómez[88], David Gómez-Cabrero[100,101], Elena Carnero-Montoro[102], Jordi Pérez-Tur[85,86,87], Israel Fernandez-Cadenas[8], Carmen L. Cadilla[103], Andrés Moreno-Estrada[104], Adriana Garmendia[104], Leire Moya[105], Bahar Sedaghati-Khayat[88], Minttu Marttila[75], Palwendé Romuald Boua[106], Guillaume Butler-Laporte[17,53], Marie-Julie Favé[107], Laurent Francioli[2], Audrey Lemaçon[108], Sophie Limou[78], Isabelle Migeotte[109], Atanu Kumar Dutta[70], Sanjay Patel[70], Reka Varnai[112], Jozsef L. Szentpeteri[112], Csilla Sipeky[113], Francesca Colombo[114], Margherita Francescatto[43], Kathrin von Hohenstaufen[115], Pietro Lio[116], Costanza Vallerga[88], Pasquale Striano[12,13], Qingbo Wang[2], Masahiro Kanai[2], Yosuke Tanigawa[46], Tomoko Nakanishi[17,56,57], Hogune Im[117], Chulho Han[117], Han Song[117], Jiwoo Lim[117], Younhee Lee[117], Sugyeong Kim[117], Sangyoon Im[1281], Biljana Atanasovska[119], Hajar Fauzan Ahmad[120], Kumar Veerapen[2], Cindy Boer[88], Philip Jansen[122], Lude Franke[123], Karolina Chwialkowska[42], Elżbieta Kaja[124], Dorota Pasko[95], Ingrid Kennis-Szilagyi[88], Sergey A. Kornilov[125], Vid Prijatelj[88], Ivana Prokić[88], Ilangkumaran Sivanadhan[126], Sarala Perumal[127], Sahar Esmaeeli[128], Nathaniel M. Pearson[129], Ruth Zárate[98] & Mohd Anisul Karim[15,38]

**Website Development**
**Website development leaders**
Huy Nguyen[2] & Matthew Solomonson[2]

**23andMe**
**Analysis team leader**
Adam Auton[130]

**Data collection leaders**
23andMe COVID-19 Team*, Janie F. Shelton[130] & Anjali J. Shastri[130]

**Analysis team members**
Teresa Filshtein-Sonmez[130], Daniella Coker[130], Antony Symons[130], Jorge Esparza-Gordillo[131], Stella Aslibekyan[130] & Jared O'Connell[130]

**Data collection members**
Chelsea Ye[130] & Catherine H. Weldon[130]

**23andMe COVID-19 Team**
Adam Auton[130]

**ACCOuNT**
**Analysis team leader**
Minoli Perera[132]

**Data collection leaders**
Kevin O'Leary[133], Matthew Tuck[134], Travis O'Brien[135], David Meltzer[136], Peter O'Donnell[137] & Edith Nutescu[138]

**Analysis team members**
Guang Yang[132]

**Data collection members**
Cristina Alarcon[132], Stefanie Herrmann[132], Sophia Mazurek[133], Jeff Banagan[134], Zacharia Hamidi[134], April Barbour[144], Noora Raffat[136] & Diana Moreno[138]

**Admin team member**
Paula Friedman[132]

**Amsterdam UMC COVID Study Group**
**Analysis team leader**
Bart Ferwerda[148]

**Data collection leaders**
Diederik van de Beek[148], Matthijs C. Brouwer[148], Alexander P. J. Vlaar[149] & W. Joost Wiersinga[150]

**Analysis team members**
Danielle Posthuma[16], Elleke Tissink[16], A. H. Koos Zwinderman[151] & Emil Uffelmann[16]

**Data collection members**
Michiel van Agtmael[150], Anne Geke Algera[149], Frank van Baarle[149], Diane Bax[152], Martijn Beudel[148], Harm Jan Bogaard[153], Marije Bomers[150], Peter I. Bonta[153], Lieuwe Bos[149], Michela Botta[149], Justin de Brabander[150], Godelieve de Bree[150], Sanne de Bruin[149], Marianna Bugiani[154], Esther Bulle[149], Osoul Chouchane[150], Alex Cloherty[152], Dave Dongelmans[149], Paul Elbers[149], Lucas Fleuren[149], Suzanne Geerlings[150], Bart Geerts[155], Theo Geijtenbeek[152], Armand Girbes[149], Bram Goorhuis[150], Martin P. Grobusch[150], Florianne Hafkamp[152], Laura Hagens[149], Jorg Hamann[156], Vanessa Harris[150], Robert Hemke[157], Sabine M. Hermans[150], Leo Heunks[149], Markus Hollmann[155], Janneke Horn[149], Joppe W. Hovius[150], Menno D. de Jong[158], Rutger Koning[148], Niels van Mourik[149], Jeannine Nellen[150], Esther J. Nossent[153], Frederique Paulus[149], Edgar Peters[150], Tom van der Poll[150], Bennedikt Preckel[152], Jan M. Prins[150], Jorinde Raasveld[149], Tom Reijnders[150], Michiel Schinkel[150], Marcus J. Schultz[149], Alex Schuurman[150], Kim Sigaloff[150], Marry Smit[149], Cornelis S. Stijnis[150], Willemke Stilma[159], Charlotte Teunissen[149], Patrick Thoral[149], Anissa Tsonas[150], Marc van der Valk[150], Denise Veelo[155], Heder de Vries[149], Michèle van Vugt[150], Dorien Wouters[159], René P. Minnaar[160], Adrie Kromhout[160], Kees W. J. van Uffelen[160] & Ruud A. Wolterman[161]

**AncestryDNA COVID-19 Research Study**
**Analysis team leader**
Genevieve Roberts[162]

**Data collection leader**
Danny Park[162]

**Admin team leader**
Catherine A. Ball[162]

**Analysis team members**
Marie Coignet[162], Shannon McCurdy[162], Spencer Knight[162], Raghavendran Partha[162] & Brooke Rhead[162]

**Data collection members**
Miao Zhang[162], Nathan Berkowitz[162], Michael Gaddis[162], Keith Noto[162], Luong Ruiz[162] & Milos Pavlovic[162]

**Admin team members**
Eurie L. Hong[162], Kristin Rand[162], Ahna Girshick[162], Harendra Guturu[162] & Asher Haug Baltzell[162]

**BelCovid**
**Analysis team leader**
Mari E. K. Niemi[1,2180]

**Data collection leaders**
Isabelle Migeotte[109], Souad Rahmouni[163] & Julien Guntz[164]

**Admin team leader**
Yves Beguin[165]

**Analysis team members**
Mattia Cordioli[1], Sara Pigazzini[1] & Lindokuhle Nkambule[3,29,167]

**Data collection members**
Youssef Bouysran[168], Adeline Busson[168], Xavier Peyrassol[168], Françoise Wilkin[168], Bruno Pichon[168], Guillaume Smits[168], Isabelle Vandernoot[168], Jean-Christophe Goffard[169], Michel Georges[163], Michel Moutschen[170], Benoit Misset[170], Gilles Darcis[170], Julien Guiot[170], Laurent Jadot[164], Samira Azarzar[170], Patricia Dellot[170], Stéphanie Gofflot[165], Sabine Claassen[164], Axelle Bertrand[170], Gilles Parzibut[170], Mathilde Clarinval[170], Catherine Moermans[170], Olivier Malaise[170], Kamilia El Kandoussi[165], Raphaël Thonon[165], Pascale Huynen[170], Alyssia Mesdagh[170], Sofia Melo[163], Nicolas Jacques[163], Emmanuel Di Valentin[163], François Giroule[163], Alice Collignon[163], Coraline Radermecker[163], Marielle Lebrun[163], Alice Collignon[163], Hélène Perée[163], Samuel Latour[163], Olivia Barada[163], Judit Sanchez[163], Claire Josse[170], Bouchra Boujemla[170], Margot Meunier[163], Emeline Mariavelle[163], Sandy Anania[163], Hélène Gazon[163], Danusia Juszczak[170], Marjorie Fadeur[170], Séverine Camby[170], Christelle Meuris[170], Marie Thys[170], Jessica Jacques[170], Monique Henket[170], Philippe Léonard[170], Frederic Frippiat[170], Jean-Baptiste Giot[170], Anne-Sophie Sauvage[170], Christian Von Frenckell[170], Myriam Mni[163], Marie Wéry[163], Alicia Staderoli[170], Yasmine Belhaj[163] & Bernard Lambermont[170]

**Biobanque Quebec COVID-19**
**Analysis team leader**
Tomoko Nakanishi[17,56,57]

**Data collection leader**
David R. Morrison[17]

**Admin team leaders**
Vincent Mooser[56,174] & J. Brent Richards[17,53,55,56]

**Analysis team members**
Guillaume Butler-Laporte[17,53], Vincenzo Forgetta[17] & Rui Li[56,174]

**Data collection members**
Biswarup Ghosh[17], Laetitia Laurent[17], Alexandre Belisle[56,174], Danielle Henry[17], Tala Abdullah[17], Olumide Adeleye[17], Noor Mamlouk[17], Nofar Kimchi[17], Zaman Afrasiabi[17], Nardin Rezk[17], Branka Vulesevic[17], Meriem Bouab[17], Charlotte Guzman[17], Louis Petitjean[17], Chris Tselios[17], Xiaoqing Xue[17], Jonathan Afilalo[17], Marc Afilalo[177,178], Maureen Oliveira[179], Bluma Brenner[180], Nathalie Brassard[181] & Madeleine Durand[182,183]

**Admin team members**
Erwin Schurr[184], Pierre Lepage[56,174], Jiannis Ragoussis[56,174], Daniel Auld[56,174], Michaël Chassé[183,185], Daniel E. Kaufmann[183,186], G. Mark Lathrop[56,174] & Darin Adra[17]

**BioVU**
**Analysis team leaders**
Lea K. Davis[187,188], Nancy J. Cox[187,188] & Jennifer E. Below[187,188]

**Analysis team members**
Julia M. Sealock[187,188], Annika B. Faucon[187,188], Megan M. Shuey[187,188], Hannah G. Polikowsky[187,188], Lauren E. Petty[187,188], Douglas M. Shaw[187,188], Hung-Hsin Chen[187,188] & Wanying Zhu[187,188]

**Bonn Study of COVID-19 Genetics**
**Data collection leader**
Kerstin U. Ludwig[189]

**Analysis team members**
Julia Schröder[189] & Carlo Maj[190]

**Data collection members**
Selina Rolker[189], Markus M. Nöthen[189], Julia Fazaal[189], Verena Keitel[191], Björn-Erik Ole Jensen[191], Torsten Feldt[191], Ingo Kurth[192], Nikolaus Marx[193], Michael Dreher[194], Isabell Pink[195], Markus Cornberg[196], Thomas Illig[197], Clara Lehmann[198,199,200], Philipp Schommers[198,199,200], Max Augustin[198], Jan Rybniker[198], Lisa Knopp[191], Thomas Eggermann[192], Sonja Volland[197], Janine Altmüller[202], Marc M. Berger[203], Thorsten Brenner[203], Anke Hinney[204], Oliver Witzke[205], Robert Bals[206], Christian Herr[206], Nicole Ludwig[207] & Jörn Walter[208]

**CHRIS**
**Analysis team leader**
Christian Fuchsberger[209]

**Data collection leaders**
Cristian Pattaro[209] & Alessandro De Grandi[209]

**Admin team leader**
Peter Pramstaller[209]

**Analysis team members**
David Emmert[209], Roberto Melotti[209] & Luisa Foco[209]

**Admin team members**
Deborah Mascalzoni[209], Martin Gögele[209], Francisco Domingues[209] & Andrew Hicks[209]

**Colorado Center for Personalized Medicine (CCPM)**
**Analysis team leader**
Christopher R. Gignoux[210]

**Data collection leaders**
Stephen J. Wicks[210] & Kristy Crooks[210]

**Admin team leader**
Kathleen C. Barnes[210]

**Analysis team members**
Michelle Daya[210], Jonathan Shortt[210], Nicholas Rafaels[210] & Sameer Chavan[210]

**Columbia University COVID-19 Biobank**
**Analysis team leaders**
David B. Goldstein[211] & Krzysztof Kiryluk[212]

**Data collection leaders**
Soumitra Sengupta[213], Wendy Chung[214] & Muredach P. Reilly[215]

**Analysis team members**
Atlas Khan[215], Chen Wang[215], Gundula Povysil[216], Nitin Bhardwaj[216], Ali G. Gharavi[215] & Iuliana Ionita-Laza[217]

**Data collection members**
Ning Shang[215], Sheila M. O'Byrne[215], Renu Nandakumar[215], Amritha Menon[213], Yat S. So[213] & Eldad Hod[218]

**Admin team member**
Danielle Pendrick[218]

**Corea (Genetics of COVID-19-related Manifestation)**
**Analysis team leader**
Han-Na Kim[219,220]

**Data collection leaders**
Soo-Kyung Park[221], Hyung-Lae Kim[222], Chang Kyung Kang[223], Hyo-Jung Lee[224] & Kyoung-Ho Song[225]

**Admin team leaders**
Kyung Jae Yoon[226,227,228] & Nam-Jong Paik[229,230]

**Analysis team members**
Woojin Seok[231] & Heejun Yoon[232]

**Data collection members**
Eun-Jeong Joo[233], Yoosoo Chang[234,235], Seungho Ryu[234,235], Wan Beom Park[223], Jeong Su Park[236], Kyoung Un Park[236], Sin Young Ham[225], Jongtak Jung[225], Eu Suk Kim[225] & Hong Bin Kim[225]

**COVID-19-Hostage**
**Analysis team leaders**
David Ellinghaus[237,238], Frauke Degenhardt[237], Mario Cáceres[239,240], Simonas Juzenas[237] & Tobias L. Lenz[241,242]

**Data collection leaders**
Agustín Albillos[243,244], Antonio Julià[245], Bettina Heidecker[246], Eva C. Schulte[9,10,11], Federico Garcia[249,250], Florian Kurth[246], Florian Tran[237], Frank Hanses[251,252], Heinz Zoller[253], Jan C. Holter[254,255], Javier Fernández[256,257], Leif Erik Sander[246], Philip Rosenstiel[237], Philipp Koehler[198,259,260], Rafael de Cid[261], Rosanna Asselta[262,263], Stefan Schreiber[237,265], Ute Hehr[266], Daniele Prati[285], Guido Baselli[285], Luca Valenti[285,316], Luis Bujanda[244,322,329], Jesus M. Banales[244,321,322], Stefano Duga[262,263], Mauro D'Amato[321,322,344], Manuel Romero-Gómez[1241,244], Maria Buti[244,292,298] & Pietro Invernizzi[286,287]

**Admin team leaders**
Andre Franke[237,267], Johannes R. Hov[254,268,269,270], Tom H. Karlsen[254,268,269,270], Trine Folseraas[254,268,269,270] & Douglas Maya-Miles[1244,1245,244]

**Analysis team members**
Ana Teles[241,242], Clinton Azuure[241], Eike Matthias Wacker[237], Florian Uellendahl-Werth[237], Hesham ElAbd[237], Jatin Arora[29,272,273,274,275], Jon Lerga-Jaso[239], Lars Wienbrandt[237], Malte Christoph Rühlemann[237], Mareike Wendorff[237], May Sissel Vadla[277], Ole Bernt Lenning[276], Onur Özer[241,242], Ronny Myhre[278], Soumya Raychaudhuri[29,272,273,274,275,279], Anja Tanck[237], Christoph Gassner[237,1240], Georg Hemmrich-Stanisak[237], Jan Kässens[237], Maria E. Figuera Basso[237], Martin Schulzky[237], Michael Wittig[237], Nicole Braun[237,267], Tanja Wesse[237], Wolfgang Albrecht[237] & Xiaoli Yi[237]

**Data collection members**
Aaron Blandino Ortiz[280], Adolfo Garrido Chercoles[281], Agustín Ruiz[282,283], Alberto Mantovani[262,263], Aleksander Rygh Holten[254,284], Alena Mayer[246], Alessandro Cherubini[285], Alessandro Protti[262,263], Alessio Aghemo[262,263], Alessio Gerussi[286,287], Alfredo Ramirez[288,289,290,291], Alice Braun[246], Ana Barreira[292], Ana Lleo[262,263], Anders Benjamin Kildal[293], Andrea Ganna[1,2,3,1280], Andreas Glück[294], Anna Carreras Nolla[261], Anna Latiano[295], Anne Ma Dyrhol-Riise[254,296], Antonio Muscatello[285], Antonio Voza[262], Ariadna Rando-Segura[297,298], Aurora Solier[299,1226], Banasik Karina[238], Beatriz Cortes[261], Beatriz Mateos[243,244], Beatriz Nafria-Jimenez[281], Benedikt Schaefer[253], Carla Bellinghausen[300], Carlos Ferrando[256], Carmen Quereda[301], Carsten Skurk[246], Charlotte Thibeault[246], Christoph D. Spinner[302], Christoph Lange[303,304,305], Cinzia Hu[285], Claudio Cappadona[263], Cristiana Bianco[285], Cristina Sancho[306], Dag Arne Lihaug Hoff[307,308], Daniela Galimberti[285], David Jiménez[299,1226], David Pestaña[309], David Toapanta[310,313], Elena Azzolini[285], Elio Scarpini[285], Elisa T. Helbig[246], Eloisa Urrechaga[311], Elvezia Maria Paraboschi[262,263], Emanuele Pontali[312], Enric Reverter[256,313], Enrique Navas[301], Eunate Arana[314], Félix García Sánchez[315], Ferruccio Ceriotti[285], Francesco Malvestiti[316], Francisco Mesonero[243,244], Gianni Pezzoli[317], Giuseppe Lamorte[285], Holger Neb[318], Ilaria My[262], Isabel Hernández[282,283], Itziar de Rojas[282,283], Iván Galván-Femenia[261], Jan Heyckendorf[303,304,305], Jan Rybniker[198,260,319], Joan Ramon Badia[256], Jochen Schneider[302], Josune Goikoetxea[323], Julia Kraft[246], Karl Erik Müller[324], Karoline I. Gaede[325,326,327], Koldo Garcia-Etxebarria[244,322,329], Kristian Tonby[254,330], Lars Heggelund[324,331], Laura Izquierdo-Sanchez[244,322], Lauro Sumoy[333], Lena J. Lippert[246], Leonardo Terranova[285], Lindokuhle Nkambule[3,29,167], Lucia Garbarino[312], Luis Téllez[243,244], Luisa Roade[298], Mahnoosh Ostadreza[285], Maider Intxausti[306], Manolis Kogevinas[335,336,337,338], Mari E. K. Niemi[1,1280], María A. Gutiérrez-Stampa[339], Maria J. G. T. Vehreschild[340], Marta Marquié[282,283], Massimo Castoldi[341], Mattia Cordioli[1], Maurizio Cecconi[262,263], Mercè Boada[282,283], Michael J. Seilmaier[345], Michela Mazzocco[312], Miguel Rodríguez-Gandía[243,244], Natale Imaz Ayo[314], Natalia Blay[261], Nilda Martínez[346], Norwegian SARS-CoV-2 Study Group*, Oliver A. Cornely[198,259,319,349], Orazio Palmieri[295], Paolo Tentorio[262], Pedro M. Rodrigues[244,322,332], Pedro P. España[311], Per Hoffmann[351], Petra Bacher[352,353,354], Phillip Suwalski[246], Raúl de Pablo[280], Rosa Nieto[299,1226], Salvatore Badalamenti[262], Sandra Ciesek[355,356], Sara Bombace[262], Sara Pigazzini[1], Sibylle Wilfling[252,266,357], Søren Brunak[238], Stefanie Heilmann-Heimbach[351], The Humanitas COVID-19 Task Force*, The Humanitas Gavazzeni COVID-19 Task Force*, Stephan Ripke[246], Thomas Bahmer[294], Ulf Landmesser[359], Ulrike Protzer[9,360], Valeria Rimoldi[263], Vegard Skogen[361,362], Victor Andrade[289,291], Victor Moreno[336,363,364,365], Wolfgang Poller[246], Xavier Farre[261], Xiaomin Wang[246], Yascha Khodamoradi[340], Zehra Karadeniz[246], Adolfo de Salazar[249,50], Adriana Palom[245,292], Alba-Estela Garcia-Fernandez[1242], Albert Blanco-Grau[1242], Alberto Zanella[285,316], Alessandra Bandera[285,316], Almut Nebel[237], Andrea Biondi[1264], Andrea Caballero-Garralda[1242], Andrea Gori[285,316], Andreas Lind[254,255], Anna Ludovica Fracanzani[285,316], Anna Peschuck[237], Antonio Pesenti[285,316], Carmen de la Horra[136,1244,1245,336,1246], Chiara Milani[286,287], Cinzia Paccapelo[285], Claudio Angelini[1247], Cristina Cea[1242], Eduardo Muñiz-Diaz[1248], Elena Sandoval[1249], Enrique J. Calderón[1243,1244,1245,366,1246], Erik Solligård[1265,1266], Fátima Aziz[1249], Filippo Martinelli-Boneschi[285,316], Flora Peyvandi[285,316], Francesco Blasi[285,1250], Francisco J. Medrano[1251,1243,1244,366,1246], Francisco Rodriguez-Frias[245,298,244,1242], Fredrik Müller[254,255], Giacomo Grasselli[285,316], Giorgio Costantino[285,316], Giulia Cardamone[1252], Giuseppe Foti[1253], Giuseppe Matullo[1254], Hayato Kurihara[1247], Jan Egil Afset[307,1255], Jan Kristian Damås[1256,1257], Javier Ampuero[1243,1244,1245,369], Javier Martín[1259], Jeanette Erdmann[1260,1261,1262], Jonas Bergan[1263], Siegfried Goerg[264], Jose Ferrusquía-Acosta[1249], Jose Hernández Quero[249,250], Juan Delgado[1243,1244,1245,366,1246], Juan M. Guerrero[1243,1244,1245], Kari Risnes[1257,1258], Laura Rachele Bettini[1257,1258], Leticia Moreira[1249], Lise Tuset Gustad[1265,1267], Luigi Santoro[285], Luigia Scudeller[285], Mar Riveiro-Barciela[292,298,244], Marco Schaefer[1268], Maria Carrabba[285], Maria G. Valsecchi[1269], María Hernandez-Tejero[256], Marialbert Acosta-Herrera[1259], Mariella D'Angiò[1264], Marina Baldini[285], Marina Cazzaniga[1270], Michele Ciccarelli[1247], Monica Bocciolone[1247], Monica Miozzo[285,316], Natalia Chueca[249], Nicola Montano[285,316], Paola Faverio[1271], Paoletta Preatoni[1247], Paolo Bonfanti[1272,1273], Paolo Omodei[1247], Pedro Castro[256], Ricard Ferrer[84,1274], Roberta Gualtierotti[285,316], Rocío Gallego-Durán[144,1245,244], Rubén Morilla[1243,1244,1245,366,1246], Sammra Haider[308], Sara Marsal[245], Serena Aneli[1254], Serena Pelusi[285,316], Silvano Bosari[285,316], Stefano Aliberti[285,1250], Susanne Dudman[254,255],

Tenghao Zheng[344], Tomas Pumarola[397], Trinidad Gonzalez Cejudo[249], Valter Monzani[285], Vicente Friaza[1243,1244,1245,366,1246], Wolfgang Peter[1268] & Ximo Dopazo[1275]

**Norwegian SARS-CoV-2 Study Group**
Tom H. Karlsen[254,268,269,270]

**Humanitas COVID-19 Task Force**
Stefano Duga[262,263,341]

**The Humanitas Gavazzeni COVID-19 Task Force**
Stefano Duga[262,263,341]

**Admin team members**
Sandra May[237] & Marit M. Grimsrud[254,269,270]

**deCODE**
**Analysis team leader**
Daniel F. Gudbjartsson[366]

**Data collection leader**
Kari Stefansson[366]

**Analysis team members**
Patrick Sulem[366], Gardar Sveinbjornsson[366], Pall Melsted[366], Gudmundur Norddahl[366] & Kristjan Helgi Swerford Moore[366]

**Data collection members**
Unnur Thorsteinsdottir[366] & Hilma Holm[366]

**Determining the Molecular Pathways & Genetic Predisposition of the Acute Inflammatory Process Caused by SARS-CoV-2**
**Analysis team leader**
Marta E. Alarcón-Riquelme[102]

**Data collection leader**
David Bernardo[368,369]

**Analysis team member**
Manuel Martínez-Bueno[102]

**Data collection member**
Silvia Rojo Rello[370]

**Estonian Biobank**
**Analysis team leader**
Reedik Mägi[371]

**Data collection leader**
Lili Milani[371]

**Admin team leader**
Andres Metspalu[371]

**Analysis team members**
Triin Laisk[371], Kristi Läll[371] & Maarja Lepamets[371]

**Data collection members**
Tõnu Esko[371], Ene Reimann[371], Paul Naaber[372], Edward Laane[373,374], Jaana Pesukova[374], Pärt Peterson[375], Kai Kisand[375], Jekaterina Tabri[377], Raili Allos[377], Kati Hensen[377], Joel Starkopf[378], Inge Ringmets[379], Anu Tamm[380] & Anne Kallaste[380]

**Admin team members**
Helene Alavere[371], Kristjan Metsalu[371] & Mairo Puusepp[371]

**FinnGen**
**Data collection members**
Kati Kristiansson[382], Juha Karjalainen[1], Sami Koskelainen[382], Markus Perola[382,383], Kati Donner[1], Katja Kivinen[1] & Aarno Palotie[1]

**Admin team member**
Mari Kaunisto[1]

**FinnGen Admin team leader**
Aarno Palotie[381]

**Functional Host Genomics in Infectious Diseases (FHoGID)**
**Analysis team leader**
Carlo Rivolta[384,385]

**Data collection leaders**
Pierre-Yves Bochud[386], Stéphanie Bibert[386], Noémie Boillat[386], Semira Gonseth Nussle[388] & Werner Albrich[389]

**Analysis team members**
Mathieu Quinodoz[384,385] & Dhryata Kamdar[384,385]

**Data collection members**
Noémie Suh[390], Dionysios Neofytos[391], Véronique Erard[392], Cathy Voide[393], FHoGID*, RegCOVID*, P-PredictUs*, SeroCOVID* & CRiPSI*

**FHoGID**
P. Y. Bochud[394,395,398], C. Rivolta[394], S. Bibert[394], M. Quinodoz[394], D. Kamdar[394], D. Neofytos[394], V. Erard[394], C. Voide[394], R. Friolet[394], P. Vollenweider[394,395], J. L. Pagani[394,395], M. Oddo[394], F. Meyer zu Bentrup[394], A. Conen[394], O. Clerc[394], O. Marchetti[394], A. Guillet[394], C. Guyat-Jacques[394], S. Foucras[394], M. Rime[394], J. Chassot[394], M. Jaquet[394], R. Merlet Viollet[394], Y. Lannepoudenx[394] & L. Portopena[394]

**RegCOVID**
P. Y. Bochud[394,395,398], P. Vollenweider[394,395], J. L. Pagani[394,395], F. Desgranges[395], P. Filippidis[395], B. Guéry[395], D. Haefliger[395], E. E. Kampouri[395], O. Manuel[395], A. Munting[395], M. Papadimitriou-Olivgeris[395], J. Regina[395], L. Rochat-Stettler[395], V. Suttels[395], E. Tadini[395], J. Tschopp[395], M. Van Singer[395] & B. Viala[395]

**P-PredictUs**
N. Boillat-Blanco[396], T. Brahier[396], O. Hügli[396], J. Y. Meuwly[396] & O. Pantet[396]

**SérocoViD**
S. Gonseth Nussle[397], M. Bochud[397], V. D'Acremont[397] & S. Estoppey Younes[397]

**CRiPSI**
W. C. Albrich[398], N. Suh[398], A. Cerny[398], L. O'Mahony[398], C. von Mering, P. Y. Bochud[394,395,398], M. Frischknecht[398], G.-R. Kleger[398], M. Filipovic[398], C. R. Kahlert[398], H. Wozniak[398], T. Rochat Negro[398], J. Pugin[398], K. Bouras[398], C. Knapp[398], T. Egger[398], A. Perret[398], P. Montillier[398], C. di Bartolomeo[398] & B. Barda[398]

**GCAT Genomes For Life**
**Analysis team leader**
Rafael de Cid[399]

**Data collection leaders**
Anna Carreras[399], Victor Moreno[400] & Manolis Kogevinas[335,336,337,338]

**Analysis team members**
Iván Galván-Femenía[399], Natalia Blay[399], Xavier Farré[399] & Lauro Sumoy[399]

**Data collection members**
Beatriz Cortés[399], Josep Maria Mercader[401,1276,1277,1278,1279], Marta Guindo-Martinez[401], David Torrents[401], Judith Garcia-Aymerich[335,336,337], Gemma Castaño-Vinyals[335,336,337,338] & Carlota Dobaño[335,336]

**GEN-COVID Multicenter Study**
**Analysis team leaders**
Marco Gori[404,405] & Mari E. K. Niemi[1,1280]

**Data collection leaders**
Alessandra Renieri[63,64,65], Francesca Mari[63,64,65], Mario Umberto Mondelli[408,409], Francesco Castelli[410], Massimo Vaghi[411], Stefano Rusconi[412,413], Francesca Montagnani[65,414], Elena Bargagli[415], Federico Franchi[416], Maria Antonietta Mazzei[417], Luca Cantarini[418], Danilo Tacconi[419], Marco Feri[420], Raffaele Scala[421], Genni Spargi[422], Cesira Nencioni[423], Maria Bandini[424], Gian Piero Caldarelli[425], Maurizio Spagnesi[424], Anna Canaccini[426], Agostino Ognibene[427], Antonella D'Arminio Monforte[428], Massimo Girardis[429], Andrea Antinori[430], Daniela Francisci[431,432], Elisabetta Schiaroli[431,432], Pier Giorgio Scotton[433], Sandro Panese[434], Renzo Scaggiante[435], Matteo Della Monica[436], Mario Capasso[437,438,439], Giuseppe Fiorentino[440], Marco Castori[441], Filippo Aucella[442], Antonio Di Biagio[443], Luca Masucci[444,445], Serafina Valente[446], Marco Mandalà[447], Patrizia Zucchi[448], Ferdinando Giannattasio[449], Domenico A. Coviello[12,450], Cristina Mussini[451], Giancarlo Bosio[452], Luisa Tavecchia[453], Lia Crotti[454,455,456,457], Marco Rizzi[458], Maria Teresa La Rovere[459], Simona Sarzi-Braga[460], Maurizio Bussotti[461], Sabrina Ravaglia[462], Rosangela Artuso[463], Antonio Perrella[464], Davide Romani[465], Paola Bergomi[466], Emanuele Catena[466], Antonella Vincenti[1231], Claudio Ferri[1232], Davide Grassi[1232], Gloria Pessina[1233], Mario Tumbarello[65,414], Massimo Di Pietro[1234], Ravaglia Sabrina[462], Sauro Luchi[1235], Chiara Barbieri[1235], Donatella Acquilini[1235], Elena Andreucci[1235], Francesco Paciosi[1235], Francesco Vladimiro Segala[1238], Giusy Tiseo[1236], Marco Falcone[1236], Mirjam Lista[63,65], Monica Poscente[1235], Oreste De Vivo[446], Paola Petrocelli[1235], Alessandra Guarnaccia[444,445], Silvia Baroni[1239] & Valentina Perticaroli[63,64,65]

**Admin team leaders**
Simone Furini[65] & Simona Dei[467]

**Analysis team members**
Elisa Benetti[65], Nicola Picchiotti[404,468], Maurizio Sanarico[469], Stefano Ceri[66], Pietro Pinoli[66], Francesco Raimondi[470], Filippo Biscarini[471], Alessandra Stella[471], Mattia Bergomi[473],

Kristina Zguro[65], Katia Capitani[65,475], Mattia Cordioli[1], Sara Pigazzini[1], Lindokuhle Nkambule[3,29,167] & Marco Tanfoni[468]

**Data collection members**
Chiara Fallerini[63,65], Sergio Daga[63,65], Margherita Baldassarri[63,65], Francesca Fava[63,64,65], Elisa Frullanti[63,65], Floriana Valentino[63,65], Gabriella Doddato[63,65], Annarita Giliberti[63,65], Rossella Tita[64], Sara Amitrano[64], Mirella Bruttini[63,65,64], Susanna Croci[63,65], Ilaria Meloni[63,65], Maria Antonietta Mencarelli[64], Caterina Lo Rizzo[64], Anna Maria Pinto[63,65], Giada Beligni[63,65], Andrea Tommasi[63,64,65], Laura Di Sarno[63,65], Maria Palmieri[63,65], Miriam Lucia Carriero[63,65], Diana Alaverdian[63,65], Nicola Iuso[63,65], Gabriele Inchingolo[63,65], Stefano Busani[429], Raffaele Bruno[408,409], Marco Vecchia[478], Mary Ann Belli[453], Stefania Mantovani[478], Serena Ludovisi[408,409], Eugenia Quiros-Roldan[410], Melania Degli Antoni[410], Isabella Zanella[479,480], Matteo Siano[413], Arianna Emiliozzi[430], Massimiliano Fabbiani[414], Barbara Rossetti[414], Giacomo Zanelli[65,414], Laura Bergantini[415], Miriana D'Alessandro[415], Paolo Cameli[415], David Bennet[415], Federico Anedda[416], Simona Marcantonio[416], Sabino Scolletta[416], Susanna Guerrini[417], Edoardo Conticini[418], Bruno Frediani[418], Chiara Spertilli[419], Alice Donati[420], Luca Guidelli[421], Marta Corridi[422], Leonardo Croci[423], Paolo Piacentini[424], Elena Desanctis[424], Silvia Cappelli[424], Agnese Verzuri[426], Valentina Anemoli[426], Alessandro Pancrazi[427], Maria Lorubbio[427], Esther Merlini[428], Federica Gaia Miraglia[428], Sophie Venturelli[429], Andrea Cossarizza[481], Alessandra Vergori[430], Arianna Gabrieli[413], Agostino Riva[412,413], Francesco Paciosi[432], Francesca Andretta[433], Francesca Gatti[435], Saverio Giuseppe Parisi[482], Stefano Baratti[482], Carmelo Piscopo[436], Roberta Russo[437,438], Immacolata Andolfo[437,438], Achille Iolascon[437,438], Massimo Carella[441], Giuseppe Merla[437,483], Gabriella Maria Squeo[483], Pamela Raggi[484], Carmen Marciano[484], Rita Perna[484], Matteo Bassetti[443,485], Maurizio Sanguinetti[444,445], Alessia Giorli[447], Lorenzo Salerni[447], Pierpaolo Parravicini[448], Elisabetta Menatti[486], Tullio Trotta[449], Gabriella Coiro[449], Fabio Lena[487], Enrico Martinelli[452], Sandro Mancarella[453], Chiara Gabbi[488], Franco Maggiolo[458], Diego Ripamonti[458], Tiziana Bachetti[489], Claudia Suardi[490], Gianfranco Parati[454,455], Giordano Bottà[491], Paolo Di Domenico[491], Ilaria Rancan[414], Francesco Bianchi[65,464], Riccardo Colombo[466], Chiara Barbieri[1235], Donatella Acquilini[1235], Elena Andreucci[1235], Francesco Paciosi[1235], Francesco Vladimiro Segala[1238], Giusy Tiseo[1236], Marco Falcone[1236], Mirjam Lista[63,65], Monica Poscente[1235], Oreste De Vivo[446], Paola Petrocelli[1235], Alessandra Guarnaccia[444,445], Silvia Baroni[1239] & Valentina Perticaroli[63,64,65]

**Genes & Health**
**Analysis team leader**
David A. van Heel[50]

**Data collection leader**
Karen A. Hunt[50]

**Admin team leader**
Richard C. Trembath[493]

**Analysis team members**
Qin Qin Huang[494] & Hilary C. Martin[494]

**Data collection members**
Dan Mason[495], Bhavi Trivedi[50] & John Wright[495]

**Admin team members**
Sarah Finer[497], Genes & Health Research Team* & Christopher J. Griffiths[499]

**Genes & Health Research Team**
Shaheen Akhtar[498], Mohammad Anwar[498], Elena Arciero[498], Samina Ashraf[498], Gerome Breen[498], Raymond Chung[498], Charles J. Curtis[498], Maharun Chowdhury[498], Grainne Colligan[498], Panos Deloukas[498], Ceri Durham[498], Sarah Finer[498], Chris Griffiths[498], Qin Qin Huang[498], Matt Hurles[498], Karen A. Hunt[498], Shapna Hussain[498], Kamrul Islam[498], Ahsan Khan[498], Amara Khan[498], Cath Lavery[498], Sang Hyuck Lee[498], Robin Lerner[498], Daniel MacArthur[498], Bev MacLaughlin[498], Hilary Martin[498], Dan Mason[498], Shefa Miah[498], Bill Newman[498], Nishat Safa[498], Farah Tahmasebi[498], Richard C. Trembath[498], Bhavi Trivedi[498], David A. van Heel[498] & John Wright[498]

**Genes for Good**
**Analysis team leader**
Albert V. Smith[500]

**Data collection members**
Andrew P. Boughton[500], Kevin W. Li[500], Jonathon LeFaive[500] & Aubrey Annis[500]

**Genetic determinants of COVID-19 complications in the Brazilian population**
**Analysis team leader**
Mari E. K. Niemi[1,1280]

**Data collection leader**
Cinthia E. Jannes[501]

**Admin team leaders**
Jose E. Krieger[501] & Alexandre C. Pereira[501]

**Analysis team members**
Mariliza Velho[501], Emanuelle Marques[501], Mattia Cordioli[1], Sara Pigazzini[1] & Lindokuhle Nkambule[3,29,167]

**Data collection members**
Isabella Ramos Lima[501], Mauricio Teruo Tada[501] & Karina Valino[501]

**Genetic influences on severity of COVID-19 illness in Korea**
**Analysis team leaders**
Mark McCarthy[502] & Carrie Rosenberger[502]

**Data collection leader**
Jong Eun Lee[503]

**Analysis team members**
Diana Chang[502], Christian Hammer[502], Julie Hunkapiller[502], Anubha Mahajan[502], Sarah Pendergrass[502], Lara Sucheston-Campbell[502] & Brian Yaspan[502]

**Data collection members**
Hyun Soo Lee[503], Eunsoon Shin[503], Hye Yoon Jang[503], Sunmie Kim[504], Sungmin Kym[505], Yeon-Sook Kim[505], Hyeongseok Jeong[505], Ki Tae Kwon[507], Shin-Woo Kim[507], Jin Yong Kim[508], Young Rock Jang[508], Hyun ah Kim[509], Ji Yeon Lee[509], Jeong Eun Lee[510], Shinwon Lee[510], Kang-Won Choe[511], Yu Min Kang[511], Sun Ha Jee[512] & Keum Ji Jung[512]

**Genomic epidemiology of SARS-CoV-2 host genetics in coronavirus disease 2019**
**Data collection leaders**
Victoria Parikh[513], Euan Ashley[514,515], Matthew Wheeler[513], Manuel Rivas[516], Carlos Bustamante[515,516], Benjamin Pinksy[518], Phillip Febbo[519], Kyle Farh[519], Gary P. Schroth[519] & Francis deSouza[519]

**Admin team leaders**
Karen Dalton[513] & Jeff Christle[513]

**Analysis team members**
Christopher Deboever[520], Sándor Szalma[520], Yosuke Tanigawa[516], Simone Rubinacci[521] & Olivier Delaneau[521]

**Data collection members**
John Gorzynski[513], Hannah de Jong[513], Shirley Sutton[513], Nathan Youlton[513], Ruchi Joshi[513], David Jimenez-Morales[513], Christopher Hughes[513], David Amar[513], Alex Ioannidis[516], Steve Hershman[513], Anna Kirillova[513], Kinya Seo[513], Yong Huang[513], Massa Shoura[518], Nathan Hammond[518], Nathaniel Watson[518], Archana Raja[513], ChunHong Huang[518], Malaya Sahoo[518] & Hannah Wang[518]

**Admin team member**
Jimmy Zhen[513]

**Genotek COVID-19 study**
**Analysis team leader**
Alexander Rakitko[974]

**Admin team leader**
Valery Ilinsky[974]

**Analysis team members**
Danat Yermakovich[974], Iaroslav Popov[974], Alexander Chernitsov[974], Elena Kovalenko[974], Anna Krasnenko[974], Nikolay Plotnikov[974], Ivan Stetsenko[974] & Anna Kim[974]

**Helix & Healthy Nevada Project Exome+ COVID-19 Phenotypes**
**Analysis team leader**
Elizabeth T. Cirulli[975]

**Analysis team members**
Kelly M. Schiabor Barrett[975], Alexandre Bolze[975], Simon White[975], Nicole L. Washington[975] & James T. Lu[975]

**Data collection members**
Stephen Riffle[975], Francisco Tanudjaja[975], Xueqing Wang[975], Jimmy M. Ramirez III[975], Nicole Leonetti[975], Efren Sandoval[975], Iva Neveux[976], Shaun Dabe[977] & Joseph J. Grzymski[976]

**24Genetics & IdiPaz Genomic Variants associated to COVID-19 infection outcome**
**Analysis team leader**
Juan Ignacio Esteban Miñano[978]

**Data collection leader**
Luis A. Aguirre[979]

**Admin team leader**
Eduardo López-Collazo[979]

**Analysis team members**
Manuel de la Mata Pazos[978], Luciano Cerrato[978] & Lasse Folkersen[978]

**Data collection members**
Roberto Lozano-Rodríguez[979], José Avendaño-Ortiz[979], Verónica Terrón Arcos[979], Karla Marina Montalbán-Hernández[979], Jaime Valentín Quiroga[979] & Alejandro Pascual-Iglesias[979]

**Admin team members**
Charbel Maroun-Eid[979] & Alejandro Martín-Quirós[979]

**Japan Coronavirus Taskforce**
**Analysis team leaders**
Ho Namkoong[980], Yukinori Okada[981,982,983] & Seiya Imoto[984]

**Data collection leaders**
Kazuhiko Katayama[985], Koichi Fukunaga[980], Yuko Kitagawa[986], Toshiro Sato[987], Naoki Hasegawa[988], Atsushi Kumanogoh[983,989,990], Akinori Kimura[991], Masumi Ai[992] & Katsushi Tokunaga[993]

**Admin team leaders**
Takanori Kanai[994], Satoru Miyano[995] & Seishi Ogawa[996,997]

**Analysis team members**
Ryuya Edahiro[981,989], Kyuto Sonehara[981], Yuya Shirai[981,989] & Masahiro Kanai[274]

**Data collection members**
Makoto Ishii[980], Hiroki Kabata[980], Katsunori Masaki[980], Hirofumi Kamata[980], Shinnosuke Ikemura[980], Shotaro Chubachi[980], Satoshi Okamori[980], Hideki Terai[980], Hiromu Tanaka[980], Atsuho Morita[980], Ho Lee[980], Takanori Asakura[980], Junichi Sasaki[999], Hiroshi Morisaki[1000], Yoshifumi Uwamino[1001], Kosaku Nanki[994], Yohei Mikami[994], Kazunori Tomono[1002], Kazuto Kato[1003], Fumihiko Matsuda[1004], Meiko Takahashi[1004], Nobuyuki Hizawa[1005], Yoshito Takeda[989], Haruhiko Hirata[989], Takayuki Shiroyama[989], Satoru Miyawaki[1006], Ken Suzuki[981], Yuichi Maeda[989,1007], Takuro Nii[989,1007], Yoshimi Noda[989], Takayuki Niitsu[989], Yuichi Adachi[989], Takatoshi Enomoto[989], Saori Amiya[989], Reina Hara[989], Kunihiko Takahashi[995], Tatsuhiko Anzai[995], Takanori Hasegawa[995], Satoshi Ito[995], Ryuji Koike[1009], Akifumi Endo[1010], Yuji Uchimura[1011], Yasunari Miyazaki[1012], Takayuki Honda[1012], Tomoya Tateishi[1012], Shuji Tohda[1013], Naoya Ichimura[1013], Kazunari Sonobe[1013], Chihiro Sassa[1013], Jun Nakajima[1013], Yasuhito Nannya[1014], Yosuke Omae[993], Kazuhisa Takahashi[1015], Norihiro Harada[1015], Makoto Hiki[1016,1017], Haruhi Takagi[1015], Ai Nakamura[1015], Etsuko Tagaya[1018], Masatoshi Kawana[1019], Ken Arimura[1018], Takashi Ishiguro[1020], Noboru Takayanagi[1020], Taisuke Isono[1020], Yotaro Takaku[1020], Kenji Takano[1020], Ryusuke Anan[1021], Yukiko Nakajima[1021], Yasushi Nakano[1021], Kazumi Nishio[1021], Soichiro Ueda[1022], Reina Hayashi[1022], Hiroki Tateno[1023], Isano Hase[1023], Shuichi Yoshida[1023], Shoji Suzuki[1023], Keiko Mitamura[1024], Fumitake Saito[1025], Tetsuya Ueda[1026], Masanori Azuma[1026], Tadao Nagasaki[1026], Yoshinori Yasui[1028], Yoshinori Hasegawa[1026], Yoshikazu Mutoh[1029], Takashi Yoshiyama[1030], Tomohisa Shoko[1031], Mitsuaki Kojima[1031], Tomohiro Adachi[1031], Motonao Ishikawa[1032], Kenichiro Takahashi[1033], Kazuyoshi Watanabe[1034], Tadashi Manabe[1035], Fumimaro Ito[1035], Takahiro Fukui[1035], Yohei Funatsu[1035], Hidefumi Koh[1035], Yoshihiro Hirai[1036], Hidetoshi Kawashima[1036], Atsuya Narita[1036], Kazuki Niwa[1037], Yoshiyuki Sekikawa[1037], Fukuki Saito[1038], Kazuhisa Yoshiya[1038], Tomoyuki Yoshihara[1038], Yusuke Suzuki[1039], Sohei Nakayama[1039], Keita Masuzawa[1039], Koichi Nishi[1040], Masaru Nishitsuji[1040], Maiko Tani[1040], Takashi Inoue[1041], Toshiyuki Hirano[1041], Keigo Kobayashi[1041], Naoki Miyazawa[1042], Yasuhiro Kimura[1042], Reiko Sado[1042], Takashi Ogura[1043], Hideya Kitamura[1043], Kota Murohashi[1043], Ichiro Nakachi[1044], Rie Baba[1044], Daisuke Arai[1044], Satoshi Fuke[1045], Hiroshi Saito[1045], Naota Kuwahara[1046], Akiko Fujiwara[1046], Takenori Okada[1046], Tomoya Baba[1047], Junya Noda[1047], Shuko Mashimo[1047], Kazuma Yagi[1048], Tetsuya Shiomi[1048], Mizuha Hashiguchi[1048], Toshio Odani[1049], Takao Mochimaru[1050,1051], Yoshitaka Oyamada[1050,1051], Nobuaki Mori[1052], Namiki Izumi[1053], Kaoru Nagata[1053], Reiko Taki[1053], Koji Murakami[1054], Mitsuhiro Yamada[1054], Hisatoshi Sugiura[1054], Kentaro Hayashi[1055], Tetsuo Shimizu[1055], Yasuhiro Gon[1055], Shigeki Fujitani[1056], Tomoya Tsuchida[1057], Toru Yoshida[1056], Takashi Kagaya[1058], Toshiyuki Kita[1058], Satoru Sakagami[1058], Yoshifumi Kimizuka[1059], Akihiko Kawana[1059], Yoshihiko Nakamura[1060], Hiroyasu Ishikura[1060], Tohru Takata[1061], Takahide Kikuchi[1062], Daisuke Taniyama[1062], Morio Nakamura[1062], Nobuhiro Kodama[1063], Yasunari Kaneyama[1063], Shunsuke Maeda[1063], Yoji Nagasaki[1064], Masaki Okamoto[1065,1027], Sayoko Ishihara[1064], Akihiro Ito[1067], Yusuke Chihara[1068], Mayumi Takeuchi[1068], Keisuke Onoi[1068], Naozumi Hashimoto[1069], Keiko Wakahara[1069], Akira Ando[1069], Makoto Masuda[1070], Aya Wakabayashi[1070], Hiroki Watanabe[1070], Hisako Sageshima[1071], Taka-Aki Nakada[1072], Ryuzo Abe[1072], Tadanaga Shimada[1072], Kodai Kawamura[1073], Kazuya Ichikado[1073], Kenta Nishiyama[1073], Masaki Yamasaki[1074], Satoru Hashimoto[1074], Yu Kusaka[1075], Takehiko Ohba[1075], Susumu Isogai[1075], Minoru Takada[1076], Hidenori Kanda[1076], Yuko Komase[1077], Fumiaki Sano[1078], Koichiro Asano[1079], Tsuyoshi Oguma[1080], Masahiro Harada[1081], Takeshi Takahashi[1081], Takayuki Shibusawa[1081], Shinji Abe[1082], Yuta Kono[1082], Yuki Togashi[1082], Takehiro Izumo[1083], Minoru Inomata[1083], Nobuyasu Awano[1083], Shinichi Ogawa[1084], Tomouki Ogata[1084], Shoichiro Ishihara[1084], Arihiko Kanehiro[1085], Shinji Ozaki[1085], Yasuko Fuchimoto[1085], Yuichiro Kitagawa[1086], Shozo Yoshida[1086], Shinji Ogura[1086], Kei Nishiyama[1088], Kousuke Yoshida[1088], Satoru Beppu[1088], Satoru Fukuyama[1089], Yoshihiro Eriguchi[1090], Akiko Yonekawa[1090], Yoshiaki Inoue[1091], Kunihiko Yamagata[1092], Shigeru Chiba[1093], Osamu Narumoto[1094], Hideaki Nagai[1094], Nobuharu Ooshima[1094], Mitsuru Motegi[1095], Hironori Sagara[1096], Akihiko Tanaka[1096], Shin Ohta[1096], Yoko Shibata[1097], Yoshinori Tanino[1097], Yuki Sato[1097], Yuichiro Yamada[1098], Takuya Nishida[1098], Masato Shinoki[1098], Hajime Iwagoe[1099], Tomonori Imamura[1100], Akira Umeda[1101], Hisato Shimada[1101], Mayu Endo[1102], Shinichi Hayashi[1103], Mai Takahashi[1103], Shigefumi Nakano[1103], Masakiyo Yatomi[1104], Toshitaka Maeno[1104], Tomoo Ishii[1105], Mitsuyoshi Utsugi[1106], Akihiro Ono[1106], Kensuke Kanaoka[1107], Shoichi Ihara[1107] & Kiyoshi Komuta[1107]

**Lifelines**
**Analysis team leader**
Lude Franke[51]

**Data collection leader**
Marike Boezen[1109]

**Analysis team members**
Patrick Deelen[51,52], Annique Claringbould[51], Esteban Lopera[51], Robert Warmerdam[51], Judith. M. Vonk[1109] & Irene van Blokland[51]

**Data collection members**
Pauline Lanting[51] & Anil P. S. Ori[1112,1113]

**Lung eQTL Consortium**
**Data collection members**
Ma'en Obeidat[1114], Ana I. Hernández Cordero[1114], Don D. Sin[1114,1115], Yohan Bossé[1116], Philippe Joubert[1116], Ke Hao[1117], David Nickle[1118,1119], Wim Timens[1120,1121] & Maarten van den Berge[1121,1122]

**Mass General Brigham-Host Vulnerability to COVID-19**
**Analysis team leaders**
Yen-Chen Anne Feng[1123] & Josep Mercader[29,1123]

**Data collection leaders**
Scott T. Weiss[1126], Elizabeth W. Karlson[1127], Jordan W. Smoller[1128], Shawn N. Murphy[1129], James B. Meigs[1130,1124,1125] & Ann E. Woolley[1127]

**Admin team leader**
Robert C. Green[2,273]

**Data collection member**
Emma F. Perez[273]

**Michigan Genomics Initiative (MGI)**
**Analysis team leader**
Brooke Wolford[1132]

**Admin team leader**
Sebastian Zöllner[500]

**Analysis team members**
Jiongming Wang[500] & Andrew Beck[500]

**Mount Sinai Health System COVID-19 Genomics Initiative**
**Analysis team leader**
Laura G. Sloofman[26]

**Data collection leaders**
Steven Ascolillo[1133], Robert P. Sebra[1117,1135], Brett L. Collins[26] & Tess Levy[26]

**Admin team leaders**
Joseph D. Buxbaum[26] & Stuart C. Sealfon[7]

**Analysis team members**
Shea J. Andrews[7], Daniel M. Jordan[1117,1137], Ryan C. Thompson[1133,1140,1141], Kyle Gettler[1117], Kumardeep Chaudhary[1117,1143], Gillian M. Belbin[1144], Michael Preuss[1143,1146], Clive Hoggart[1147,1148,1142], Sam Choi[1147,1148,1142,1149] & Slayton J. Underwood[26]

**Data collection members**
Irene Salib[1117], Bari Britvan[26], Katherine Keller[26], Lara Tang[26], Michael Peruggia[26], Liam L. Hiester[26], Kristi Niblo[26], Alexandra Aksentijevich[26], Alexander Labkowsky[26], Avrohom Karp[26], Menachem Zlatopolsky[26] & Marissa Zyndorf[1117]

**Admin team members**
Alexander W. Charney[1141,1150], Noam D. Beckmann[1133], Eric E. Schadt[1117,1135], Noura S. Abul-Husn[1144], Judy H. Cho[1117,1143], Yuval Itan[1117,1143], Eimear E. Kenny[1144], Ruth J. F. Loos[1143,1146,1151], Girish N. Nadkarni[1133,1143,1153,1154,1155], Ron Do[1117,1143], Paul O'Reilly[1147,1148,1142,1149] & Laura M. Huckins[1147,1148,1142]

**MyCode Health Initiative**
**Analysis team leaders**
Manuel A. R. Ferreira[1157] & Goncalo R. Abecasis[1157]

**Data collection leaders**
Joseph B. Leader[1158] & Michael N. Cantor[1157]

**Admin team leaders**
Anne E. Justice[1159] & Dave J. Carey[1160]

**Analysis team members**
Geetha Chittoor[1159], Navya Shilpa Josyula[1159], Jack A. Kosmicki[1157], Julie E. Horowitz[1157] & Aris Baras[1157]

**Data collection members**
Matthew C. Gass[1158] & Ashish Yadav[1157]

**Admin team member**
Tooraj Mirshahi[1160]

**Netherlands Twin Register**
**Analysis team leader**
Jouke Jan Hottenga[122]

**Data collection leader**
Meike Bartels[122]

**Admin team leader**
Eco J. C. de Geus[122]

**Analysis team member**
Michel G. Nivard[122]

**Penn Medicine Biobank**
**Analysis team leaders**
Anurag Verma[1162] & Marylyn D. Ritchie[1162]

**Admin team leader**
Daniel Rader[1162]

**Analysis team members**
Binglan Li[1163], Shefali S. Verma[1162], Anastasia Lucas[1162] & Yuki Bradford[1162]

**Population controls**
**Analysis team leader**
Federico Zara[12]

**Analysis team members**
Vincenzo Salpietro[12], Marcello Scala[1172], Michele Iacomino[12], Paolo Scudieri[12] & Renata Bocciardi[12]

**Data collection members**
Carlo Minetti[12], Antonella Riva[1172], Maria Stella Vari[12], Myriam Mni[163], Jean-François Rahier[1173], Elisa Giorgio[1174], Federico Zara[12] & Diana Carli[1175]

**Data collection leaders**
Pasquale Striano[12,13], Edoud Louis[170], Michel Georges[163], Souad Rahmouni[163], Cynthia M. Bulik[709,1166,1167], Mikael Landén[709,1168], Alfredo Brusco[1169] & Giovanni Battista Ferrero[1170]

**Admin team leaders**
Francesca Madia[12] & Bengt Fundín[709]

**Qatar Genome Program**
**Analysis team leader**
Hamdi Mbarek[80]

**Data collection leader**
Said I. Ismail[80]

**Analysis team members**
Chadi Saad[80] & Yaser Al-Sarraj[80]

**Data collection members**
Radja Messai Badji[80], Wadha Al-Muftah[80], Asma Al Thani[80] & Nahla Afifi[1176]

**Study of the COVID-19 host genetics in the population of Latvia**
**Analysis team leader**
Janis Klovins[1177]

**Data collection leader**
Vita Rovite[1177]

**Analysis team members**
Raimonds Rescenko[1177] & Raitis Peculis[1177]

**Data collection member**
Monta Ustinova[1177]

**The genetic predisposition to severe COVID-19**
**Analysis team leader**
Mari E. K. Niemi[1,1280]

**Data collection leader**
Hugo Zeberg[1178,1179]

**Analysis team members**
Mattia Cordioli[1], Sara Pigazzini[1] & Lindokuhle Nkambule[3,29,167]

**Data collection members**
Robert Frithiof[1180], Michael Hultström[1180,1181] & Miklos Lipcsey[1180,1182]

**UCLA Precision Health COVID-19 Host Genomics Biobank**
**Analysis team leader**
Ruth Johnson[1183]

**Data collection leader**
UCLA Health ATLAS & Data Mart Working Group*

**UCLA Health ATLAS & Data Mart Working Group**
Daniel H. Geschwind[1184]

**Admin team leaders**
Nelson Freimer[1185], Manish J. Butte[1186,1171,1184], Daniel H. Geschwind[1188,1187,1152] &
Bogdan Pasaniuc[1189,1190,1139]

**Analysis team members**
Yi Ding[1191], Alec Chiu[1191], Timothy S. Chang[1192] & Paul Boutros[1193,1139]

**UK 100,000 Genomes Project (Genomics England)**
**Analysis team leader**
Loukas Moutsianas[14,95]

**Data collection leaders**
Mark J. Caulfield[95,695] & Richard H. Scott[95,1195,1196]

**Analysis team members**
Athanasios Kousathanas[95], Dorota Pasko[95], Susan Walker[95], Alex Stuckey[95],
Christopher A. Odhams[95] & Daniel Rhodes[95]

**Data collection members**
Tom Fowler[95], Augusto Rendon[95,1197], Georgia Chan[95] & Prabhu Arumugam[95]

**UK Biobank**
**Analysis team leaders**
Tomoko Nakanishi[17,56,57], Konrad J. Karczewski[3,29], Alicia R. Martin[3,29], Daniel J. Wilson[1199] &
Chris A. Spencer[91]

**Data collection leaders**
Derrick W. Crook[1201], David H. Wyllie[1201,1202] & Anne Marie O'Connell[1203]

**Admin team leader**
J. Brent Richards[17,53,55,56]

**Analysis team members**
Guillaume Butler-Laporte[17,53], Vincenzo Forgetta[17], Elizabeth G. Atkinson[3,29],
Masahiro Kanai[3,29,1204], Kristin Tsuo[3,29,1205], Nikolas Baya[3,29], Patrick Turley[3,29], Rahul Gupta[3,29],
Raymond K. Walters[3,29], Duncan S. Palmer[3,29], Gopal Sarma[3,29], Matthew Solomonson[3,29],
Nathan Cheng[3,29], Wenhan Lu[3,29], Claire Churchhouse[3,29], Jacqueline I. Goldstein[3,29],
Daniel King[3,29], Wei Zhou[3,29], Cotton Seed[3,29], Mark J. Daly[1,2,3], Benjamin M. Neale[3,29],
Hilary Finucane[3,29], Sam Bryant[2], F. Kyle Satterstrom[3,29], Gavin Band[700], Sarah G. Earle[1199],
Shang-Kuan Lin[1199], Nicolas Arning[1199] & Nils Koelling[91]

**Data collection members**
Jacob Armstrong[1199] & Justine K. Rudkin[1199]

**Admin team members**
Shawneequa Callier[1207], Sam Bryant[3,29] & Caroline Cusick[29]

**UK Blood Donors Cohort**
**Analysis team leaders**
Nicole Soranzo[1208,1209,1210] & Jing Hua Zhao[1211]

**Data collection leaders**
John Danesh[1211,1212,1213,1214,1215] & Emanuele Di Angelantonio[1211,1212,1213,1214]

**Analysis team member**
Adam S. Butterworth[1211,1212,1213,1214]

**VA Million Veteran Program (MVP)**
**Analysis team leaders**
Yan V. Sun[1216,1217] & Jennifer E. Huffman[1218]

**Data collection leader**
Kelly Cho[1219]

**Admin team leaders**
Christopher J. O'Donnell[1218], Phil Tsao[1220,1221] & J. Michael Gaziano[1219]

**Analysis team member**
Gina Peloso[1218,1222]

**Data collection member**
Yuk-Lam Ho[1219]

**Val Gardena**
**Analysis team leader**
Christian Fuchsberger[209]

**Data collection leader**
Michael Mian[1223]

**Data collection member**
Federica Scaggiante[1224]

**Admin team members**
Cristian Pattaro[209] & Peter Pramstaller[209]

**CHOP_CAG**
Xiao Chang[1227], Joseph R. Glessner[1227,1228] & Hakon Hakonarson[1227,1228,1229]

**GenOMICC/ISARIC4C**
**Data collection leaders**
J. Kenneth Baillie[60,61,62], Peter J. McGuigan[523], Luke Stephen Prockter Moore[524],
Marcela Paola Vizcaychipi[524], Kathryn Hall[525], Andy Campbell[526], Ailstair Nichol[527],
Geraldine Ward[528], Valerie Joan Page[529], Malcolm G. Semple[530], Kayode Adeniji[531],
Daniel Agranoff[532], Ken Agwuh[533], Dhiraj Ail[534], Erin L. Aldera[535], Ana Alegria[536,506],
Brian Angus[537], Abdul Ashish[538], Dougal Atkinson[539], Shahedal Bari[540], Gavin Barlow[541],
Stella Barnass[542], Nicholas Barrett[543], Christopher Bassford[544], Sneha Basude[545],
David Baxter[546], Michael Beadsworth[547], Jolanta Bernatoniene[548], John Berridge[549],
Nicola Best[550], Pieter Bothma[551], David Chadwick[552], Robin Brittain-Long[553], Naomi Bulteel[554],
Tom Burden[555], Andrew Burtenshaw[556], Vikki Caruth[557], David Chadwick[552],
Duncan Chambler[558], Nigel Chee[559], Jenny Child[560], Srikanth Chukkambotla[561], Tom Clark[562],
Paul Collini[563], Catherine Cosgrove[564], Jason Cupitt[565], Maria-Teresa Cutino-Moguel[566],
Paul Dark[567], Chris Dawson[568], Samir Dervisevic[569], Phil Donnison[837], Sam Douthwaite[543],
Andrew Drummond[572,773], Ingrid DuRand[573], Ahilanadan Dushianthan[574], Tristan Dyer[575],
Cariad Evans[563], Chi Eziefula[532], Christopher Fegan[576], Adam Finn[577], Duncan Fullerton[578],
Sanjeev Garg[579], Atul Garg[580], Effrossyni Gkrania-Klotsas[581], Jo Godden[582], Arthur Goldsmith[583],
Clive Graham[584], Elaine Hardy[585], Stuart Hartshorn[586], Daniel Harvey[587], Peter Havalda[588],
Daniel B. Hawcutt[589], Maria Hobrok[590], Luke Hodgson[591], Anil Hormis[592], Michael Jacobs[593],
Susan Jain[594], Paul Jennings[595], Agilan Kaliappan[596], Vidya Kasipandian[597], Stephen Kegg[598],
Michael Kelsey[599], Jason Kendall[1281], Caroline Kerrison[600], Ian Kerslake[601], Oliver Koch[602],
Gouri Koduri[603], George Koshy[604], Shondipon Laha[605], Steven Laird[606], Susan Larkin[607],
Tamas Leiner[604], Patrick Lillie[608], James Limb[609], Vanessa Linnett[610], Jeff Little[611], Mark Lyttle[612],
Michael MacMahon[1281], Emily MacNaughton[613], Ravish Mankregod[614], Huw Masson[615],
Elijah Matovu[578], Katherine McCullough[616], Ruth McEwen[617], Manjula Meda[618], Gary H. Mills[563],
Jane Minton[620], Karl Ward[620], Mariyam Mirfenderesky[621], Kavya Mohandas[622], Quen Mok[623],
James Moon[624], Elinoor Moore[581], Patrick Morgan[625], Craig Morris[626], Katherine Mortimore[604],
Samuel Moses[627], Mbiye Mpenge[628], Rohinton Mulla[629], Michael Murphy[630], Megan Nagel[631],
Thapas Nagarajan[632], Mark Nelson[633], Matthew K. O'Shea[634], Igor Otahal[635],
Marlies Ostermann[543], Mark Pais[536], Selva Panchatsharam[637], Danai Papakonstantinou[638],
Hassan Paraiso[639], Brij Patel[640], Natalie Pattison[641], Justin Pepperell[642], Mark Peters[1281],
Mandeep Phull[643], Stefania Pintus[544], Jagtur Singh Pooni[645], Frank Post[646], David Price[647],
Rachel Prout[648], Nikolas Rae[649], Henrik Reschreiter[650], Tim Reynolds[651], Neil Richardson[652],
Mark Roberts[653], Devender Roberts[654], Alistair Rose[655], Guy Rousseau[656], Brendan Ryan[657],
Taranprit Saluja[658], Aarti Shah[659], Prad Shanmuga[660], Anil Sharma[661], Anna Shawcross[662],
Jeremy Sizer[663], Manu Shankar-Hari[543], Richard Smith[664], Catherine Snelson[665], Nick Spittle[666],
Nikki Staines[667], Tom Stambach[668], Richard Stewart[669], Pradeep Subudhi[670],
Tamas Szakmany[671], Kate Tatham[672], Jo Thomas[673], Chris Thompson[674], Robert Thompson[1281],
Ascanio Tridente[675], Darell Tupper-Carey[551], Mary Twagira[676], Andrew Ustianowski[572],
Nick Vallotton[677], Lisa Vincent-Smith[678], Shico Visuvanathan[667], Alan Vuylsteke[679],
Sam Waddy[680], Rachel Wake[681], Andrew Walden[682], Ingeborg Welters[547], Tony Whitehouse[665],
Paul Whittaker[683], Ashley Whittington[684], Padmasayee Papineni[685], Meme Wijesinghe[686],
Martin Williams[1281], Lawrence Wilson[617], Sarah Cole[966], Stephen Winchester[687],
Martin Wiselka[688], Adam Wolverson[689], Daniel G. Wooton[690], Andrew Workman[588],
Bryan Yates[691] & Peter Young[692]

**Analysis team members**
J. Kenneth Baillie[60,61,62], Rupert Beale[693], Andrew D. Bretherick[62], Mark J. Caulfield[95,695],
Sara Clohisey[60], Max Head Fourman[60], James Furniss[60], Elvina Gountouna[696],
Graeme Grimes[62], Chris Haley[60], David Harrison[697], Caroline Hayward[62,696], Sean Keating[61],
Lucija Klaric[62], Paul Klenerman[700], Athanasios Kousathanas[95], Andy Law[60], Alison M. Meynert[62],
Jonathan Millar[60], Loukas Moutsianas[14,95], Erola Pairo-Castineira[60,62], Nicholas Parkinson[60],
Dorota Pasko[95], Chris P. Ponting[62], David J. Porteous[696], Konrad Rawlik[60], Anne Richmond[62],
Kathy Rowan[697], Clark D. Russell[60,705], Richard H. Scott[95,706], Xia Shen[707,708,709], Barbara Shih[60],
Albert Tenesa[60,62,708], Veronique Vitart[62], Susan Walker[95], Bo Wang[60], James F. Wilson[62,708],
Yang Wu[710], Jian Yang[711,712], Zhijian Yang[707], Marie Zechner[60], Ranran Zhai[707], Chenqing Zheng[707],
Lisa Norman[714], Riinu Pius[714], Thomas M. Drake[714], Cameron J. Fairfield[714], Stephen R. Knight[714],
Kenneth A. Mclean[714], Derek Murphy[714], Catherine A. Shaw[714], Jo Dalton[715], Michelle Girvan[715],
Egle Saviciute[715], Stephanie Roberts[715], Janet Harrison[715], Laura Marsh[715], Marie Connor[715],

Sophie Halpin[715], Clare Jackson[715], Carrol Gamble[715], Gary Leeming[716], Andrew Law[60], Murray Wham[717], Sara Clohisey[60], Ross Hendry[60] & James Scott-Brown[718]

**Data collection members**

Colin Begg[719], Sara Clohisey[60], Charles Hinds[695], Antonia Ying Wai Ho[721], Peter W. Horby[722], Julian Knight[700], Lowell Ling[724], David Maslove[725], Danny McAuley[726,727], Jonathan Millar[60], Hugh Montgomery[728], Alistair Nichol[729], Peter J. M. Openshaw[730,731], Chris P. Ponting[62], Kathy Rowan[697], Malcolm G. Semple[732,733], Manu Shankar-Hari[734], Charlotte Summers[735], Timothy Walsh[61], Lisa Armstrong[736], Hayley Bates[736], Emma Dooks[736], Fiona Farquhar[736], Brigid Hairsine[736], C. McParland[736], Sophie Packham[736], Zoe Alldis[737], Raine Astin-Chamberlain[737], Fatima Bibi[737], Jack Biddle[737], Sarah Blow[737], Matthew Bolton[737], Catherine Borra[737], Ruth Bowles[737], Maudrian Burton[737], Yasmin Choudhury[737], David Collier[737], Amber Cox[737], Amy Easthope[737], Patrizia Ebano[737], Stavros Fotiadis[737], Jana Gurasashvili[737], Rosslyn Halls[737], Pippa Hartridge[737], Delordson Kallon[737], Jamila Kassam[737], Ivone Lancoma-Malcolm[737], Maninderpal Matharu[737], Peter May[737], Oliver Mitchelmore[737], Tabitha Newman[737], Mital Patel[737], Jane Pheby[737], Irene Pinzuti[737], Zoe Prime[737], Oleksandra Prysyazhna[737], Julian Shiel[737], Melanie Taylor[737], Carey Tierney[737], Suzanne Wood[737], Anne Zak[737], Olivier Zongo[737], Miranda Forsey[738], Agilan Kaliappan[738], Anne Nicholson[738], Joanne Riches[738], Mark Vertue[738], Christopher Wasson[523], Stephanie Finn[523], Jackie Green[523], Erin Collins[523], Bernadette King[523], Lina Grauslyte[739], Musarat Hussain[739], Mandeep Phull[739], Tatiana Pogreban[739], Lace Rosaroso[739], Erika Salciute[739], George Franke[739], Joanna Wong[739], Aparna George[739], Louise Akeroyd[740], Shereen Bano[740], Matt Bromley[740], Lucy Gurr[740], Tom Lawton[740], James Morgan[740], Kirsten Sellick[740], Deborah Warren[740], Brian Wilkinson[740], Janet McGowan[740], Camilla Ledgard[740], Amelia Stacey[740], Kate Pye[740], Ruth Bellwood[740], Michael Bentley[740], Maria Hobrok[741], Ronda Loosley[741], Heather McGuinness[741], Helen Tench[741], Rebecca Wolf-Roberts[741], Sian Gibson[742], Amanda Lyle[742], Fiona McNeela[742], Jayachandran Radhakrishnan[742], Alistair Hughes[742], Asifa Ali[743], Megan Brady[743], Sam Dale[743], Annalisa Dance[743], Lisa Gledhill[743], Jill Greig[743], Kathryn Hanson[743], Kelly Holdroyd[743], Marie Home[743], Diane Kelly[743], Ross Kitson[743], Lear Matapure[743], Deborah Melia[743], Samantha Mellor[743], Tonicha Nortcliffe[743], Jez Pinnell[743], Matthew Robinson[743], Lisa Shaw[743], Ryan Shaw[743], Lesley Thomis[743], Alison Wilson[743], Tracy Wood[743], Lee-Ann Bayo[743], Ekta Merwaha[743], Tahira Ishaq[743], Sarah Hanley[743], David Antcliffe[744], Dorota Banach[744], Stephen Brett[744], Phoebe Coghlan[744], Ziortza Fernandez[744], Anthony Gordon[744], Roceld Rojo[744], Sonia Sousa Arias[744], Maie Templeton[744], Rajeev Jha[745], Vinodh Krishnamurthy[745], Lai Lim[745], Rehana Bi[746], Barney Scholefield[746], Lydia Ashton[746], Alison Williams[747], Claire Cheyne[747], Anne Saunderson[747], Angela Allan[748], Felicity Anderson[748], Callum Kaye[748], Jade Liew[748], Jasmine Medhora[748], Teresa Scott[748], Erin Trumper[748], Adriana Botello[748], Petra Polgarova[749], Katerina Stroud[749], Eoghan Meaney[749], Megan Jones[749], Anthony Ng[749], Shruti Agrawal[749], Nazima Pathan[749], Deborah White[749], Esther Daubney[749], Kay Elston[749], Robert Parker[750], Amie Reddy[750], Ian Turner-Bone[750], Laura Wilding[750], Peter Harding[750], Reni Jacob[752], Cathy Jones[752], Craig Denmade[752], Maria Croft[753], Ian White[753], Rajeev Jha[745], Vinodh Krishnamurthy[745], Li Lim[745], Denise Griffin[754], Nycola Muchenje[754], Mcdonald Mupudzi[754], Richard Partridge[754], Jo-Anna Conyngham[754], Rachel Thomas[754], Mary Wright[754], Maria Alvarez Corral[754], Victoria Bastion[663], Daphene Clarke[663], Beena David[663], Harriet Kent[663], Rachel Lorusso[663], Gamu Lubimbi[663], Sophie Murdoch[663], Melchizedek Penacerrada[663], Alastair Thomas[663], Jennifer Valentine[663], Ana Vochin[663], Retno Wulandari[663], Brice Djeugam[663], Joy Dawson[755], Sweyn Garrioch[755], Melanie Tolson[755], Jonathan Aldridge[755], Laura Gomes de Almeida Martins[524], Jaime Carungcong[524], Sarah Beavis[756], Katie Dale[756], Rachel Gascoyne[756], Joanne Hawes[756], Kelly Pritchard[756], Lesley Stevenson[756], Amanda Whileman[756], Anne Cowley[757], Judith Highgate[757], Rikki Crawley[758], Abigail Crew[758], Mishell Cunningham[758], Allison Daniels[758], Laura Harrison[758], Susan Hope[758], Ken Inweregbu[758], Sian Jones[758], Nicola Lancaster[758], Jamie Matthews[758], Alice Nicholson[758], Gemma Wray[758], Leonie Benham[759], Zena Bradshaw[759], Joanna Brown[759], Melanie Caswell[759], Jason Cupitt[759], Sarah Melling[759], Stephen Preston[759], Nicola Slawson[759], Emma Stoddard[759], Scott Warden[759], Edward Combes[760], Teishel Joefield[760], Sonja Monnery[760], Valerie Beech[760], Sallyanne Trotman[760], Bridget Hopkins[761], James Scriven[761], Laura Thrasyvoulou[761], Heather Willis[761], Susan Anderson[762], Janine Birch[762], Emma Collins[762], Kate Hammerton[762], Ryan O'Leary[762], Caroline Abernathy[763], Louise Foster[763], Andrew Gratrix[763], Vicky Martinson[763], Priyai Parkinson[763], Elizabeth Stones[763], Llucia Carbral-Ortega[763], Ritoo Kapoor[765], David Loader[765], Karen Castle[765], Craig Brandwood[766], Lara Smith[766], Richard Clark[766], Katie Birchall[766], Laurel Kolakaluri[766], Deborah Baines[766], Anila Sukumaran[766], Isheunesu Mapfunde[525], Megan Meredith[767], Lucy Morris[767], Lucy Ryan[767], Amy Clark[767], Julia Sampson[767], Cecilia Peters[767], Martin Dent[767], Margaret Langley[767], Saira Ashraf[767], Shuying Wei[767], Angela Andrew[767], Manish Chablani[767], Amy Kirkby[768], Kimberley Netherton[768], Michelle Bates[769], Jo Dasgin[769], Jaspret Gill[769], Annette Nilsson[769], James Scriven[769], Elena Apetri[770], Cathrine Basikolo[770], Bethan Blackledge[770], Laura Catlow[770], Bethan Charles[770], Paul Dark[770], Reece Doonan[770], Jade Harris[770], Alice Harvey[770], Daniel Horner[770], Karen Knowles[770], Stephanie Lee[770], Diane Lomas[770], Chloe Lyons[770], Tracy Marsden[770], Danielle McLaughlan[770], Liam McMorrow[770], Jessica Pendlebury[770], Jane Perez[770], Maria Poulaka[770], Nicola Proudfoot[770], Melanie Slaughter[770], Kathryn Slevin[770], Melanie Taylor[770], Vicky Thomas[770], Danielle Walker[770], Angiy Michael[770], Matthew Collis[770], Martyn Clark[771], Martina Coulding[771], Edward Jude[771], Jacqueline McCormick[771], Oliver Mercer[771], Darsh Potla[771], Hafiz Rehman[771], Heather Savill[771], Victoria Turner[771], Miriam Davey[772], David Golden[772], Rebecca Seaman[772], Jodie Hunt[773], Joy Dearden[773], Emma Dobson[773], Andrew Drummond[572,773], Michelle Mulcahy[773], Sheila Munt[773], Grainne O'Connor[773], Jennifer Philbin[773], Chloe Rishton[773], Redmond Tully[773], Sarah Winnard[773], Lenka Cagova[774], Adama Fofano[774], Lucie Garner[774], Helen Holcombe[774], Sue Mepham[774], Alice Michael Mitchell[774], Lucy Mwaura[774], K. Praman[774], Alain Vuylsteke[774], Julie Zamikula[774], Miriam Davey[772], David Golden[772], Rebecca Seaman[772], Georgia Bercades[775], David Brealey[775], Ingrid Hass[775], Niall MacCallum[775], Gladys Martir[775], Eamon Raith[775], Anna Reyes[775], Deborah Smyth[775], Abigail Taylor[776], Rachel Anne Hughes[776], Helen Thomas[776], Alun Rees[776], Michaela Duskova[776], Janet Phipps[776], Suzanne Brooks[776], Michelle Edwards[776], Peter Alexander[777], Schvearn Allen[777], Joanne Bradley-Potts[777], Craig Brantwood[777], Jasmine Egan[777], Timothy Felton[777], Grace Padden[777], Luke Ward[777], Stuart Moss[777], Susannah Glasgow[777], Kate Beesley[778], Sarah Board[778], Agnieszka Kubisz-Pudelko[778], Alison Lewis[778], Jess Perry[778], Lucy Pippard[778], Di Wood[778], Clare Buckley[778], Alison Brown[779], Jane Gregory[779], Susan O'Connell[779], Tim Smith[779], Zakaula Belagodu[780], Bridget Fuller[780], Anca Gherman[780], Olumide Olufuwa[780],

Remi Paramsothy[780], Carmel Stuart[780], Naomi Oakley[780], Charlotte Kamundi[780], David Tyl[780], Katy Collins[780], Pedro Silva[780], June Taylor[780], Laura King[780], Charlotte Coates[780], Maria Crowley[780], Phillipa Wakefield[780], Jane Beadle[780], Laura Johnson[780], Janet Sargeant[780], Madeleine Anderson[780], Catherine Jardine[781], Dewi Williams[781], Victoria Parris[782], Sheena Quaid[782], Ekaterina Watson[782], Julie Melville[783], Jay Naisbitt[783], Rosane Joseph[783], Maria Lazo[783], Olivia Walton[783], Alan Neal[783], Michaela Hill[784], Thogulava Kannan[784], Laura Wild[784], Elizabeth Allan[785], Kate Darlington[785], Ffyon Davies[785], Jack Easton[785], Sumit Kumar[785], Richard Lean[785], Daniel Menzies[785], Richard Pugh[785], Xinyi Qiu[785], Llinos Davies[785], Hannah Williams[785], Jeremy Scanlon[785], Gwyneth Davies[785], Callum Mackay[785], Joanne Lewis[785], Stephanie Rees[785], Samantha Coetzee[786], Alistair Gales[786], Igor Otahal[786], Meena Raj[786], Craig Sell[786], Helen Langton[787], Rachel Prout[787], Malcolm Watters[787], Catherine Novis[787], Gill Arbane[788], Aneta Bociek[788], Sara Campos[788], Neus Grau[788], Tim Owen Jones[788], Rosario Lim[788], Martina Marotti[788], Marlies Ostermann[788], Manu Shankar-Hari[788], Christopher Whitton[788], Anthony Barron[789], Ciara Collins[789], Sundeep Kaul[789], Heather Passmore[789], Claire Prendergast[789], Anna Reed[789], Paula Rogers[789], Rajvinder Shokkar[789], Meriel Woodruff[789], Hayley Middleton[789], Oliver Polgar[789], Claire Nolan[789], Vicky Thwaites[789], Kanta Mahay[789], Chunda Sri-Chandana[790], Joslan Scherewode[790], Lorraine Stephenson[790], Sarah Marsh[790], Hollie Bancroft[638], Mary Bellamy[638], Margaret Carmody[638], Jacqueline Daglish[638], Faye Moore[638], Joanne Rhodes[638], Mirriam Sangombe[638], Salma Kadiri[638], James Scriven[638], Amanda Ayers[792], Wendy Harrison[792], Julie North[792], Anna Cavazza[792], Maeve Cockrell[792], Eleanor Corcoran[646], Maria Depante[646], Clare Finney[646], Ellen Jerome[646], Mark McPhail[646], Monalisa Nayak[646], Harriet Noble[646], Kevin O'Reilly[646], Evita Pappa[646], Rohit Saha[646], Sian Saha[646], John Smith[646], Abigail Knighton[646], Mandy Gill[794], Paul Paul[794], Valli Ratnam[794], Sarah Shelton[794], Inez Wynter[794], David Baptista[795], Rebecca Crowe[795], Rita Fernandes[795], Rosaleen Herdman-Grant[795], Anna Joseph[795], Adam Loveridge[795], India McKenley[795], Eriko Morino[795], Andres Naranjo[795], Richard Simms[795], Kathryn Sollesta[795], Andrew Swain[795], Harish Venkatesh[795], Jacyntha Khera[795], Jonathan Fox[795], Russell Barber[796], Claire Hewitt[796], Annette Hilldrith[796], Karen Jackson-Lawrence[796], Sarah Shepardson[796], Maryanne Wills[796], Susan Butler[796], Silvia Tavares[796], Amy Cunningham[796], Julia Hindale[796], Sarwat Arif[796], Linsha George[796], Sophie Twiss[797], David Wright[797], Maureen Holland[798], Natalie Keenan[798], Marc Lyons[798], Helen Wassall[798], Chris Marsh[798], Mervin Mahenthran[798], Emma Carter[798], Thomas Kong[798], Oluronke Adanini[799], Nikhil Bhatia[799], Maines Msiska[799], Miranda Forsey[738], Agilan Kaliappan[738], Anne Nicholson[738], Joanne Riches[738], Mark Vertue[738], Louise Mew[800], Esther Mwaura[800], Richard Stewart[800], Felicity Williams[800], Lynn Wren[800], Sara-Beth Sutherland[800], Ceri Battle[801], Elaine Brinkworth[801], Rachel Harford[801], Carl Murphy[801], Luke Newey[801], Tabitha Rees[801], Marie Williams[801], Sophie Arnold[801], David Brealey[802], John Hardy[802], Henry Houlden[802], Eleanor Moncur[802], Eamon Raith[802], Ambreen Tariq[802], Arianna Tucci[802], Karen Convery[803], Deirdre Fottrell-Gould[803], Lisa Hudig[803], Jocelyn Keshet-Price[803], Georgina Randell[803], Katie Stammers[803], Marwa Abdelrazik[804], Dhanalakshmi Bakthavatsalam[804], Munzir Elhassan[804], Arunkumar Ganesan[804], Anne Haldeos[804], Jeronimo Moreno-Cuesta[804], Dharam Purohit[804], Rachel Vincent[804], Kugan Xavier[804], Kumar Rohit[804], Frater Alasdair[804], Malik Saleem[804], Carter Denise[804], Samuel Jenkins[804], Zoe Lamond[804], Alanna Wall[804], Bryan Yates[806], Jessica Reynolds[806], Helen Campbell[806], Maria Thompsom[806], Steve Dodds[806], Stacey Duffy[806], Deborah Butcher[807], Susie O'Sullivan[807], Nicola Butterworth-Cowin[807], Bethan Deacon[808], Meg Hibbert[808], Carla Pothecary[808], Dariusz Tetla[808], Christopher Woodford[808], Latha Durga[808], Gareth Kennard-Holden[808], Laura Ortiz-Ruiz de Gordoa[809], Emily Peasgood[809], Claire Phillips[809], Denise Skinner[810], Jane Gaylard[810], Dee Mullan[810], Julie Newman[810], Ellie Davies[811], Lisa Roche[811], Sonia Sathe[811], Lutece Brimfield[812], Zoe Daly[812], David Pogson[812], Steve Rose[812], Amy Collins[813], Waqas Khaliq[813,853], Estefania Treus Gude[813], Louise Allen[536], Eva Beranova[536], Nikki Crisp[536], Joanne Deery[536], Tracy Hazelton[536], Alicia Knight[536], Carly Price[536], Sorrell Tilbey[536], Salah Turki[536], Sharon Turney[536], Julian Giles[815], Simon Booth[815], Gillian Bell[816], Katy English[816], Amro Katary[816], Louise Wilcox[816], Rachael Campbell[817], Noreen Clarke[817], Jonathan Whiteside[817], Mairi Mascarenhas[817], Avril Donaldson[817], Joanna Matheson[817], Fiona Barrett[817], Marianne O'Hara[817], Laura O'Keefe[817], Clare Bradley[817], Dawn Collier[818], Anil Hormis[818], Rachel Walker[818], Victoria Maynard[818], Tahera Patel[819], Matthew Smith[819], Srikanth Chukkambotla[819], Aayesha Kazi[819], Janice Hartley[819], Joseph Dykes[819], Muhammad Hijazi[819], Sarah Keith[819], Meherunnisa Khan[819], Janet Ryan-Smith[819], Philippa Springle[819], Jacqueline Thomas[819], Samuel Saad[819], Dabheoc Coleman[819], Christopher Fine[819], Roseanna Matt[819], Bethan Gay[819], Jack Dalziel[819], Syamlan Ali[819], Drew Goodchild[819], Rhiannan Harling[819], Ravi Bhatterjee[819], Wendy Goddard[819], Chloe Davison[819], Stephen Duberly[819], Jeanette Hargreaves[819], Rachel Bolton[819], Shondipon Laha[820], Mark Verlander[820], Alexandra Williams[820], Helen Blackman[820], Ben Creagh-Brown[821], Sinead Donlon[821], Natalia Michalak-Glinska[821], Sheila Mtuwa[821], Veronika Pristopan[821], Armorel Salberg[821], Eleanor Smith[821], Sarah Stone[821], Charles Piercy[821], Jerik Verula[821], Dorota Burda[821], Rugia Montaser[821], Lesley Harden[821], Irving Mayangao[821], Cheryl Marriott[821], Paul Bradley[821], Celia Harris[821], Joshua Cooper[822], Cheryl Finch[822], Sarah Liderth[822], Alison Quinn[822], Natalia Waddington[822], Katy Fidler[823], Emma Tagliavini[823], Kevin Donnelly[823], Lynn Abel[824], Michael Brett[824], Brian Digby[824], Lisa Gemmell[824], James Hornsby[824], Patrick MacGoey[824], Pauline O'Neil[824], Richard Price[824], Natalie Rodden[824], Kevin Rooney[824], Radha Sundaram[824], Nicola Thomson[824], Rebecca Flanagan[825], Gareth Hughes[825], Scott Latham[825], Emma McKenna[825], Jennifer Anderson[825], Robert Hull[825], Kat Rhead[825], Debbie Branney[826], Jordan Frankham[826], Sally Pitts[826], Nigel White[826], Daniele Cristiano[827], Natalie Dormand[827], Zohreh Farzad[827], Mahitha Gummadi[827], Kamal Liyanage[827], Brijesh V. Patel[828], Sara Salmi[827], Geraldine Sloane[827], Vicky Thwaites[827], Mathew Varghese[827], Anelise C. Zborowski[827], Sarah Bean[829], Karen Burt[829], Michael Spivey[829], Christine Eastgate-Jackson[830], Helder Filipe[830], Daniel Martin[830], Amitaa Maharajh[830], Sara Mingo Garcia[830], Mark De Neef[830], Bethan Deacon[831], Ceri Lynch[831], Carla Pothecary[831], Lisa Roche[831], Gwenllian Sera Howe[831], Jayaprakash Singh[831], Keri Turner[831], Hannah Ellis[831], Natalie Stroud[832], Shiney Cherian[832], Sean Cutler[832], Anne Emma Heron[832], Anna Roynon-Reed[832], Tamas Szakmany[832], Gemma Williams[832], Owen Richards[832], Yusuf Cheema[832], Norfaizan Ahmad[563,833], Joann Barker[563,833], Kris Bauchmuller[563,833], Sarah Bird[563,833], Kay Cawthron[563,833], Kate Harrington[563,833], Yvonne Jackson[563,833], Faith Kibutu[563,833], Becky Lenagh[563,833], Shamiso Masuko[563,833], Gary H. Mills[563,833], Ajay Raithatha[563,833], Matthew Wiles[563,833], Jayne Willson[563,833], Helen Newell[563,833], Alison Lye[563,833], Lorenza Nwafor[563,833], Claire Jarman[563,833], Sarah Rowland-Jones[563,833], David Foote[563,833], Joby Cole[563,833], Roger Thompson[563,833], James Watson[563,833],

Lisa Hesseldon[563,833], Irene Macharia[563,833], Luke Chetam[563,833], Jacqui Smith[563,833], Amber Ford[563,833], Samantha Anderson[563,833], Kathryn Birchall[563,833], Kay Housley[563,833], Sara Walker[563,833], Leanne Milner[563,833], Helena Hanratty[563,833], Helen Trower[563,833], Patrick Phillips[563,833], Simon Oxspring[563,833], Ben Donne[563,833], Emily Bevan[834], Jane Martin[834], Dawn Trodd[834], Geoff Watson[834], Caroline Wrey Brown[834], Lara Bunni[835], Claire Jennings[835], Monica Latif[835], Rebecca Marshall[835], Gayathri Subramanian[835], Nageswar Bandla[836], Minnie Gellamucho[836], Michelle Davies[836], Christopher Thompson[836], Laura Ortiz-Ruiz de Gordoa[809], Emily Peasgood[809], Claire Phillips[809], Denise Skinner[810], Jane Gaylard[810], Dee Mullan[810], Julie Newman[810], Phil Donnison[837], Fiona Trim[837], Beena Eapen[837], Cecilia Ahmed[838], Balvinder Baines[838], Sarah Clamp[838], Julie Colley[838], Risna Haq[838], Anne Hayes[838], Jonathan Hulme[838], Samia Hussain[838], Sibet Joseph[838], Rita Kumar[838], Zahira Maqsood[838], Manjit Purewal[838], Ben Chandler[839], Kerry Elliott[839], Janine Mallinson[839], Alison Turnbull[839], Kathy Dent[840], Elizabeth Horsley[840], Muhmmad Nauman Akhtar[840], Sandra Pearson[840], Dorota Potoczna[840], Sue Spencer[840], Hayley Blakemore[841], Borislava Borislavova[841], Beverley Faulkner[841], Emma Gendall[841], Elizabeth Goff[841], Kati Hayes[841], Matt Thomas[841], Ruth Worner[841], Kerry Smith[841], Deanna Stephens[841], Carlos Castro Delgado[842], Deborah Dawson[842], Lijun Ding[842], Georgia Durrant[842], Obiageri Ezeobu[842], Sarah Farnell-Ward[842], Abiola Harrison[842], Rebecca Kanu[842], Susannah Leaver[842], Elena Maccacari[842], Soumendu Manna[842], Romina Pepermans Saluzzio[842], Joana Queiroz[842], Tinashe Samakomva[842], Christine Sicat[842], Joana Texeira[842], Edna Fernandes Da Gloria[842], Ana Lisboa[842], John Rawlins[842], Jisha Mathew[842], Ashley Kinch[842], William James Hurt[842], Nirav Shah[842], Victoria Clark[842], Maria Thanasi[842], Nikki Yun[842], Kamal Patel[842], Alison Brown[843], Vikki Crickmore[843], Gabor Debreceni[843], Joy Wilkins[843], Liz Nicol[843], Iona Burn[844], Geraldine Hambrook[844], Katarina Manso[844], Ruth Penn[844], Pradeep Shanmugasundaram[844], Julie Tebbutt[844], Danielle Thornton[844], Anthony Rostron[845], Alistair Roy[845], Lindsey Woods[845], Sarah Cornell[845], Fiona Wakinshaw[845], Kimberley Rogerson[845], Jordan Jarmain[845], Peter Anderson[846], Katie Archer[846], Karen Austin[846], Caroline Davis[846], Alison Durie[846], Olivia Kelsall[846], Jessica Thrush[846], Charlie Vigurs[846], Laura Wild[846], Hannah-Louise Wood[846], Helen Tranter[846], Karen Harrison[846], Nicholas Cowley[846], Michael McAlindon[846], Andrew Burtenshaw[846], Stephen Digby[846], Emma Low[846], Aled Morgan[846], Naiara Cother[846], Tobias Rankin[846], Sarah Clayton[846], Alex McCurdy[846], Suzanne Allibone[847], Roman Mary-Genetu[847], Vidya Kasipandian[847], Amit Patel[847], Ainhi Mac[847], Anthony Murphy[847], Parisa Mahjoob[847], Roonak Nazari[847], Lucy Worsley[847], Andrew Fagan[847], Inthakab Ali Mohamed Ali[848], Karen Beaumont[848], Mark Blunt[848], Zoe Coton[848], Hollie Curgenven[848], Mohamed Elsaadany[848], Kay Fernandes[848], Sameena Mohamed Ally[848], Harini Rangarajan[848], Varun Sarathy[848], Sivarupan Selvanayagam[848], Dave Vedage[848], Matthew White[848], Jaime Fernandez-Roman[849], David O. Hamilton[849], Emily Johnson[849], Brian Johnston[849], Maria Lopez Martinez[849], Suleman Mulla[849], David Shaw[849], Alicia A. C. Waite[849], Victoria Waugh[849], Ingeborg D. Welters[849], Karen Williams[849], Thomas Bemand[850], Ethel Black[850], Arnold Dela Rosa[850], Ryan Howle[850], Shaman Jhanji[850], Ravishankar Rao Baikady[850], Kate Colette Tatham[850], Benjamin Thomas[850], Matthew Halkes[851], Pauline Mercer[851], Lorraine Thornton[851], Joe West[852], Tracy Baird[852], Jim Ruddy[852], Rosie Reece-Anthony[853], Mark Birt[854], Amanda Cowton[854], Andrea Kay[854], Melanie Kent[854], Kathryn Potts[854], Ami Wilkinson[854], Suzanne Naylor[854], Ellen Brown[854], Michele Clark[855], Sarah Purvis[855], Jade Cole[856], Michelle Davies[856], Rhys Davies[856], Donna Duffin[856], Helen Hill[856], Ben Player[856], Emma Thomas[856], Angharad Williams[856], Claire Marie Beith[857], Karen Black[857], Suzanne Clements[857], Alan Morrison[857], Dominic Strachan[857], Margaret Taylor[857], Michelle Clarkson[857], Stuart D'Sylva[857], Kathryn Norman[857], Tina Coventry[858], Susan Fowler[858], Michael MacMahon[858], Amanda McGregor[858], Ailbhe Brady[859], Rebekah Chan[859], Jeff Little[859], Shane McIvor[859], Helena Prady[859], Helen Whittle[859], Bijoy Mathew[859], Melanie Clapham[860], Rosemary Harper[860], Una Poultney[860], Polly Rice[860], Tim Smith[860], Rachel Mutch[860], Yolanda Baird[861], Aaron Butler[861], Indra Chadbourn[861], Linda Folkes[861], Heather Fox[861], Amy Gardner[861], Raquel Gomez[861], Gillian Hobden[861], Luke Hodgson[861], Kirsten King[861], Michael Margarson[861], Tim Martindale[861], Emma Meadows[861], Dana Raynard[861], Yvette Thirlwall[861], David Helm[861], Jordi Margalef[861], Karen Shuker[862], Ascanio Tridente[862], Sara Smuts[526], Joseph Duffield[526], Oliver Smith[526], Lewis Mallon[526], Watkins Claire[526], Isobel Birkinshaw[863], Joseph Carter[863], Kate Howard[863], Joanne Ingham[863], Rosie Joy[863], Harriet Pearson[863], Samantha Roche[863], Zoe Scott[863], Ellen Knights[864], Alicia Price[864], Alice Thomas[864], Chris Thorpe[864], Azmerelda Abraheem[865], Peter Bamford[865], Kathryn Cawley[865], Charlie Dunmore[865], Maria Faulkner[865], Rumanah Girach[865], Helen Jeffrey[865], Rhianna Jones[865], Emily London[865], Imrun Nagra[865], Farah Nasir[865], Hannah Sainsbury[865], Clare Smedley[865], Reena Khade[866], Ashok Sundar[866], George Tsinaslanidis[866], Teresa Behan[867], Caroline Burnett[867], Jonathan Hatton[867], Elaine Heeney[867], Atideb Mitra[867], Maria Newton[867], Rachel Pollard[867], Rachael Stead[867], Jenny Birch[867], Laura Bough[868], Josie Goodsell[868], Rebecca Tutton[868], Patricia Williams[868], Sarah Williams[868], Barbara Winter-Goodwin[868], Anne Cowley[757], Judith Highgate[757], Fiona Auld[869], Joanne Donnachie[869], Ian Edmond[869], Lynn Prentice[869], Nikole Runciman[869], Dario Salutous[869], Lesley Symon[869], Anne Todd[869], Patricia Turner[869], Abigail Short[869], Laura Sweeney[869], Euan Murdoch[869], Dhaneesha Senaratne[869], Karen Burns[870], Andrew Higham[870,905], Taya Anderson[871], Dan Hawcutt[871], Laura O'Malley[871], Laura Rad[871], Naomi Rogers[871], Paula Saunderson[871], Kathryn Sian Allison[871], Deborah Afolabi[871], Jennifer Whitbread[871], Dawn Jones[871], Rachael Dore[871], Liana Lankester[872], Nikitas Nikitas[872], Colin Wells[872], Bethan Stowe[872], Kayleigh Spencer[872], Susanne Cathcart[873], Katharine Duffy[873], Alex Puxty[873], Kathryn Puxty[873], Lynne Turner[873], Jane Ireland[873], Gary Semple[873], Peter Barry[874], Paula Hilltout[875], Jayne Evitts[875], Amanda Tyler[875], Joanne Waldron[875], Val Irvine[875], Benjamin Shelley[876], Olugbenga Akinkugbe[877], Alasdair Bamford[877], Emily Beech[877], Holly Belfield[877], Michael Bell[877], Charlene Davies[877], Gareth A. L. Jones[877], Tara McHugh[877], Hamza Meghari[877], Lauran O'Neill[877], Mark J. Peters[877], Samiran Ray[877], Ana Luisa Tomas[877], Amy Easthope[878], Claire Gorman[878], Abhinav Gupta[878], Elizabeth Timlick[878], Rebecca Brady[878], Stephen Bonner[879], Keith Hugill[879], Jessica Jones[879], Steven Liggett[879], Archana Bashyal[880], Neil Davidson[880], Paula Hutton[880], Stuart McKechnie[880], Jean Wilson[880], Neil Flint[881], Patel Rekha[881], Dawn Hales[881], Carina Cruz[882], Natalie Pattison[882], Shameer Gopal[883], Nichola Harris[883], Victoria Lake[883], Stella Metherell[883], Elizabeth Radford[883], Ian Clement[884], Bijal Patel[884], A. Gulati[884], Carole Hays[884], K. Webster[884], Anne Hudson[884], Andrea Webster[884], Elaine Stephenson[884], Louise McCormack[884], Victoria Slater[884], Rachel Nixon[884], Helen Hanson[884], Maggie Fearby[884], Sinead Kelly[884], Victoria Bridgett[884], Philip Robinson[884], Christine Almaden-Boyle[885], Pauline Austin[885], Louise Cabrelli[885], Stephen Cole[885], Matt Casey[885], Susan Chapman[885], Stephen Cole[885], Clare Whyte[885], Adam Brayne[886], Emma Fisher[886], Jane Hunt[886], Peter Jackson[886], Duncan Kaye[886], Nicholas Love[886], Juliet Parkin[886], Victoria Tuckey[886], Lynne van Koutrik[886], Sasha Carter[886], Benedict Andrew[886], Louise Findlay[886], Katie Adams[886], Michelle Bruce[887], Karen Connolly[887], Tracy Duncan[887], Helen T.-Michael[887], Gabriella Lindergard[887], Samuel Hey[887], Claire Fox[887], Jordan Alfonso[887], Laura Jayne Durrans[887], Jacinta Guerin[887], Bethan Blackledge[887], Jade Harris[887], Martin Hruska[887], Ayaa Eltayeb[887], Thomas Lamb[887], Tracey Hodgkiss[887], Lisa Cooper[887], Joanne Rothwell[887], Catherine Dennis[888], Alastair McGregor[888], Victoria Parris[888], Sinduya Srikaran[888], Anisha Sukha[888], Kim Davies[889], Linda O'Brien[889], Zohra Omar[889], Igor Otahal[889], Emma Perkins[889], Tracy Lewis[889], Isobel Sutherland[889], Hollie Brooke[890], Sarah Buckley[890], Jose Cebrian Suarez[890], Ruth Charlesworth[890], Karen Hansson[890], John Norris[890], Alice Poole[890], Alastair Rose[890], Rajdeep Sandhu[890], Brendan Sloan[890], Elizabeth Smithson[890], Muthu Thirumaran[890], Veronica Wagstaff[890], Alexandra Metcalfe[890], Julie Camsooksai[891], Charlotte Humphrey[891], Sarah Jenkins[891], Henrik Reschreiter[891], Beverley Wadams[891], Yasmin DeAth[891], Colene Adams[892], Anita Agasou[892], Tracie Arden[893], Amy Bowes[892], Pauline Boyle[892], Mandy Beekes[893], Heather Button[893], Nigel Capps[624,893], Mandy Carnahan[892], Anne Carter[892], Danielle Childs[892], Denise Donaldson[892], Kelly Hard[892], Fran Hurford[893], Yasmin Hussain[892], Ayesha Javaid[893], James Jones[893], Sanal Jose[624,893], Michael Leigh[892], Terry Martin[893], Helen Millward[893], Nichola Motherwell[893], Rachel Rikunenko[892], Jo Stickley[892], Julie Summers[893], Louise Ting[893], Helen Tivenan[892], Louise Tonks[893], Rebecca Wilcox[892], Maria Bokhari[899], Vanessa Linnett[899], Rachael Lucas[899], Wendy McCormick[899], Jenny Ritzema[899], Amanda Sanderson[899], Helen Wild[899], Nicola Baxter[900], Steven Henderson[900], Sophie Kennedy-Hay[900], Christopher McParland[900], Laura Rooney[900], Malcolm Sim[900], Gordan McCreath[900], Mark Brunton[901], Jess Caterson[901], Holly Coles[901], Matthew Frise[901], Sabi Gurung Rai[901], Nicola Jacques[901], Liza Keating[901], Emma Tilney[901], Shauna Bartley[901], Parminder Bhuie[901], Charlotte Downes[902], Kathleen Holding[902], Katie Riches[902], Mary Hilton[902], Mel Hayman[902], Deepak Subramanian[902], Priya Daniel[902], Letizia Zitter[903], Sarah Benyon[903], Suzie Marriott[903], Linda Park[903], Samantha Keenan[903], Elizabeth Gordon[903], Helen Quinn[903], Kizzy Baines[903], Gillian Andrew[904], J. Kenneth Baillie[904], Lucy Barclay[904], Marie Callaghan[904], Rachael Campbell[904], Sarah Clark[904], Dave Hope[904], Lucy Marshall[904], Corrienne McCulloch[904], Kate Briton[904], Jo Singleton[904], Sophie Birch[904], Andrew Higham[905], Kerry Simpson[905], Jayne Craig[905], Carrie Demetriou[905], Charlotte Eckbad[906], Sarah Hierons[906], Lucy Howie[906], Sarah Mitchard[906], Lidia Ramos[906], Alfredo Serrano-Ruiz[906], Willie Holme[906], Fiona Kelly[906], Vishal Amin[907], Elena Anastasescu[907], Vikram Anumakonda[907], Komala Karthik[907], Rizwana Kausar[907], Karen Reid[907], Jacqueline Smith[907], Janet Imeson-Wood[907], Arianna Bellini[908], Jade Bryant[908], Anton Mayer[908], Amy Pickard[908], Nicholas Roe[908], Jason Sowter[908], Alex Howlett[908], Kristine Criste[909], Rebecca Cusack[909], Kim Golder[909], Hannah Golding[909], Oliver Jones[909], Samantha Leggett[909], Michelle Male[909], Martyna Marani[909], Kirsty Prager[909], Toran Williams[909], Belinda Roberts[909], Karen Salmon[909], Prisca Gondo[910], B. Hadebe[910], Abdul Kayani[910], Bridgett Masunda[910], Ashar Ahmed[911], Anna Morris[911], Srinivas Jakkula[911], Kate Long[912], Simon Whiteley[912], Elizabeth Wilby[912], Bethan Ogg[912], Sam Moultrie[747], M. Odam[747], Jeremy Bewley[913], Zoe Garland[913], Lisa Grimmer[913], Bethany Gumbrill[913], Rebekah Johnson[913], Katie Sweet[913], Denise Webster[913], Georgia Efford[913], Sara Bennett[914], Emma Goodwin[914], Matthew Jackson[914], Alissa Kent[914], Clare Tibke[914], Wiesia Woodyatt[914], Ahmed Zaki[914], Amelia Daniel[915], Joanne Finn[915], Rajnish Saha[915], Nikki Staines[915], Amy Easthope[915], Pamela Bremmer[528], J. Allan[916], T. Geary[916], Gordon Houston[916], A. Meikle[916], P. O'Brien[916], Dina Bell[917], Rosalind Boyle[917], Katie Douglas[917], Lynn Glass[917], Emma Lee[917], Liz Lennon[917], Austin Rattray[917], Rob Charnock[918], Denise McFarland[918], Denise Cosgrove[918], Ben Attwood[919], Penny Parsons[919], Siobhain Carmody[529], Metod Oblak[920], Monica Popescu[920], Mini Thankachen[920], Rosie Baruah[920], Sheila Morris[920], Susie Ferguson[602], Amy Shepherd[602], Abdelhakim Altabaibeh[922], Ana Alvaro[922], Kayleigh Gilbert[922], Louise Ma[922], Loreta Mostoles[922], Chetan Parmar[922], Kathryn Simpson[922], Champa Jetha[922], Lauren Booker[922], Anezka Pratley[922], Tracey Cosier[923], Gemma Millen[923], Neil Richardson[923], Natasha Schumacher[923], Heather Weston[923], James Rand[923], Beatrice Alex[718], Benjamin Bach[718], Wendy S. Barclay[932], Debby Bogaert[705], Meera Chand[933], Graham S. Cooke[934], Annemarie B. Docherty[714], Jake Dunning[714], Ana da Silva Filipe[936], Tom Fletcher[937], Christoper A. Green[934], Ewen M. Harrison[714], Julian A. Hiscox[338], Samreen Ijaz[940], Saye Khoo[941], Paul Klenerman[942], Andrew Law[60], Wei Shen Lim[944], Alexander J. Mentzer[945], Laura Merson[946], Alison M. Meynert[62], Mahdad Noursadeghi[947], Shona C. Moore[948], Massimo Palmarini[948], William A. Paxton[948], Georgios Pollakis[948], Nicholas Price[949], Andrew Rambaut[950], David L. Robertson[936], Clark D. Russell[705], Vanessa Sancho-Shimizu[951], Janet T. Scott[936], Thushan de Silva[952], Louise Sigfrid[946], Tom Solomon[530], Shiranee Sriskandan[934], David Stuart[953], Richard S. Tedder[955], Emma C. Thomson[936], A. A. Roger Thompson[956], Ryan S. Thwaites[730], Lance C. W. Turtle[530,732], Rishi K. Gupta[957], Carlo Palmieri[958], Olivia V. Swann[959], Maria Zambon[935], Marc-Emmanuel Dumas[960], Julian L. Griffin[960], Zoltan Takats[960], Kanta Chechi[961], Petros Andrikopoulos[960], Anthonia Osagie[960], Michael Olanipekun[960], Sonia Liggi[960], Matthew R. Lewis[962], Gonçalo dos Santos Correia[962], Caroline J. Sands[962], Panteleimon Takis[962], Lynn Maslen[962], William Greenhalf[963], Victoria Shaw[964], Sarah E. McDonald[965], Seán Keating[965], Katie A. Ahmed[966], Jane A. Armstrong[966], Milton Ashworth[966], Innocent G. Asiimwe[966], Siddharth Bakshi[966], Samantha L. Barlow[966], Laura Booth[966], Benjamin Brennan[967], Katie Bullock[966], Benjamin W. A. Catterall[966], Jordan J. Clark[966], Emily A. Clarke[966], Louise Cooper[966], Helen Cox[966], Christopher Davis[966], Oslem Dincarslan[966], Chris Dunn[966], Philip Dyer[966], Angela Elliott[966], Anthony Evans[966], Lorna Finch[966], Lewis W. S. Fisher[966], Terry Foster[966], Isabel Garcia-Dorival[966], William Greenhalf[966], Philip Gunning[966], Catherine Hartley[966], Rebecca L. Jensen[966], Christopher B. Jones[966], Trevor R. Jones[966], Shadia Khandaker[966], Katharine King[966], Chrysa Koukorava[966], Annette Lake[967], Suzannah Lant[966], Diane Latawiec[966], Lara Lavelle-Langham[966], Daniella Lefteri[966], Lauren Lett[966], Lucia A. Livoti[966], Maria Mancini[966], Sarah McDonald[966], Laurence McEvoy[966], John McLauchlan[967], Soeren Metelmann[966], Nahida S. Miah[966], Joanna Middleton[966], Joyce Mitchell[966], Shona C. Moore[966], Ellen G. Murphy[966], Rebekah Penrice-Randal[966], Jack Pilgrim[966], Tessa Prince[966], Will Reynolds[966], P. Matthew Ridley[966], Debby Sales[966], Victoria E. Shaw[966], Rebecca K. Shears[966], Benjamin Small[966], Krishanthi S. Subramaniam[966], Agnieska Szemiel[966], Aislynn Taggart[967], Jolanta Tanianis-Hughes[966], Jordan Thomas[966], Erwan Trochu[966], Libby van Tonder[966], Eve Wilcock[966], J. Eunice Zhang[966], Lisa Flaherty[966], Nicole Maziere[966], Emily Cass[966], Alejandra Doce Carracedo[966], Nicola Carlucci[966], Anthony Holmes[966], Hannah Massey[966], Lee Murphy[554], Nicola Wrobel[554], Sarah McCafferty[554], Kirstie Morrice[554] & Alan MacLean[554]

**Admin team members**

Ruth Armstrong[60], J. Kenneth Baillie[60,61,62], Ceilia Boz[60], Adam Brown[60], Richard Clark[968], Sara Clohisey[60], Audrey Coutts[968], Louise Cullum[60], Nicky Day[60], Lorna Donnelly[968], Esther Duncan[60], Angie Fawkes[968], Paul Finernan[60], Max Head Fourman[60], James Furniss[60], Tammy Gilchrist[968], Ailsa Golightly[60], Katarzyna Hafezi[968], Ross Hendry[60], Andy Law[60], Dawn Law[60], Rachel Law[60], Sarah Law[60], Louise Macgillivray[968], Alan Maclean[968], Hanning Mal[60], Sarah McCafferty[968], Ellie Mcmaster[60], Jen Meikle[60], Shona C. Moore[732], Kirstie Morrice[968], Lee Murphy[968], Wilna Oosthuyzen[60], Nicholas Parkinson[60], Trevor Paterson[60], Andrew Stenhouse[60], Maaike Swets[60,970], Helen Szoor-McElhinney[60], Filip Taneski[60], Lance C. W. Turtle[530,732], Tony Wackett[60], Mairi Ward[60], Jane Weaver[60], Nicola Wrobel[968], Marie Zechner[60], Judy Coyle[60], Bernadette Gallagher[60], Rebecca Lidstone-Scott[60], Debbie Hamilton[60], Katherine Schon[971], Anita Furlong[971], Heather Biggs[971], Fiona Griffiths[60], Eleanor Andrews[762], Kathy Brickell[527], Michelle Smyth[527], Lorna Murphy[527], Gail Carson[946], Hayley Hardwick[530] & Chloe Donohue[715]

**COVID-19 HGI corresponding authors**

Benjamin M. Neale[4,1280]✉, Mark Daly[1,2,3,1280]✉ & Andrea Ganna[1,2,3,1280]✉

[1]Institute for Molecular Medicine Finland (FIMM), University of Helsinki, Helsinki, Finland. [2]Broad Institute of MIT and Harvard, Cambridge, MA, USA. [3]Analytic and Translational Genetics Unit, Massachusetts General Hospital, Boston, MA, USA. [4]Massachusetts General Hospital, Broad Institute of MIT and Harvard, Cambridge, MA, USA. [6]Yale University, New Haven, CT, USA. [7]Icahn School of Medicine at Mount Sinai, New York, NY, USA. [8]Stroke Pharmacogenomics and Genetics, Biomedical Research Institute Sant Pau (IIB Sant Pau), Sant Pau Hospital, Inmungen-CoV2, Barcelona, Spain. [9]Institute of Virology, Technical University Munich and Helmholtz Zentrum München, Munich, Germany. [10]Institute of Psychiatric Phenomics and Genomics, Medical Center of the University of Munich, Munich, Germany. [11]Department of Psychiatry, Medical Center of the University of Munich, Munich, Germany. [12]IRCCS, Istituto Giannina Gaslini, Genova, Italy. [13]Department of Neurosciences, Rehabilitation, Ophthalmology, Genetics, Maternal and Child Health, University of Genova, Genova, Italy. [14]Queen Mary University of London, London, UK. [15]Open Targets, Wellcome Genome Campus, Hinxton, UK. [16]Department of Complex Trait Genetics, Center for Neurogenomics and Cognitive Research, Amsterdam Neuroscience, Vrije Universiteit Amsterdam, Amsterdam, The Netherlands. [17]Lady Davis Institute, Jewish General Hospital, McGill University, Montreal, Quebec, Canada. [18]Medical Research Institute, Kangbuk Samsung Hospital, Sungkyunkwan University School of Medicine, Suwon, Republic of Korea. [19]Osaka University Graduate School of Medicine, Osaka, Japan. [20]Baylor College of Medicine, Houston, TX, USA. [21]Mohammed Bin Rashid University of Medicine and Health Sciences, Dubai, United Arab Emirates. [22]MRC Integrative Epidemiology Unit (IEU), University of Bristol, Bristol, UK. [23]Department of Internal Medicine, Division of Cardiovascular Medicine, Michigan Medicine, Ann Arbor, MI, USA. [24]Department of Human Genetics, University of Michigan Medical School, Ann Arbor, MI, USA. [25]Department of Computational Medicine and Bioinformatics, University of Michigan Medical School, Ann Arbor, MI, USA. [26]Seaver Autism Center for Research and Treatment, Department of Psychiatry, Icahn School of Medicine at Mount Sinai, New York, NY, USA. [29]Program in Medical and Population Genetics, Broad Institute of MIT and Harvard, Cambridge, MA, USA. [32]David Geffen School of Medicine at UCLA, Los Angeles, CA, USA. [33]Institut Pasteur, Paris, France. [34]Harvard School of Public Health, Boston, MA, USA. [35]Institute for Molecular Bioscience, The University of Queensland, Brisbane, Queensland, Australia. [38]Wellcome Sanger Institute, Wellcome Genome Campus, Hinxton, UK. [40]European Molecular Biology Laboratory, European Bioinformatics Institute (EMBL-EBI), Wellcome Genome Campus, Hinxton, UK. [42]Centre for Bioinformatics and Data Analysis, Medical University of Bialystok, Bialystok, Poland. [43]Trieste University, Trieste, Italy. [44]Vanderbilt University Medical Center, Nashville, TN, USA. [45]University of California San Francisco, San Francisco, CA, USA. [46]Stanford University, Stanford, CA, USA. [47]University of Siena, Siena, Italy. [49]Boston Children's Hospital, Broad Institute of MIT and Harvard, Cambridge, MA, USA. [50]Blizard Institute, Queen Mary University of London, London, UK. [51]Department of Genetics, University Medical Centre Groningen, Groningen, The Netherlands. [52]Department of Genetics, University Medical Centre Utrecht, Utrecht, The Netherlands. [53]Department of Epidemiology, Biostatistics and Occupational Health, McGill University, Montreal, Quebec, Canada. [55]Department of Twin Research, King's College London, London, UK. [56]Department of Human Genetics, McGill University, Montreal, Quebec, Canada. [57]Kyoto-McGill International Collaborative School in Genomic Medicine, Graduate School of Medicine, Kyoto University, Kyoto, Japan. [59]National Institutes of Health, Bethesda, MD, USA. [60]The Roslin Institute, University of Edinburgh, Edinburgh, UK. [61]Intensive Care Unit, Royal Infirmary of Edinburgh, Edinburgh, UK. [62]MRC Human Genetics Unit, Institute of Genetics and Molecular Medicine, University of Edinburgh, Western General Hospital, Edinburgh, UK. [63]Medical Genetics, University of Siena, Siena, Italy. [64]Genetica Medica, Azienda Ospedaliero-Universitaria Senese, Siena, Italy. [65]Med Biotech Hub and Competence Center, Department of Medical Biotechnologies, University of Siena, Siena, Italy. [66]Department of Electronics, Information and Bioengineering (DEIB), Politecnico di Milano, Milano, Italy. [67]Politecnico di Milano, Milan, Italy. [68]University of Michigan, Ann Arbor, MI, USA. [69]Vanderbilt School of Medicine, Nashville, TN, USA. [70]All India Institute of Medical Sciences Kalyani, Kalyani, India. [71]Hasso Plattner Institute, New York, NY, USA. [72]Naina Tech, Hyderabad, India. [73]EMBL-European Bioinformatics Institute, Hinxton, UK. [74]University of Northampton, Northampton, UK. [75]University of Helsinki, Helsinki, Finland. [76]University of Miami, Miami, FL, USA. [78]Ecole Centrale de Nantes, Inserm, Centre de Recherche en Transplantation et Immunologie, Nantes University, UMR1064, ITUN, Nantes, France. [79]University of Liège, Liège, Belgium. [80]Qatar Genome Program, Qatar Foundation Research, Development and Innovation, Qatar Foundation, Doha, Qatar. [82]Medical and Population Genetics and Cardiovascular Disease Initiative, Broad Institute of Harvard and MIT, Cambridge, Cambridge, MA, USA. [83]Cardiovascular Research Center, Massachusetts General Hospital, Boston, MA, USA. [84]Intensive Care Unit, Vall d'Hebron Hospital, Barcelona, Spain. [85]Institut de Biomedicina de València - CSIC, València, Spain. [86]Centro de Investigación Biomédica en Red en Enfermedades Neurodegenerativas (CIBERNED), València, Spain. [87]Unidad Mixta de Neurología y Genética, Instituto de Investigación Sanitaria La Fe, València, Spain. [88]Erasmus Medical Center, Rotterdam, The Netherlands. [89]National Genome Center, Copenhagen, Denmark. [90]University of Copenhagen, Copenhagen, Denmark. [91]Genomics PLC, Oxford, UK. [93]Institute for Community Medicine, University Medicine Greifswald, Greifswald, Germany. [94]Department of Population Medicine and Lifestyle Diseases Prevention, Medical University of Bialystok, Bialystok, Poland. [95]Genomics England, London, UK. [96]Junta de Andalucía, Seville, Spain. [97]Human Genetics Program of ICBM and Department of Basic-Clinical Oncology, University of Chile, Santiago, Chile. [98]Center for the Development of Scientific Research (CEDIC), Asunción, Paraguay. [100]Translational Bioinformatics Unit, Navarrabiomed, Complejo Hospitalario de Navarra (CHN), Universidad Pública de Navarra (UPNA), IdiSNA, Pamplona, Spain. [101]Mucosal & Salivary Biology Division, King's College London Dental Institute, London, UK. [102]GENYO, Center for Genomics and Oncological Research Pfizer, University of Granada, Andalusian Regional Government, Granada, Spain. [103]University of Puerto Rico, San Juan, Puerto Rico. [104]National Laboratory of Genomics for Biodiversity (LANGEBIO), Advanced Genomics Unit, CINVESTAV, Irapuato, Mexico. [105]Queensland University of Technology, Brisbane, Queensland, Australia. [106]Clinical Research Unit of Nanoro, Institut de Recherche en Sciences de la Santé, CNRST, Ouagadougou, Burkina Faso. [107]McGill University, Montreal, Quebec, Canada. [108]Université de Montréal, Montreal, Quebec, Canada. [109]Fonds de la Recherche Scientifique (FNRS) & Centre de Génétique Humaine, Hôpital Erasme, Université Libre de Bruxelles, Brussels, Belgium. [112]University of Pecs Medical School, Pécs, Hungary. [113]Institute of Biomedicine and Cancer Research Laboratories, Western Cancer Centre FICAN West, University of Turku, Turku, Finland. [114]Institute of Biomedical Technologies, National Research Council, Segrate, Italy. [115]Immediate, Milan, Italy. [116]University of Cambridge, Cambridge, UK. [117]Genome Opinion, Seoul, Republic of Korea. [119]University of Groningen, Groningen, The Netherlands. [120]Universiti Malaysia Pahang, Gambang, Malaysia. [122]Vrije Universiteit Amsterdam, Amsterdam, The Netherlands. [123]University Medical Centre Groningen, University of Groningen, Groningen, The Netherlands. [124]MNM DIAGNOSTICS, Pozna?, Poland. [125]Institute for Systems Biology, Seattle, WA, USA. [126]Sultan Idris Education University, Tanjung Malim, Malaysia. [127]Hospital Kulim, Kedah, Malaysia. [128]AbbVie, Lake Buff, IL, USA. [129]Root Deep Insight, Boston MA, USA. [130]23andMe, Sunnyvale, CA, USA. [131]GSK, Stevenage, UK. [132]Department of Pharmacology, Feinberg School of Medicine, Northwestern University, Chicago, IL, USA. [133]Department of Medicine, Northwestern University, Chicago, IL, USA. [134]Washington DC Veterans Affairs Medical Center, Hospital Medicine, Washington, DC, USA. [135]Department of Medicine, George Washington University, Washington, DC, USA. [136]Section of Hospital Medicine, Department of Medicine, University of Chicago, Chicago, IL, USA. [137]Section of Hematology and Oncology, Department of Medicine, University of Chicago, Chicago, IL, USA. [138]College of Pharmacy, University of Illinois at Chicago, Chicago, IL, USA. [144]Department of Pharmacology, George Washington University, Washington, DC, USA. [148]Department of Neurology, Amsterdam UMC, Amsterdam Neuroscience, Amsterdam, The Netherlands. [149]Department of Intensive Care, Amsterdam UMC, Amsterdam, The Netherlands. [150]Department of Infectious Diseases, Amsterdam UMC, Amsterdam, The Netherlands. [151]Department of Clinical Epidemiology, Biostatistics and Bioinformatics, Amsterdam UMC, Amsterdam, The Netherlands. [152]Experimental Immunology, Amsterdam UMC, Amsterdam, The Netherlands. [153]Department of Pulmonology, Amsterdam UMC, Amsterdam, The Netherlands. [154]Department of Pathology, Amsterdam UMC, Amsterdam, The Netherlands. [155]Department of Anesthesiology, Amsterdam UMC, Amsterdam, The Netherlands. [156]Amsterdam UMC Biobank Core Facility, Amsterdam UMC, Amsterdam, The Netherlands. [157]Department of Radiology, Amsterdam UMC, Amsterdam, The Netherlands. [158]Department of Medical Microbiology, Amsterdam UMC, Amsterdam, The Netherlands. [159]Department of Clinical Chemistry, Amsterdam UMC, Amsterdam, The Netherlands. [160]Amsterdam UMC Biobank, Amsterdam UMC, Amsterdam, The Netherlands. [161]Core Facility Genomics, Amsterdam UMC, Amsterdam, The Netherlands. [162]Ancestry, Lehi, UT, USA. [163]GIGA-Institute, University of Liège, Liège, Belgium. [164]CHC Mont-Légia, Liège, Belgium. [165]BHUL (Liège Biobank), CHU of Liège, Liège, Belgium. [167]Stanley Center for Psychiatric Research, Broad Institute of MIT and Harvard, Cambridge, MA, USA. [168]Centre de Génétique Humaine, Hôpital Erasme, Université Libre de Bruxelles, Brussels, Belgium. [169]Service de Médecine Interne, Hôpital Erasme, Université Libre de Bruxelles, Brussels, Belgium. [170]CHU of Liège, University of Liège, Liège, Belgium. [174]McGill Genome Centre, McGill University, Montréal, Québec, Canada. [177]Department of Emergency Medicine, McGill University, Montreal, Quebec, Canada. [178]Emergency Department, Jewish General Hospital, McGill University, Montreal, Quebec, Canada. [179]McGill AIDS Centre, Department of Microbiology and Immunology, Lady Davis Institute for Medical Research, Jewish General Hospital, McGill University, Montreal, Quebec, Canada. [180]McGill Centre for Viral Diseases, Department of Infectious Disease, Lady Davis Institute, Jewish General Hospital, Montreal, Quebec, Canada. [181]Research Centre of the Centre Hospitalier de l'Université de Montréal, Montreal, Quebec, Canada. [182]Department of Medicine, Research Centre of the Centre Hospitalier de l'Université de Montréal, Montreal, Quebec, Canada. [183]Department of Medicine, Université de Montréal, Montreal, Quebec, Canada. [184]Department of Medicine and Human Genetics, McGill University, Montreal, Quebec, Canada. [185]Department of Intensive Care, Research Centre of the Centre Hospitalier de l'Université de Montréal, Montreal, Quebec, Canada. [186]Division of Infectious Diseases, Research Centre of the Centre Hospitalier de l'Université de Montréal, Montréal, Quebec, Canada. [187]Division of Genetic Medicine, Department of Medicine, Vanderbilt University Medical Center, Nashville, TN, USA. [188]Vanderbilt Genetics Institute, Vanderbilt University Medical Center, Nashville, TN, USA. [189]Institute of Human Genetics, University Hospital Bonn, Medical Faculty University of Bonn, Bonn, Germany. [190]Institute of Genomic Statistics and Bioinformatics, University Hospital Bonn, Medical Faculty University of Bonn, Bonn, Germany. [191]Department of Gastroenterology, Hepatology and Infectious Diseases, University Hospital Düsseldorf, Medical Faculty Heinrich Heine University, Düsseldorf, Germany. [192]Institute of Human Genetics, Medical Faculty, RWTH Aachen University, Aachen, Germany. [193]Clinic for Cardiology, Angiology and Internal Intensive Medicine, Medical Clinic I, RWTH Aachen University, Aachen, Germany. [194]Department of Pneumology and Intensive Care Medicine, Faculty of Medicine, RWTH Aachen University, Aachen, Germany. [195]Department of Pneumology, Hannover Medical School, Hannover, Germany. [196]Department of Gastroenterology, Hepatology and Endocrinology, Hannover Medical School, Hannover, Germany. [197]Hannover Unified Biobank, Hannover Medical School, Hannover, Germany. [198]Department I of Internal Medicine, Faculty of Medicine and University Hospital of Cologne, University of Cologne, Cologne, Germany. [199]Center for Molecular Medicine Cologne (CMMC), University of Cologne, Cologne, Germany. [200]German Center for Infection Research (DZIF), Partner Site Bonn-Cologne, Cologne, Germany. [202]Cologne Center

for Genomics (CCG), University of Cologne, Cologne, Germany. [203]Department of Anesthesiology and Intensive Care Medicine, University Hospital Essen, University Duisburg-Essen, Essen, Germany. [204]Department of Child and Adolescent Psychiatry, University Hospital Essen, University of Duisburg-Essen, Essen, Germany. [205]Department of Infectious Diseases, University Hospital Essen, University Duisburg-Essen, Essen, Germany. [206]Department of Pneumology, Allergology and Respiratory Medicine, University Hospital Saarland, Homburg/Saar, Germany. [207]Center of Human and Molecular Biology, Department of Human Genetics, University Hospital Saarland, Homburg/Saar, Germany. [208]Department of Genetics & Epigenetics, Saarland University, Saarbrücken, Germany. [209]Eurac Research, Institute for Biomedicine (affiliated to the University of Lübeck), Bolzano, Italy. [210]University of Colorado Anschutz Medical Campus, Aurora, CO, USA. [211]Department of Genetics and Development, Institute for Genomic Medicine, Columbia University, New York, NY, USA. [212]Department of Medicine, Institute for Genomic Medicine, Columbia University, New York, NY, USA. [213]Department of Biomedical Informatics, Columbia University, New York, NY, USA. [214]Department of Pediatrics, Columbia University, New York, NY, USA. [215]Department of Medicine, Columbia University, New York, NY, USA. [216]Institute for Genomic Medicine, Columbia University, New York, NY, USA. [217]Department of Biostatistics, Mailman School of Public Health, Columbia University, New York, NY, USA. [218]Department of Pathology and Cell Biology, Columbia University, New York, NY, USA. [219]Medical Research Institute, Kangbuk Samsung Hospital, Sungkyunkwan University School of Medicine, Seoul, Republic of Korea. [220]Department of Clinical Research Design and Evaluation, SAIHST, Sungkyunkwan University, Seoul, Republic of Korea. [221]Division of Gastroenterology, Department of Medicine, Kangbuk Samsung Hospital, Sungkyunkwan University, School of Medicine, Seoul, Republic of Korea. [222]Department of Biochemistry, College of Medicine, Ewha Womans University, Seoul, Republic of Korea. [223]Department of Internal Medicine, Seoul National University Hospital, Seoul National University College of Medicine, Seoul, Republic of Korea. [224]Department of Periodontology, Section of Dentistry, Seoul National University Bundang Hospital, Seongnam, Republic of Korea. [225]Department of Internal Medicine, Seoul National University Bundang Hospital, Seoul National University College of Medicine, Seongnam, Republic of Korea. [226]Department of Physical & Rehabilitation Medicine, Kangbuk Samsung Hospital, Sungkyunkwan University School of Medicine, Seoul, Republic of Korea. [227]Department of Clinical Research Design & Evaluation, SAIHST, Sungkyunkwan University, Seoul, Republic of Korea. [228]Biomedical Institute for Convergence at SKKU, Sungkyunkwan University School of Medicine, Suwon, Republic of Korea. [229]Department of Public Health Service, Seoul National University Bundang Hospital, Seongnam, Republic of Korea. [230]Department of Rehabilitation Medicine, Seoul National University College of Medicine, Seoul, Republic of Korea. [231]Korea Research Environment Open NETwork, Korea Institute of Science and Technology Information, Daejeon, Republic of Korea. [232]Global Science Experimental Data Hub Center, Korea Institute of Science and Technology Information, Daejeon, Republic of Korea. [233]Division of Infectious Diseases, Department of Medicine, Kangbuk Samsung Hospital, Sungkyunkwan University School of Medicine, Seoul, Republic of Korea. [234]Center for Cohort Studies, Kangbuk Samsung Hospital, Sungkyunkwan University School of Medicine, Seoul, Republic of Korea. [235]Department of Occupational and Environmental Medicine, Sungkyunkwan University School of Medicine, Seoul, Republic of Korea. [236]Department of Laboratory Medicine, Seoul National University Bundang Hospital, Seoul National University College of Medicine, Seongnam, Republic of Korea. [237]Institute of Clinical Molecular Biology, Christian-Albrechts-University, Kiel, Germany. [238]Novo Nordisk Foundation Center for Protein Research, Disease Systems Biology, Faculty of Health and Medical Sciences, University of Copenhagen, Copenhagen, Denmark. [239]Institut de Biotecnologia i de Biomedicina, Universitat Autònoma de Barcelona, Barcelona, Spain. [240]ICREA, Barcelona, Spain. [241]Research Group for Evolutionary Immunogenomics, Max Planck Institute for Evolutionary Biology, Plön, Germany. [242]Research Unit for Evolutionary Immunogenomics, Department of Biology, University of Hamburg, Hamburg, Germany. [243]Department of Gastroenterology, Hospital Universitario Ramón y Cajal, University of Alcalá, Instituto Ramón y Cajal de Investigación Sanitaria (IRYCIS), Madrid, Spain. [244]Centro de Investigación Biomédica en Red en Enfermedades Hepáticas y Digestivas (CIBEREHD), Instituto de Salud Carlos III (ISCIII), Madrid, Spain. [245]Vall d'Hebron Institut de Recerca (VHIR), Vall d'Hebron Hospital Universitari, Barcelona, Spain. [246]Charite Universitätsmedizin Berlin, Berlin, Germany. [249]Hospital Universitario Clinico San Cecilio, Granada, Spain. [250]Instituto de Investigación Ibs.Granada, Granada, Spain. [251]Emergency Department, University Hospital Regensburg, Regensburg, Germany. [252]Department for Infectious Diseases and Infection Control, University Hospital Regensburg, Regensburg, Germany. [253]Medical University of Innsbruck, Department of Medicine and Christian Doppler Laboratory on Iron and Phosphate Biology, Innsbruck, Austria. [254]Institute of Clinical Medicine, University of Oslo, Oslo, Norway. [255]Department of Microbiology, Oslo University Hospital, Oslo, Norway. [256]Hospital Clinic, University of Barcelona and IDIBAPS, Barcelona, Spain. [257]European Foundation for the Study of Chronic Liver Failure (EF-CLIF), Barcelona, Spain. [259]Cologne Excellence Cluster on Cellular Stress Responses in Aging-Associated Diseases (CECAD), University of Cologne, Cologne, Germany. [260]Center for Molecular Medicine Cologne (CMMC), University of Cologne, Cologne, Germany. [261]Genomes for Life-GCAT labGermans Trias i Pujol Research Institute (IGTP), Badalona, Spain. [262]IRCCS Humanitas Research Hospital, Milan, Italy. [263]Department of Biomedical Sciences, Humanitas University, Pieve Emanuele, Milan, Italy. [264]Institute of Transfusionsmedicine, University Hospital Schleswig-Holstein (UKSH), Kiel, Germany. [265]Klinik für Innere Medizin I, Universitätsklinikum Schleswig-Holstein, Kiel Campus, Kiel, Germany. [266]Zentrum für Humangenetik Regensburg, Regensburg, Germany. [267]University Hospital Schleswig-Holstein (UKSH), Kiel Campus, Kiel, Germany. [268]Section for Gastroenterology, Department of Transplantation Medicine, Division for Cancer Medicine, Surgery and Transplantation, Oslo University Hospital Rikshospitalet, Oslo, Norway. [269]Research Institute for Internal Medicine, Division of Surgery, Inflammatory Diseases and Transplantation, Oslo University Hospital Rikshospitalet and University of Oslo, Oslo, Norway. [270]Norwegian PSC Research Center, Department of Transplantation Medicine, Division of Surgery, Inflammatory Diseases and Transplantation, Oslo University Hospital Rikshospitalet, Oslo, Norway. [272]Division of Rheumatology, Inflammation and Immunity, Brigham and Women's Hospital and Harvard Medical School, Boston, MA, USA. [273]Division of Genetics, Department of Medicine, Brigham and Women's Hospital, Boston, MA, USA. [274]Department of Biomedical Informatics, Harvard Medical School, Boston, MA, USA. [275]Center for Data Sciences, Brigham and Women's Hospital, Boston, MA, USA. [276]Randaberg Municipality, Randaberg, Norway. [277]Department of Quality and Health Technology, Faculty of Health Sciences, University of Stavanger, Stavanger, Norway. [278]Department of Genetics and Bioinformatics (HDGB), Division of Health Data and Digitalization, Norwegian Institute of Public Health, Oslo, Norway. [279]Centre for Genetics and Genomics Versus Arthritis, Centre for Musculoskeletal Research, Manchester Academic Health Science Centre, The University of Manchester, Manchester, UK. [280]Department of Intensive Care, Hospital Universitario Ramón y Cajal, Instituto Ramón y Cajal de Investigación Sanitaria (IRYCIS), University of Alcalá, Madrid, Spain. [281]Osakidetza Basque Health Service, Donostialdea Integrated Health Organisation, Clinical Biochemistry Department, San Sebastian, Spain. [282]Research Center and Memory Clinic, Fundació ACE, Institut Català de Neurociències Aplicades, Universitat Internacional de Catalunya, Barcelona, Spain. [283]Networking Research Center on Neurodegenerative Diseases (CIBERNED), Instituto de Salud Carlos III, Madrid, Spain. [284]Department of Acute Medicine, Oslo University Hospital, Oslo, Norway. [285]Fondazione IRCCS Ca' Granda Ospedale Maggiore Policlinico, Milan, Italy. [286]European Reference Network on Hepatological Diseases (ERN RARE LIVER), San Gerardo Hospital, Monza, Italy. [287]Division of Gastroenterology, Center for Autoimmune Liver Diseases, Department of Medicine and Surgery, University of Milan Bicocca, Milan, Italy. [288]German Center for Neurodegenerative Diseases (DZNE Bonn), Bonn, Germany. [289]Division of Neurogenetics and Molecular Psychiatry, Department of Psychiatry and Psychotherapy, Medical Faculty, University of Cologne, Cologne, Germany. [290]Department of Psychiatry, Glenn Biggs Institute for Alzheimer's and Neurodegenerative Diseases, San Antonio, TX, USA. [291]Department of Neurodegenerative Diseases and Geriatric Psychiatry, University Hospital Bonn, Bonn, Germany. [292]Liver Unit, Department of Internal Medicine, Hospital Universitari Vall d'Hebron, Vall d'Hebron Barcelona Hospital Campus, Barcelona, Spain. [293]Department of Anesthesiology and Intensive Care, University Hospital of North Norway, Tromsø, Norway. [294]Klinik für Innere Medizin I, Universitätsklinikum Schleswig-Holstein, Kiel Campus, Kiel, Germany. [295]Gastroenterology Unit, Fondazione IRCCS Casa Sollievo della Sofferenza, San Giovanni Rotondo, Italy. [296]Department of Infectious Diseases, Oslo University Hospital, Oslo, Norway. [297]Microbiology Department, Hospital Universitari Vall d'Hebron, Barcelona, Spain. [298]Universitat Autònoma de Barcelona, Bellaterra, Spain. [299]Department of Respiratory Diseases, Hospital Universitario Ramón y Cajal, Instituto Ramón y Cajal de Investigación Sanitaria (IRYCIS), Madrid, Spain. [300]Department of Respiratory Medicine and Allergology, University Hospital, Goethe University, Frankfurt am Main, Germany. [301]Department of Infectious Diseases, Hospital Universitario Ramón y Cajal, Instituto Ramón y Cajal de Investigación Sanitaria (IRYCIS), University of Alcalá, Madrid, Spain. [302]Department of Internal Medicine II, Technical University of Munich, School of Medicine, University Hospital rechts der Isar, Munich, Germany. [303]Division of Clinical Infectious Diseases, Research Center Borstel, Borstel, Germany. [304]German Center for Infection Research (DZIF) Clinical Tuberculosis Unit, Borstel, Germany. [305]Respiratory Medicine & International Health, University of Lübeck, Lübeck, Germany. [306]Osakidetza Basque Health Service, Basurto University Hospital, Respiratory Service, Bilbao, Spain. [307]Department of Clinical and Molecular Medicine, Faculty of Medicine and Health Science, Norwegian University of Science and Technology, Trondheim, Norway. [308]Clinic Ålesund Hospital, Department of Medicine, Møre & Romsdal Hospital Trust, Ålesund, Norway. [309]Department of Anesthesiology, Hospital Universitario Ramón y Cajal, Instituto Ramón y Cajal de Investigación Sanitaria (IRYCIS), University of Alcalá, Madrid, Spain. [310]Spain Hospital Clinic, University of Barcelona and IDIBAPS, Barcelona, Spain. [311]Osakidetza Basque Health Service, Galdakao Hospital, Respiratory Service, Galdakao, Spain. [312]IBMDR - E.OOspedali Galliera, Genova, Italy. [313]Liver ICU, Hospital Clinic Barcelona, Barcelona, Spain. [314]Biocruces Bizkaia Health Research Institute, Barakaldo, Spain. [315]Histocompatibilidad y Biologia Molecular, Centro de Transfusion de Madrid, Madrid, Spain. [316]University of Milan, Milan, Italy. [317]Fondazione Grigioni per il Morbo di Parkinson, Milan, Italy. [318]Department of Anesthesiology, Intensive Care Medicine and Pain Therapy, University Hospital Frankfurt, Frankfurt am Main, Germany. [319]German Center for Infection Research (DZIF), Medical Faculty and University Hospital Cologne, University of Cologne, Partner Site Bonn-Cologne, Cologne, Germany. [321]Ikerbasque, Basque Foundation for Science, Bilbao, Spain. [322]Department of Liver and Gastrointestinal Diseases, Biodonostia Health Research Institute, Donostia University Hospital, University of the Basque Country (UPV/EHU), San Sebastian, Spain. [323]Infectious Diseases Service, Osakidetza, Biocruces Bizkaia Health Research Institute, Barakaldo, Spain. [324]Medical Department, Drammen Hospital, Vestre Viken Hospital Trust, Drammen, Norway. [325]Research Center Borstel, BioMaterialBank Nord, Borstel, Germany. [326]German Center for Lung Research (DZL), Airway Research Center North (ARCN), Giessen, Germany. [327]Popgen 2.0 Network (P2N), Kiel, Germany. [329]Department of Liver and Gastrointestinal Diseases, Biodonostia Health Research Institute, Donostia University Hospital, University of the Basque Country (UPV/EHU), CIBERehd, San Sebastian, Spain. [330]Department of Infectious Diseases, Oslo University Hospital, Oslo, Norway. [331]Department of Clinical Science, University of Bergen, Bergen, Norway. [332]Biodonostia Health Research Institute, Donostia University Hospital, San Sebastian, Spain. [333]Germans Trias i Pujol Research Institute (IGTP), Badalona, Spain. [335]ISGlobal, Barcelona, Spain. [336]CIBER Epidemiología y Salud Pública (CIBERESP), Madrid, Spain. [337]Universitat Pompeu Fabra (UPF), Barcelona, Spain. [338]Hospital del Mar Medical Research Institute (IMIM), Barcelona, Spain. [339]Osakidetza Basque Health Service, Donostialdea Integrated Health Organization, San Sebastian, Spain. [340]Department of Internal Medicine, Infectious Diseases, University H.ospital Frankfurt and Goethe University Frankfurt, Frankfurt am Main, Germany. [341]Humanitas Gavazzeni-Castelli, Bergamo, Italy. [344]School of Biological Sciences, Monash University, Clayton, Victoria, Australia. [345]Munich Clinic Schwabing, Academic Teaching Hospital, Ludwig-Maximilians-University (LMU), Munich, Germany. [346]Department of Anesthesiology, Hospital Universitario Ramón y Cajal, Instituto Ramón y Cajal de Investigación Sanitaria (IRYCIS), Madrid, Spain. [349]Clinical Trials Centre Cologne, ZKS Köln, Cologne, Germany. [351]Institute of Human Genetics, University of Bonn School of Medicine, University Hospital Bonn, Bonn, Germany. [352]Institute of Clinical Molecular Biology, Christian-Albrechts-University of Kiel, Kiel, Germany. [353]UKSH Schleswig-Holstein, Kiel, Germany. [354]Institute of Immunology, Christian-Albrechts-University of Kiel, Kiel, Germany. [355]Institute of Medical Virology, University Hospital Frankfurt, Goethe University, Frankfurt am Main, Germany. [356]German Centre for Infection Research (DZIF), External Partner Site Frankfurt, Frankfurt am Main, Germany. [357]Department of Neurology, Bezirksklinikum Regensburg, University of Regensburg, Regensburg, Germany. [359]Charite Universitätsmedizin Berlin, Berlin Institute of Health, Berlin, Germany. [360]German Center for Infection Research (DZIF), Partner Site Munich, Munich, Germany. [361]Department of Infectious Diseases,

University Hospital of North Norway, Tromsø, Norway. [362]Faculty of Health Sciences, UIT The Arctic University of Norway, Tromsø, Norway. [363]Catalan Institute of Oncology (ICO), Barcelona, Spain. [364]Bellvitge Biomedical Research Institute (IDIBELL), Barcelona, Spain. [365]Universitat de Barcelona (UB), Barcelona, Spain. [366]deCODE genetics, Reykjavik, Iceland. [368]Mucosal Immunology Lab, Unidad de Excelencia Instituto de Biomedicina y Genética Molecular de Valladolid (IBGM), Universidad de Valladolid-CSIC, Valladolid, Spain. [369]Centro de Investigaciones Biomédicas en Red de Enfermedades Hepáticas y Digestivas (CIBERehd), Madrid, Spain. [370]Valladolid University Hospital, Valladolid, Spain. [371]Estonian Genome Centre, Institute of Genomics, University of Tartu, Tartu, Estonia. [372]SYNLAB Estonia, University of Tartu, Tartu, Estonia. [373]University of Tartu, Tartu, Estonia. [374]Kuressaare Hospital, Kuressaare, Estonia. [375]Institute of Biomedicine and Translational Medicine, University of Tartu, Tartu, Estonia. [377]West Tallinn Central Hospital, Tallinn, Estonia. [378]University of Tartu, Tartu University Hospital, Tartu, Estonia. [379]Estonian Health Insurance Fund, Tallinn, Estonia. [380]Tartu University Hospital, Tartu, Estonia. [381]FinnGen, Helsinki, Finland. [382]Finnish Institute for Health and Welfare (THL), Helsinki, Finland. [383]University of Helsinki, Faculty of Medicine, Clinical and Molecular Metabolism Research Program, Helsinki, Finland. [384]Institute of Molecular and Clinical Ophthalmology Basel (IOB), Basel, Switzerland. [385]Department of Ophthalmology, University of Basel, Basel, Switzerland. [386]Infectious Diseases Service, Department of Medicine, University Hospital and University of Lausanne, Lausanne, Switzerland. [388]Centre for Primary Care and Public Health, University of Lausanne, Lausanne, Switzerland. [389]Division of Infectious Diseases and Hospital Epidemiology, Cantonal Hospital St Gallen, St Gallen, Switzerland. [390]Division of Intensive Care, Geneva University Hospitals and the University of Geneva Faculty of Medicine, Geneva, Switzerland. [391]Infectious Disease Service, Department of Internal Medicine, Geneva University Hospital, Geneva, Switzerland. [392]Clinique de Médecine et spécialités, Infectiologie, HFR-Fribourg, Fribourg, Switzerland. [393]Infectious Diseases Division, University Hospital Centre of the Canton of Vaud, Hospital of Valais, Sion, Switzerland. [394]Functional Host Genomics of Infectious Diseases, University Hospital and University of Lausanne, Lausanne, Switzerland. [395]Registry COVID, University Hospital and University of Lausanne, Lausanne, Switzerland. [396]Pneumonia Prediction using Lung Ultrasound, University Hospital and University of Lausanne, Lausanne, Switzerland. [397]Center for Primary Care and Public Health (Unisanté), University of Lausanne, Lausanne, Switzerland. [398]COVID-19 Risk Prediction in Swiss ICUs-Trial, Division of Infectious Diseases and Hospital Epidemiology, Cantonal Hospital St Gallen, St Gallen, Switzerland. [399]GCAT-Genomes for Life, Germans Trias i Pujol Health Sciences Research Institute (IGTP), Badalona, Spain. [400]Catalan Institute of Oncology, Bellvitge Biomedical Research Institute, Consortium for Biomedical Research in Epidemiology and Public Health, University of Barcelona, Barcelona, Spain. [401]Barcelona Supercomputing Center, Centro Nacional de Supercomputación (BSC-CNS), Life & Medical Sciences, Barcelona, Spain. [404]University of Siena, DIISM-SAILAB, Siena, Italy. [405]Université Côte d'Azur, Inria, CNRS, I3S, Maasai, Nice, France. [408]Division of Infectious Diseases and Immunology, Department of Medical Sciences and Infectious Diseases, Fondazione IRCCS Policlinico San Matteo, Pavia, Italy. [409]Department of Internal Medicine and Therapeutics, University of Pavia, Pavia, Italy. [410]Department of Infectious and Tropical Diseases, University of Brescia and ASST Spedali Civili Hospital, Brescia, Italy. [411]Chirurgia Vascolare, Ospedale Maggiore di Crema, Crema, Italy. [412]III Infectious Diseases Unit, ASST-FBF-Sacco, Milan, Italy. [413]Department of Biomedical and Clinical Sciences Luigi Sacco, University of Milan, Milan, Italy. [414]Department of Specialized and Internal Medicine, Tropical and Infectious Diseases Unit, Azienda Ospedaliera Universitaria Senese, Siena, Italy. [415]Unit of Respiratory Diseases and Lung Transplantation, Department of Internal and Specialist Medicine, University of Siena, Siena, Italy. [416]Department of Emergency and Urgency, Medicine, Surgery and Neurosciences, Unit of Intensive Care Medicine, Siena University Hospital, Siena, Italy. [417]Department of Medical, Surgical and Neurosciences and Radiological Sciences, Unit of Diagnostic Imaging, University of Siena, Siena, Italy. [418]Rheumatology Unit, Department of Medicine, Surgery and Neurosciences, University of Siena, Policlinico Le Scotte, Siena, Italy. [419]Department of Specialized and Internal Medicine, Infectious Diseases Unit, San Donato Hospital Arezzo, Arezzo, Italy. [420]Department of Emergency, Anesthesia Unit, San Donato Hospital, Arezzo, Italy. [421]Department of Specialized and Internal Medicine, Pneumology Unit and UTIP, San Donato Hospital, Arezzo, Italy. [422]Department of Emergency, Anesthesia Unit, Misericordia Hospital, Grosseto, Italy. [423]Department of Specialized and Internal Medicine, Infectious Diseases Unit, Misericordia Hospital, Grosseto, Italy. [424]Department of Preventive Medicine, Azienda USL Toscana Sud Est, Arezzo, Italy. [425]Clinical Chemical Analysis Laboratory, Misericordia Hospital, Grosseto, Italy. [426]Territorial Scientific Technician Department, Azienda USL Toscana Sud Est, Arezzo, Italy. [427]Clinical Chemical Analysis Laboratory, San Donato Hospital, Arezzo, Italy. [428]Department of Health Sciences, Clinic of Infectious Diseases, ASST Santi Paolo e Carlo, University of Milan, Milan, Italy. [429]Department of Anesthesia and Intensive Care, University of Modena and Reggio Emilia, Modena, Italy. [430]HIV/AIDS Department, National Institute for Infectious Diseases, IRCCS, Lazzaro Spallanzani, Rome, Italy. [431]Infectious Diseases Clinic, Department of Medicine, Azienda Ospedaliera di Perugia, Perugia, Italy. [432]Infectious Diseases Clinic, Santa Maria Hospital, University of Perugia, Perugia, Italy. [433]Department of Infectious Diseases, Treviso Hospital, Treviso, Italy. [434]Clinical Infectious Diseases, Mestre Hospital, Venezia, Italy. [435]Infectious Diseases Clinic, ULSS1, Belluno, Italy. [436]Medical Genetics and Laboratory of Medical Genetics Unit, A.O.R.N "Antonio Cardarelli", Naples, Italy. [437]Department of Molecular Medicine and Medical Biotechnology, University of Naples Federico II, Naples, Italy. [438]CEINGE Biotecnologie Avanzate, Naples, Italy. [439]IRCCS SDN, Naples, Italy. [440]Unit of Respiratory Physiopathology, AORN dei Colli, Monaldi Hospital, Naples, Italy. [441]Division of Medical Genetics, Fondazione IRCCS Casa Sollievo della Sofferenza Hospital, San Giovanni Rotondo, Italy. [442]Department of Medical Sciences, Fondazione IRCCS Casa Sollievo della Sofferenza Hospital, San Giovanni Rotondo, Italy. [443]Infectious Diseases Clinic, Policlinico San Martino Hospital, IRCCS for Cancer Research, Genova, Italy. [444]Microbiology, Fondazione Policlinico Universitario Agostino Gemelli IRCCS, Catholic University of Medicine, Rome, Italy. [445]Department of Laboratory Sciences and Infectious Diseases, Fondazione Policlinico Universitario AGemelli IRCCS, Rome, Italy. [446]Department of Cardiovascular Diseases, University of Siena, Siena, Italy. [447]Otolaryngology Unit, University of Siena, Siena, Italy. [448]Department of Internal Medicine, ASST Valtellina e Alto Lario, Sondrio, Italy. [449]First Aid Department, Luigi Curto Hospital, Polla, Italy. [450]U.O.C. Laboratorio di Genetica Umana, Genova, Italy. [451]Infectious Diseases Clinics, University of Modena and Reggio Emilia, Modena, Italy. [452]Department of Respiratory Diseases, Azienda Ospedaliera di Cremona, Cremona, Italy. [453]U.O.C. Medicina, ASST Nord Milano, Ospedale Bassini, Milan, Italy. [454]Department of Cardiovascular, Neural and Metabolic Sciences, Istituto Auxologico Italiano, IRCCS, San Luca Hospital, Milan, Italy. [455]Department of Medicine and Surgery, University of Milano-Bicocca, Milan, Italy. [456]Center for Cardiac Arrhythmias of Genetic Origin, Istituto Auxologico Italiano, IRCCS, Milan, Italy. [457]Laboratory of Cardiovascular Genetics, Istituto Auxologico Italiano, IRCCS, Milan, Italy. [458]Unit of Infectious Diseases, ASST Papa Giovanni XXIII Hospital, Bergamo, Italy. [459]Department of Cardiology, Institute of Montescano, Istituti Clinici Scientifici Maugeri, IRCCS, Pavia, Italy. [460]Department of Cardiac Rehabilitation, Institute of Tradate (VA), Istituti Clinici Scientifici Maugeri, IRCCS, Pavia, Italy. [461]Cardiac Rehabilitation Unit, Fondazione Salvatore Maugeri, IRCCS, Scientific Institute of Milan, Milan, Italy. [462]IRCCS CMondino Foundation, Pavia, Italy. [463]Medical Genetics Unit, Meyer Children's University Hospital, Florence, Italy. [464]Department of Medicine, Pneumology Unit, Misericordia Hospital, Grosseto, Italy. [465]Department of Preventive Medicine, Azienda USL Toscana Sud Est, Arezzo, Italy. [466]Department of Anesthesia and Intensive Care Unit, ASST Fatebenefratelli Sacco, Luigi Sacco Hospital, Polo Universitario, University of Milan, Milan, Italy. [467]Health Management, Azienda USL Toscana Sudest, Arezzo, Italy. [468]Department of Mathematics, University of Pavia, Pavia, Italy. [469]Independent researcher, Milan, Italy. [470]Scuola Normale Superiore, Pisa, Italy. [471]CNR-Consiglio Nazionale delle Ricerche, Istituto di Biologia e Biotecnologia Agraria (IBBA), Milano, Italy. [473]Veos Digital, Milan, Italy. [475]Core Research Laboratory, ISPRO, Florence, Italy. [478]Division of Infectious Diseases and Immunology, Fondazione IRCCS Policlinico San Matteo, Pavia, Italy. [479]Department of Molecular and Translational Medicine, University of Brescia, Brescia, Italy. [480]Clinical Chemistry Laboratory, Cytogenetics and Molecular Genetics Section, Diagnostic Department, ASST Spedali Civili di Brescia, Brescia, Italy. [481]Department of Medical and Surgical Sciences for Children and Adults, University of Modena and Reggio Emilia, Modena, Italy. [482]Department of Molecular Medicine, University of Padova, Padua, Italy. [483]Laboratory of Regulatory and Functional Genomics, Fondazione IRCCS Casa Sollievo della Sofferenza, San Giovanni Rotondo, Italy. [484]Clinical Trial Office, Fondazione IRCCS Casa Sollievo della Sofferenza Hospital, San Giovanni Rotondo, Italy. [485]Department of Health Sciences, University of Genova, Genova, Italy. [486]Oncologia Medica e Ufficio Flussi Sondrio, Sondrio, Italy. [487]Local Health Unit, Pharmaceutical Department of Grosseto, Toscana Sud Est Local Health Unit, Grosseto, Italy. [488]Independent researcher, Milan, Italy. [489]Direzione Scientifica, Istituti Clinici Scientifici Maugeri IRCCS, Pavia, Italy. [490]Fondazione per la ricerca Ospedale di Bergamo, Bergamo, Italy. [491]Allelica, New York, NY, USA. [493]School of Basic and Medical Biosciences, Faculty of Life Sciences and Medicine, King's College London, London, UK. [494]Medical and Population Genomics, Wellcome Sanger Institute, Hinxton, UK. [495]Bradford Institute for Health Research, Bradford Teaching Hospitals National Health Service (NHS) Foundation Trust, Bradford, UK. [497]Institute of Population Health Sciences, Queen Mary University of London, London, UK. [498]Genes & Health, Blizard Institute, Queen Mary University of London, London, UK. [499]Institute of Population Health Sciences, Queen Mary University of London, London, UK. [500]Department of Biostatistics, University of Michigan, Ann Arbor, MI, USA. [501]Heart Institute (InCor), University of Sao Paulo Med School, São Paulo, Brazil. [502]Genentech, San Francisco, CA, USA. [503]DNA Link Inc., Seoul, Republic of Korea. [504]Seoul National University Hospital Gangnam Center, Seoul, Republic of Korea. [505]Division of Infectious Diseases, Department of Internal Medicine, Chungnam National University School of Medicine, Daejeon, Republic of Korea. [506]East Kent Hospitals NHS Foundation Trust, Canterbury, UK. [507]Department of Internal Medicine, School of Medicine, Kyungpook National University, Daegu, Republic of Korea. [508]Division of Infectious Diseases, Department of Internal Medicine, Incheon Medical Center, Incheon, Republic of Korea. [509]Department of Infectious Diseases, Keimyung University Dongsan Hospital, Keimyung University School of Medicine, Daegu, Republic of Korea. [510]Department of Internal Medicine, Pusan National University School of Medicine and Medical Research Institute, Pusan National University Hospital, Busan, Republic of Korea. [511]Division of Infectious Diseases, Department of Internal Medicine, Myongji Hospital, Goyang, Republic of Korea. [512]Institute for Health Promotion, Graduate School of Public Health, Yonsei University, Seoul, Republic of Korea. [513]Division of Cardiovascular Medicine, Stanford University, Stanford, CA, USA. [514]Department of Medicine, Stanford University, Stanford, CA, USA. [515]Department of Genetics, Stanford University, Stanford, CA, USA. [516]Department of Biomedical Data Science, Stanford University, Stanford, CA, USA. [518]Department of Pathology, Stanford University, Stanford, CA, USA. [519]Illumina, San Diego, CA, USA. [520]Computational Biology, Drug Discovery Sciences, Takeda Pharmaceuticals, Boston, MA, USA. [521]Department of Computational Biology, Swiss Institute of Bioinformatics (SIB), University of Lausanne, Lausanne, Switzerland. [523]Royal Victoria Hospital, Belfast, UK. [524]Chelsea & Westminster NHS Foundation Trust, London, UK. [525]Northampton General Hospital NHS Trust, Northampton, UK. [526]Wrexham Maelor Hospital, Wrexham, UK. [527]University College Dublin, St Vincent's University Hospital, Dublin, Ireland. [528]University Hospitals Coventry & Warwickshire NHS Trust, Coventry, UK. [529]Watford General Hospital, Watford, UK. [530]NIHR Health Protection Research Unit, Institute of Infection, Veterinary and Ecological Sciences, Faculty of Health and Life Sciences, University of Liverpool, Liverpool, UK. [531]Queen Alexandra Hospital (Hampshire), Portsmouth Hospital Trust, Portsmouth, UK. [532]Princess Royal Hospital, Brighton & Sussex Universities Hospitals NHS Trust, Brighton, UK. [533]Bassetlaw Hospital, Doncaster and Bassetlaw, Worksop, UK. [534]Darent Valley Hospital, Dartford & Gravesham NHS Trust, Dartford, UK. [535]High Containment Laboratories, University of Birmingham, Birmingham, UK. [536]Queen Elizabeth the Queen Mother Hospital, Margate, UK. [537]John Radcliffe Hospital, Oxford University Hospitals NHS Foundation Trust, Oxford, UK. [538]Royal Albert Edward Infirmary (Wigan), Wrightington, Wigan and Leigh, Wigan, UK. [539]Manchester Royal Infirmary, Manchester University Hospitals NHS Foundation Trust, Manchester, UK. [540]Furness General Hospital, Morecambe Bay NHS Foundation Trust, Barrow-in-Furness, UK. [541]Castle Hill Hospital, Hull University Teaching Hospital Trust, Hull, UK. [542]Hillingdon Hospital, Hillingdon Hospital, London, UK. [543]St Thomas Hospital, Guys and St Thomas Foundation Trust, London, UK. [544]University Hospitals Coventry and Warwickshire, Coventry, UK. [545]St Michaels Hospital (Bristol), University Hospitals Bristol and Weston NHS Foundation Trust, Bristol, UK. [546]Stepping Hill Hospital, Stockport NHS Foundation Trust, Manchester, UK. [547]Royal Liverpool Hospital, Liverpool University Hospitals NHS Foundation Trust, Liverpool, UK. [548]Bristol Royal Hospital (Children's), University Hospitals Bristol and Weston NHS Foundation Trust, Bristol, UK. [549]Scarborough Hospital, York Teaching Hospitals NHS Foundation Trust, York, UK. [550]Liverpool Heart & Chest Hospital, Liverpool Heart & Chest NHS Foundation Trust, Liverpool, UK. [551]James Paget University Hospital, James Paget University Hospitals NHS Foundation Trust, Great Yarmouth, UK. [552]The

James Cook University Hospital, South Tees NHS Foundation Trust, Middlesbrough, UK. [553]Aberdeen Royal Infirmary, Grampian, Aberdeen, UK. [554]University of Edinburgh, Edinburgh, UK. [555]Royal Devon and Exeter Hospital, Royal Devon and Exeter NHS Foudation Trust, Exeter, UK. [556]Worcestershire Royal Hospital, Worcestershire Acute Hospitals NHS Trust, Worcester, UK. [557]Conquest Hospital, Hastings, East Sussex Healthcare NHS Trust, Seaford, UK. [558]Dorset County Hospital, Dorset County Hospital NHS Foundation Trust, Dorchester, UK. [559]Royal Bournemouth General Hospital, University Hospitals Dorset NHS Foundation Trust, Bournemouth, UK. [560]Harrogate Hospital, Harrogate and District NHS Foundation Trust, Harrogate, UK. [561]Burnley General Teaching Hospital, East Lancashire Hospitals NHS Hospitals, Burnley, UK. [562]Torbay Hospital, Torbay & South Devon NHS Foundation Trust, Torquay, UK. [563]Royal Hallamshire Hospital, Sheffield Teaching Hospitals NHS Foundation Trust, Sheffield, UK. [564]St Georges Hospital (Tooting), St Georges University Hospitals NHS Foundation Trust, London, UK. [565]Blackpool Victoria Hospital, Blackpool Teaching Hospitals NHS Foundation Trust, Blackpool, UK. [566]The Royal London Hospital, Barts Health NHS Trust, London, UK. [567]Salford Royal NHS Foundation Trust, Salford Royal NHS Foundation Trust, Manchester, UK. [568]University Hospital of North Durham, County Durham and Darlington Foundation Trust, Durham, UK. [569]Norfolk and Norwich University Hospital, Norfolk and Norwich University Hospital NHS Foundation Trust, Norwich, UK. [572]Fairfield General Hospital, Pennine Acute Hospitals NHS Trust, Manchester, UK. [573]Hereford County Hospital, Wye Valley NHS Trust, Hereford, UK. [574]Southampton General Hospital, University Hospital Southampton NHS Foundation Trust, Southampton, UK. [575]Northampton General Hospital, Northampton General Hospital NHS Trust, Northampton, UK. [576]University Hospital of Wales, Cardiff and Vale University Health Board, Cardiff, UK. [577]University of Bristol, Bristol, UK. [578]Leighton Hospital, Mid Cheshire Hospitals NHS Foundation Trust, Crewe, UK. [579]Diana Princess of Wales Hospital (Grimsby), North Lincolnshire & Goole, Grimsby, UK. [580]Manor Hospital, Walsall Healthcare NHS Trust, Walsall, UK. [581]Addenbrookes Hospital, Cambridge University Hospital NHS Foundation Trust, Cambridge, UK. [582]West Suffolk Hospital, West Suffolk Hospital NHS Foundation Trust, Bury St Edmunds, UK. [583]Basingstoke and North Hampshire Hospital, Hampshire Hospitals NHS Foundation Trust, Basingstoke, UK. [584]North Cumbria Integrated Care NHS Foundation Trust, Carlisle, UK. [585]Warwick Hospital, South Warwickshire NHS Foundation Trust, Warwick, UK. [586]Birmingham Women's and Children's Hospital, Birmingham Women's and Children's Hospital NHS Foundation Trust, Birmingham, UK. [587]Nottingham City Hospital, Nottingham University Hospitals NHS Trust, Nottingham, UK. [588]Glangwili Hospital Child Health Section, Hywel Dda University Health Board, Carmarthen, UK. [589]Alder Hey Children's Hospital, Alder Hey Children's NHS Foundation Trust, Liverpool, UK. [590]Bronglais General Hospital, Hywel Dda University Health Board, Aberystwyth, UK. [591]Worthing Hospital, Western Sussex Hospitals NHS Foundation Trust, Worthing, UK. [592]Rotheram District General Hospital, The Rotheram NHS Foundation Trust, Rotherham, UK. [593]Royal Free Hospital, Royal Free London NHS Foundation Trust, London, UK. [594]Homerton Hospital, Homerton University Hospital NHS Foundation Trust, London, UK. [595]Airedale Hospital, Airedale NHS Foundation Trust, Keighley, UK. [596]Basildon Hospital, Basildon and Thurrock University Hospitals NHS Foundation Trust, Basildon, UK. [597]The Christie NHS Foundation Trust, Manchester, UK. [598]Queen Elizabeth Hospital (Greenwich), Lewisham and Greenwich NHS Trust, London, UK. [599]The Whittington Hospital, Whittington Health NHS Trust, London, UK. [600]Sheffield Children's Hospital, Sheffield Children's NHS Foundation Trust, Sheffield, UK. [601]Royal United Hospital, Bath, Royal United Hospitals Bath NHS Foundation Trust, Bath, UK. [602]Western General Hospital, Edinburgh, UK. [603]Mid and South Essex NHS Foundation Trust, Basildon, UK. [604]Hinchingbrooke Hospital, North West Anglia NHS Foundation Trust, Peterborough, UK. [605]Royal Preston Hospital, Lancashire Teaching Hospitals NHS Foundation Trust, Preston, UK. [606]University Hospital (Coventry), University Hospitals Coventry and Warwickshire, Coventry, UK. [607]The Walton Centre, The Walton Centre, Liverpool, UK. [608]Hull Royal Infirmary, Hull University Teaching Hospital Trust, Hull, UK. [609]Darlington Memorial Hospital, County Durham and Darlington Foundation Trust, Darlington, UK. [610]Queen Elizabeth Hospital (Gateshead), Gateshead NHS Foundation Trust, Newcastle, UK. [611]Warrington Hospital, Warrington & Halton Hospitals NHS Foundation Trust, Warrington, UK. [612]University Hospitals Bristol and Weston NHS Foundation Trust, Bristol, UK. [613]St Mary's Hospital (Isle of Wight), Isle of Wight NHS Trust, Isle of Wight, UK. [614]The Maidstone Hospital, Maidstone & Tunbridge Wells NHS Trust, Maidstone, UK. [615]Huddersfield Royal, Calderdale and Huddersfield NHS Foundation Trust, Huddersfield, UK. [616]Royal Surrey County Hospital, Guildford, UK. [617]Countess of Chester Hospital, Countess of Chester Hospital NHS Foundation Trust, Chester, UK. [618]Frimley Park Hospital, Frimley Health Foundation Trust, Frimley, UK. [620]Leeds General Infirmary, Leeds Teaching Hospitals, Leeds, UK. [621]North Middlesex Hospital, North Middlesex University Hospital NHS Trust, London, UK. [622]Arrowe Park Hospital, Wirral University Teaching Hospital NHS Foundation Trust, Wirral, UK. [623]Great Ormond Street Hospital, Great Ormond Street Hospital for Children NHS Foundation Trust, London, UK. [624]Royal Shrewsbury Hospital, Shrewsbury and Telford Hospital NHS Trust, Shrewsbury, UK. [625]East Surrey Hospital (Redhill), Surrey & Sussex Healthcare, Redhill, UK. [626]Burton Hospital, University Hospitals of Derby & Burton NHS Foundation Trust, Burton-on-Trent, UK. [627]Kent and Canterbury Hospital, East Kent Hospitals NHS Foundation Trust, Canterbury, UK. [628]Weston Area General Trust, University Hospitals Bristol and Weston NHS Foundation Trust, Bristol, UK. [629]Luton and Dunstable University Hospital, Luton, UK. [630]Glasgow Royal Infirmary, Greater Glasgow and Clyde, Glasgow, UK. [631]Derbyshire Healthcare, Derbyshire Healthcare NHS Foundation Trust, Derby, UK. [632]Macclesfield General Hospital, East Cheshire NHS Foundation Trust, Macclesfield, UK. [633]Chelsea and Westminster Hospital, Chelsea and Westminster NHS Trust, London, UK. [634]Institute of Microbiology and Infection, University of Birmingham, Birmingham, UK. [635]Prince Philip Hospital, Hywel Dda University Health Board, Llanelli, UK. [636]George Eliot Hospital - Acute Services, George Eliot Hospital, Nuneaton, UK. [637]Kettering General Hospital, Kettering General Hospital NHS Foundation Trust, Kettering, UK. [638]Birmingham Heartlands Hospital, Birmingham, UK. [639]Russells Hall Hospital, The Dudley Group NHS Foundation Trust, Dudley, UK. [640]Harefield Hospital, Royal Brompton & Harefield Trust, London, UK. [641]Lister Hospital, East and North Hertfordshire NHS Trust, Stevenage, UK. [642]Musgrove Park Hospital (Taunton & Somerset), Somerset NHS Foundation Trust, Taunton, UK. [643]Queen's Hospital, Havering (Romford), Barking, Havering and Redbridge University Hospitals NHS Trust, London, UK. [644]Southport & Formby District General Hospital, Southport and Ormskirk Hospital NHS Trust, Southport, UK. [645]New Cross Hospital, The Royal Wolverhampton NHS Trust, Wolverhampton, UK. [646]King's College Hospital, London, UK. [647]The Royal Victoria Infirmary, Newcastle Hospitals NHS Trust, Newcastle, UK. [648]The Great Western Hospital, Great Western Hospitals NHS Foundation Trust, Swindon, UK. [649]Ninewells Hospital, Tayside, Dundee, UK. [650]Poole Hospital NHS Trust, Poole, UK. [651]Burton Hospital, University Hospitals of Derby & Burton NHS Foundation Trust, Derby, UK. [652]William Harvey Hospital, Ashford, East Kent Hospitals NHS Foundation Trust, Willesborough, UK. [653]King's Mill Hospital, Sherwood Forest Hospitals NHS Foundation Trust, Sutton-in-Ashfield, UK. [654]Liverpool Women's NHS Foundation Trust, Liverpool, UK. [655]Dewsbury Hospital, Mid Yorkshire Hospitals NHS Trust, Dewsbury, UK. [656]Northern Devon District Hospital, Northern Devon Healthcare NHS Trust, Barnstaple, UK. [657]Tameside General Hospital, Tameside and Glossop Integrated Care NHS Foundation Trust, Manchester, UK. [658]Sandwell General Hospital, Sandwell and West Birmingham Hospitals NHS Trust, Birmingham, UK. [659]Broomfield Hospital, Mid and South Essex University Hospitals Group, Broomfield, UK. [660]Wycombe Hospital, Buckingham Healthcare NHS Trust, Wycombe, UK. [661]University Hospital of North Tees, North Tees and Hartlepool NHS Trust, Stockton-on-Tees, UK. [662]Royal Manchester Children's Hospital, Manchester University Hospitals NHS Foundation Trust, Manchester, UK. [663]Bedford Hospital, Bedford, UK. [664]Colchester General Hospital, East Suffolk and North Essex Foundation Trust, Colchester, UK. [665]Queen Elizabeth Hospital (Birmingham) and Heartlands, University Hospital Birmingham NHS Foundation Trust, Birmingham, UK. [666]Chesterfield Royal Hospital, Chesterfield Royal Hospital NHS Foundation Trust, Chesterfield, UK. [667]Princess Alexandra Hospital, The Princess Alexandra Hospital NHS Trust, Harlow, UK. [668]Watford General Hospital, West Hertfordshire Hospitals NHS Trust, Watford, UK. [669]Milton Keynes Hospital, Milton Keynes University Hospital NHS Foundation Trust, Milton Keynes, UK. [670]Royal Bolton General Hospital, Bolton Foundation Trust, Bolton, UK. [671]Royal Gwent (Newport), Aneurin Bevan University Health Board, Newport, UK. [672]The Royal Marsden Hospital (London), The Royal Marsden NHS Foundation Trust, London, UK. [673]Queen Victoria Hospital (East Grinstead), Queen Victoria Hospital NHS Foundation Trust, East Grinstead, UK. [674]County Hospital (Stafford), University Hospitals of North Midlands NHS Trust, Stafford, UK. [675]Whiston Hospital, St Helen's & Knowlsey Hospitals NHS Trust, Prescot, UK. [676]Croydon University Hospital, London, UK. [677]Gloucester Royal, Gloucestershire Hospitals NHS Foundation Trust, Gloucester, UK. [678]Medway Maritime Hospital, Medway Maritime NHS Trust, Gillingham, UK. [679]Royal Papworth Hospital Everard, Royal Papworth Hospital NHS Foundation Trust, Cambridge, UK. [680]Derriford (Plymouth), University Hospital Plymouth NHS Trust, Plymouth, UK. [681]St Helier Hospital, Epsom and St Helier University Hospital NHS Trust, London, UK. [682]Royal Berkshire Hospital, Royal Berkshire Foundation Trust, London, UK. [683]Bradford Royal Infirmary, Bradford Teaching Hospitals NHS Foundation Trust, Bradford, UK. [684]Northwick Park, London North West University Hospital Trust, London, UK. [685]Ealing Hospital, London North West University Hospital Trust, London, UK. [686]Royal Cornwall Hospital (Tresliske), Royal Cornwall NHS Trust, Truro, UK. [687]Ashford Hospital, Ashford & St Peter's Hospital, Stanwell, UK. [688]Leicester Royal Infirmary (includes Glenfield Site), University Hospitals of Leicester, Leicester, UK. [689]Grantham and District Hospital, United Lincolnshire Hospitals NHS Trust, Grantham, UK. [690]University Hospital Aintree, Liverpool University Hospitals NHS Foundation Trust, Liverpool, UK. [691]North Tyneside General Hospital, Northumbria Healthcare NHS Trust, North Shields, UK. [692]Queen Elizabeth Hospital (King's Lynn), Queen Elizabeth Hospital King's Lynn NHS Foundation Trust, King's Lynn, UK. [693]The Crick Institute, London, UK. [695]William Harvey Research Institute, Barts and the London School of Medicine and Dentistry, Queen Mary University of London, London, UK. [696]Centre for Genomic and Experimental Medicine, Institute of Genetics and Molecular Medicine, University of Edinburgh, Western General Hospital, Edinburgh, UK. [697]Intensive Care National Audit & Research Centre, London, UK. [700]Wellcome Centre for Human Genetics, University of Oxford, Oxford, UK. [705]Centre for Inflammation Research, The Queen's Medical Research Institute, University of Edinburgh, Edinburgh, UK. [706]Great Ormond Street Hospital for Children NHS Foundation Trust, London, UK. [707]Biostatistics Group, School of Life Sciences, Sun Yat-sen University, Guangzhou, China. [708]Centre for Global Health Research, Usher Institute of Population Health Sciences and Informatics, Edinburgh, UK. [709]Department of Medical Epidemiology and Biostatistics, Karolinska Institutet, Stockholm, Sweden. [710]Institute for Molecular Bioscience, The University of Queensland, Brisbane, Queensland, Australia. [711]School of Life Sciences, Westlake University, Hangzhou, China. [712]Westlake Laboratory of Life Sciences and Biomedicine, Westlake University, Hangzhou, China. [714]Centre for Medical Informatics, The Usher Institute, University of Edinburgh, Edinburgh, UK. [715]Liverpool Clinical Trials Centre, University of Liverpool, Liverpool, UK. [716]Centre for Health Informatics, Division of Informatics, Imaging and Data Science, School of Health Sciences, Faculty of Biology, Medicine and Health, University of Manchester, Manchester Academic Health Science Centre, Manchester, UK. [717]MRC Human Genetics Unit, MRC Institute of Genetics and Molecular Medicine, University of Edinburgh, Edinburgh, UK. [718]School of Informatics, University of Edinburgh, Edinburgh, UK. [719]Royal Hospital for Children, Glasgow, UK. [721]MRC-University of Glasgow Centre for Virus Research, Institute of Infection, Immunity and Inflammation, College of Medical, Veterinary and Life Sciences, University of Glasgow, Glasgow, UK. [722]Centre for Tropical Medicine and Global Health, Nuffield Department of Medicine, University of Oxford, Oxford, UK. [724]Department of Anaesthesia and Intensive Care, The Chinese University of Hong Kong, Prince of Wales Hospital, Hong Kong, China. [725]Department of Critical Care Medicine, Queen's University and Kingston Health Sciences Centre, Kingston, Ontario, Canada. [726]Wellcome-Wolfson Institute for Experimental Medicine, Queen's University Belfast, Belfast, UK. [727]Department of Intensive Care Medicine, Royal Victoria Hospital, Belfast, UK. [728]UCL Centre for Human Health and Performance, London, UK. [729]Clinical Research Centre at St Vincent's University Hospital, University College Dublin, Dublin, Ireland. [730]National Heart and Lung Institute, Imperial College London, London, UK. [731]Imperial College Healthcare NHS Trust London, London, UK. [732]NIHR Health Protection Research Unit for Emerging and Zoonotic Infections, Institute of Infection, Veterinary and Ecological Sciences University of Liverpool, Liverpool, UK. [733]Respiratory Medicine, Alder Hey Children's Hospital, Institute in The Park, University of Liverpool, Alder Hey Children's Hospital, Liverpool, UK. [734]Department of Intensive Care Medicine, Guy's and St Thomas NHS Foundation Trust, London, UK. [735]Department of Medicine, University of Cambridge, Cambridge, UK. [736]Airedale General Hospital, Keighley, UK. [737]Barts Health NHS Trust, London, UK. [738]Basildon Hospital, Basildon, UK. [739]BHRUT (Barking Havering) - Queens Hospital and King George Hospital, Romford, UK. [740]Bradford Royal Infirmary, Bradford, UK. [741]Bronglais General Hospital, Aberystwyth, UK. [742]Broomfield Hospital, Chelmsford, UK. [743]Calderdale Royal Hospital, Halifax, UK. [744]Charing Cross Hospital, St Mary's Hospital and Hammersmith Hospital, London, UK. [745]Barnet Hospital, London, UK. [746]Birmingham Children's Hospital, Birmingham, UK. [747]St John's Hospital Livingston,

Livingston, UK. [748]Aberdeen Royal Infirmary, Aberdeen, UK. [749]Addenbrooke's Hospital, Cambridge, UK. [750]Aintree University Hospital, Liverpool, UK. [752]Arrowe Park Hospital, Wirral, UK. [753]Ashford and St Peter's Hospital, Lyne, UK. [754]Basingstoke and North Hampshire Hospital, Basingstoke, UK. [755]Borders General Hospital, Melrose, UK. [756]Chesterfield Royal Hospital Foundation Trust, Chesterfield, UK. [757]Eastbourne District General Hospital, East Sussex, UK and Conquest Hospital, Eastbourne, UK. [758]Barnsley Hospital, Barnsley, UK. [759]Blackpool Victoria Hospital, Blackpool, UK. [760]East Surrey Hospital, Redhill, UK. [761]Good Hope Hospital, Birmingham, UK. [762]Hereford County Hospital, Hereford, UK. [763]Hull Royal Infirmary, Hull, UK. [765]Kent & Canterbury Hospital, Canterbury, UK. [766]Manchester Royal Infirmary, Manchester, UK. [767]Nottingham University Hospital, Nottingham, UK. [768]Pilgrim Hospital, Lincoln, UK. [769]Queen Elizabeth Hospital, Birmingham, UK. [770]Salford Royal Hospital, Manchester, UK. [771]Tameside General Hospital, Ashton-under-Lyne, UK. [772]The Tunbridge Wells Hospital and Maidstone Hospital, Maidstone, UK. [773]The Royal Oldham Hospital, Manchester, UK. [774]The Royal Papworth Hospital, Cambridge, UK. [775]University College Hospital, London, UK. [776]Withybush General Hospital, Haverfordwest, UK. [777]Wythenshawe Hospital, Manchester, UK. [778]Yeovil Hospital, Yeovil, UK. [779]Cumberland Infirmary, Carlisle, UK. [780]Darent Valley Hospital, Dartford, UK. [781]Dumfries and Galloway Royal Infirmary, Dumfries, UK. [782]Ealing Hospital, London, UK. [783]Fairfield General Hospital, Bury, UK. [784]George Eliot Hospital NHS Trust, Nuneaton, UK. [785]Glan Clwyd Hospital, Bodelwyddan, UK. [786]Glangwili General Hospital, Camarthen, UK. [787]The Great Western Hospital, Swindon, UK. [788]Guys and St Thomas' Hospital, London, UK. [789]Harefield Hospital, London, UK. [790]Harrogate and District NHS Foundation Trust, Harrogate, UK. [792]James Paget University Hospital NHS Trust, Great Yarmouth, UK. [794]King's Mill Hospital, Nottingham, UK. [795]Kingston Hospital, Kingston, UK. [796]Lincoln County Hospital, Lincoln, UK. [797]Liverpool Heart and Chest Hospital, Liverpool, UK. [798]Macclesfield District General Hospital, Macclesfield, UK. [799]Medway Maritime Hospital, Gillingham, UK. [800]Milton Keynes University Hospital, Milton Keynes, UK. [801]Morriston Hospital, Swansea, UK. [802]National Hospital for Neurology and Neurosurgery, London, UK. [803]Norfolk and Norwich University hospital (NNUH), Norwich, UK. [804]North Middlesex University Hospital NHS Trust, London, UK. [806]Northumbria Healthcare NHS Foundation Trust, North Shields, UK. [807]Peterborough City Hospital, Peterborough, UK. [808]Prince Charles Hospital, Merthyr Tydfil, UK. [809]Royal Sussex County Hospital, Brighton, UK. [810]Princess Royal Hospital, Haywards Heath, UK. [811]Princess of Wales Hospital, Llantrisant, UK. [812]Queen Alexandra Hospital, Portsmouth, UK. [813]Queen Elizabeth Hospital, London, UK. [815]Queen Victoria Hospital, East Grinstead, UK. [816]Queen's Hospital Burton, Burton-On-Trent, UK. [817]Raigmore Hospital, Inverness, UK. [818]Rotherham General Hospital, Rotherham, UK. [819]Royal Blackburn Teaching Hospital, Blackburn, UK. [820]Royal Preston Hospital, Preston, UK. [821]Royal Surrey County Hospital, Guildford, UK. [822]Royal Albert Edward Infirmary, Wigan, UK. [823]The Royal Alexandra Children's Hospital, Brighton, UK. [824]Royal Alexandra Hospital, Paisley, UK. [825]Royal Bolton Hospital, Bolton, UK. [826]University Hospitals Dorset NHS Foundation Trust, Dorchester, UK. [827]Royal Brompton Hospital, London, UK. [828]Imperial College London, London, UK. [829]Royal Cornwall Hospital, Truro, UK. [830]Royal Free Hospital, London, UK. [831]Royal Glamorgan Hospital, Pontyclun, UK. [832]Royal Gwent Hospital, Newport, UK. [833]Northern General Hospital, Sheffield, UK. [834]Royal Hampshire County Hospital, Winchester, UK. [835]Royal Manchester Children's Hospital, Manchester, UK. [836]Royal Stoke University Hospital, Stoke-on-Trent, UK. [837]Salisbury District Hospital, Salisbury, UK. [838]Sandwell General Hospital, Birmingham, UK. [839]Scarborough General Hospital, Scarborough, UK. [840]Scunthorpe General Hospital, Scunthorpe, UK. [841]Southmead Hospital, Bristol, UK. [842]St George's Hospital, London, UK. [843]St Mary's Hospital, Newport, UK. [844]Stoke Mandeville Hospital, Aylesbury, UK. [845]Sunderland Royal Hospital, Sunderland, UK. [846]Alexandra Hospital, Redditch and Worcester Royal Hospital, Worcester, UK. [847]The Christie NHS Foundation Trust, Manchester, UK. [848]The Queen Elizabeth Hospital, King's Lynn, UK. [849]The Royal Liverpool University Hospital, Liverpool, UK. [850]The Royal Marsden NHS Foundation Trust, London, UK. [851]Torbay Hospital, Torquay, UK. [852]University Hospital Monklands, Airdrie, UK. [853]University Hospital Lewisham, London, UK. [854]University Hospital North Durham, Darlington, UK. [855]University Hospital of North Tees, Stockton-on-Tees, UK. [856]University Hospital of Wales, Cardiff, UK. [857]University Hospital Wishaw, Wishaw, UK. [858]Victoria Hospital, Kirkcaldy, UK. [859]Warrington General Hospital, Warrington, UK. [860]West Cumberland Hospital, Whitehaven, UK. [861]Western Sussex Hospitals, Chichester, UK. [862]Whiston Hospital, Prescot, UK. [863]York Hospital, York, UK. [864]Ysbyty Gwynedd, Bangor, UK. [865]Countess of Chester Hospital, Chester, UK. [866]Croydon University Hospital, Croydon, UK. [867]Diana Princess of Wales Hospital, Grimsby, UK. [868]Dorset County Hospital, Dorchester, UK. [869]Forth Valley Royal Hospital, Falkirk, UK. [870]Furness General Hospital, Barrow-in-Furness, UK. [871]Alder Hey Children's Hospital, Liverpool, UK. [872]Derriford Hospital, Plymouth, UK. [873]Glasgow Royal Infirmary, Glasgow, UK. [874]Glenfield Hospital, Leicester, UK. [875]Gloucestershire Royal Hospital, Gloucester, UK. [876]Golden Jubilee National Hospital, Clydebank, UK. [877]Great Ormond St Hospital and UCL Great Ormond St Institute of Child Health NIHR Biomedical Research Centre, London, UK. [878]Homerton University Hospital Foundation NHS Trust, London, UK. [879]James Cook University Hospital, Middlesbrough, UK. [880]John Radcliffe Hospital, Oxford, UK. [881]Leicester Royal Infirmary, Leicester, UK. [882]Lister Hospital, Stevenage, UK. [883]New Cross Hospital, Wolverhampton, UK. [884]Royal Victoria Infirmary, Newcastle Upon Tyne, UK. [885]Ninewells Hospital, Dundee, UK. [886]North Devon District Hospital, Barnstaple, UK. [887]North Manchester General Hospital, Manchester, UK. [888]Northwick Park Hospital, London, UK. [889]Prince Philip Hospital, Lianelli, UK. [890]Pinderfields General Hospital, Wakefield, UK. [891]Poole Hospital, Poole, UK. [892]Royal Shrewsbury Hospital, Shrewsbury, UK. [893]Princess Royal Hospital, Telford, UK. [899]Queen Elizabeth Hospital Gateshead, Gateshead, UK. [900]Queen Elizabeth University Hospital, Glasgow, UK. [901]Royal Berkshire NHS Foundation Trust, Reading, UK. [902]Royal Derby Hospital, Derby, UK. [903]Royal Devon and Exeter Hospital, Exeter, UK. [904]Royal Infirmary of Edinburgh, Edinburgh, UK. [905]Royal Lancaster Infirmary, Lancaster, UK. [906]Royal United Hospital, Bath, UK. [907]Russells Hall Hospital, Dudley, UK. [908]Sheffield Children's Hospital, Sheffield, UK. [909]Southampton General Hospital, Southampton, UK. [910]Southend University Hospital, Westcliff-on-Sea, UK. [911]Southport and Formby District General Hospital, Ormskirk, UK. [912]St James's University Hospital and Leeds General Infirmary, Leeds, UK. [913]Bristol Royal Infirmary, Bristol, UK. [914]Stepping Hill Hospital, Stockport, UK. [915]The Princess Alexandra Hospital, Harlow, UK. [916]University Hospital Crosshouse, Kilmarnock, UK. [917]University Hospital Hairmyres, East Kilbride, UK. [918]Craigavon Area Hospital, Craigavon, UK. [919]Warwick Hospital, Warwick, UK. [920]West Middlesex Hospital, Isleworth, UK. [922]Whittington Hospital, London, UK. [923]William Harvey Hospital, Ashford, UK. [932]Section of Molecular Virology, Imperial College London, London, UK. [933]Antimicrobial Resistance and Hospital Acquired Infection Department, Public Health England, London, UK. [934]Department of Infectious Disease, Imperial College London, London, UK. [935]National Infection Service, Public Health England, London, UK. [936]MRC-University of Glasgow Centre for Virus Research, Glasgow, UK. [937]Liverpool School of Tropical Medicine, Liverpool, UK. [938]Institute of Infection and Global Health, University of Liverpool, Liverpool, UK. [940]Virology Reference Department, National Infection Service, Public Health England, London, UK. [941]Department of Pharmacology, University of Liverpool, Liverpool, UK. [942]Nuffield Department of Medicine, University of Oxford, Oxford, UK. [944]Nottingham University Hospitals NHS Trust, Nottingham, UK. [945]Nuffield Department of Medicine, John Radcliffe Hospital, Oxford, UK. [946]ISARIC Global Support Centre, Centre for Tropical Medicine and Global Health, Nuffield Department of Medicine, University of Oxford, Oxford, UK. [947]Division of Infection and Immunity, University College London, London, UK. [948]Institute of Infection, Veterinary and Ecological Sciences, University of Liverpool, Liverpool, UK. [949]Centre for Clinical Infection and Diagnostics Research, Department of Infectious Diseases, School of Immunology and Microbial Sciences, King's College London, London, UK. [950]Institute of Evolutionary Biology, University of Edinburgh, Edinburgh, UK. [951]Department of Pediatrics and Virology, Imperial College London, London, UK. [952]The Florey Institute for Host-Pathogen Interactions, Department of Infection, Immunity and Cardiovascular Disease, University of Sheffield, Sheffield, UK. [953]Division of Structural Biology, The Wellcome Centre for Human Genetics, University of Oxford, Oxford, UK. [955]Blood Borne Virus Unit, Virus Reference Department, National Infection Service, Public Health England, London, UK. [956]Department of Infection, Immunity and Cardiovascular Disease, University of Sheffield, Sheffield, UK. [957]Institute for Global Health, University College London, London, UK. [958]Molecular and Clinical Cancer Medicine, Institute of Systems, Molecular and Integrative Biology, University of Liverpool, Liverpool, UK. [959]Department of Child Life and Health, University of Edinburgh, Edinburgh, UK. [960]Section of Biomolecular Medicine, Division of Systems Medicine, Department of Metabolism, Digestion and Reproduction, Imperial College London, London, UK. [961]Department of Epidemiology and Biostatistics, School of Public Health, Faculty of Medicine, Imperial College London, London, UK. [962]National Phenome Centre, Department of Metabolism, Digestion and Reproduction, Imperial College London, London, UK. [963]Department of Molecular and Clinical Cancer Medicine, University of Liverpool, Liverpool, UK. [964]Institute of Translational Medicine, University of Liverpool, Liverpool, UK. [965]Intensive Care Unit, Royal Infirmary Edinburgh, Edinburgh, UK. [966]University of Liverpool, Liverpool, UK. [967]University of Glasgow, Glasgow, UK. [968]Edinburgh Clinical Research Facility, Western General Hospital, University of Edinburgh, Edinburgh, UK. [970]Department of Infectious Diseases, Leiden University Medical Center, Leiden, The Netherlands. [971]Cambridge University Hospitals NHS Foundation Trust, Cambridge, UK. [974]Genotek, Moscow, Russia. [975]Helix, San Mateo, CA, USA. [976]Center for Genomic Medicine, Desert Research Institute, Reno, NV, USA. [977]Renown Health, Reno, NV, USA. [978]24Genetics, Boston, MA, USA. [979]Hospital La Paz Institute for Health Research, Madrid, Spain. [980]Division of Pulmonary Medicine, Department of Medicine, Keio University School of Medicine, Tokyo, Japan. [981]Department of Statistical Genetics, Osaka University Graduate School of Medicine, Suita, Japan. [982]Laboratory of Statistical Immunology, Immunology Frontier Research Center (WPI-IFReC), Osaka University, Suita, Japan. [983]Integrated Frontier Research for Medical Science Division, Institute for Open and Transdisciplinary Research Initiatives, Osaka University, Suita, Japan. [984]Division of Health Medical Intelligence, Human Genome Center, the Institute of Medical Science, The University of Tokyo, Tokyo, Japan. [985]Laboratory of Viral Infection I, Department of Infection Control and Immunology, Ōmura Satoshi Memorial Institute & Graduate School of Infection Control Sciences, Kitasato University, Tokyo, Japan. [986]Department of Surgery, Keio University School of Medicine, Tokyo, Japan. [987]Department of Organoid Medicine, Keio University School of Medicine, Tokyo, Japan. [988]Department of Infectious Diseases, Keio University School of Medicine, Tokyo, Japan. [989]Department of Respiratory Medicine and Clinical Immunology, Osaka University Graduate School of Medicine, Suita, Japan. [990]Department of Immunopathology, Immunology Frontier Research Center (WPI-IFReC), Osaka University, Suita, Japan. [991]Institute of Research, Tokyo Medical and Dental University, Tokyo, Japan. [992]Department of Insured Medical Care Management, Tokyo Medical and Dental University Hospital of Medicine, Tokyo, Japan. [993]Genome Medical Science Project (Toyama), National Center for Global Health and Medicine, Chiba, Japan. [994]Division of Gastroenterology and Hepatology, Department of Medicine, Keio University School of Medicine, Tokyo, Japan. [995]M&D Data Science Center, Tokyo Medical and Dental University, Tokyo, Japan. [996]Department of Pathology and Tumor Biology Institute for the Advanced Study of Human Biology (WPI-ASHBi), Kyoto University, Kyoto, Japan. [997]Department of Medicine, Center for Hematology and Regenerative Medicine, Karolinska Institute, Stockholm, Sweden. [999]Department of Emergency and Critical Care Medicine, Keio University School of Medicine, Tokyo, Japan. [1000]Department of Anesthesiology, Keio University School of Medicine, Tokyo, Japan. [1001]Department of Laboratory Medicine, Keio University School of Medicine, Tokyo, Japan. [1002]Division of Infection Control and Prevention, Osaka University Hospital, Suita, Japan. [1003]Department of Biomedical Ethics and Public Policy, Osaka University Graduate School of Medicine, Suita, Japan. [1004]Center for Genomic Medicine, Kyoto University Graduate School of Medicine, Kyoto, Japan. [1005]Department of Pulmonary Medicine, Faculty of Medicine, University of Tsukuba, Tsukuba, Japan. [1006]Department of Neurosurgery, Faculty of Medicine, The University of Tokyo, Tokyo, Japan. [1007]Laboratory of Immune Regulation, Department of Microbiology and Immunology, Osaka University Graduate School of Medicine, Suita, Japan. [1009]Medical Innovation Promotion Center, Tokyo Medical and Dental University, Tokyo, Japan. [1010]Clinical Research Center, Tokyo Medical and Dental University Hospital of Medicine, Tokyo, Japan. [1011]Department of Medical Informatics, Tokyo Medical and Dental University Hospital of Medicine, Tokyo, Japan. [1012]Respiratory Medicine, Tokyo Medical and Dental University, Tokyo, Japan. [1013]Clinical Laboratory, Tokyo Medical and Dental University Hospital of Medicine, Tokyo, Japan. [1014]Department of Pathology and Tumor Biology, Kyoto University, Kyoto, Japan. [1015]Department of Respiratory Medicine, Graduate School of Medicine, Faculty of Medicine, Juntendo University, Tokyo, Japan. [1016]Department of Emergency and Disaster Medicine, Graduate School of Medicine, Faculty of Medicine, Juntendo University, Tokyo, Japan. [1017]Department of Cardiovascular Biology and Medicine, Graduate School of Medicine, Faculty of Medicine, Juntendo University, Tokyo, Japan. [1018]Department of Respiratory Medicine, Tokyo Women's Medical University, Tokyo, Japan. [1019]Department of General Medicine, Tokyo Women's Medical University, Tokyo, Japan. [1020]Department of Respiratory

Medicine, Saitama Cardiovascular and Respiratory Center, Saitama, Japan. [1021]Kawasaki Municipal Ida Hospital, Kanagawa, Japan. [1022]Saitama Medical Center, Internal Medicine, Japan Community Healthcare Organization (JCHO), Saitama, Japan. [1023]Saitama City Hospital, Saitama, Japan. [1024]Division of Infection Control, Eiju General Hospital, Tokyo, Japan. [1025]Department of Pulmonary Medicine, Eiju General Hospital, Tokyo, Japan. [1026]Department of Respiratory Medicine, Osaka Saiseikai Nakatsu Hospital, Osaka, Japan. [1027]Division of Respirology, Rheumatology, and Neurology, Department of Internal Medicine Kurume University School of Medicine, Fukuoka, Japan. [1028]Department of Infection Control, Osaka Saiseikai Nakatsu Hospital, Osaka, Japan. [1029]Department of Infectious Diseases, Tosei General Hospital, Aichi, Japan. [1030]Fukujuji Hospital, Kiyose, Japan. [1031]Department of Emergency and Critical Care Medicine, Tokyo Women's Medical University Medical Center East, Tokyo, Japan. [1032]Department of Medicine, Tokyo Women's Medical University Medical Center East, Tokyo, Japan. [1033]Department of Pediatrics, Tokyo Women's Medical University Medical Center East, Tokyo, Japan. [1034]Japan Community Healthcare Organization Kanazawa Hospital, Kanazawa, Japan. [1035]Division of Pulmonary Medicine, Department of Internal Medicine, Federation of National Public Service Personnel Mutual Aid Associations, Tachikawa Hospital, Tachikawa, Japan. [1036]Department of Respiratory Medicine, Japan Organization of Occupational Health and Safety, Kanto Rosai Hospital, Kawasaki, Japan. [1037]Department of General Internal Medicine, Japan Organization of Occupational Health and Safety, Kanto Rosai Hospital, Kawasaki, Japan. [1038]Department of Emergency and Critical Care Medicine, Kansai Medical University General Medical Center, Kirakata, Japan. [1039]Department of Respiratory Medicine, Kitasato University, Kitasato Institute Hospital, Tokyo, Japan. [1040]Ishikawa Prefectural Central Hospital, Kanazawa, Japan. [1041]Internal Medicine, Sano Kosei General Hospital, Sano, Japan. [1042]Saiseikai Yokohamashi Nanbu Hospital, Yokohama, Japan. [1043]Kanagawa Cardiovascular and Respiratory Center, Yokohama, Japan. [1044]Saiseikai Utsunomiya Hospital, Utsunomiya, Japan. [1045]Department of Respiratory Medicine, KKR Sapporo Medical Center, Sapporo, Japan. [1046]Internal Medicine, Internal Medicine Center, Showa University Koto Toyosu Hospital, Tokyo, Japan. [1047]Department of Respiratory Medicine, Toyohashi Municipal Hospital, Toyohashi, Japan. [1048]Keiyu Hospital, Yokohama, Japan. [1049]Department of Rheumatology, National Hospital Organization Hokkaido Medical Center, Sapporo, Japan. [1050]Department of Respiratory Medicine, National Hospital Organization Tokyo Medical Center, Tokyo, Japan. [1051]Department of Allergy, National Hospital Organization Tokyo Medical Center, Tokyo, Japan. [1052]Department of General Internal Medicine and Infectious Diseases, National Hospital Organization Tokyo Medical Center, Tokyo, Japan. [1053]Japanese Red Cross Musashino Hospital, Musashino, Japan. [1054]Department of Respiratory Medicine, Tohoku University Graduate School of Medicine, Sendai, Japan. [1055]Division of Respiratory Medicine, Department of Internal Medicine, Nihon University School of Medicine, Tokyo, Japan. [1056]Department of Emergency and Critical Care Medicine, St Marianna University School of Medicine, Kawasaki, Japan. [1057]Division of General Internal Medicine, Department of Internal Medicine, St Marianna University School of Medicine, Kawasaki, Japan. [1058]National Hospital Organization Kanazawa Medical Center, Kanazawa, Japan. [1059]Division of Infectious Diseases and Respiratory Medicine, Department of Internal Medicine, National Defense Medical College, Tokorozawa, Japan. [1060]Department of Emergency and Critical Care Medicine, Faculty of Medicine, Fukuoka University, Fukuoka, Japan. [1061]Department of Infection Control, Fukuoka University Hospital, Fukuoka, Japan. [1062]Tokyo Saiseikai Central Hospital, Tokyo, Japan. [1063]Department of Internal Medicine, Fukuoka Tokushukai Hospital, Kasuga, Japan. [1064]Department of Infectious Disease and Clinical Research Institute, National Hospital Organization Kyushu Medical Center, Fukuoka, Japan. [1065]Department of Respirology, National Hospital Organization Kyushu Medical Center, Fukuoka, Japan. [1067]Matsumoto City Hospital, Matsumoto, Japan. [1068]Uji-Tokushukai Medical Center, Uji, Japan. [1069]Department of Respiratory Medicine, Nagoya University Graduate School of Medicine, Nagoya, Japan. [1070]Department of Respiratory Medicine, Fujisawa City Hospital, Fujisawa, Japan. [1071]Sapporo City General Hospital, Sapporo, Japan. [1072]Department of Emergency and Critical Care Medicine, Chiba University Graduate School of Medicine, Chiba, Japan. [1073]Division of Respiratory Medicine, Social Welfare Organization Saiseikai Imperial Gift Foundation, Saiseikai Kumamoto Hospital, Kumamoto, Japan. [1074]Department of Anesthesiology and Intensive Care Medicine, Kyoto Prefectural University of Medicine, Kyoto, Japan. [1075]Ome Municipal General Hospital, Ome, Japan. [1076]Hanwa Daini Hospital, Osaka, Japan. [1077]Department of Respiratory Internal Medicine, St Marianna University School of Medicine, Yokohama-City Seibu Hospital, Yokohama, Japan. [1078]Division of Hematology, Department of Internal Medicine, St Marianna University Yokohama-City Seibu Hospital, Yokohama, Japan. [1079]Division of Pulmonary Medicine, Department of Medicine, Tokai University School of Medicine, Tokai University School of Medicine, Tokyo, Japan. [1080]Division of Pulmonary Medicine, Department of Medicine, Tokai University School of Medicine, Tokyo, Japan. [1081]National Hospital Organization Kumamoto Medical Center, Kumamoto, Japan. [1082]Department of Respiratory Medicine, Tokyo Medical University Hospital, Tokyo, Japan. [1083]Department of Respiratory Medicine, Japanese Red Cross Medical Center, Tokyo, Japan. [1084]JA Toride Medical Hospital, Toride, Japan. [1085]Japan Organization of Occupational Health and Safety Okayama Rosai Hospital, Okayama, Japan. [1086]Emergency and Disaster Medicine, Graduate School of Medicine, Gifu University School of Medicine, Gifu, Japan. [1087]Niigata University, Niigata, Japan. [1088]National Hospital Organization Kyoto Medical Center, Kyoto, Japan. [1089]Research Institute for Diseases of the Chest, Graduate School of Medical Sciences, Kyushu University, Fukuoka, Japan. [1090]Department of Medicine and Biosystemic Science, Kyushu University Graduate School of Medical Sciences, Fukuoka, Japan. [1091]Department of Emergency and Critical Care Medicine, Tsukuba University, Tsukuba, Japan. [1092]Department of Nephrology, Faculty of Medicine, University of Tsukuba, Tsukuba, Japan. [1093]Department of Hematology, Faculty of Medicine, University of Tsukuba, Tsukuba, Japan. [1094]National Hospital Organization Tokyo Hospital, Tokyo, Japan. [1095]Fujioka General Hospital, Fujioka, Japan. [1096]Division of Respiratory Medicine and Allergology, Department of Medicine, School of Medicine, Showa University, Tokyo, Japan. [1097]Department of Pulmonary Medicine, Fukushima Medical University, Fukushima, Japan. [1098]Kansai Electric Power Hospital, Osaka, Japan. [1099]Kumamoto City Hospital, Kumamoto, Japan. [1100]Department of Emergency and Critical Care Medicine, Tokyo Metropolitan Police Hospital, Tokyo, Japan. [1101]Department of Respiratory Medicine, International University of Health and Welfare, Shioya Hospital, Narita, Japan. [1102]Department of Clinical Laboratory, International University of Health and Welfare, Shioya Hospital, Narita, Japan. [1103]National Hospital Organization Saitama Hospital, Saitama, Japan. [1104]Department of Respiratory Medicine, Gunma University Graduate School of Medicine, Maebashi, Japan. [1105]Department of Orthopedic Surgery, Tokyo Medical University, Ibaraki Medical Center, Tokyo, Japan. [1106]Department of Internal Medicine, Kiryu Kosei General Hospital, Kiryu, Japan. [1107]Daini Osaka Police Hospital, Osaka, Japan. [1109]Department of Epidemiology, University Medical Centre Groningen, University of Groningen, Groningen, The Netherlands. [1112]Department of Psychiatry, University Medical Center Groningen, Groningen, The Netherlands. [1113]Department of Genetics, University Medical Center Groningen, Groningen, The Netherlands. [1114]Centre for Heart Lung Innovation, University of British Columbia, Vancouver, British Columbia, Canada. [1115]Division of Respiratory Medicine, Faculty of Medicine, University of British Columbia, Vancouver, British Columbia, Canada. [1116]Institut Universitaire de Cardiologie et de Pneumologie de Québec, Université Laval, Quebec, Quebec, Canada. [1117]Department of Genetics and Genomic Sciences, Icahn School of Medicine at Mount Sinai, New York, NY, USA. [1118]University of Washington, Global Health, Seattle, WA, USA. [1119]Gossamer Bio, San Diego, CA, USA. [1120]Department of Pathology and Medical Biology, University Medical Centre Groningen, University of Groningen, Groningen, The Netherlands. [1121]GRIAC Research Institute, University Medical Centre Groningen, University of Groningen, Groningen, The Netherlands. [1122]Department of Pulmonary Diseases, University Medical Center Groningen, University of Groningen, Groningen, The Netherlands. [1123]Center for Genomic Medicine, Massachusetts General Hospital, Boston, MA, USA. [1124]Harvard Medical School, Cambridge, MA, USA. [1125]Program in Medical and Population Genetics, Broad Institute, Boston, MA, USA. [1126]Channing Division of Network Medicine, Department of Medicine, Brigham and Women's Hospital, Boston, MA, USA. [1127]Brigham and Women's Hospital, Boston, MA, USA. [1128]Psychiatric and Neurodevelopmental Genetics Unit, Center for Genomic Medicine, Massachusetts General Hospital, Boston, MA, USA. [1129]Department of Neurology, Massachusetts General Hospital, Boston, MA, USA. [1130]Division of General Internal Medicine, Massachusetts General Hospital and Department of Medicine, Boston, MA, USA. [1132]Department of Human Genetics, University of Michigan, Ann Arbor, MI, USA. [1133]Mount Sinai Clinical Intelligence Center, Department of Genetics and Genomic Sciences, Icahn School of Medicine at Mount Sinai, New York, NY, USA. [1135]Sema4, a Mount Sinai venture, Stamford, CT, USA. [1137]Mount Sinai Clinical Intelligence Center, Charles Bronfman Institute for Personalized Medicine, New York, NY, USA. [1139]Department of Human Genetics, David Geffen School of Medicine at UCLA, Los Angeles, CA, USA. [1140]Icahn Institute of Data Science and Genomics Technology, Icahn School of Medicine, New York, NY, USA. [1141]Mount Sinai Clinical Intelligence Center, Icahn School of Medicine, New York, NY, USA. [1142]Department of Genetic and Genomic Sciences, Icahn School of Medicine at Mount Sinai, New York, NY, USA. [1143]Charles Bronfman Institute for Personalized Medicine, Icahn School of Medicine at Mount Sinai, New York, NY, USA. [1144]Institute for Genomic Health, Icahn School of Medicine at Mount Sinai, New York, NY, USA. [1146]The Mindich Child Health and Development Institute, Icahn School of Medicine at Mount Sinai, New York, NY, USA. [1147]Pamela Sklar Division of Psychiatric Genomics, Icahn School of Medicine at Mount Sinai, New York, NY, USA. [1148]Department of Psychiatry, Icahn School of Medicine at Mount Sinai, New York, NY, USA. [1149]Icahn School of Medicine at Mount Sinai, New York, NY, USA. [1150]Department of Psychiatry, Department of Genetic and Genomic Sciences, Icahn School of Medicine at Mount Sinai, New York, NY, USA. [1151]Department of Environmental Medicine and Public Health, Icahn School of Medicine at Mount Sinai, New York, NY, USA. [1152]Department of Human Genetics, Center for Autism Research and Treatment, Institute for Precision Health, University of California Los Angeles, Los Angeles, CA, USA. [1153]The Hasso Plattner Institute of Digital Health at Mount Sinai, Icahn School of Medicine at Mount Sinai, New York, NY, USA. [1154]BioMe Phenomics Center, >Icahn School of Medicine at Mount Sinai, New York, NY, USA. [1155]Department of Medicine, Icahn School of Medicine at Mount Sinai, New York, NY, USA. [1157]Regeneron Genetics Center, Tarrytown, NY, USA. [1158]Phenomic Analytics & Clinical Data Core, Geisinger Health System, Danville, PA, USA. [1159]Department of Population Health Sciences, Geisinger Health System, Danville, PA, USA. [1160]Department of Molecular and Functional Genomics, Geisinger Health System, Danville, PA, USA. [1162]Department of Genetics, University of Pennsylvania Perelman School of Medicine, Philadelphia, PA, USA. [1163]Department of Biomedical Data Science, Stanford University, Stanford, CA, USA. [1166]Department of Psychiatry, University of North Carolina at Chapel Hill, Chapel Hill, USA. [1167]Department of Nutrition, University of North Carolina at Chapel Hill, Chapel Hill, USA. [1168]Institute of Neuroscience and Physiology, University of Gothenburg, Gothenburg, Sweden. [1169]Department of Medical Sciences, University of Turin, Turin, Italy. [1170]Department of Clinical and Biological Sciences, University of Turin, Orbassano, Italy. [1171]Department of Pediatrics, Department of Microbiology, Immunology and Molecular Genetics, University of California Los Angeles, Los Angeles, CA, USA. [1172]University of Genova, Genova, Italy. [1173]Hopital Mont-Godinne, Yvoir, Belgium. [1174]Department of Molecular Medicine, University of Pavia, Pavia, Italy. [1175]Department of Public Health and Pediatric Sciences, University of Turin, Turin, Italy. [1176]Qatar Biobank for Medical Research, Qatar Foundation Research, Development and Innovation, Qatar Foundation, Doha, Qatar. [1177]Latvian Biomedical Research and Study Centre, Riga, Latvia. [1178]Department of Neuroscience, Karolinska Institutet, Stockholm, Sweden. [1179]Max Planck Institute for Evolutionary Anthropology, Leipzig, Germany. [1180]Anaesthesiology and Intensive Care Medicine, Department of Surgical Sciences, Uppsala University, Uppsala, Sweden. [1181]Integrative Physiology, Department of Medical Cell Biology, Uppsala University, Uppsala, Sweden. [1182]Hedenstierna Laboratory, CIRRUS, Anaesthesiology and Intensive Care Medicine, Department of Surgical Sciences, Uppsala University, Uppsala, Sweden. [1183]Department of Computer Science, School of Engineering, University of California Los Angeles, Los Angeles, CA, USA. [1184]University of California Los Angeles, Los Angeles, CA, USA. [1185]Department of Psychiatry and Biobehavioral Sciences, David Geffen School of Medicine at University of California Los Angeles, Los Angeles, CA, USA. [1186]Division of Immunology, Allergy, and Rheumatology, University of California Los Angeles, Los Angeles, CA, USA. [1187]Department of Psychiatry, University of California Los Angeles, Los Angeles, CA, USA. [1188]Department of Neurology, University of California Los Angeles, Los Angeles, CA, USA. [1189]Department of Computational Medicine, University of California Los Angeles, Los Angeles, CA, USA. [1190]Department of Pathology and Laboratory Medicine, University of California Los Angeles, Los Angeles, CA, USA. [1191]Bioinformatics IDP, UCLA, Los Angeles, CA, USA. [1192]Department of Neurology, David Geffen School of Medicine at UCLA, Los Angeles, CA, USA. [1193]Department of Urology, David Geffen School of Medicine at UCLA, Los Angeles, CA, USA. [1196]Queen Mary University, London, UK. [1196]UCL Great Ormond Street Institute of Child Health, London, UK. [1197]University of Cambridge, Cambridge, UK. [1199]Big Data Institute, Nuffield Department of Population Health, Li Ka Shing Centre for Health Information and Discovery, University of Oxford, Oxford, UK. [1201]Experimental Medicine Division, Nuffield Department of

Medicine, John Radcliffe Hospital, University of Oxford, Oxford, UK. [1202]Public Health England, Field Service, Addenbrooke's Hospital, Cambridge, UK. [1203]Public Health England, Data and Analytical Services, National Infection Service, London, UK. [1204]Program in Bioinformatics and Integrative Genomics, Harvard Medical School, Boston, MA, USA. [1205]Program in Biological and Biomedical Sciences, Harvard Medical School, Boston, MA, USA. [1207]Department of Clinical Research and Leadership, George Washington University, Washington, DC, USA. [1208]Department of Human Genetics, The Wellcome Sanger Institute, Wellcome Genome Campus, Hinxton, Cambridge, UK. [1209]Strangeways Research Laboratory, The National Institute for Health Research Blood and Transplant Unit in Donor Health and Genomics, University of Cambridge, Cambridge, UK. [1210]Department of Haematology, University of Cambridge, Cambridge Biomedical Campus, Cambridge, UK. [1211]British Heart Foundation Cardiovascular Epidemiology Unit, Department of Public Health and Primary Care, University of Cambridge, Cambridge, UK. [1212]British Heart Foundation Centre of Research Excellence, University of Cambridge, Cambridge, UK. [1213]The National Institute for Health Research Blood and Transplant Research Unit in Donor Health and Genomics, University of Cambridge, Cambridge, UK. [1214]Health Data Research UK Cambridge, Wellcome Genome Campus and University of Cambridge, Cambridge, UK. [1215]Department of Human Genetics, Wellcome Sanger Institute, Hinxton, UK. [1216]Department of Epidemiology, Emory University Rollins School of Public Health, North Druid Hills, GA, USA. [1217]Atlanta CA Health Care System, North Druid Hills, GA, USA. [1218]Center for Population Genomics, MAVERIC, VA Boston Healthcare System, Boston, MA, USA. [1219]MAVERIC, VA Boston Healthcare System, Boston, MA, USA. [1220]Stanford University, Stanford, CA, USA. [1221]Palo Alto VA Healthcare System, Stanford, CA, USA. [1222]Department of Biostatistics, Boston University School of Public Health, Boston, MA, USA. [1223]Department of Haematology, Central Hospital of Bolzano (SABES-ASDAA), Bolzano, Italy. [1224]Laboratory of Clinical Pathology, Hospital of Bressanone (SABES-ASDAA), Bressanone, Italy. [1226]University of Alcalá, Centro de Investigación Biomédica en Red en Enfermedades Respiratorias (CIBERES), Madrid, Spain. [1227]Center for Applied Genomics, The Children's Hospital of Philadelphia, Philadelphia, PA, USA. [1228]Division of Human Genetics, Department of Pediatrics, The Perelman School of Medicine, University of Pennsylvania, Philadelphia, PA, USA. [1229]Faculty of Medicine, University of Iceland, Reykjavik, Iceland. [1231]Infectious Disease Unit, Hospital of Massa, Massa, Italy. [1232]Department of Clinical Medicine, Public Health, Life and Environment Sciences, University of L'Aquila, L'Aquila, Italy. [1233]UOSD Laboratorio di Genetica Medica - ASL Viterbo, San Lorenzo, Italy. [1234]Unit of Infectious Diseases, S. M. Annunziata Hospital, Florence, Italy. [1235]Infectious Disease Unit, Hospital of Lucca, Lucca, Italy. [1236]Department of Clinical and Experimental Medicine, Infectious Diseases Unit, University of Pisa, Pisa, Italy. [1238]Clinic of Infectious Diseases, Catholic University of the Sacred Heart, Rome, Italy. [1239]Department of Diagnostic and Laboratory Medicine, Institute of Biochemistry and Clinical Biochemistry, Fondazione Policlinico Universitario A. Gemelli IRCCS, Catholic University of the Sacred Heart, Rome, Italy. [1240]Private University in the Principality of Liechtenstein, Triesen, Liechtenstein. [1241]Digestive Diseases Unit, Virgen del Rocio University Hospital, Institute of Biomedicine of Seville, University of Seville, Seville, Spain. [1242]Department of Biochemistry, University Hospital Vall d'Hebron, Barcelona, Spain. [1243]University of Sevilla, Sevilla, Spain. [1244]Instituto de Biomedicina de Sevilla, Sevilla, Spain. [1245]Hospital Universitario Virgen del Rocío de Sevilla, Sevilla, Spain. [1246]Consejo Superior de Investigaciones científicas, Madrid, Spain. [1247]Humanitas Clinical and Research Center, IRCCS, Milan, Italy. [1248]Immunohematology Department, Banc de Sang i Teixits, Autonomous University of Barcelona, Barcelona, Spain. [1249]August Pi i Sunyer Biomedical Research Institute, Hospital Clinic, University of Barcelona, Barcelona, Spain. [1250]Department of Pathophysiology and Transplantation, Università degli Studi di Milano, Milan, Italy. [1251]Internal Medicine Department, Virgen del Rocio University Hospital, Sevilla, Spain. [1252]Department of Biomedical Sciences, Humanitas University, Milan, Italy. [1253]Department Emergency, Anesthesia and Intensive Care, University Milano-Bicocca, Monza, Italy. [1254]Department of Medical Sciences, Università degli Studi di Torino, Turin, Italy. [1255]Department of Medical Microbiology, Clinic of Laboratory Medicine, St Olav's Hospital, Trondheim, Norway. [1256]Department of Infectious Diseases, St Olav's Hospital, Trondheim University Hospital, Trondheim, Norway. [1257]Department of Clinical and Molecular Medicine, NTNU, Trondheim, Norway. [1258]Department of Research, St Olav's Hospital, Trondheim University Hospital, Trondheim, Norway. [1259]Institute of Parasitology and Biomedicine Lopez-Neyra, Granada, Spain. [1260]Institute for Cardiogenetics, University of Lübeck, Lübeck, Germany. [1261]German Research Center for Cardiovascular Research, partner site Hamburg-Lübeck-Kiel, Lübeck, Germany. [1262]University Heart Center Lübeck, Lübeck, Germany. [1263]Department of Research, Ostfold Hospital Trust, Gralum, Norway. [1264]Pediatric Departement, Centro Tettamanti- European Reference Network (ERN) PaedCan, EuroBloodNet, MetabERN-University of Milano-Bicocca-Fondazione MBBM/Ospedale San Gerardo, Milan, Italy. [1265]Geminicenter for Sepsis Research, Institute of Circulation and Medical Imaging (ISB), NTNU, Trondheim, Norway. [1266]Clinic of Anesthesia and Intensive Care, St Olav's Hospital, Trondheim University Hospital, Trondheim, Norway. [1267]Clinic of Medicine and Rehabilitation, Levanger Hospital, Nord-Trondelag Hospital Trust, Levanger, Norway. [1268]Stefan-Morsch-Stiftung, Birkenfeld, Germany. [1269]Center of Bioinformatics, Biostatistics, and Bioimaging, School of Medicine and Surgery, University of Milano Bicocca, Milan, Italy. [1270]Phase 1 Research Centre, ASST Monza, School of Medicine and Surgery, University of Milano-Bicocca, Milan, Italy. [1271]Pneumologia ASST-Monza, University of Milano-Bicocca, Milano, Italy. [1272]School of Medicine and Surgery, University of Milano-Bicocca, Milano, Italy. [1273]Infectious Diseases Unit, San Gerardo Hospital, Monza, Italy. [1274]SODIR-VHIR research group, Barcelona, Spain. [1275]Bioinformatics area, Fundación progreso y Salud, Andalucia, Spain. [1276]Present address: Program in Metabolism, Broad Institute of MIT and Harvard, Cambridge, MA, USA. [1277]Present address: Program in Medical and Population Genetics, Broad Institute of MIT and Harvard, Cambridge, MA, USA. [1278]Present address: Diabetes Unit, Center for Genomic Medicine, Massachusetts General Hospital, Boston, MA, USA. [1279]Present address: Harvard Medical School, Boston, MA, USA. [1280]These authors contributed equally: Mari E. K. Niemi, Juha Karjalainen, Benjamin M. Neale, Mark Daly, Andrea Ganna. [1281]Unaffiliated: Sangyoon Im, Jason Kendall, Michael MacMahon, Mark Peters, Robert Thompson, Martin Williams. ✉e-mail: bneale@broadinstitute.org; mark.daly@helsinki.fi; andrea.ganna@helsinki.fi

## Methods

### Contributing studies

All of the participants were recruited following protocols approved by local Institutional Review Boards; this information is collected in Supplementary Table 1 for all 46 studies. All protocols followed local ethics recommendations and informed consent was obtained when required. Information about sample numbers, sex and age from for each contributing study is given in Supplementary Table 1. In total, 16 studies contributed data to the analysis of critical illness due to COVID-19, 29 studies contributed data to hospitalized COVID-19 analysis and 44 studies contributed to the analysis of all cases of COVID-19. Each individual study that contributed data to a particular analysis met a minimum threshold of 50 cases, as defined by the phenotypic criteria, for statistical robustness. The effective sample sizes for each ancestry group shown in Fig. 1 were calculated for display using the formula: $((4 \times N_{case} \times N_{control})/(N_{case} + N_{control}))$. Details of contributing research groups are provided in Supplementary Table 1.

### Phenotype definitions

COVID-19 disease status (critical illness and hospitalization status) was assessed following the Diagnosis and Treatment Protocol for Novel Coronavirus Pneumonia[38]. The critically ill COVID-19 group included patients who were hospitalized owing to symptoms associated with laboratory-confirmed SARS-CoV-2 infection and who required respiratory support or whose cause of death was associated with COVID-19. The hospitalized COVID-19 group included patients who were hospitalized owing to symptoms associated with laboratory-confirmed SARS-CoV-2 infection.

The reported SARS-CoV-2 infection group included individuals with laboratory-confirmed SARS-CoV-2 infection or electronic health record, ICD coding or clinically confirmed COVID-19, or self-reported COVID-19 (for example, by questionnaire), with or without symptoms of any severity. Genetic-ancestry-matched control individuals for the three case definitions were sourced from population-based cohorts, including individuals whose exposure status to SARS-CoV-2 was either unknown or infection-negative for questionnaire/electronic-health-record-based cohorts. Additional information regarding individual studies contributing to the consortium are described in Supplementary Table 1.

### Genome-wide association studies and meta-analyses

Each contributing study genotyped the samples and performed quality controls, data imputation and analysis independently, but following the consortium recommendations (information is available at https://www.covid19hg.org/). We recommended that genome-wide association study (GWAS) analyses were run using Scalable and Accurate Implementation of GEneralized mixed model (SAIGE)[39] on chromosomes 1–22 and X. The recommended analysis tool was SAIGE, but studies also used other software such as PLINK[40]. The suggested covariates were age, age$^2$, sex, age × sex and the 20 first principal components. Any other study-specific covariates to account for known technical artefacts could be added. SAIGE automatically accounts for sample relatedness and case–control imbalances. Quality-control and analysis approaches for individual studies are reported in Supplementary Table 1.

Study-specific summary statistics were then processed for meta-analysis. Potential false positives, inflation and deflation were examined for each submitted GWAS. Allele frequency plots against gnomAD 3.0 genomes were manually inspected for each study. Standard error values as a function of the effective sample size were used to find studies that deviated from the expected trend. Summary statistics passing this manual quality control were included in the meta-analysis. Variants with an allele frequency of >0.1% and an imputation INFO score of >0.6 were carried forward from each study. Variants and alleles were lifted over to genome build GRCh38, if needed, and harmonized to gnomAD 3.0 genomes[41] by finding matching variants by strand flipping or switching the ordering of alleles. If multiple matching variants were included, the best match was chosen according to the minimum fold change in absolute allele frequency. Meta-analysis was performed using the inverse-variance-weighted (IVW) method on variants that were present in at least two-thirds of the studies contributing to the phenotype analysis. The method summarizes effect sizes across the multiple studies by computing the mean of the effect sizes weighted by the inverse variance in each individual study.

We report 13 meta-analysis variants that pass the genome-wide significance threshold after adjusting the threshold for multiple traits tested ($P < 5 \times 10^{-8}/3$). We report the unadjusted $P$ values for each variant. We tested for heterogeneity between estimates from contributing studies using Cochran's $Q$-test[42,43]. This is calculated for each variant as the weighted sum of squared differences between the effects sizes and their meta-analysis effect, the weights being the inverse variance of the effect size. $Q$ is distributed as a $\chi^2$ statistic with $k$ (number of studies) minus one degrees of freedom. Two loci reached genome-wide significance but were excluded from the significant results in Supplementary Table 2 due to heterogeneity between estimates from contributing studies and missingness between studies at chr. 6: 31057940–31380334 and chr. 7: 54671568–54759789; however, these regions are not excluded from the corresponding summary statistics in data release 5 (COVID-19 HGI (https://www.covid19hg.org/results/r5/) and GWAS Catalog (study code GCST011074)). For each of the lead variants reported in Supplementary Table 2, we aimed to find loci specific to susceptibility or severity by testing whether there was heterogeneity between the effect sizes associated with hospitalized COVID-19 (progression to severe disease) and reported SARS-CoV-2 infection. We used the Cochran's $Q$ measure[42,43], calculated for each variant as the weighted sum of squared differences between the two analysis effect sizes and their meta-analysis effect with the weights being the inverse variance of the effect size. A significant $P$ value of $P < 0.004$ ((0.05/13 loci) for multiple tests) indicates that the effect sizes for a particular variant are significantly different in the two analyses (Supplementary Table 2). For the nine loci, in which the lead variant effect size was significantly higher for hospitalized COVID-19, we carried out the same test again but comparing effect sizes from hospitalized COVID-19 with critically ill COVID-19 (Supplementary Table 4). Furthermore, we carried out the same test comparing meta-analysed hospitalized COVID-19 (population as controls) and hospitalized COVID-19 (SARS-CoV-2-positive but non-hospitalized as controls) (Supplementary Table 4). For these pairs of phenotype comparisons, we generated new meta-analysis summary statistics to use; including only those studies that could contribute data to both phenotypes that were under comparison.

### Principal component projection

To project every GWAS participant into the same principal component (PC) space, we used pre-computed PC loadings and reference allele frequencies. For reference, we used unrelated samples from the 1000 Genomes Project and the Human Genome Diversity Project and computed PC loadings and allele frequencies for the 117,221 single-nucleotide polymorphisms (SNPs) that (1) are available in every cohort; (2) have a minor allele frequency of >0.1% in the reference; and (3) are LD-pruned ($r^2 < 0.8$; 500-kb window). We then asked each cohort to project their samples using our automated script provided at https://github.com/covid19-hg/. It internally uses the PLINK2[44] --score function with the variance-standardize option and reference allele frequencies (--read-freq); so that each cohort-specific genotype/dosage matrix is mean-centred and variance-standardized with respect to reference allele frequencies, but not cohort-specific allele frequencies. We further normalized the projected PC scores by dividing the values by a square root of the number of variants used for projection to account for a subtle difference due to missing variants.

## Gene prioritization

To prioritize candidate causal genes reported in full in Supplementary Table 2, we used various gene prioritization approaches using both locus-based and similarity-based methods. Because we only describe the in silico gene prioritization results without characterizing the actual functional activity in vitro or in vivo, we aimed to provide a systematic approach to nominate potential causal genes in a locus using the following criteria.

(1) The closest gene: a gene that is closest to a lead variant by distance to the gene body.

(2) Genes in the LD region: genes that overlap with a genomic range containing any variants in LD ($r^2 > 0.6$) with a lead variant. For LD computation, we retrieved LD matrices provided by gnomAD v.2.1.1[41] for each population analysed in this study (except for admixed American, Middle Eastern and South Asian genetic ancestry populations, for whom data are not available). We then constructed a weighted-average LD matrix by per-population sample sizes in each meta-analysis, which we used as a LD reference.

(3) Genes with coding variants: genes with at least one loss-of-function or missense variant (annotated by VEP[45] v.95 with GENCODE v.29) that is in LD with a lead variant ($r^2 > 0.6$).

(4) eGenes: genes with at least one fine-mapped *cis*-eQTL variant (PIP > 0.1) that is in LD with a lead variant ($r^2 > 0.6$) (Supplementary Table 5). We retrieved fine-mapped variants from the GTEx v.8[20] (https://www.finucanelab.org/) and eQTL catalogue[46]. In addition, we looked up significant associations in the Lung eQTL Consortium[21] ($n = 1,103$) to further support our findings in lung with a larger sample size (Supplementary Table 7). We note that, in contrast to the GTEx or eQTL catalogue, we only looked at associations and did not fine-map our data to the Lung eQTL Consortium data.

(5) V2G: a gene with the highest overall V2G score based on Open Targets Genetics (OTG)[26]. For each variant, the overall V2G score aggregates differentially weighted evidence of variant–gene associations from several data sources, including molecular *cis*-QTL data (for example, *cis*-protein QTLs from ref. [47], *cis*-eQTLs from GTEx v.7 and so on), interaction-based datasets (for example, promoter capture Hi-C), genomic distance and variant effect predictions (VEP) from Ensembl. A detailed description of the evidence sources and weights used is provided in the OTG documentation (https://genetics-docs.opentargets.org/our-approach/data-pipeline)[26].

## Phenome-wide association study

To investigate the evidence of shared effects of 15 index variants for COVID-19 and previously reported phenotypes, we performed a phenome-wide association study. We considered phenotypes in OTG obtained from the GWAS catalogue (this included studies with and without full summary statistics, $n = 300$ and 14,013, respectively)[48] and from the UK Biobank. Summary statistics for UK Biobank traits were extracted from SAIGE[39] for binary outcomes ($n = 1,283$ traits) and Neale v.2 ($n = 2,139$ traits) for both binary and quantitative traits (http://www.nealelab.is/uk-biobank/) and FinnGen Freeze 4 cohort (https://www.finngen.fi/en/access_results). We report PheWAS results for phenotypes for which the lead variants were in high LD ($r^2 > 0.8$) with the 13 genome-wide significant lead variants from our main COVID-19 meta-analysis (Supplementary Table 6). This conservative approach allowed spurious signals primarily driven by proximity rather than actual colocalization to be removed (see Methods).

To remove plausible spurious associations, we retrieved phenotypes for GWAS lead variants that were in LD ($r^2 > 0.8$) with COVID-19 index variants.

## Heritability

LD score regression v.1.0.1[49] was used to estimate the SNP heritability of the phenotypes from the meta-analysis summary statistic files. As this method depends on matching the LD structure of the analysis sample to a reference panel, the summary statistics of European ancestry only were used. Sample sizes were $n = 5,101$ critically ill cases of COVID-19 and $n = 1,383,241$ control participants, $n = 9,986$ hospitalized cases of COVID-19 and $n = 1,877,672$ control participants, and $n = 38,984$ cases and $n = 1,644,784$ control participants for the analysis of all cases— all including the 23andMe cohort. Pre-calculated LD scores from the 1000 Genomes European reference population were obtained online (https://data.broadinstitute.org/alkesgroup/LDSCORE/). Analyses were conducted using the standard program settings for variant filtering (removal of non-HapMap3 SNPs, the HLA region on chromosome 6, non-autosomal, $\chi^2 > 30$, minor allele frequency of <1%, or allele mismatch with reference). We additionally report SNP heritability estimates for the all-ancestries meta-analyses, calculated using European panel LD scores, in Supplementary Table 8.

## Partitioned heritability

We used partitioned LD score regression[50] to partition COVID-19 SNP heritability in cell types in our summary statistics for European ancestry only. We ran the analysis using the baseline model LD scores calculated for European populations and regression weights that are available online (https://github.com/bulik/ldsc). We used the COVID-19 summary statistics for European ancestry only for the analysis.

## Genome-wide association summary statistics

We obtained genome-wide association summary statistics for 43 complex-disease, neuropsychiatric, behavioural or biomarker phenotypes (Supplementary Table 10). These phenotypes were selected based on their putative relevance to COVID-19 susceptibility, severity or mortality, with 19 selected based on the Centers for Disease Control list of underlying medical conditions associated with COVID-19 severity[51] or traits reported to be associated with increased risk of COVID-19 mortality by OpenSafely[52]. Summary statistics generated from GWAS using individuals of European ancestry were preferentially selected if available. These summary statistics were used in subsequent genetic correlation and Mendelian randomization analyses.

## Genetic correlation

LD score regression[50] was also used to estimate the genetic correlations between our COVID-19 meta-analysis phenotypes reported using samples of only European ancestry, and between these and the curated set of 38 summary statistics. Genetic correlations were estimated using the same LD score regression settings as for heritability calculations. Differences between the observed genetic correlations of SARS-CoV-2 infection and COVID-19 severity were compared using a $z$-score method[53].

## Mendelian randomization

Two-sample Mendelian randomization was used to evaluate the potential for causal association of the 38 traits on COVID-19 hospitalization, on COVID-19 severity and reported SARS-CoV-2 infection using samples of only European ancestry. Independent genome-wide significant SNPs robustly associated with the exposures of interest ($P < 5 \times 10^{-8}$) were selected as genetic instruments by performing LD clumping using PLINK[40]. We used a strict $r^2$ threshold of 0.001, a 10-Mb clumping window, and the European reference panel from the 1000 Genomes Project[54] to discard SNPs in LD with another variant with a smaller $P$-value association. For genetic variants that were not present in the hospitalized COVID-19 analysis, PLINK was used to identify proxy variants that were in LD ($r^2 > 0.8$). Next, the exposure and outcome datasets were harmonized using the R package TwoSampleMR[55]. Namely, we ensured that the effect of a variant on the exposure and outcome corresponded to the same allele, we inferred positive-strand alleles and dropped palindromes with ambiguous allele frequencies, as well as incompatible alleles. Supplementary Table 10 includes the harmonized datasets used in the analyses.

The global test from Mendelian randomization pleiotropy residual sum and outlier (MR-PRESSO)[56] software was used to investigate overall horizontal pleiotropy. In brief, the standard IVW meta-analytic framework was used to calculate the average causal effect by excluding each genetic variant used to instrument the analysis. A global statistic was calculated by summing the observed residual sum of squares, that is, the difference between the effect predicted by the IVW slope excluding the SNP, and the observed effect of the SNP on the outcome. Overall horizontal pleiotropy was subsequently analysed by comparing the observed residual sum of squares, with the residual sum of squares expected under the null hypothesis of no pleiotropy. The MR-PRESSO global test was shown to perform well when the outcome and exposure GWASs are not disjoint (although the power to detect horizontal pleiotropy is slightly reduced by complete sample overlap). We also used the regression intercept in MR-Egger[57] to evaluate potential bias due to directional pleiotropic effects. This additional check was used in Mendelian randomization analyses with an $I_{GX}^2$ index surpassing the recommended threshold ($I_{GX}^2 > 90\%$)[58]. Contingent on the MR-PRESSO global test results we analysed the causal effect of each exposure on COVID-19 hospitalization by using a fixed-effect IVW meta-analysis as the primary analysis, or, if pleiotropy was present, the MR-PRESSO outlier-corrected test. The IVW approach estimates the causal effect by aggregating the single-SNP causal effects (obtained using the ratio of coefficients method—that is, the ratio of the effect of the SNP on the outcome over the effect of the SNP on the exposure) in a fixed-effects meta-analysis. The SNPs were assigned weights based on their inverse variance. The IVW method confers the greatest statistical power for estimating causal associations[59], but assumes that all variants are valid instruments and can produce biased estimates if the average pleiotropic effect differs from zero. Alternatively, when horizontal pleiotropy was present, we used the MR-PRESSO outlier-corrected method to correct the IVW test by removing outlier SNPs. We conducted further sensitivity analyses using alternative Mendelian randomization methods that provide consistent estimates of the causal effect even when some instrumental variables are invalid, at the cost of reduced statistical power including: (1) Weighted median estimator (WME); (2) weighted mode-based estimator (WMBE); and (3) MR-Egger regression. Robust causal estimates were defined as those that were significant at an FDR of 5% and either (1) showed no evidence of heterogeneity (MR-PRESSO global test $P > 0.05$) or horizontal pleiotropy (Egger intercept $P > 0.05$); or (2) in the presence of heterogeneity or horizontal pleiotropy, the WME-, WMBE-, MR-Egger- or MR-PRESSO-corrected estimates were significant ($P < 0.05$). All statistical analyses were conducted using R v.4.0.3. Mendelian randomization analysis was performed using the 'TwoSampleMR' v.0.5.5 package[55].

## Website and data distribution

In anticipation of the need to coordinate many international partners around a single meta-analysis effort, we created the COVID-19 HGI website (https://covid19hg.org). We were able to centralize information, recruit partner studies, rapidly distribute summary statistics and present preliminary interpretations of the results to the public. Open meetings are held on a monthly basis to discuss future plans and new results; video recordings and supporting documents are shared (https://covid19hg.org/meeting-archive). This centralized resource provides a conceptual and technological framework for organizing global academic and industry groups around a shared goal. The website source code and additional technical details are available at https://github.com/covid19-hg/covid19hg.

To recruit new international partner studies, we developed a workflow in which new studies are registered and verified by a curation team (https://covid19hg.org/register). Users can explore the registered studies using a customized interface to find and contact studies with similar goals or approaches (https://covid19hg.org/partners). This helps to promote organic assembly around focused projects that are adjacent to the centralized effort (https://covid19hg.org/projects). Visitors can query study information, including study design and research questions. Registered studies are visualized on a world map and are searchable by institutional affiliation, city and country.

To encourage data sharing and other forms of participation, we created a rolling acknowledgements page (https://covid19hg.org/acknowledgements) and directions on how to contribute data to the central meta-analysis effort (https://covid19hg.org/data-sharing). Upon the completion of each data freeze, we post summary statistics, plots and sample size breakdowns for each phenotype and contributing cohort (https://covid19hg.org/results). The results can be explored using an interactive web browser (https://app.covid19hg.org). Several computational research groups carry out follow-up analyses, which are made available for download (https://covid19hg.org/in-silico). To enhance scientific communication to the public, preliminary results are described in blog posts by the scientific communications team and shared on Twitter. The first post was translated to 30 languages with the help of 85 volunteer translators. We compile publications and pre-prints submitted by participating groups and summarize genome-wide significant findings from these publications (https://covid19hg.org/publications).

## Reporting summary

Further information on research design is available in the Nature Research Reporting Summary linked to this paper.

## Data availability

Summary statistics generated by the COVID-19 HGI are available at https://www.covid19hg.org/results/r5/ and are available in the GWAS Catalog (study code GCST011074). The analyses described here include the freeze-5 data. COVID-19 HGI continues to regularly release new data freezes. Summary statistics for non-European ancestry samples are not currently available due to the small individual sample sizes of these groups, but results for lead variants of 13 loci are reported in Supplementary Table 3. Individual level data can be requested directly from contributing studies, listed in Supplementary Table 1. We used publicly available data from GTEx (https://gtexportal.org/home/), the Neale lab (http://www.nealelab.is/uk-biobank/), Finucane lab (https://www.finucanelab.org), the FinnGen Freeze 4 cohort (https://www.finngen.fi/en/access_results) and the eQTL catalogue release 3 (http://www.ebi.ac.uk/eqtl/).

## Code availability

The code for summary statistics lift-over, the projection PCA pipeline including precomputed loadings and meta-analyses are available on GitHub (https://github.com/covid19-hg/) and the code for the Mendelian randomization and genetic correlation pipeline is available on GitHub at https://github.com/marcoralab/MRcovid.

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

**Acknowledgements** We thank the entire COVID-19 HGI community for their contributions and continued collaboration. The work of the contributing studies was supported by numerous grants from governmental and charitable bodies. Acknowledgements specific to contributing studies are provided in Supplementary Table 13. We thank G. Butler-Laporte, G. Wojcik, M.-G. Hollm-Delgado, C. Willer and G. Davey Smith for their extensive feedback and discussion.

**Author contributions** Author contributions are provided within the author list.

**Competing interests** A full list of competing interests is supplied as Supplementary Table 13.

**Additional information**
**Correspondence and requests for materials** should be addressed to Benjamin M. Neale, Mark Daly, Andrea Ganna, Benjamin M. Neale or Mark Daly.

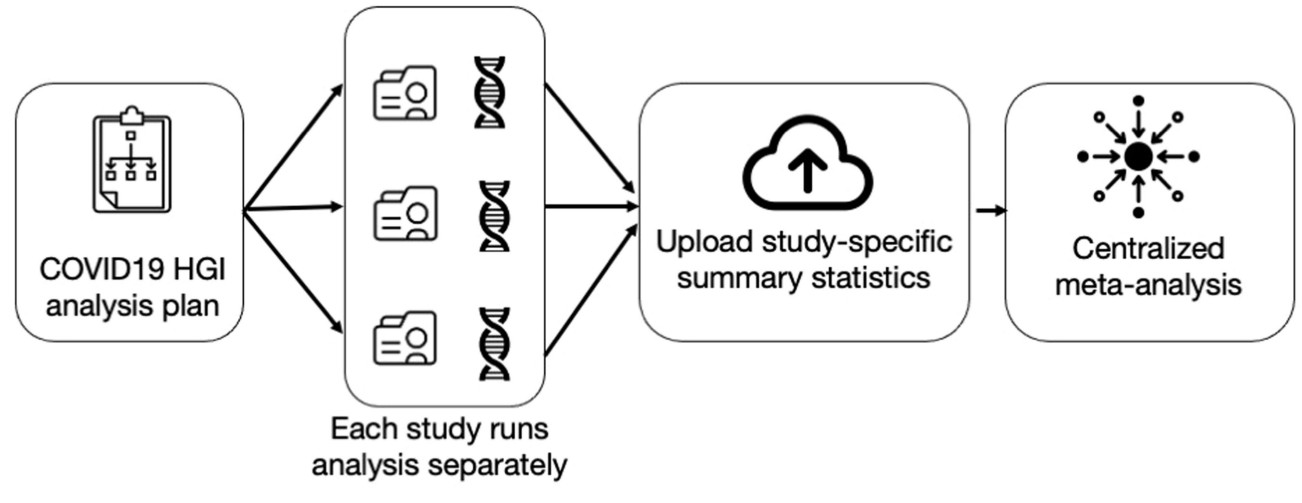

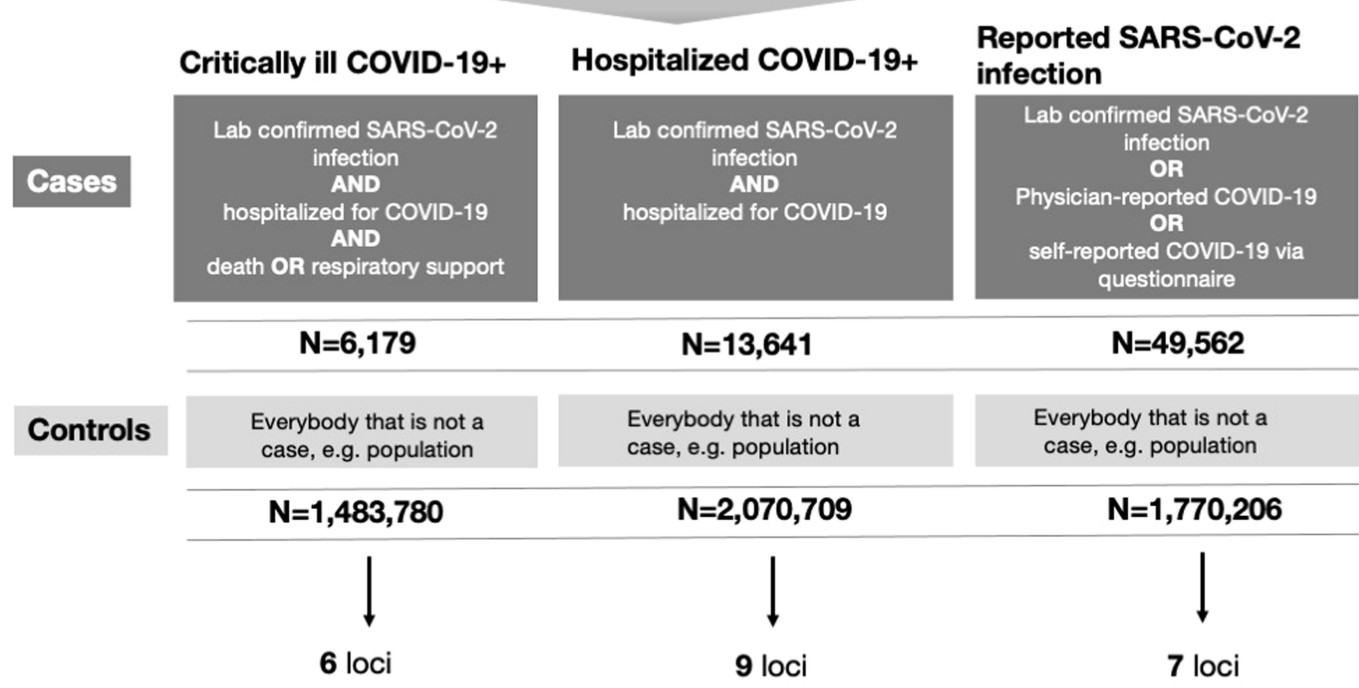

**Extended Data Fig. 1 | Analytical summary of the COVID-19 HGI meta-analysis.** Using the analytical plan set by the COVID-19 HGI, each individual study runs their analyses and uploads the results to the Initiative, who then runs the meta-analysis. There are three main analyses that each study can contribute summary statistics to: critically ill COVID-19, hospitalized COVID-19 and reported SARS-CoV-2 infection. The phenotypic criteria used to define cases are listed in the dark grey boxes, along with the numbers of cases (*N*) included in the final all-ancestries meta-analysis. Controls were defined in the same way across all three analyses as everybody that is not a case—for example, population controls (light grey box). Sensitivity analyses—not reported in this extended data figure—also included mild and/or asymptomatic cases of COVID-19 as control individuals. Sample number (*N*) of control individuals differed between the analyses due to the difference in the number of studies contributing data to these.

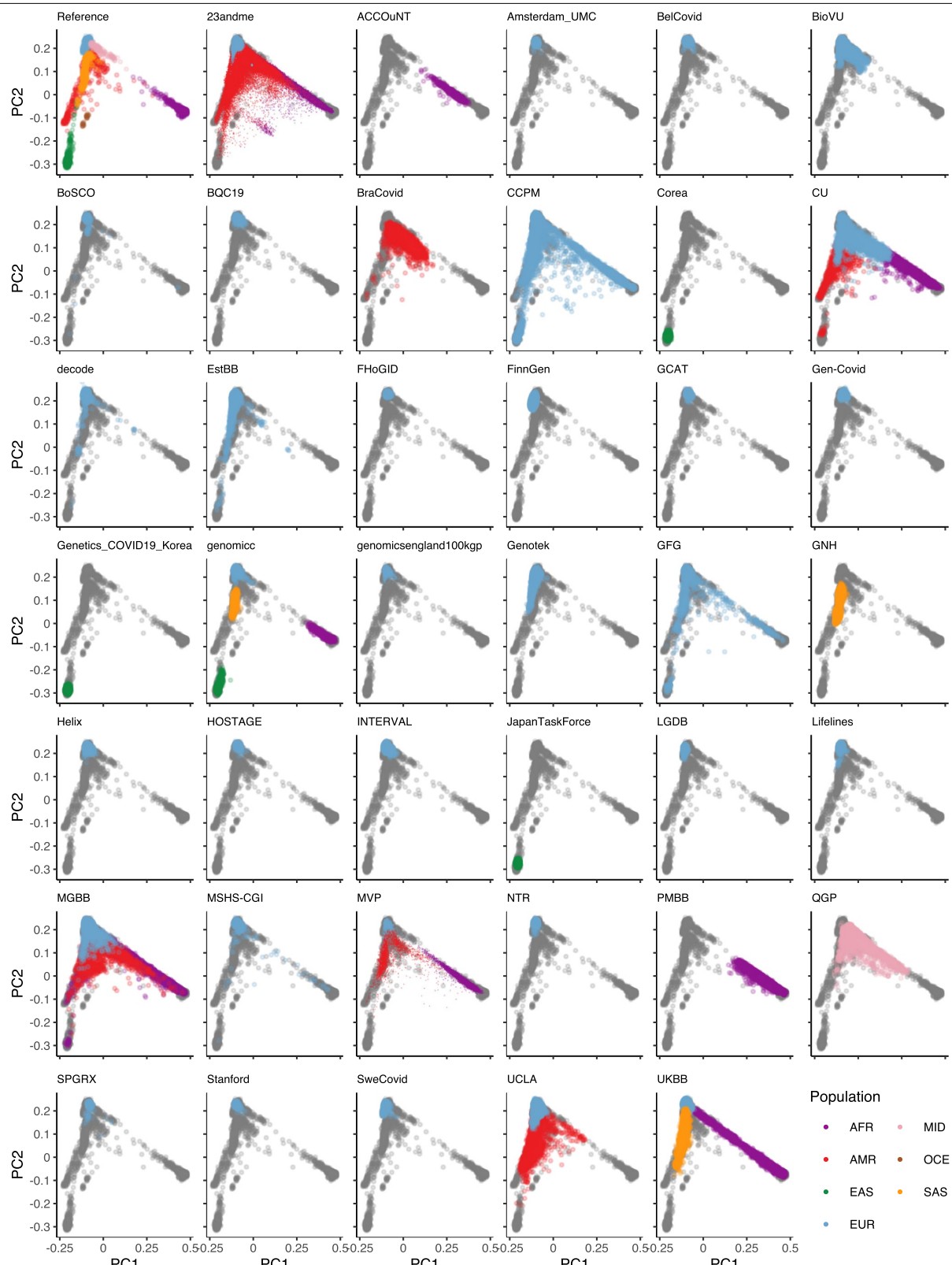

**Extended Data Fig. 2 | Projection of contributing studies samples into the same PC space.** We asked participating studies to perform a PC projection using the 1000 Genomes Project and Human Genome Diversity Project as a reference, with a common set of variants. For each panel (except for the reference), coloured points correspond to contributed samples from each cohort, whereas grey points correspond to the reference samples from the 1000 Genomes Project. Colour represents a genetic population that each cohort specified. As 23andMe, Genomics England 100,000 Genomes Project (GenomicsEngland100kgp), and Million Veterans Program (MVP) only submitted PCA images, we overlaid their submitted transparent images using the same coordinates, instead of directly plotting them. Populations are defined as African (AFR), admixed American (AMR), East Asian (EAS), European (EUR), Middle Eastern (MID) and South Asian (SAS), Oceanian (OCE).

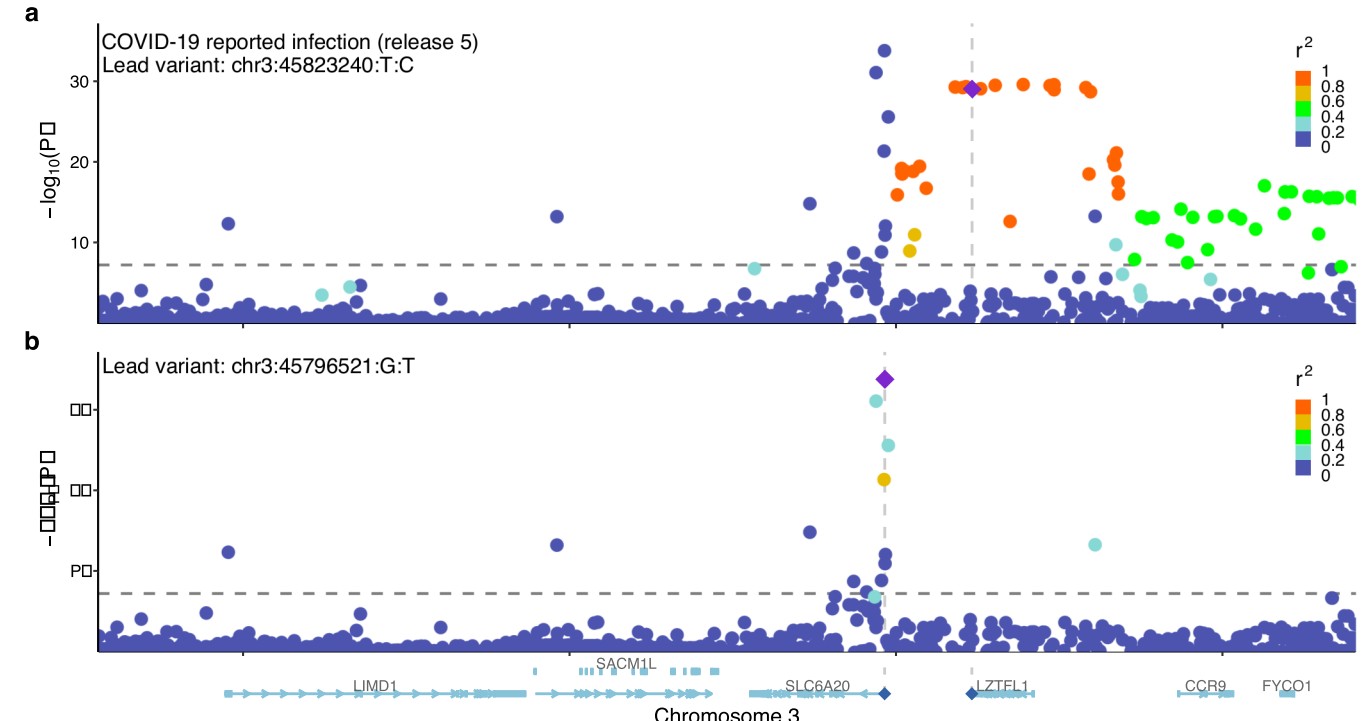

**Extended Data Fig. 3 | Locus-zoom plots of the 3p21.31 region for reported SARS-CoV-2 infection. a**, A standard plot without exclusion. Here, the severity lead variant rs10490770 (chr. 3: 45823240T:C) is shown as a lead variant. **b**, Additional independent susceptibility signal(s) after excluding variants with $r^2 > 0.05$ with rs10490770. The susceptibility lead variant rs2271616 (chr. 3: 45796521G:T) is highlighted.

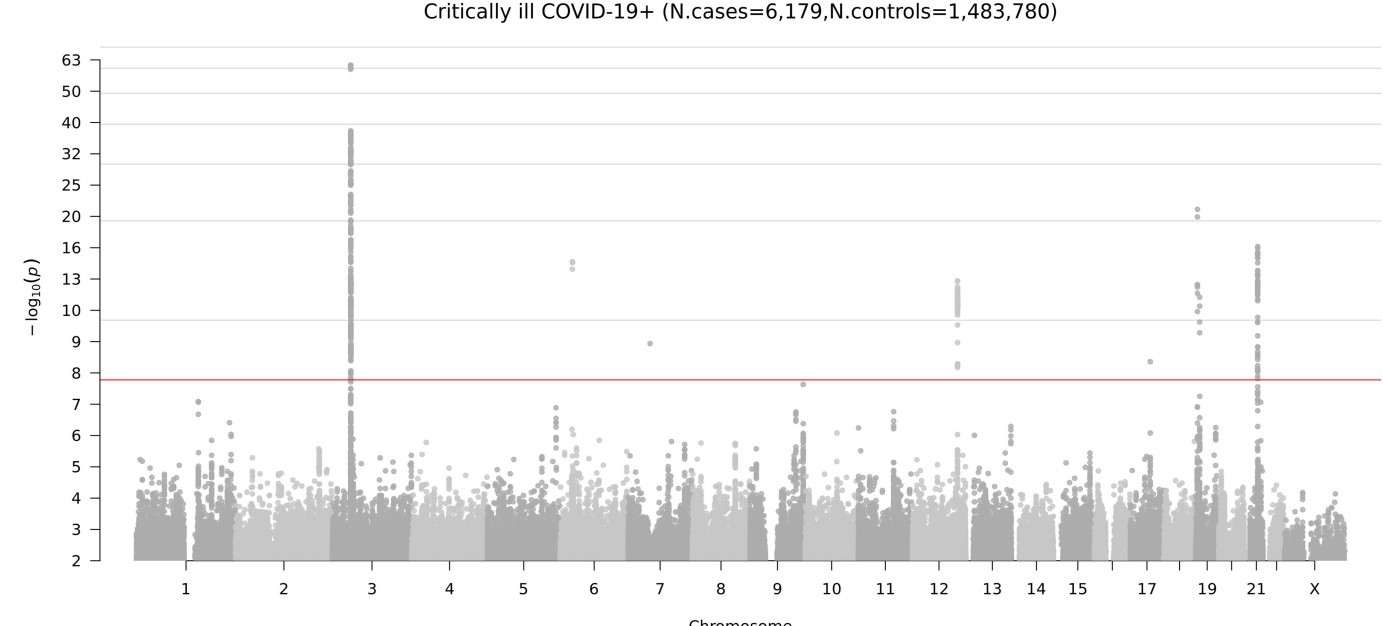

Critically ill COVID-19+ (N.cases=6,179,N.controls=1,483,780)

**Extended Data Fig. 4 | Genome-wide meta-analysis association results for critical illness due to COVID-19.** The locus on chromosome 6 is the HLA locus, which was removed from the list of reported loci in Supplementary Table 2 due to the high heterogeneity in effect size estimated between studies included in the analysis. The locus on chromosome 7 was also not reported in Supplementary Table 2 due to missingness across studies—that is, the high number of studies in the meta-analysis that did not report summary statistics for this region. There are two association peaks on chromosome 19.

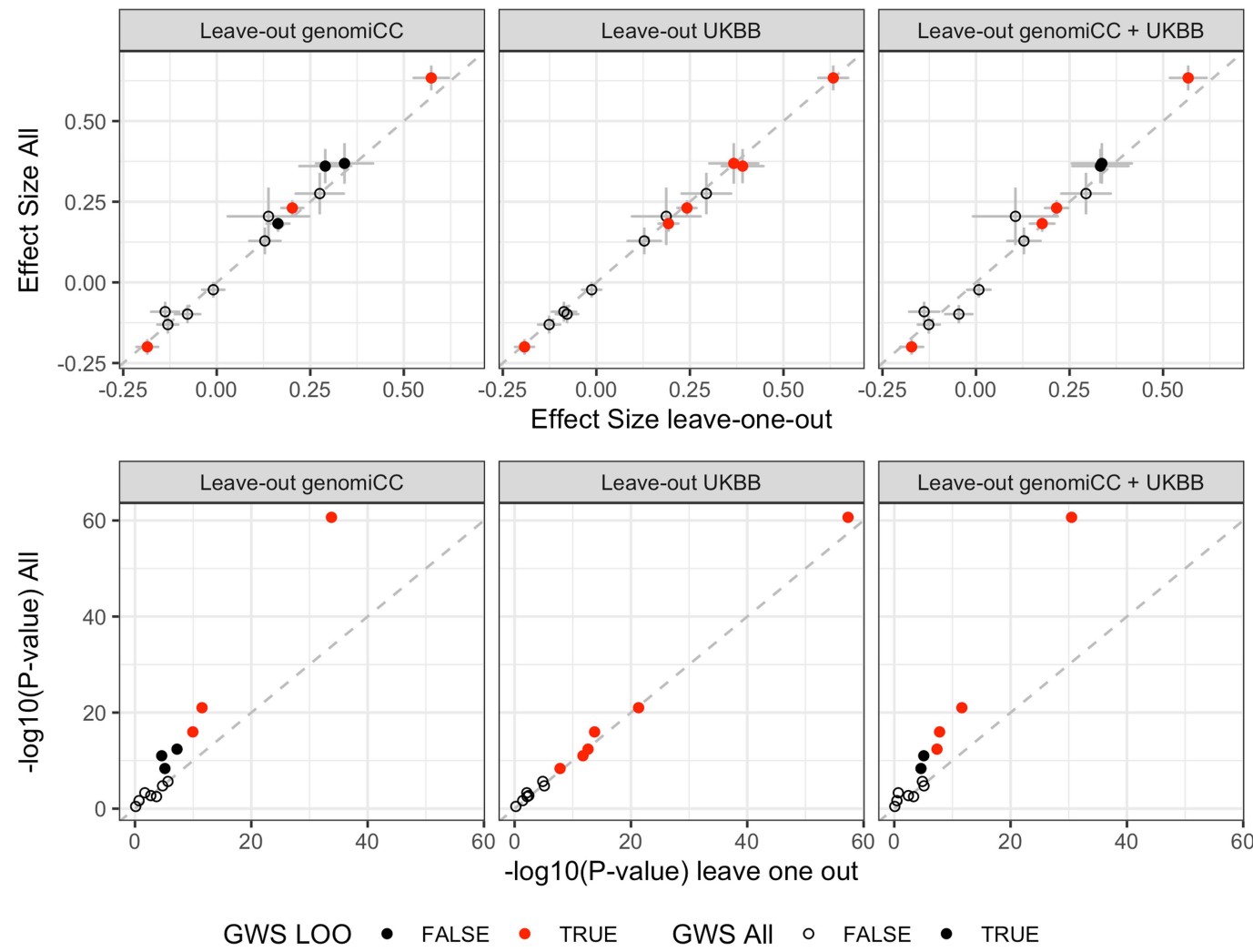

**Extended Data Fig. 5 | Sensitivity analyses for overlapping controls in genomiCC and UK Biobank.** Comparison of the beta effect sizes (top) and unadjusted *P* values (bottom) of the 13 lead variants, using data from the COVID-19 critical illness meta-analysis in all the cohorts to leaving out genomiCC (cases, *n* = 4,354; controls, *n* = 1,474,655; total, *n* = 1,479,009), leaving out the UK Biobank (UKBB; cases, *n* = 5,870; controls, *n* = 1,155,203; total, *n* = 1,161,073) and leaving out both genomiCC and UK Biobank (cases, *n* = 4,045; controls, *n* = 1,146,078; total, *n* = 1,150,123) (from left to right, respectively). Top, dots and grey bars represent the beta effect size estimates ± standard error from the corresponding GWAS meta-analysis. Bottom, dots represent two-sided *P* values from the corresponding GWAS meta-analysis. Filled dots indicate variants that showed genome-wide significance in the full meta-analysis of critical illness due to COVID-19, and empty dots represent variants that were not significant for critical illness but were significant for either hospitalization due to COVID-19 or reported SARS-CoV-2 infection. Red dots represent variants that showed genome-wide significance in the leave-one-out analysis for genomiCC, UK Biobank or genomiCC and UK Biobank.

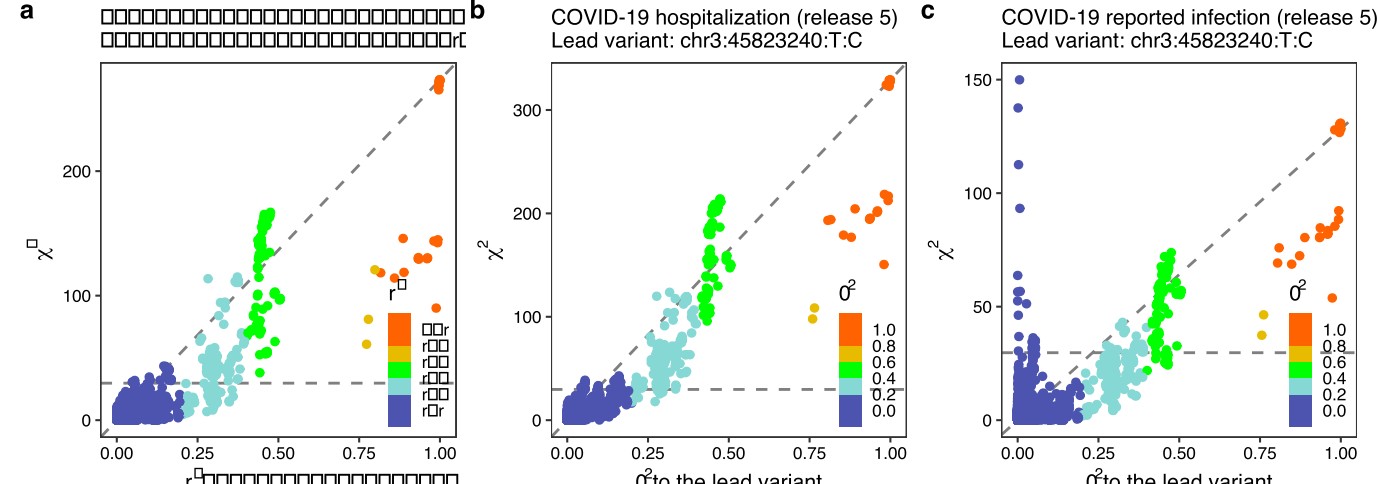

**a** COVID-19 critical illness (release 5)
Lead variant: chr3:45823240:T:C

**b** COVID-19 hospitalization (release 5)
Lead variant: chr3:45823240:T:C

**c** COVID-19 reported infection (release 5)
Lead variant: chr3:45823240:T:C

**Extended Data Fig. 6 | Comparison of $\chi^2$ statistics and $r^2$ values to the lead variant in the 3p21.31 region. a–c,** Data are shown for critical illness (**a**), hospitalization (**b**) and reported SARS-CoV-2 infection (**c**). The left blue peak in **c**, which is uncorrelated with the lead variants in the region, indicates that there are independent signals.

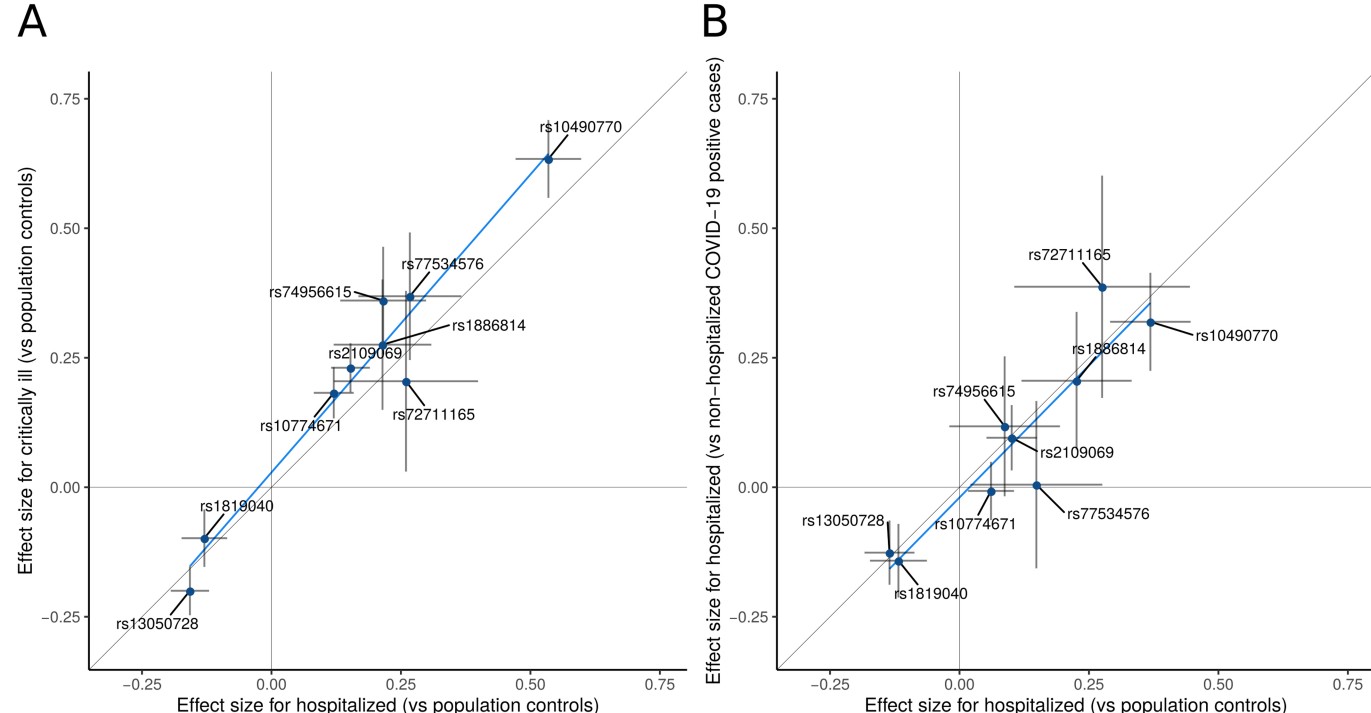

**Extended Data Fig. 7 | Comparison of the effect sizes of lead variants between pairs of COVID-19 meta-analyses.** Comparison of effect sizes for the nine variants associated with severity of COVID-19 disease. **a**, Comparing hospitalized cases of COVID-19 versus population controls ($n$ = 10,428 cases and $n$ = 1,483,270 controls) and critically ill cases of COVID-19 versus population controls ($n$ = 6,179 cases and $n$ = 1,483,780 controls). **b**, Hospitalized cases of COVID-19 versus population controls ($n$ = 5,806 cases and $n$ = 1,144,263 controls) and hospitalized cases of COVID-19 versus non-hospitalized cases of COVID-19 ($n$ = 5,773 cases and $n$ = 15,497 controls). Sample sizes for hospitalized cases of COVID-19 versus population controls differ between **a** and **b** due to differences in the sampling of studies selected for the analysis. This selection included all studies that were able to contribute data to the respective analyses that the data were compared to (shown on the $y$ axis) in each panel. Dots represent the effect size beta estimates, bars represent the 95% confidence interval of the estimates. Effect size estimates and $P$ values for heterogeneity tests (Cochran's $Q$, two-tailed test) are reported in Supplementary Table 3.

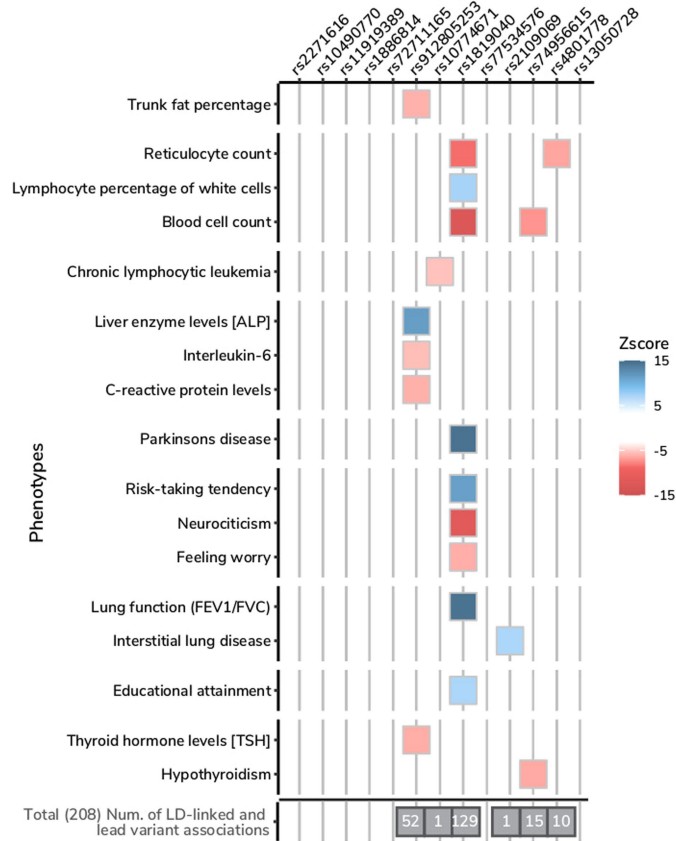

**Extended Data Fig. 8 | PheWAS for genome-wide significant lead variants.**
Selected phenotypes associated with genome-wide significant COVID-19
variants (see Supplementary Table 6 for a complete list). We report those
associations for which a lead variant from a previous GWAS result was in high
LD ($r^2 > 0.8$) with the index COVID-19 variants. The colour represents the
$z$-scores of correlated risk increasing alleles for the trait. The total number of
associations for each COVID-19 variant is highlighted in the grey box.

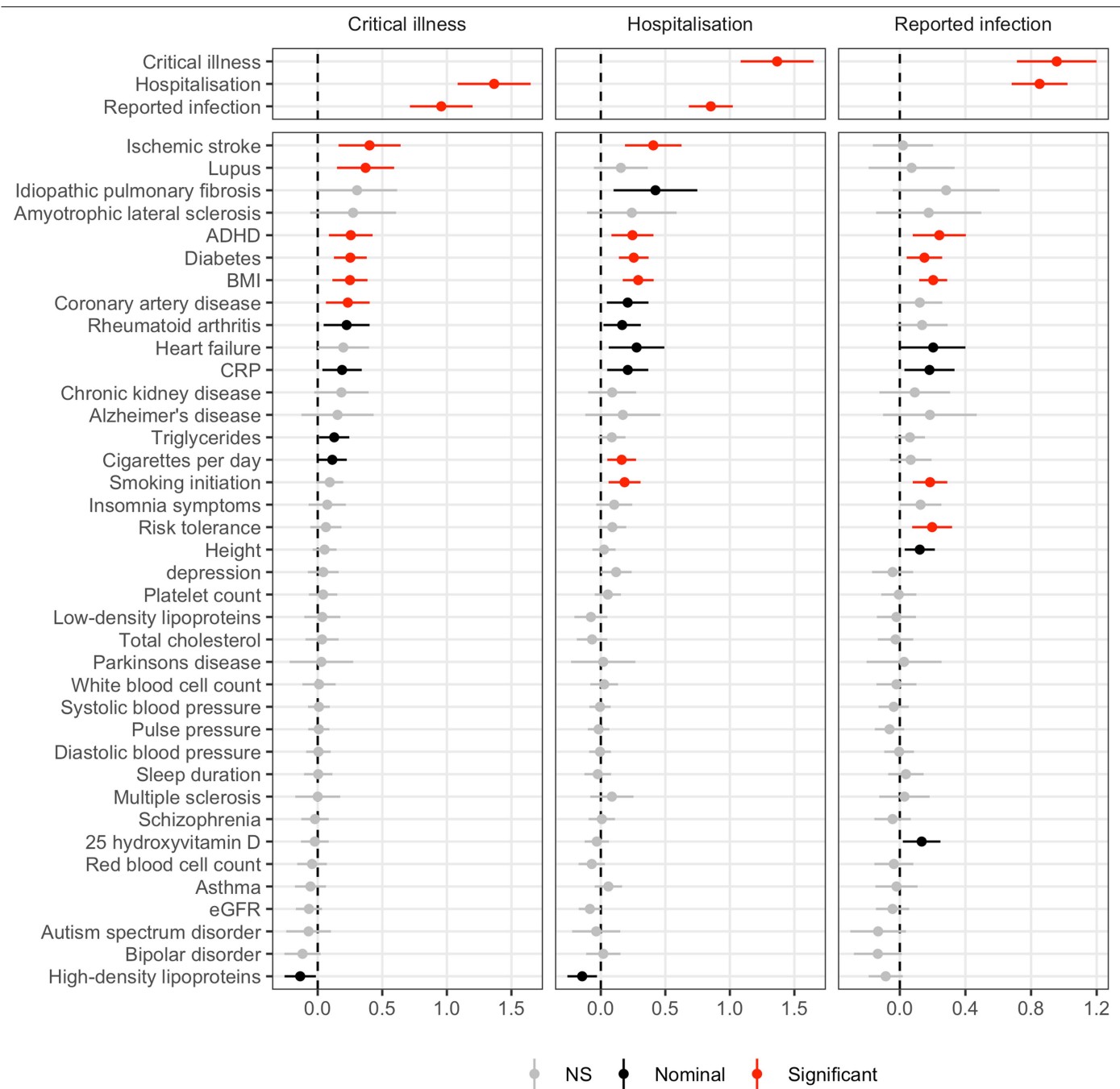

**Extended Data Fig. 9 | Genetic correlation with COVID-19 phenotypes.** Each column shows the genetic correlation results for the three COVID-19 phenotypes (European-ancestry analyses only): critical illness, hospitalization and reported SARS-CoV-2 infection. The traits that the genetic correlation is run against are listed on the left. Significant correlations (FDR < 0.05) are shown with their 95% confidence intervals in red, nominally significant correlations (P < 0.05) are in black and non-significant correlations are in grey. Two-sided P values were calculated using LDSC for genetic correlations and exact estimates, unadjusted standard errors and two-sided P values are available in Supplementary Table 11.

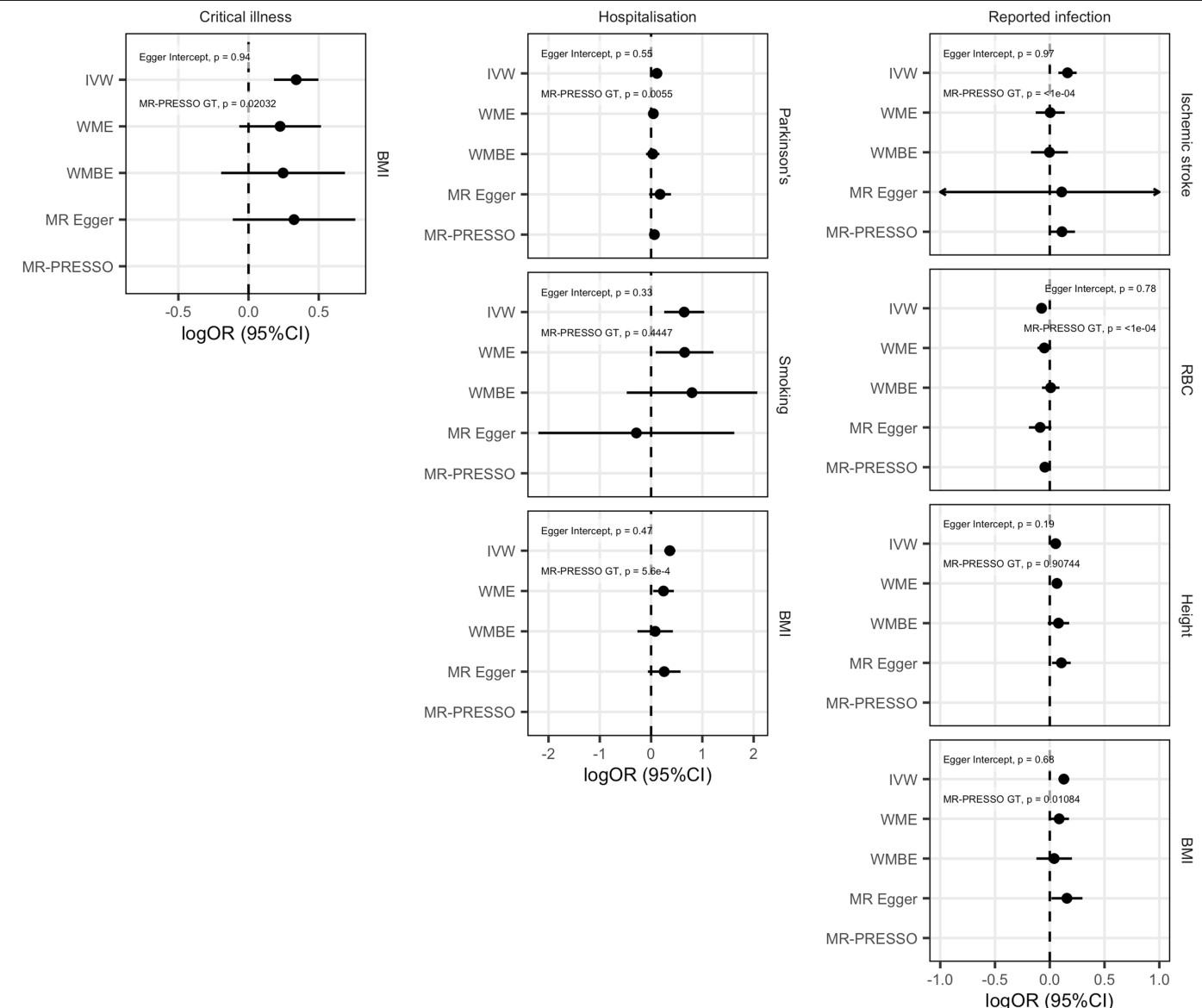

**Extended Data Fig. 10 | Mendelian randomization sensitivity analyses.**
Genetic correlations and Forest plots displaying the causal estimates for each of the sensitivity analyses used in the Mendelian randomization analysis for trait pairs that were significant at an FDR of 5%. Two-sided *P* values were estimated using IVW, WME, WMBE and MR-PRESSO analyses. RBC, red blood cell count.

# Reporting Summary

Nature Research wishes to improve the reproducibility of the work that we publish. This form provides structure for consistency and transparency in reporting. For further information on Nature Research policies, see our Editorial Policies and the Editorial Policy Checklist.

## Statistics

For all statistical analyses, confirm that the following items are present in the figure legend, table legend, main text, or Methods section.

| n/a | Confirmed | |
|---|---|---|
| ☐ | ☒ | The exact sample size (*n*) for each experimental group/condition, given as a discrete number and unit of measurement |
| ☐ | ☒ | A statement on whether measurements were taken from distinct samples or whether the same sample was measured repeatedly |
| ☐ | ☒ | The statistical test(s) used AND whether they are one- or two-sided <br> *Only common tests should be described solely by name; describe more complex techniques in the Methods section.* |
| ☐ | ☒ | A description of all covariates tested |
| ☐ | ☒ | A description of any assumptions or corrections, such as tests of normality and adjustment for multiple comparisons |
| ☐ | ☒ | A full description of the statistical parameters including central tendency (e.g. means) or other basic estimates (e.g. regression coefficient) AND variation (e.g. standard deviation) or associated estimates of uncertainty (e.g. confidence intervals) |
| ☐ | ☒ | For null hypothesis testing, the test statistic (e.g. *F*, *t*, *r*) with confidence intervals, effect sizes, degrees of freedom and *P* value noted <br> *Give P values as exact values whenever suitable.* |
| ☒ | ☐ | For Bayesian analysis, information on the choice of priors and Markov chain Monte Carlo settings |
| ☒ | ☐ | For hierarchical and complex designs, identification of the appropriate level for tests and full reporting of outcomes |
| ☐ | ☒ | Estimates of effect sizes (e.g. Cohen's *d*, Pearson's *r*), indicating how they were calculated |

*Our web collection on statistics for biologists contains articles on many of the points above.*

## Software and code

Policy information about availability of computer code

| | |
|---|---|
| Data collection | No code was used to collect data in the study. |
| Data analysis | Each individual study that contributed genetic-phenotype association summary statistics to the consortium carried out their association analyses independently of the consortium (study-specific information outlined in Supplementary Table 1). However, the consortium did release phenotyping and analysis guidelines as a recommendation (https://www.covid19hg.org/). For quality control of genotype data we recommended using the Ricopili pipeline (PMID: 31393554). For genotype phasing and imputation we recommended the TopMed Imputation Server (PMID: 27571263) or Michigan Imputation Server (PMID: 27571263). For genome-wide association study (GWAS), we recommended SAIGE (PMID: 30104761), but some studies used PLINK  (PMID: 17701901). Each study then submitted their GWAS summary statistics to the consortium for meta-analysis. <br><br> LD score regression v 1.0.1 [PMID: 25642630] was used for heritability and partitioned heritability analyses. Variants for Mendelian randomization instruments were selected using PLINK version 1.90b6.18 (PMID: 17701901). Exposure and outcome datasets were harmonized, and MR statistical analysis conducted using R version 4.0.3. with the R-package TwoSampleMR version 0.5.5 (PMID: 29846171) (which included Fixed-effects IVW analysis (PMID: 24114802), weighted median estimator (WME) (PMID: 27061298), weighted mode based estimator (WMBE) and MR Egger regression (PMID: 26050253)) and additionally MR-PRESSO version 1.0 (PMID: 29686387). <br><br> Code availability statement: The code for summary statistics liftover, projection PCA pipeline including precomputed loadings and meta-analysis are available at https://github.com/covid19-hg/ and the code for Mendelian randomization and genetic correlation pipeline at https://github.com/marcoralab/MRcovid. |

For manuscripts utilizing custom algorithms or software that are central to the research but not yet described in published literature, software must be made available to editors and reviewers. We strongly encourage code deposition in a community repository (e.g. GitHub). See the Nature Research guidelines for submitting code & software for further information.

## Data

Policy information about availability of data

All manuscripts must include a data availability statement. This statement should provide the following information, where applicable:

- Accession codes, unique identifiers, or web links for publicly available datasets
- A list of figures that have associated raw data
- A description of any restrictions on data availability

Data availability statement:
Summary statistics generated by COVID-19 HGI are available at https://www.covid19hg.org/results/r5/ and are available on GWAS Catalog (study code GCST011074). The analyses described here utilize the freeze 5 data. COVID-19 HGI continues to regularly release new data freezes. Summary statistics for non-European ancestry samples are not currently available due to the small individual sample sizes of these groups, but results for 13 loci lead variants are reported in Supplementary Table 3. Individual level data can be requested directly from contributing studies, listed in Supplementary Table 1. We used publicly available data from GTEx (https://gtexportal.org/home/), the Neale lab (http://www.nealelab.is/uk-biobank/), Finucane lab (https://www.finucanelab.org), FinnGen Freeze 4 cohort (https://www.finngen.fi/en/access_results), and eQTL catalogue release 3 (http://www.ebi.ac.uk/eqtl/).

# Field-specific reporting

Please select the one below that is the best fit for your research. If you are not sure, read the appropriate sections before making your selection.

☒ Life sciences          ☐ Behavioural & social sciences          ☐ Ecological, evolutionary & environmental sciences

For a reference copy of the document with all sections, see nature.com/documents/nr-reporting-summary-flat.pdf

# Life sciences study design

All studies must disclose on these points even when the disclosure is negative.

| | |
|---|---|
| Sample size | The consortium meta-analysed genome-wide association study (GWAS) summary statistics from any individual study that had included a minimum of n=50 cases and n=50 controls in their analysis. The cutoff at n=50 cases and n=50 controls was aimed at reducing noise to the meta-analysis, but also to be inclusive of studies that had not yet accumulated large numbers of COVID-19 patient data. No statistical calculation for adequate sample size was performed, but the results identifying multiple genomic regions at genome-wide significance threshold indicates adequate power for genetic discovery. |
| Data exclusions | Individual level phenotype and genotype data exclusions were performed by each individual study, following the consortium analysis plan recommendations (www.covid19hg.org). Possible reasons for sample exclusion included removing genetic ancestry outliers within a study (using principal components analysis), poor quality of genetic data or lack of phenotypic data for a sample.<br><br>The consortium manually examined GWAS summary statistics data submitted by each study (for each submitted analysis separately), including sample size used for analysis, allele frequency check against gnomad reference panel, and distribution of test statistics. After meta-analysis, the results were checked for heterogeneity variant effects between contributing studies, and Table 1 excludes two genome-wide significant loci that were deemed to have extremely heterogeneous effects, but these variants are reported in the released consortium summary statistics (with heterogeneity test values). |
| Replication | No replication was performed. The consortium meta-analysed GWAS summary statistics, bringing together as many studies as possible to achieve the largest possible sample size and statistical power for association. this meant that the consortium included most large studies of COVID-19 host genetics that have been performed to date, so it was not possible to perform replication analyses in external cohorts. Therefore we performed manual checks on each study contributing summary statistics before entering them into the meta-analysis. In addition, after meta-analysis, we performed a check for heterogeneity between variant association estimates across studies contributing data. This allowed us to better understand whether the variant effects differed much between individual studies. |
| Randomization | No randomization was performed because there was no allocation of samples to experimental groups. |
| Blinding | Blinding was not relevant to the study. The case status and severity of symptoms was evaluated for each sample by investigators from each study respectively. The consortium recommended using covariates to control for confounding: age + age2 + sex + age*sex + 20 principal components (obtained using genetic data) + study specific covariates (if any). The consortium meta-analysed summary statistics from these case/control studies, not individual level data. Details of which variables each study used and how the calculated PCs for their analysis are available in Supplementary Table 1. |

# Reporting for specific materials, systems and methods

We require information from authors about some types of materials, experimental systems and methods used in many studies. Here, indicate whether each material, system or method listed is relevant to your study. If you are not sure if a list item applies to your research, read the appropriate section before selecting a response.

## Materials & experimental systems

| n/a | Involved in the study |
|-----|------------------------|
| ☒ | ☐ Antibodies |
| ☒ | ☐ Eukaryotic cell lines |
| ☒ | ☐ Palaeontology and archaeology |
| ☒ | ☐ Animals and other organisms |
| ☐ | ☒ Human research participants |
| ☒ | ☐ Clinical data |
| ☒ | ☐ Dual use research of concern |

## Methods

| n/a | Involved in the study |
|-----|------------------------|
| ☒ | ☐ ChIP-seq |
| ☒ | ☐ Flow cytometry |
| ☒ | ☐ MRI-based neuroimaging |

## Human research participants

Policy information about studies involving human research participants

| | |
|---|---|
| Population characteristics | Summary statistics from 46 independent studies were included in consortium meta-analyses. Mean age of cases across studies was 55.3 years. The effective sample size for genetic ancestry populations was: n=11,598 Middle Eastern; n=28,918 South Asian; 43,332 East Asian; 48,714 African; 70,902 Ad-mixed American; 738,538 European. Population characteristics regarding age, sex and exact case and control sample numbers for each contributing study are given in Supplementary Table 1. |
| Recruitment | The consortium pre-defined phenotype criteria for cases and controls, but the specific recruitment was carried out independently by each contributing study. COVID-19 disease status (critical illness, hospitalization status) was assessed following the Diagnosis and Treatment Protocol for Novel Coronavirus Pneumonia (PMID: 32358325). The critically ill COVID-19 group included patients who were hospitalized due to symptoms associated with laboratory-confirmed SARS-CoV-2 infection and who required respiratory support or whose cause of death was associated with COVID-19. The hospitalized COVID-19 group included patients who were hospitalized due to symptoms associated with laboratory-confirmed SARS-CoV-2 infection. The reported infection cases group included individuals with laboratory-confirmed SARS-CoV-2 infection or electronic health record, ICD coding or clinically confirmed COVID-19, or self-reported COVID-19 (e.g. by questionnaire), with or without symptoms of any severity. Genetic ancestry-matched controls for the three case definitions were sourced from population-based cohorts, including individuals whose exposure status to SARS-CoV-2 was either unknown or infection- negative for questionnaire/electronic health record based cohorts. |
| Ethics oversight | Ethical statements for each contributing study are given in Supplementary Table 1. |

Note that full information on the approval of the study protocol must also be provided in the manuscript.

