## [Peer Review File · Nature]

Manuscript Title: Mapping the human genetic architecture of COVID-19

Editorial Notes:

Redactions – Third Party Material

Redactions – confidential information

Parts of this Peer Review File have been redacted as indicated to maintain the confidentiality of information from personal communication

Reviewer Comments & Author Rebuttals

Reviewer Reports on the Initial Version:

Referee #1 (Remarks to the Author):

TITLE

Mapping the human genetic architecture of COVID-19 by worldwide meta-analysis

AUTHORS

The COVID-19 Host Genetics Initiative

SUMMARY

The authors describe a GWAS meta-analysis of three COVID phenotypes (infection, hospitalisation & critical illness) including multiple ancestries (predominantly Europeans) and perform multiple downstream post-GWAS analyses to characterise the genetic architecture of COVID phenotypes and link to biological function. They acknowledged study limitations particularly in phenotype heterogeneity and hence, results should be interpreted with caution. I would recommend the article for publication given it is a well-run and through study which will add value to the field. They also highlight the value of international collaboration to streamline analyses and rapidly share knowledge particularly in pandemics. However, it is unclear what elements are particularly novel, some genetic loci have previously been reported (eg. <https://doi.org/10.1038/s41586-020-03065-y>), as have links between, BMI, smoking and disease severity (eg. <https://www.ahajournals.org/doi/10.1161/CIRCULATIONAHA.120.050753>). The contribution of genetics to phenotypes is also hard to quantify/interpret given such low SNP heritability reported. I suggest a number of adjustments (outlined below) that I feel would improve the clarity and confidence of findings in the manuscript.

Comments

1. The use of “worldwide” in the title/text is misleading given the relatively small number of non-European ancestry compared to Europeans, additionally, most downstream analyses has been presented using European data and thus cannot be fully representative of a “worldwide” sample.
2. In the abstract line 19 the N of 49,562 is misleading as only one phenotype had that sample size. Kindly rephrase for clarity.

3. In the introduction, please add relevant reference(s) to line 50 following "...most robust finding at locus 3p21.31".
4. Genome-wide significance needs to be adjusted. The threshold of $P < 5 \times 10^{-8}$ is no longer suitable, particularly given differences in LD patterns across populations and use of modern genotyping arrays plus imputation ref panels a more stringent and realistic threshold of at least $P < 1 \times 10^{-8}$ is recommended to declare association signals.
5. The authors don't clearly acknowledge previously identified GWAS loci associated with COVID phenotypes. In the results and discussion, it would be useful to highlight/distinguish novel loci from those that have been identified in recent COVID GWASs (eg <https://doi.org/10.1038/s41586-020-03065-y>). If previously identified loci have not been replicated, suggestions should be made as to why.
6. In the results section entitled "Comparison of effect for genome-wide significant results across studies and phenotype definition" the authors display effect sizes (assuming beta) in extended figures and supplementary however report odds ratios in some instances in text (eg. lines 118-126) which is slightly confusing. I would suggest for consistency to report and display ORs and 95% in C.I. in text, tables and plots respectively, which is the standard for a case-control study.
7. In extended data figure 2, why is the sample size for cases and controls in hospitalized COVID-19 cases vs population controls different for A and B?
8. This statement in lines 136-138 "We noted that two loci, tagged by lead variants rs1886814 and rs72711165, were identified primarily from East Asian genetic ancestry samples (n = 1,414 cases hospitalized due to COVID-19) with minor allele frequencies in European populations being < 3%." is unclear. Were these loci not significant in other populations? Is there allelic heterogeneity occurring for variants in the loci across ancestries?
9. In line 139 "Another locus at 3p21.31..." Do you mean "another variant(s)" in this locus? If so, please rephrase and also add the variant id.
10. The paragraph lines 141-146 is unclear: when you say effect of this locus do you mean odds ratio/effect size of specific variants across this locus or was this a locus-based test? Was this effect significantly different in other ancestries?
11. With regards to phenotype definition of controls (lines 148), if information is available on exposure to cases/suspected cases, a suggestion would be to use that as a proxy for exposure to refine control status as the controls are quite a noisy phenotype.
12. The Figure 2 legend states "Eleven loci highlighted in yellow", only 9 loci are in yellow, kindly adjust. If the loci in green are associated with infection, what about the significant loci in grey? This is slightly confusing.
13. In table 1 results are presented from all GWAS but in the figure 2 the critical illness phenotype is omitted. I would suggest the addition of Manhattan plot to main text or supplementary and also of QQ plots for all studies to supplementary.
14. Unclear what genetic associations are being referred to (lines 369-370), also previous publications need to be cited.
15. The extremely low heritability ($h^2 \sim 0.002$) reported is intriguing. Heritability of COVID19 critical illness has been previously reported by Pairo-Castineira & Colleagues (reference above), why is the heritability so different here?
16. Statement in lines 373-375, conclusion is not robust given overall heritability is extremely low $\sim 0.02\%$.
17. For all tables/figures, kindly add SEs and 95% C.I.s for betas and ORs reported, respectively.

Referee #2 (Remarks to the Author):

In this manuscript, Ganna and colleagues at the COVID-19 host genetic initiative present the result of a multi-ethnic meta-analysis of three COVID-19 phenotypes related to susceptibility to SARS-CoV-2 infection and disease progression/severity. Controls are from public databases in whom the infection status is not ascertained. The authors report 15 genome-wide significant loci in testing the three traits.

The authors use heterogeneity test to show that potentially different genomic loci are involved in susceptibility to infection vs. disease severity progression. They use in silico analysis approaches to identify the plausible target gene(s) for each locus and speculate about the functional relevance of the identified loci. Finally, the authors look into the association of COVID-19 risk loci with other traits, investigate the genetic heritability of the three COVID-19 susceptibility phenotypes, test the genetic correlation between these phenotypes and other relevant phenotypes, and look into the causal relationship between these other phenotypes and susceptibility to COVID-19.

This manuscript tackles an important problem. Understanding the genetic basis of COVID-19 can lead to a better understanding of COVID-19 disease mechanism, help patient risk stratification, and help improve clinical care. This work also adds to existing evidence about the genetic basis of infectious diseases and underlies the importance of large, multi-ethnic cohorts and international collaborations in human genomics of infectious disease research. Despite these, I found the paper unsuitable for publication in its current form. My main concerns include the lack of sufficient correction (or even discussion about the role of) potential, non-genetic confounders that can derive the results. I was disappointed to see that, little effort was dedicated to exploring/leveraging the multi-ethnic aspect of this cohort. A lost opportunity, given the lack of genetic studies in non-European populations and the high burden of COVID-19 in these populations. Finally, some crucial information type definition about study design and phenotype definition is missing.

Study design:

- 1- Given the multi-layer and parallel analyses a figure to explain the study design and the case/control comparisons that are performed is needed to make following the paper easier.
- 2- In figure 1, the ancestry breakdown for the whole cohort is shown but not for each GWAS? It worth knowing whether there are major differences in ancestry composition of cases for each GWAS analysis and if the differences in ORs/results can be driven by differences in cohort genetic composition, LD, or non-genetic factors that correlate with genetic ancestry.
- 3- In line with the above point is it possible to provide figures/supplementary figures of the PCA analyses for the combined cases and cases and controls for each phenotype? Projecting the PCs for each dataset to a common PC space (1KG for example) will allow visual comparisons between different analyses.
- 4- The phenotype definition presented in the paper is not detailed enough, given that heterogeneity in phenotype definitions between cohorts can lead to spurious findings it helps the reader judge the validity of the results to know the case phenotype definition (Table S1) and the harmonization method.
- 5- The answer to some of the points above might be available in the CHGI website but this flagship paper should provide enough information for the reader to be able to judge the validity of the methods and result. Adding more details can also help future similar studies to use this paper as a guideline.
- 6- The methods indicate: "Standard error values as a function of effective sample size was used to find studies which deviated from the expected trend." It would be helpful to provide a figure for this analysis.

GWAS analysis:

- 1- There is no correction (or no discussion around) the role of sociodemographic and environmental factors (or their interaction with genetic factors) in susceptibility to SARS-CoV-2 exposure and progression. Is it possible that the differences observed between phenotypes and/or cases and controls are, at least in part, driven by these non-genetic factors?
- 2- Extended figure 2 shows that restricting the analyses to infected controls does not affect the OR of the 15 loci described in the manuscript. How do the genome-wide results change if you restrict the analyses to infected controls? Are there any new associations (apart from the reported 15 loci)? How do the genome-wide Z-scores correlate between the analyses that I suggest here and GWAS for 1- critical illness vs. population controls or 2- hospitalized vs. population control? Such a comparison can alleviate some of the concerns about the impact of confounding factors related to exposure status.
- 3- Extended figure 2A also shows that rs72711165's OR is higher in hospitalized than severe cases. If this SNP is associated with severity, as the authors claim, shouldn't the OR increase with the more carefully defined and more severe phenotype (i.e critical illness)?

- 4- rs72711165's OR increases further when controls are restricted to infected controls. This change can be the result of a reduced noise (unlikely, as the authors claim because we don't see the same increase for other loci in Extended figure 2B), changes in ancestry composition and the effect allele frequency (for example having proportionally less East Asians in the infected controls vs population controls), or changes in non-genetic confounders that affect COVID risk and correlate with genetic ancestry. Comparing rs72711165's MAF in population controls and infected controls can help distinguish between the two latter possibilities. If the MAF is the same my worry is that other findings can also be biased due to factors not captured by the covariates included in the association analyses.
- 5- It is a shame that no formal fine-mapping is performed for the GWAS loci. The multi-ethnic nature of this work can be leveraged for fine-mapping (compared to a single ancestry study) and help to find potential causal variants (which as the authors know is not necessarily the index variant)?
- 6- The paper, does not present any conditional analysis for the identified loci. This is a basic analysis and may help to find additional association signals.
- 7- Please add the figure for the third GWAS analysis to fig 2.
- 8- Another potential confounding factor is the time of infection. New reports show a considerable decrease in in-hospital mortality rate that is largely attributable to non-genetic factors (here is a summary: <https://www.nature.com/articles/d41586-020-03132-4>). So people with the same genetic predisposition can have largely different outcomes depends on when they got infected. I honestly don't know how to account for this (unless you have the hospitalization dates) but it worth considering such confounding factors in the interpretation of the results. At a minimum, this should be listed as a limitation in the discussion.
- 9- There is no signal at the HLA locus, is HLA excluded from the analyses? If yes there is no mention of this in the methods section. If no, given the role of this locus in infectious diseases please expand the methods to include HLA typing and analysis and consider expanding the discussion to speculate why no HLA signal was observed.
- 10- The genome-wide significance threshold needs adjustment for the number of traits tested.

Multi-ethnic nature of the work:

- 1- What are the ancestry-specific ORs/Z-scores for the reported loci? Two SNPs were mentioned with variable MAF in different populations. The ancestry-dependent heterogeneity in MAF, OR for other loci are not discussed. A table to show the MAF and ORs of variants among different genetic ancestries and formal heterogeneity test is needed and can help the downstream utility of this study.
- 2- It was shown recently that the TYK2 P1104A variant, involved in susceptibility to TB, went under negative selection in the European population (Kerner et al, AJHG, 2021), doo the authors see any difference in the frequency and OR of this variant among different ethnicities?

Heritability, genetic correlation, and in silico functional analyses:

- 1- If you perform the analyses for critical and hospitalized COVID-19 using infected controls only, do you see any change in heritability estimates?
- 2- What is the intercept for each phenotype in your LDSC heritability analyses.
- 3- Please include the figures for heritability and genetic correlation analyses in the manuscript/supplement. Beyond the presented numbers, seeing these figures will help the reader judge the potential inflations/abnormalities in the data.
- 4- If you use a colocalization test such as COLOC for PheWAS GWAS colocalization analysis does your colocalization results change?
- 5- In which tissues the candidate genes listed in table 1 are expressed (for example in GTEx)?

Manuscript text:

- 1- Methods lack information about how "ancestry matched" controls were matched and selected.
- 2- Similarly, methods for effective sample size calculation is missing.
- 3- There is no mention of previous work of human genomics of respiratory viruses (influenza, RSV, etc.) in the paper. It worth expanding the discussion to previously acknowledge the previous work and compare the current manuscript's findings with what is known regarding human genomics of

susceptibility respiratory viruses.

4- In the discussion, there is a reference to the FOXP4 locus and its frequency in different ancestral groups (lines 381 – 385). The information about the frequency of this locus in different groups (admixed Americans, middle eastern, etc,) is not present in the text.

5- Lines 369-379 of discussion lack any reference to previous work.

6- The wording for the different phenotypes used in the paper needs more clarification. According to table 1, the three phenotypes are referred to as 1- critical illness, 2- hospitalized, and 3- reported infection. Throughout the paper, there are multiple references to the “severity” phenotype but this phenotype is not clearly defined in the text (I understand 1+2 vs 3 is the severity but it needs to be stated somewhere in the main text).

7- The last sentence of the abstract seems to miss a word (provides maybe?).

8- Table 1 legend mentions V2G column but this column doesn't exist in the table.

9- The authors need to carefully check the manuscript to keep the wording consistent (e.g SARS-CoV-2 in the main text vs. SARS Cov 2 in the discussion)

10- Line 328 is it p or FDR?

Referee #3 (Remarks to the Author):

Summary: The authors describe the flagship analysis of the COVID-19 HGI assessing the human genetic component in determining susceptibility to SARS-CoV-2 infection and COVID-19 severity through a large, multi-ethnic meta-analysis of genome-wide data. Combining data from 46 studies across 19 countries the consortium amassed a sample size of ~50,000 COVID-19 cases and 2 million, mostly population-level, controls, making it by far the largest genomic study of an infectious disease. Using 3 a priori defined phenotypes, they identify 15 genome-wide significant loci with consistent effect sizes across cohorts (i.e. no significant heterogeneity). Of these, 4 loci are deemed to be associated with enhanced susceptibility while the remaining 11 associate with severity after infection. At each of these loci, the authors apply various heuristic gene prioritization approaches to implicate causal genes and investigate the variant impact on additional traits using a PheWAS approach, implicating lung dysfunction in COVID-19 severity. Finally, the authors perform a series of analyses looking at genetic correlation between COVID-19 severity and other traits, the most striking of which are causal relationships between increased BMI and smoking and COVID-19 severity. Overall this is very well performed study of a critical issue in global health. The scope of the study and the speed with which it was performed are, to my mind, unparalleled and the consortium organizers and participants are to be commended. I do however have several suggestions to be considered prior to publication.

1) As the authors correctly point, out the varying study designs and ascertainment make combining these studies in such a way as to avoid all confounding difficult. I feel that adding some supplementary information or extended data figures describing how the ancestry matching was performed and how the test statistics were calibrated per study would be beneficial. Presentation of PCA plots and test statistic distributions and inflation factors for each cohort and the full meta-analysis could address this.

2) More detail should be provided on the decision to remove 2 associated loci based on “extreme heterogeneity across contributing studies” but to retain other loci with “significant heterogeneous effects ... likely reflecting heterogeneous ascertainment of cases across studies”. In particular, the 3p21.31 hit (reported previously and represented here by rs10490770) does have strong evidence for heterogeneity, in particular in the reported infection analysis. Is there a reason this is expected at this locus and can the authors comment on how the heterogeneous ascertainment of cases doesn't impact the other associated loci, or indeed the full genome-wide results.

3) Given the multi-ethnic nature of the study, were sub-analyses performed to identify any ancestry-specific effects that may not have been observed in the full analysis? The authors point out a few loci with stronger effects in Asian ancestry samples and that the 3p21.31 haplotype is most common in

Bangladeshi populations but what about variants that are exclusive to ancestry groups? Would these be captured in the present analysis and is it possible that population-specific causal variants would be missed when analyzing tag SNPs across a multi-ethnic sample?

4) I also find the analysis of the effect at 3p21.31 in the Bangladeshi population lacking. It appears that this analysis is simply testing for association at a single proxy variant to provide additional replication of the association. If the claim is that the causal variants at this locus should be more pronounced due to increased frequency in this population then wouldn't a more in-depth analysis of haplotype structure, presence of rare, coding variants including a conditional analysis be more informative?

5) Although full functional characterization of the associated loci will likely require years of dedicated research and is beyond the scope of this report, I do feel some additional analyses could be done with the data on hand to help guide functional follow-up studies. Firstly, for loci with two or more protein altering variants linked to the lead SNP a conditional analysis could be performed to determine if the combination of functional variants statistically out-performs the lead SNP. This could also help to understand how many independent signals are expected at each associated locus. For the eQTL analysis including GTEx and the Lung eQTL Consortium, a full colocalization analysis (using eCAVIAR or something similar) is warranted. I would also be interested to know if the overlap of effects on lung expression and COVID-19 severity go in an expected direction? And if so, is there evidence that other expression-influencing variants also impact COVID-19 severity? This is very hard for me to parse from Table S11 and would be of interest to labs following up on these genes.

6) As I mentioned in my summary, I believe the authors and organizers are to be commended for this global effort. However, I do think that the lack of inclusion of any groups or cohorts from continental Africa should be commented on. Although I don't think any one study can solve the continued lack of diversity in genomics, it would be useful for the authors include a comment highlighting the relative benefit of prioritizing inclusion of African and other non-European ancestry populations in further efforts. Particularly given the large potential for ancestry-specific genetic effects and the apparent reduced severity of COVID-19 across Africa.

A few minor comments are included here:

Line 58, what is meant by "variable ascertainment strategies" this is worth expanding on in Methods
Should Table 1 contain a V2G column? Seems so from the legend but didn't come through in the manuscript pdf

Typos throughout, a few examples:

Line 26 typo "international collaboration *is* a blue- print"

Line 238 typo "Lastly, there are *six remaining* loci"

Line 355 typo "to to"

Line 428 clarify "and with the enrollment of vaccines"

Author Rebuttals to Initial Comments:

Referee #1

Summary: The authors describe a GWAS meta-analysis of three COVID phenotypes (infection, hospitalisation & critical illness) including multiple ancestries (predominantly Europeans) and perform multiple downstream post-GWAS analyses to characterise the genetic architecture of COVID phenotypes and link to biological function. They acknowledged study limitations particularly in phenotype heterogeneity and hence, results should be interpreted with caution. I

would recommend the article for publication given it is a well-run and through study which will add value to the field. They also highlight the value of international collaboration to streamline analyses and rapidly share knowledge particularly in pandemics. However, it is unclear what elements are particularly novel, some genetic loci have previously been reported (eg. Pairo-Castinerira et al. 2020. Nature), as have links between, BMI, smoking and disease severity (eg. Ponsford et al. 2020. Circulation). The contribution of genetics to phenotypes is also hard to quantify/interpret given such low SNP heritability reported. I suggest a number of adjustments (outlined below) that I feel would improve the clarity and confidence of findings in the manuscript.

Comments

1. The use of “worldwide” in the title/text is misleading given the relatively small number of non-European ancestry compared to Europeans, additionally, most downstream analyses has been presented using European data and thus cannot be fully representative of a “worldwide” sample.

Answer: We would first like to thank reviewer 1 for their thorough comments and highly constructive feedback.

We agree with the reviewer that the current data freeze includes mostly European data. Nonetheless we are expanding to more non-US, non-EU countries and we expect the "worldwide" aspect of our initiative to be more relevant in the following data freezes. Given the stylistic nature of the comment, we will prefer to defer to the Editor's decision on this.

2. In the abstract line 19 the N of 49,562 is misleading as only one phenotype had that sample size. Kindly rephrase for clarity.

Answer: We thank the reviewer for pointing out this misleading phrasing, and have corrected the wording on page 1:

We describe the results of three genome-wide association meta-analyses comprising up to 49,562 COVID-19 patients from 46 studies across 19 countries worldwide.

3. In the introduction, please add relevant reference(s) to line 50 following “...most robust finding at locus 3p21.31”.

Answer: We have added the references on page 1: ¹⁻⁵

4. Genome-wide significance needs to be adjusted. The threshold of $P < 5 \times 10^{-8}$ is no longer suitable, particularly given differences in LD patterns across populations and use of modern genotyping arrays plus imputation ref panels a more stringent and realistic threshold of at least $P < 1 \times 10^{-8}$ is recommended to declare association signals.

Answer: We understand the reviewer's point of view on this matter, and have considered which threshold might be best to use here. We feel that there is no definitive consensus on what the threshold should be adjusted to⁶, and considering that the majority of our samples are still of European ancestry, we decided that a compromise approach would be to divide the typical threshold used in the field by the three COVID-19 phenotypes that we test here: $P < (5 \times 10^{-8})/3$. This threshold, 1.67×10^{-8} is similar to what the reviewer recommends and consistent with the recommendation from **reviewer 2, "GWAS analysis" section comment**

10. We have edited the manuscript text on page 3:

Across our three analyses, we reported a total of 13 independent genome-wide significant loci associated with COVID-19 ($P < 1.67 \times 10^{-8}$; threshold adjusted for multiple trait testing).

This new threshold removes three loci from the genome-wide significant list on chromosomes 1, 2 and 5. We have edited the manuscript, tables and figures accordingly. However, we want to note here that during the review process we identified a new independent signal - associated with susceptibility, not severity - at the chromosome 3 locus. We have extensively discussed these new findings in **reviewer 2, "GWAS analysis" section comment 6**. In total, with these new variants, we have 13 genome-wide significant loci at $P < 1.67 \times 10^{-8}$ threshold.

5. The authors don't clearly acknowledge previously identified GWAS loci associated with COVID phenotypes. In the results and discussion, it would be useful to highlight/distinguish novel loci from those that have been identified in recent COVID GWASs e.g.⁴. If previously identified loci have not been replicated, suggestions should be made as to why.

Answer: We acknowledge that this has been common practice in GWA studies, but in this instance we decided not to include this type of discussion. Because the identification of GW significant loci is based on a statistical threshold, we feel that claims over having "discovered" new loci are misplaced and based on a somehow arbitrary dichotomy. Instead, we prefer to focus on delivering the largest and most robust analysis on the genetic architecture of COVID-19. Thus, we aim to reduce speculative results (see also comment **reviewer 2 "GWAS analysis" point 5** and **reviewer 2 "Heritability, genetic correlation, and in silico functional analyses" point 5** on follow-up in-silico analysis) or discussions. In

this spirit, we appreciate the reviewer's suggestion to use a more stringent P-value threshold. There are many reasons for why some previously reported loci might not be significant in this analysis or vice versa: differences in sample size, phenotype definitions for cases or controls, and others. This disease only emerged a year ago, and therefore sample sizes in previous studies have been low, and phenotypic data has not been easy to collect, and therefore differences in findings between studies are to be expected. Additionally, our meta-analysis includes all of these previous studies (incl. Pairo-Castineira & Colleagues), and therefore we feel that emphasizing replication/nonreplication or distinguishing between 'old' and 'new' associations in our discussion is not appropriate.

6. In the results section entitled "Comparison of effect for genome-wide significant results across studies and phenotype definition" the authors display effect sizes (assuming beta) in extended figures and supplementary however report odds ratios in some instances in text (eg. lines 118-126) which is slightly confusing. I would suggest for consistency to report and display ORs and 95% in C.I. in text, tables and plots respectively, which is the standard for a case-control study.

Answer: We agree with the reviewer and have attempted to report odds ratios where possible throughout in our main text and Table 1, but as the reviewer notes, we are missing CI's in Table 1 and we have added these now.

In our PheWas section in the main text, we had also twice reported a beta-estimate for the non-COVID-19 trait's effect by mistake, and have now changed these to OR[95% CI].

In supplementary tables and figures we report a betas (e.g. comparison of effect sizes to distinguish severity vs. susceptibility locus; Supplementary Table 2, or effect of using virus-exposed controls; Supplementary Figure 7B) because this allows us to more easily compare the effect sizes between analyses.

7. In extended data figure 2, why is the sample size for cases and controls in hospitalized COVID-19 cases vs population controls different for A and B?

Answer: The phenotype used in panels A and B is indeed the same, but the difference in sample size comes from how we selected the subset of studies to include in these analyses. In panel A, we are including only studies that also contributed to the analysis of critically ill COVID-19 phenotype (not all studies were able to distinguish hospitalized and critically ill OR they did not have critically ill patients). In panel B, we are including studies that were able to contribute data also to the analysis using controls whose exposure to the virus was known (i.e. infected by no hospitalization). In summary, the study selection for analyses in panels A and B had different criteria depending on the availability of information for the

analysis presented on the y-axis. The number of studies that had information available on the virus exposure status of their controls was very limited, and therefore, panel B analyses have lower sample sizes than panel A.

We have attempted to clarify this by adding to the Extended Data Figure 2 (new order #7) legend:

Sample sizes for hospitalized COVID-19 cases vs population controls differ between panels A and B due to difference in the sampling of studies selected for the analysis. This selection included all studies that were able to contribute data to the respective analysis that the data were compared to (on the y-axis) in each panel.

8. This statement in lines 136-138 “We noted that two loci, tagged by lead variants rs1886814 and rs72711165, were identified primarily from East Asian genetic ancestry samples (n = 1,414 cases hospitalized due to COVID-19) with minor allele frequencies in European populations being < 3%.” is unclear. Were these loci not significant in other populations? Is there allelic heterogeneity occurring for variants in the loci across ancestries?

Answer: We thank the reviewer for pointing this out, and we have worded more clearly that our aim was to highlight that the allele frequencies of some of the lead variants are higher in non-European populations. The loci are also significant in the European population, but despite the low sample size in non-European populations, e.g. the variant rs1886814 has an OR significantly different from 1 in South East Asian (OR[95%CI]= 1.54[1.12-2.11], P-value=0.008). Our power for genetic analysis in non-European populations is still too low for confidently testing heterogeneity across ancestries, but we wanted to highlight how diverse populations are important to contributing data.

We noted that two loci, tagged by lead variants rs1886814 and rs72711165, had higher allele frequencies in South East Asian (rs1886814, 15%) and East Asian genetic ancestry (rs72711165, 8%) whilst the minor allele frequencies in European populations were < 3%. This highlights the value of including data from diverse populations for genetic discovery, but challenges over inter-cohort heterogeneity and lack of appropriate LD reference for multi-ancestry analysis remain, therefore hindering proper fine-mapping and conditional analysis in these data.

9. In line 139 “Another locus at 3p21.31...” Do you mean “another variant(s)” in this locus? If so, please rephrase and also add the variant id.

Answer: We decided to remove this section altogether because this was causing confusion amongst the reviewers. Please also see our response to the question below (**comment 10**).

10. The paragraph lines 141-146 is unclear: when you say effect of this locus do you mean odds ratio/effect size of specific variants across this locus or was this a locus-based test? Was this effect significantly different in other ancestries?

Answer: Our aim was not to discuss effect size differences between populations, but just to provide a demonstrative example of how the power for association discovery is higher in the Bangladeshi population compared to European populations due to higher allele frequency among individuals of Bangladeshi ancestry. However, because we have discussed these themes already at page 4 of the manuscript and because this sentence was causing confusion amongst the reviewers, we have decided to remove this particular section.

11. With regards to phenotype definition of controls (lines 148), if information is available on exposure to cases/suspected cases, a suggestion would be to use that as a proxy for exposure to refine control status as the controls are quite a noisy phenotype.

Answer: We thank the reviewer for this idea. Unfortunately, most studies that are part of the Initiative do not have access to this type of data. However, for some studies there could be an opportunity to gather such information through e.g. questionnaires such as the AncestryDNA study has done ⁷ to collect information about exposure status of controls. The AncestryDNA dataset study was able to ask participants whether they had tested positive or negative in the study but also collected information about exposure through community or to individuals who had been diagnosed with COVID-19. They could replicate our main susceptibility signals using this better phenotype definition.

12. The Figure 2 legend states “Eleven loci highlighted in yellow”, only 9 loci are in yellow, kindly adjust. If the loci in green are associated with infection, what about the significant loci in grey? This is slightly confusing.

Answer: We acknowledge that this figure is rather busy and takes effort to interpret. There are in fact eleven loci plotted in yellow in the top panel, but at chromosomes 17 and 19 two independent loci are very close to each other and appear to the eye as one peak. We have therefore attempted to make this clearer by annotating each locus with the lead variant. However, we now realize one of these lead variant annotations on chromosome 17 is missing. We also have three new variants on chromosome 3 in the highlighted region. We have modified figure 2 to add this annotation, and added to the legend a clarifying statement:

Note that top panel regions on chromosomes 17 and 19 contain two statistically independent loci within proximity of each other, and on the bottom panel the first signal on chromosome 3 contains three peaks within the same region.

The peaks in grey represent loci that were indeed genome-wide significant in the respective analysis, but were not highlighted because not specific to that analysis. For example, the ABO locus on chromosome 9 is genome-wide significant in both hospitalized COVID-19 and SARS-CoV-2 reported infection analysis, but it is highlighted only in the SARS-CoV-2 reported infection analysis. This is because the effect size was not different between the two analyses, which suggests that the locus is more likely to be specific to reported SARS-CoV-2 rather than progression to severe disease. We have added a clarifying statement about the significant loci in grey into the legend of figure 2:

We highlight only loci that were specific to each analysis. That is, some genome-wide significant loci are not colored in yellow or green if they are not specific to severity (top panel) or susceptibility (bottom panel).

13. In table 1 results are presented from all GWAS but in the figure2 the critical illness phenotype is omitted. I would suggest the addition of Manhattan plot to main text or supplementary and also of QQ plots for all studies to supplementary.

Answer: We thank the reviewer for this suggestion. We have added a Manhattan plot for the critical illness phenotype as new Extended Data Figure 4.

Moreover, we have added QQ-plots as a new Extended Data Figure 2 for all studies included. As an example we show below the plot for FinnGen reported infection analysis:

Reviewer Figure 1 (example for Extended Data Figure 2 - too large to include in this response). QQ plot showing the expected $-\log_{10}(P\text{-values})$ on the x-axis and the observed values on the y-axis (red line showing no deviation from the expected) for each study contributing data to the analyses. Sample size of cases and controls is listed for each study in the plot title, as well as the median lambda value.

14. Unclear what genetic associations are being referred to (lines 369-370), also previous publications need to be cited.

Answer: We have now clarified this sentence. In particular, we were alluding to the associations pointing towards genes such as DPP9 and TYK2, which have interesting prior associations with lung and autoimmune phenotypes. We have modified the text and added references to the original studies reporting the COVID-19 associations:

*Several of the loci reported here, as noted in previous publications ^{1,4}, intersect with well-known genetic variants that have established genetic associations. **Examples of these include variants at DPP9 which show prior evidence of increasing risk for interstitial lung disease ⁸, and missense variants within TYK2 that show a protective effect on several autoimmune-related diseases ⁹⁻¹².***

15. The extremely low heritability ($h^2 \sim 0.002$) reported is intriguing. Heritability of COVID19 critical illness has been previously reported by Pairo-Castineira & Colleagues (reference above), why is the heritability so different here?

Answer: We acknowledge that our reported heritability is low, but we do not find this to be unexpected given our study design or being a problem for downstream analyses. We warn about the interpretation of this estimate in the text at page 10:

Despite these low values, which interpretation is complicated by the use of population controls and variation in the disease prevalence estimates, we found that heritability for reported infection was significantly enriched in genes specifically expressed in the lung ($P = 5.0 \times 10^{-4}$)

Little is known about the heritability of infectious diseases. Some attempts to robustly estimate this quantity have been done in animals [<https://pubmed.ncbi.nlm.nih.gov/29331471/>, <https://www.ncbi.nlm.nih.gov/pmc/articles/PMC4319771/>, <https://www.sciencedirect.com/science/article/pii/S0022030219304333>], where heritability has been shown to be low. Little is known in humans, for example a review by ¹³ shows that immune traits are heritable, but it does not report any estimates of SNP heritability of infectious disease, so it is difficult to assess what we should expect. Moreover, methodological challenges remain in estimating heritability of infectious diseases, as current methods are based on a dichotomous phenotype (infected/non infected) and do not include epidemiological models of transmission [<https://academic.oup.com/genetics/advance-article/doi/10.1093/genetics/iyab024/6137839>]. Nonetheless, the fact that we obtain 12 genome-wide significant loci (and several close to significant ones), some with large ORs, convinces us that host genetics does play a role in modifying susceptibility to the virus and COVID-19 severity, but capturing genome wide heritability may be more difficult.

There could be many reasons for this and the difference between our study and Pairo- Castineira

et al. Although we have done our best to harmonize case and control definitions, it is still possible that there are major differences between studies that contributed data. For example, they have more characteristics available for their controls than we did, and could therefore perform better matching of cases and controls. In addition, they used the HDL ¹⁴ method for estimating heritability. We used LD score regression ¹⁵ and report the observed scale heritability, which will likely be lower than true heritability. We decided not to convert our estimate to the liability scale, due to the fact that this would be affected by the population prevalence for the phenotype, which we felt we were not able to reliably define. We attempted to run the HDL method on our data, but the heritability was likely too low to be able to run the software.

We ran our analysis using LD score regression for hospitalized COVID-19 again using only controls who had been infected but were not hospitalized. In this analysis, we found that our analysis was underpowered to detect significant heritability due to the small sample size

(European ancestry N=4,829 cases, N=11,816 controls), as the observed scale heritability was $h^2 = -0.0019$ (SE=0.0273), 1.0004 Intercept: 1.001 (SE=0.0062).

Pairo-Castineira and colleagues used controls from the UK Biobank, which is known to be biased for more highly educated and healthier individuals than the general population ¹⁶, whereas their cases were samples from around the UK, and it is known that individuals in lower socioeconomic positions are more exposed to the virus ¹⁷⁻¹⁹. Pairo-Castineira et al. report a high negative genetic correlation with higher intelligence and educational attainment. This could potentially explain some of the SNP heritability that they have detected compared to our study, despite their best efforts to match cases and controls.

16. Statement in lines 373-375, conclusion is not robust given overall heritability is extremely low ~0.02%.

Answer: We understand that this phrase may sound like an overstatement, but due to the reasons we discuss above (**comment 15**), we feel that simply looking at the observed scale heritability estimate is misleading.

17. For all tables/figures, kindly add SEs and 95% C.I.s for betas and ORs reported, respectively.

Answer: As described in response to **reviewer 1's comment 6**, we have changed the beta estimates that still appeared in our main text to OR[95%CI], and added the 95%CI to our Table 1. For Extended Data Figures 4 and 5, we argue that it is more interpretable to keep allele effects as beta estimates.

Referee #2

Expertise: infectious disease genomics, statistical genetics

Summary: In this manuscript, Ganna and colleagues at the COVID-19 host genetic initiative present the result of a multi-ethnic meta-analysis of three COVID-19 phenotypes related to susceptibility to SARS-CoV-2 infection and disease progression/severity. Controls are from public databases in whom the infection status is not ascertained. The authors report 15 genome-wide significant loci in testing the three traits. The authors use heterogeneity test to show that potentially different genomic loci are involved in susceptibility to infection vs. disease severity progression. They use in silico analysis approaches to identify the plausible target gene(s) for each locus and speculate about the functional relevance of the identified loci. Finally, the authors look into the association of COVID-19 risk loci with other traits, investigate the genetic heritability of the three COVID-19 susceptibility phenotypes, test the genetic correlation between these phenotypes and other relevant phenotypes, and look into the causal relationship between these other phenotypes and susceptibility to COVID-19.

This manuscript tackles an important problem. Understanding the genetic basis of COVID-19 can lead to a better understanding of COVID-19 disease mechanism, help patient risk stratification, and help improve clinical care. This work also adds to existing evidence about the genetic basis of infectious diseases and underlies the importance of large, multi-ethnic cohorts and international collaborations in human genomics of infectious disease research. Despite these, I found the paper unsuitable for publication in its current form. My main concerns include the lack of sufficient correction (or even discussion about the role of) potential, non-genetic confounders that can derive the results. I was disappointed to see that, little effort was dedicated to exploring/leveraging the multi-ethnic aspect of this cohort. A lost opportunity, given the lack of genetic studies in non-European populations and the high burden of COVID-19 in these populations. Finally, some crucial information type definition about study design and phenotype definition is missing.

Study design:

1. Given the multi-layer and parallel analyses a figure to explain the study design and the case/control comparisons that are performed is needed to make following the paper easier.

Answer: We would like to thank the reviewer for their extremely thorough review and constructive feedback on our manuscript.

We agree that a figure would complement the paper well, and have added the figure below as new Extended Data Figure 1.

[redacted]

New Extended Data Figure 1. Analytical summary of the COVID-19 HGI worldwide meta-analysis. Using the analytical plan set by the COVID-19 HGI, each individual study runs their analyses and uploads the results to the Initiative, who then runs the meta-analysis. There are three main analyses that each study can contribute summary statistics to; critically ill COVID-19, hospitalized COVID-19 and reported SARS-CoV-2 infection. The phenotypic criteria used to define cases are listed in the dark grey boxes, along with the numbers of cases (N) included in the final all ancestries meta-analysis. Controls were defined in the same way across all three analyses; as everybody that is not a case e.g. population controls (light grey box). Sensitivity analyses, not reported in this Figure, also used mild/asymptomatic COVID-19 cases as controls. Sample number (N) of controls differed between the analyses due to the difference in number of studies contributing data to these.

2. In figure 1, the ancestry breakdown for the whole cohort is shown but not for each GWAS? It worth knowing whether there are major differences in ancestry composition of cases for each GWAS analysis and if the differences in ORs/results can be driven by differences in cohort genetic composition, LD, or non-genetic factors that correlate with genetic ancestry.

Answer: We thank the reviewer for this notion, and agree that we should better discuss this matter throughout our paper, as it is an important point.

The ancestry breakdown is slightly different for each of the three main meta-analyses, and we have added the following phrase at page 3:

The proportion of cases with non-European genetic ancestry for each of the three analyses was 23%, 29% and 22%, respectively

The proportion of European genetic ancestry in the controls is higher because the largest number of controls is obtained from direct-to-consumer datasets, where the prevailing genetic ancestry is European.

Moreover, we have addressed the reviewer's concerns more in depth in our responses to comments adding QQ-plots for all studies (new Extended Data Figure 2), ancestry PCA plots for all studies that were willing to share these (new Extended Data Figure 3) (**reviewer 2's point below**).

3. In line with the above point is it possible to provide figures/supplementary figures of the PCA analyses for the combined cases and cases and controls for each phenotype? Projecting the PCs for each dataset to a common PC space (1KG for example) will allow visual comparisons between different analyses.

Answer: We appreciate the reviewer for raising this point. To project every GWAS participant in the same PC space, we pre-computed PC loadings using the 1000 Genomes Project and the Human Genome Diversity Project as a reference and asked each cohort to run projection using an automated script we provided (https://github.com/covid19-hg/pca_projection). In total, 40 cohorts (including three cohorts which submitted a PCA figure, not raw projected values) submitted projection results. We assembled these results and provided as a new Extended Data Figure 3.

Although we observed different degrees of genetic diversity for each cohort, we confirmed population stratification was appropriately controlled within each analysis based on QQ plots (Extended Data Figure 2) and per-cohort descriptions of their analytic procedure.

New Extended Data Figure 3. PC projection into the same PC space using the 1000 Genomes Project and Human Genome Diversity Project as a reference. For each panel (except for the reference), colored points correspond to participated samples from each cohort, whereas gray points correspond to reference samples. Color represents a genetic population that each cohort specified. Since 23andme, genomicsengland100kpg, and MVP only submitted PCA images, we overlaid their submitted transparent images using the same coordinates, instead of directly plotting them.

4. The phenotype definition presented in the paper is not detailed enough, given that heterogeneity in phenotype definitions between cohorts can lead to spurious findings it helps the reader judge the validity of the results to know the case phenotype definition (Table S1) and the harmonization method.

Answer: The phenotype definitions used by the different studies were pre-defined using simple criteria described in Methods.

Each group independently defined the phenotypes according to the provided inclusion/exclusion criteria. We expect slight differences for the same phenotype definition across studies due to the different levels of infection detection, severity of hospitalization and admission to ICU. We extensively described these challenges in our Discussion at page 14:

Nevertheless, the differences in study sample size, ascertainment and phenotyping of COVID-19 cases are unavoidable and care should be taken when interpreting the results from a meta-analysis. First, studies enriched with severe cases or studies with antibody-tested controls may disproportionately contribute to genetic discovery despite potentially smaller sample sizes. Second, differences in genomic profiling technology, imputation, and sample size across the constituent studies can have dramatic impacts on replication and downstream analyses (particularly fine-mapping where differential missing patterns in the reported results can muddy the signal). Third, the use of population controls with no complete information about SARS-CoV-2 exposure might result in cases of misclassification or reflect ascertainment biases in testing and reporting rather than true susceptibility to infection

We have also revised the use of the word “harmonization” in the manuscript and only use it where data was harmonized, e.g. we remove this wording at page 14 when describing the process of defining phenotypes.

5. The answer to some of the points above might be available in the CHGI website but this flagship paper should provide enough information for the reader to be able to judge the validity of the methods and result. Adding more details can also help future similar studies to use this paper as a guideline.

Answer: We agree with the reviewer, and thank them for the extensive improvements that they have suggested. We have done our best to provide more detailed information, where available, on the methods and data from each study. We include new figures for QQ-plots (Extended Data Figure 2), PCA ancestry projections produced with an harmonized code that each contributing study ran on their own data (Extended Data Figure 3) and have added details to the PCA approach each study has undertaken (Supplementary Table 1). We have also made our genome-wide significance threshold more stringent and provide additional information in the Methods section for why specific studies were not included, or why some loci did not make it into Table 1 (heterogeneity across studies, missing data). We hope that these changes will help the readers in their judgement of the results from the analyses.

6. The methods indicate: “Standard error values as a function of effective sample size was used to find studies which deviated from the expected trend.” It would be helpful to provide a figure for this analysis.

Answer: We have supplied the new Extended Data Figure 11 (not reported in this rebuttal because too large) and reference it in a sentence at page 16:

Standard error values as a function of effective sample size was used to find studies which deviated from the expected trend (Extended Data Figure 11).

Below we provide an example to the reviewer to illustrate how this QC step can inform about problematic studies.

New Supplementary Figure 10 illustrative extract. The figure panel shows for the top SNP on the chromosome 3 locus the standard error plotted as a function of effective sample size in each study that contributed data to the meta-analysis of the three phenotypes (from the left): critical illness due to COVID-19, hospitalization and reported infection. Because allele frequency can impact the relationship between standard error and effective sample size, we have also colored the different studies by ancestry group.

[redacted]

GWAS analysis:

1. There is no correction (or no discussion around) the role of sociodemographic and environmental factors (or their interaction with genetic factors) in susceptibility to SARS-CoV-2 exposure and progression. Is it possible that the differences observed between phenotypes and/or cases and controls are, at least in part, driven by these non-genetic factors?

Answer: We agree with the reviewer that sociodemographic factors can influence differences observed between cases and controls. We have included this as an additional limitation on page 14:

Finally, sociodemographic factors may influence an individual's susceptibility to SARS-CoV-2 infection and COVID-19 severity. In particular, lower socioeconomic level is associated with a higher risk of infection and hospitalization¹⁷. This can result in collider bias which distorts the relationship between genetic variation and the phenotypes being examined¹⁹.

Additionally, other factors such as time of infection, which affects mortality and critical care admissions²⁰), or differences in vaccination schemes may change the sociodemographic characteristics of COVID-19 positive participants.

2. Extended figure 2 shows that restricting the analyses to infected controls does not affect the OR of the 15 loci described in the manuscript. How do the genome-wide results change if you restrict the analyses to infected controls? Are there any new associations (apart from the reported 15 loci)? How do the genome-wide Z-scores correlate between the analyses that I suggest here and GWAS for 1- critical illness vs. population controls or 2- hospitalized vs. population control? Such a comparison can alleviate some of the concerns about the impact of confounding factors related to exposure status.

Answer: We agree with the reviewer that exploring analyses with infected controls is interesting. However, our analyses using these controls are unfortunately underpowered for genetic discovery.

As we described in response to **reviewer 1's comment 15**, we ran LD score regression on the analysis of European ancestry hospitalized COVID-19 cases vs. non-hospitalized infected controls (N=4,829 cases, N=11,816 controls). We found the observed scale heritability to be $h^2 = -0.0019$ (SE=0.0273), 1.0004 Intercept: 1.001 (SE=0.0062). In this analysis, the only genome-wide significant locus ($P < 5 \times 10^{-8}$) was the chromosome 3 locus.

We further ran the meta-analysis on European subset hospitalized COVID-19 vs. population controls (N=4,862 cases, N=1,064,497 controls), and plotted the z-scores from this analysis in the figure below (x-axis) and compared to the z-scores from the non-hospitalized infected cases (y-axis), as the reviewer suggested. This shows that the z-scores are relatively well correlated across the board, but we lack power to detect any interesting findings in our refined controls analysis. The z-scores deviating to the right on the x-axis are from the chromosome 3 locus.

Reviewer Figure 2. The figure shows the z-scores plotted from the meta-analysis of European ancestry hospitalized COVID-19 cases vs. population controls (on x-axis) and the z-scores from a meta-analysis of the same cases vs. infected but not hospitalized controls (y-axis). The red line is the regression line ($y \sim x$).

- Extended figure 2A also shows that rs72711165's OR is higher in hospitalized than severe cases. If this SNP is associated with severity, as the authors claim, shouldn't the OR increase with the more carefully defined and more severe phenotype (i.e critical illness)?

Answer: We thank the reviewer for pointing out these detailed findings on the rs72711165 (lead variant in chromosome 8 locus). We understand the reviewer's concern about the OR not increasing when refining our cases to the most critically ill. However, we believe that this comparison should not be read into in much depth, since the critically ill COVID-19 analysis is underpowered, and the 95% confidence intervals from these analyses overlap for most of the lead variants: e.g. for rs72711165 the ORs were 1.23[1.03,1.46] for critical illness and 1.37[1.23, 1.52] for hospitalized COVID-19. Our aim was to merely illustrate, in Supplementary Figure 2A (now 6A), that there is an overall trend when comparing these effects.

We believe the more informative comparison is between COVID-19 hospitalization and reported infection, because these analyses are much better powered. For example, the variant rs72711165 OR 95% confidence intervals were: 1.08[1.02,1.14] for reported infection and 1.37[1.23, 1.52] for hospitalized COVID-19. We decided to use a test of heterogeneity between effect size estimated at each locus lead variant to determine whether there was a significant difference in severe phenotype vs. reported infection. From these results we

conclude that ,based on the current evidence, some loci are potentially affecting severity (significant difference) or susceptibility (no difference) to COVID-19, and report the *P*-value from this test in Supplementary Table 2.

We have now added a note to the main text (page 4) about the interpretation of the results from the critically ill vs. hospitalized comparison in attempt to highlight that we are aware of the power issues with this particular analysis. We also note that with the new threshold for genome-wide significance, we have 9 severity loci instead of the previous 11:

*We further compared the ORs for these **nine** loci for critical illness due to COVID-19 vs. hospitalized due to COVID-19, and found that these loci exhibited a general increase in effect risk for critical illness (**Methods**) (**Extended Data Fig. 7A, Supplementary Table 3**), **but we note that due to the lower power for association in the critically ill analysis these results should be considered as suggestive.***

4. rs72711165's OR increases further when controls are restricted to infected controls. This change can be the result of a reduced noise (unlikely, as the authors claim because we don't see the same increase for other loci in Extended figure 2B), changes in ancestry composition and the effect allele frequency (for example having proportionally less East Asians in the infected controls vs population controls), or changes in non-genetic confounders that affect COVID risk and correlate with genetic ancestry. Comparing rs72711165's MAF in population controls and infected controls can help distinguish between the two latter possibilities. If the MAF is the same my worry is that other findings can also be biased due to factors not captured by the covariates included in the association analyses.

Answer: We appreciate the reviewer's concern over this lead variant rs72711165. We do however believe that it is difficult to draw definitive conclusions from comparing the critically ill COVID-19 analysis to hospitalized COVID-19 due to the relatively low power of the former analysis, as we have described in detail in the response above. Although the shift in effect size seems to stand out more than the other lead variants, the confidence interval for this variant overlaps with the null that there is no significant difference in both Extended Data Figures 2A and 2B (now 7A and 7B).

We replotted the Extended Data figure 2B (now 7B) with European-only meta-analysis results for both hospitalized vs. population controls (x-axis) and hospitalized vs. infected controls (y-axis), below. Again, we couldn't draw any solid conclusion because of the large variability around these estimates. Nonetheless, we agree that ancestry confounding has the potential to bias the results. We have addressed this concern in **reviewer 1 comment 13** and **reviewer 2 "Study design" comment 5** .

Reviewer Figure 3. The figure shows the Extended Data Figure 2B (now 7B) panel replotted using only European ancestry studies. Hospitalized COVID-19 vs. population controls on x-axis (N=4,862 cases, N=1,064,497 controls) and hospitalized COVID-19 vs. controls with reported infection but not hospitalized on y-axis (N=4,829 cases, N=11,816 controls).

5. It is a shame that no formal fine-mapping is performed for the GWAS loci. The multi-ethnic nature of this work can be leveraged for fine-mapping (compared to a single ancestry study) and help to find potential causal variants (which as the authors know is not necessarily the index variant)?

Answer: We appreciate the reviewer's point about potential opportunities in fine-mapping by leveraging differences in allele frequencies and LD across multiple ancestries. However, we would like to emphasize that fine-mapping a meta-analysis is extremely challenging with the current setting where each cohort was originally established with heterogeneous study design, case ascertainment, and control populations. Although we tried our best to harmonize all the phenotype definitions and analytic pipelines across cohorts, we observed that a subtle heterogeneity across cohorts, such as difference in phenotyping, genotyping, or imputation, could lead to challenges in fine-mapping. This is because, within a locus, variants might be meta-analyzed across different numbers of studies (minimum 25 studies), resulting in different power for each variant. In addition, the previous literature suggested that fine-mapping with external LD reference (instead of in-sample LD) is extremely error-prone and should be avoided ^{21,22}.

Below we show two illustrative examples from the 3p21.31 and 19p13.2 loci. For the 3p21.31 locus, the risk haplotype is known to show strong LD across populations, resulting in almost perfectly correlated associations among the variants in its core haplotype spanning ~50 kb²³. However, we observed that a subset of variants showed significantly weaker associations (because less studies reported these variants), which would falsely affect fine-mapping posterior probabilities, despite being in nearly perfect LD in gnomAD with the rest of variants. Likewise, for the 19p13.2 locus, we are very confident that a missense variant rs34536443:G>C (p.Pro1104Ala) in *TYK2* is the causal variant in this locus based on the prior literatures^{9–12}. However, the data suggests the lead variant rs74956615:T>A ($r^2 = 0.82$ with rs34536443) is more significant than rs34536443, which would falsely implicates rs74956615 is more likely to be causal.

With our aim to provide the most robust results to the community, we therefore would like to avoid any *in-silico* analysis that potentially produces spurious results. We confirmed and demonstrated that our associations are robust at locus-level resolution; yet, further large-scale harmonized efforts are required to robustly identify signals at variant-level resolution, including functional characterization for mechanistic insights. We have addressed this point in Discussion at page 14:

Second, differences in genomic profiling technology, imputation, and sample size across the constituent studies can have dramatic impacts on replication and downstream analyses (particularly fine-mapping where differential missing patterns in the reported results can muddy the signal).

Reviewer Figure 4. LocusZoom plots for **a.** the 3p21.31 locus and **b.** the 19p13.2 locus for hospitalization (excerpt from the Extended Data Figure 7). For each panel, we showed 1) a manhattan plot of each locus where a color represents a weighted-average r^2 value to a lead variant; 2) r^2 values to a lead variant across gnomAD v2 populations; 3) genes at a locus; and 4) genes prioritized by each gene prioritization metric. Any

discrepancies between observed significance in meta-analysis (1) and r^2 values in general populations (2) might indicate potential existence of other factors (including inter-cohort heterogeneity) influencing association statistics.

6. The paper, does not present any conditional analysis for the identified loci. This is a basic analysis and may help to find additional association signals.

Answer: We thank the reviewer for raising this point. As we mentioned in the previous comment, we avoided statistical conditional analysis based on the concerns over inter-cohort heterogeneity and lack of appropriate LD reference.

However, we agree with the reviewer that identifying additional independent signals would benefit the community. We thus employed an alternative, more straightforward approach by comparing chi-squared association statistics and r^2 values to the lead variant for each locus. Chi-squared statistics should correlate with r^2 values to a causal variant and any deviation might be a sign of additional causal variants in a locus. With all the aforementioned caveats in mind, we rationalized this approach to be particularly useful to distinguish loci where a single causal variant is primarily driving a signal from ones where multiple independent signals might exist.

By systematically evaluating every locus across the three analyses, we identified an additional independent signal in the 3p21.31 region for SARS-CoV-2 reported infection (**Extended Data Figure 12, panel c**), but not for critical illness or hospitalization (**Extended Data Figure 12, panel a,b**). We describe this signal in depth in the new **Supplementary Note**, which we have provided in this revision. This signal was reported by ⁵ based on our previous release version 4, but they were missed in our manuscript. We reported this signal as an independent locus in **Table 1** and the locus zoom plots are reported in **Extended Data Figure 5**. We call this a new "locus" because it is associated with a different phenotype than the main locus in the main 3p21.31 region, but discussed limitations of this approach in the Discussion on page 13.

Four out of the 13 genome-wide significant loci showed similar effects in the reported infection analysis (a proxy for disease susceptibility) and all-hospitalized COVID-19 (a proxy for disease severity). Of these, one locus was in close proximity, but yet independent, to the major genetic signal for COVID-19 severity at 3p21.31. Surprisingly, this locus was associated with COVID-19 susceptibility rather than severity. The locus, which comprises at least three independent signals all associated with susceptibility, overlaps SLC6A20, which encodes an amino acid transporter that interacts with ACE2. Nonetheless we caution that more data is needed to resolve the structure of the locus. In particular, the physical proximity and the lack of expected relationship between P-

value and linkage disequilibrium structure raises the suspicion that untagged genetic variation might be underlying this cluster of loci.

Extended Data Figure 12. Comparison of chi-squared statistics vs r^2 values to the lead variant in the 3p21.31 region for **a.** critical illness, **b.** hospitalization, and **c.** reported infection. The left blue peak in panel **c**, which is uncorrelated with the lead variants in the region, indicates that there are independent signals.

Extended Data Figure 5. Locuszoom plots of the 3p21.31 region for reported infection. **a.** A standard plot without exclusion. Here, the severity lead variant rs10490770 (chr3:45823240:T:C) is shown as a lead variant.

b. Additional independent susceptibility signal(s) after excluding variants with $r^2 > 0.05$ with rs10490770. The susceptibility lead variant rs2271616 (chr3:45796521:G:T) is highlighted.

We describe these new findings in more detail in the results section on page 4:

*Interestingly, and in agreement with the report by Robert and colleagues¹⁸, we reported a locus within the 3p21.31 region that was more strongly associated with susceptibility to SARS-CoV-2 than progression to more severe COVID-19 phenotypes. Rs2271616 showed a stronger association with reported infection ($P=1.79\times 10^{-34}$; $OR[95\%CI]= 1.15 [1.13-1.18]$) than hospitalization ($P=1.05\times 10^{-5}$; $OR[95\%CI]=1.12[1.06-1.19]$). For this locus, which contains additional independent signals, the linkage-disequilibrium pattern is discordant with the P-value expectation (**Methods**), pointing to a key missing causal variant or to a potentially complicated structural variation in this locus. More research into understanding the biology of these signals is needed.*

We have additionally added more information about these variants into the gene prioritization results section at page 8:

*Lastly, there are two loci in the 3p21.31 region with varying genes prioritized by different methods for different independent signals. For the severity lead variant rs10490770:T>C, we prioritized CXCR6 with the Variant2Gene (V2G) algorithm²¹, while LZTFL1 is the closest gene. The CXCR6 plays a role in chemokine signaling³¹, and LZTFL1 has been implicated in lung cancer³². Rs2271616:G>T, associated with susceptibility, tags a complex region including several independent signals (**Supplementary Text**) all located within a gene body of SLC6A20 which is known to functionally interact with the SARS-CoV-2 receptor ACE2³³. However, none of the lead variants in the 3p21.31 region has been previously associated with other traits or diseases in our PheWAS analysis. While these results provide supporting in-silico evidence for candidate causal gene prioritization, further functional characterization is strongly needed. Detailed locus descriptions and LocusZoom plots are provided in **Extended Data Fig. 8**.*

7. Please add the figure for the third GWAS analysis to fig 2.

Answer: We have added the manhattan plot for critically ill COVID-19 now as a new Extended Data Figure 4 (see figure below). We choose to add this as Extended Data Figure and not as main figure because we don't think this adds much on top of the existing manuscript Figure 2, but we are happy to reconsider if the reviewer thinks this should be included as main figure.

New Extended Data Figure 4. Genome-wide meta-analysis association results for critical illness due to COVID-19. The locus on chromosome 6 is the HLA locus, which was removed from the list of reported loci in Table 1 due to the high heterogeneity in effect size estimated between studies included in the analysis. The locus on chromosome 7 was also not reported in Table 1 due to missingness across studies,

i.e. the high number of studies in the meta-analysis that did not report summary statistics for this region. There are two association peaks on chromosome 19.

8. Another potential confounding factor is the time of infection. New reports show a considerable decrease in in-hospital mortality rate that is largely attributable to non-genetic factors (here is a summary: <https://www.nature.com/articles/d41586-020-03132-4>). So people with the same genetic predisposition can have largely different outcomes depends on when they got infected. I honestly don't know how to account for this (unless you have the hospitalization dates) but it worth considering such confounding factors in the interpretation of the results. At a minimum, this should be listed as a limitation in the discussion.

Answer: We thank the reviewer for this important consideration. We do not have information on the time of infection across the different studies, and are therefore limited in what we can do in terms of accounting for this in the analysis. As the reviewer suggests, we have added this limitation to the discussion at page 14:

Finally, sociodemographic factors may influence an individual's susceptibility to SARS-CoV-2 infection and COVID-19 severity. In particular, lower socioeconomic level is associated with a higher risk of infection and hospitalization¹⁷. This can result in collider bias which distorts the relationship between genetic variation and the phenotypes being examined¹⁹. Additionally, other factors such as time of infection, which affects mortality and critical care admissions²⁰, or

differences in vaccination schemes may change the sociodemographic characteristics of COVID-19 positive participants.

9. There is no signal at the HLA locus, is HLA excluded from the analyses? If yes there is no mention of this in the methods section. If no, given the role of this locus in infectious diseases please expand the methods to include HLA typing and analysis and consider expanding the discussion to speculate why no HLA signal was observed.

Answer: We did not exclude the HLA region from our three meta-analyses. In the hospitalized COVID-19 vs. population controls, we did see an association signal for the trait at the LD region chr6:31057940-31380334, but this was one of the two loci that were not reported in Table 1 due to high heterogeneity between contributing studies ($P < 0.0005$). We have made a note about this in the Methods section under “GWAS and meta-analysis”:

Two loci reached genome-wide significance but were excluded from Table 1 significant results due to heterogeneity and missingness between studies chr6:31057940-31380334 and chr7:54671568-54759789; however these regions are not excluded from the corresponding summary statistics data release.

To explore in more detail the role of HLA, specific imputation of HLA alleles should be carried out by each study. At the time of the writing of this manuscript, such a widely accessible panel was not available, but it has been recently added to the Michigan imputation server. For future data freezes of the COVID-19 HGI, we will consider focusing specifically on HLA, but we think this is beyond the scope of this paper.

[redacted]

The LD score regression software does remove the HLA region, and we have added a note about this in the methods section under “Heritability” when we first describe the LD score software settings used:

Analyses were conducted using the standard program settings for variant filtering (removal of non-HapMap3 SNPs, the HLA region on chromosome 6, non-autosomal, chi-square > 30, MAF < 1%, or allele mismatch with reference).

10. The genome-wide significance threshold needs adjustment for the number of traits tested.

Answer: We agree with the reviewer and have made this change and used a threshold of $P < 1.67 \times 10^{-8} ((5 \times 10^{-8}) / 3)$ (please see answer to **reviewer 1 - comment 4**). This change resulted in 3 less genome-wide significant loci from the original 15 loci (note: we have one new susceptibility locus on chromosome 3, bringing the total of independent loci to 13 in the revised manuscript).

Multi-ethnic nature of the work:

1. What are the ancestry-specific ORs/Z-scores for the reported loci? Two SNPs were mentioned with variable MAF in different populations. The ancestry-dependent heterogeneity in MAF, OR for other loci are not discussed. A table to show the MAF and ORs of variants among different genetic ancestries and formal heterogeneity test is needed and can help the downstream utility of this study.

Answer: As we have explained in **reviewer 3, comment 6**, we are not yet comfortable in releasing underpowered meta-analysis summary statistics for non-European ancestries in this data release (release 5). Nonetheless, we have compiled a new **Supplementary Table 12** with the frequencies and effect sizes for the ancestry-specific analyses only for the 13 genome-wide significant lead variants.

2. It was shown recently that the TYK2 P1104A variant, involved in susceptibility to TB, went under negative selection in the European population (Kerner et al, AJHG, 2021), do the authors see any difference in the frequency and OR of this variant among different ethnicities?

Answer: This is an interesting question, but like with the other cross-ancestry variants highlighted before, we do not have good power for doing proper analysis of heterogeneity between ancestries. We are planning to include these analyses in future data releases with larger samples size. Below we report a forest plot of the effect sizes for different ancestries for the hospitalized COVID-19 vs. population. Results are only significant in Europeans and Ad-mixed American, but confidence intervals are too large to test differences in effect size between populations.

Reviewer figure 5. Forest plot of variant rs74956615 (19:10317045:T:A) at the TYK2 locus for hospitalized COVID-19 vs. population controls across different ancestry groups.

Heritability, genetic correlation, and in silico functional analyses:

1. If you perform the analyses for critical and hospitalized COVID-19 using infected controls only, do you see any change in heritability estimates?

Answer: This is an important question, and we report the results of this analysis in response to **reviewer 1’s comment 15, and reviewer 2’s comment 2** under section “GWAS analysis”. In summary, the analyses were underpowered for detecting significant observed scale heritability in European samples. We repeated the analysis in the all generic ancestries data (not shown) with a few hundred more samples, but these were also underpowered for the heritability analysis.

2. What is the intercept for each phenotype in your LDSC heritability analyses.

Answer: We had supplied the LDSC intercepts in Supplementary Table 6, but have now also added these to the main text at page 10:

We detected a low, but significant heritability across all three analyses (<1% on observed scale, all P-values < 0.0001, LDSC intercept range 1.0024-1.0137; Supplementary Table 6).

3. Please include the figures for heritability and genetic correlation analyses in the manuscript/supplement. Beyond the presented numbers, seeing these figures will help the reader judge the potential inflations/abnormalities in the data.

Answer: We thank the reviewer for this suggestion, and hopefully we have interpreted the request correctly; we have supplied an additional Extended Data Figure 9, showing the forest plots of the genetic correlations:

Extended Data Figure 9. Each column shows genetic correlation results for the three COVID-19 phenotypes (European ancestry analyses only): critical illness, hospitalization and reported infection. The traits the genetic correlation is run against are listed on the left. Significant correlations (FDR<0.05) are shown with their 95% confidence intervals in red, nominally significant (P<0.05) in black and non-significant in grey.

We have not provided separate figures for SNP heritability estimates due to the low heritability because we do not think this is a particularly useful result to focus on (please refer to our responses to **reviewer 1's comment 15** about heritability analyses).

4. If you use a colocalization test such as COLOC for PheWAS GWAS colocalization analysis does your colocalization results change?

Answer: We thank the reviewers for this suggestion. The colocalization between traits would ideally require finemapped credible set, however due to signal heterogeneity across studies we are unable to perform finemapping with consistent results. We have extensively illustrated the reason why finemapping in our study is challenging in **reviewer 2, comment section "GWAS analysis" point 5**.

We also considered phenotypes for which the lead variants were in high LD ($r^2 > 0.8$) with the now 13 genome-wide significant lead variants from our main COVID-19 meta-analysis and in the credible set for that phenotype. This conservative approach identified relatively few pheWAS associations, mostly dominated by ABO locus, thus yielding limited insights on the phenotypic associations of the reported SNPs.

5. In which tissues the candidate genes listed in table 1 are expressed (for example in GTEx)?

Answer: We performed a tissue enrichment of all the genes reported in Table 1 using the tissue-specific expression data from GTExv8. The enrichment analysis did not report any significant results. We report the results for reviewer's satisfaction but we choose to not report these findings in the main text as they add little on top of the tissue-enrichment LDSC analysis that we have already performed.

Reviewer Figure 6. Heatmap of the genes tested for the tissue enrichment analysis using FUMA. The colors refer to the expression values of genes (y-axis) against the 54 tissues (x-axis) from the GTEx v8.

Reviewer Figure 7. Bar plot for tissue enrichment for up-regulation (top), down-regulation (middle), and differential expression (bottom) of genes tested (shown in **Reviewer Figure 6**). The y-axis shows log transformed p-values of the enrichment test, and tissues (x-axis). The tissues on x-axis are sorted from lowest to highest p-value, however the results are not significant after multiple-testing correction.

Manuscript text:

1. Methods lack information about how “ancestry matched” controls were matched and selected.

Answer: Since the analysis of each dataset was performed by the respective contributing studies, the consortium could only give guidelines to how to perform this step. We requested that all samples would prior to analysis be checked for genetic ancestry and labelled based on the ancestry abbreviations following the 1000 Genomes definition: African: AFR Admixed American: AMR European: EUR East Asian: EAS South Asian: SAS, or Other (to be defined by the researchers). Each ancestry group would then be analysed on its own, and the minimum sample size was N=50 cases and N=50 controls. We requested that each study use 20 PCs in their analysis.

Guided by the reviewer's comment, we have now requested more detail from each study about their methods to account for population structure in the analyses. We have added this information to Supplementary Table 1. Additionally, we have provided each study with a harmonized script to obtain PCA projection. We describe the results in response to the **reviewer 2 comment 3** (new Extended Data Figure 3). The majority of studies have tightly clustering samples for each ancestry that they have run a separate analysis for.

2. Similarly, methods for effective sample size calculation is missing.

Answer: We use the formula below and have added this to our Methods section on page 15:

In total 16 studies contributed data to analysis of critical illness due to COVID-19, 29 studies contributed data to hospitalized COVID-19 analysis, and 44 studies contributed to the analysis of all COVID-19 cases. The effective sample sizes for each ancestry group shown in Figure 1 were calculated for display using the formula: $(4 \times N_{\text{cases}} \times N_{\text{controls}}) / (N_{\text{cases}} + N_{\text{controls}})$.

3. There is no mention of previous work of human genomics of respiratory viruses (influenza, RSV, etc.) in the paper. It worth expanding the discussion to previously acknowledge the previous work and compare the current manuscript’s findings with what is known regarding human genomics of susceptibility respiratory viruses.

Answer: we thank the reviewer for this suggestion. Despite only a few genome-wide studies have been conducted on the topic, we have included a few references in the manuscript Introduction to cover previous findings on this topic:

The contribution of host genetics to susceptibility and severity of infectious disease, including respiratory viruses, is well-documented, and encompasses rare inborn errors of immunity^{3,4} as well as common genetic variation⁵⁻¹⁰.

4. In the discussion, there is a reference to the FOXP4 locus and its frequency in different ancestral groups (lines 381 – 385). The information about the frequency of this locus in different groups (admixed Americans, middle eastern, etc.) is not present in the text.

Answer: Indeed, we had missed out this information, and have now added it to the sentence at page 13:

One of these loci, close to FOXP4, is common particularly in East Asian (32%) as well as Admixed American in the Americas (20%) and Middle Eastern samples (7%), but has a low frequency in most European populations (2-3%) in our data.

5. Lines 369-379 of discussion lack any reference to previous work.

Answer: We thank the reviewer for noticing this, and have clarified this section and added references:

Several of the loci reported here, as noted in previous publications^{1,4}, intersect with well-known genetic variants that have established genetic associations. Examples of these include variants at DPP9 which show prior evidence of increasing risk for interstitial lung disease⁸, and missense variants within TYK2 that show a protective effect on several autoimmune-related diseases⁹⁻¹².

6. The wording for the different phenotypes used in the paper needs more clarification. According to table 1, the three phenotypes are referred to as 1- critical illness, 2- hospitalized, and 3- reported infection. Throughout the paper, there are multiple references to the “severity” phenotype but this phenotype is not clearly defined in the text (I understand 1+2 vs 3 is the severity but it needs to be stated somewhere in the main text).

Answer: We thank the reviewer for pointing out this lack of clarity on the severity vs. susceptibility definition. We in fact tried to avoid labelling any particular analysis as severity or susceptibility to COVID-19, and instead strive to use the labels critical illness due to COVID-19, hospitalized COVID-19 and reported infection. We note that we did in fact have some remaining mentions of ‘severity analysis’ or ‘susceptibility analysis’ in our manuscript,

and have now removed these unless we are specifically referring to a previous publication or to the comparison of ORs between analyses of the different COVID-19 phenotypes.

7. The last sentence of the abstract seems to miss a word (provides maybe?).

Answer: Indeed, there was a word missing. We have added:

*This working model of international collaboration **provides** a blue-print for future genetic discoveries in the event of pandemics or for any complex human disease.*

8. Table 1 legend mentions V2G column but this column doesn't exist in the table.

Answer: This was our mistake; we decided to move this column to Supplementary Table 2, and have now removed the phrase from the Table 1 legend.

9. The authors need to carefully check the manuscript to keep the wording consistent (e.g SARS-CoV-2 in the main text vs. SARS Cov 2 in the discussion)

Answer: We thank the reviewer for spotting this, and have changed all mentions of the virus to "SARS-CoV-2".

10. Line 328 is it p or FDR?

Answer: This was a *P*-value for the sensitivity analysis, and therefore is not multiple-testing corrected. We have added "(*unadjusted P-value* < 0.05)" to the text to help clarify this.

Referee #3

Expertise: infectious disease genomics

Summary: The authors describe the flagship analysis of the COVID-19 HGI assessing the human genetic component in determining susceptibility to SARS-CoV-2 infection and COVID-19 severity through a large, multi-ethnic meta-analysis of genome-wide data. Combining data from 46 studies across 19 countries the consortium amassed a sample size of ~50,000 COVID-19 cases and 2 million, mostly population-level, controls, making it by far the largest genomic study of an infectious disease. Using 3 a priori defined phenotypes, they identify 15 genome-wide significant loci with consistent effect sizes across cohorts (i.e. no significant heterogeneity). Of these, 4 loci are deemed to be associated with enhanced

susceptibility while the remaining 11 associate with severity after infection. At each of these loci, the authors apply various heuristic gene prioritization approaches to implicate causal genes and investigate the variant impact on additional traits using a PheWAS approach, implicating lung dysfunction in COVID-19 severity. Finally, the authors perform a series of analyses looking at genetic correlation between COVID-19 severity and other traits, the most striking of which are causal relationships between increased BMI and smoking and COVID-19 severity. Overall this is very well performed study of a critical issue in global health. The scope of the study and the speed with which it was performed are, to my mind, unparalleled and the consortium organizers and participants are to be commended. I do however have several suggestions to be considered prior to publication.

1. As the authors correctly point, out the varying study designs and ascertainment make combining these studies in such a way as to avoid all confounding difficult. I feel that adding some supplementary information or extended data figures describing how the ancestry matching was performed and how the test statistics were calibrated per study would be beneficial. Presentation of PCA plots and test statistic distributions and inflation factors for each cohort and the full meta-analysis could address this.

Answer: We would like to thank the reviewer for their insights and constructive feedback.

We agree about the value of additional information on how ancestry control was performed by the different studies and discussed this in detail in response to **reviewer 2's comment 3**.

In short, we have taken three actions. First, we have requested more detail from each study about their methods to account for population structure in the analyses. We have added this information to Supplementary Table 1. Second, we have provided each study with a harmonized script to obtain PCA projection. We have reported PCA projections in Extended Data Figure 3 for those studies that were willing to share their plots. Third, we have provided QQ-plots for all studies and for each meta-analysis phenotype in Extended Data Figure 2, all which were manually checked by the meta-analysis team before performing the meta-analysis.

2. More detail should be provided on the decision to remove 2 associated loci based on “extreme heterogeneity across contributing studies” but to retain other loci with “significant heterogeneous effects ... likely reflecting heterogeneous ascertainment of cases across studies”. In particular, the 3p21.31 hit (reported previously and represented here by rs10490770) does have strong evidence for heterogeneity, in particular in the reported infection analysis. Is there a reason this is expected at this locus and can the authors comment on how the heterogeneous ascertainment of cases doesn't impact the other associated loci, or indeed the full genome-wide results.

Answer: The justification we used in our manuscript was admittedly lacking in detail.

Heterogeneity was not the only decision for excluding these two loci, but we also considered previous evidence of association, distribution of effect sizes across studies and number of studies reporting these variants. [redacted]

One of the two loci overlaps with the HLA region and we invite to check our answer to **reviewer 2 “GWAS analysis” point 9** for a detailed explanation of exclusion of this locus. The other locus on chromosome 7 was observed in fewer studies than the other GW-significant loci. Taken together, and to improve the robustness of our results, we decided to exclude these results from the current analysis. On the contrary, and despite the high heterogeneity, we kept the chromosome 3 locus because it is a well known and established signal. Moreover, because of such a strong signal, we would expect to have more power to detect even very small heterogeneity effects, which are inherent when including studies with heterogeneous ascertainment. We see that these are driven by certain previously published studies.

3. Given the multi-ethnic nature of the study, were sub-analyses performed to identify any ancestry-specific effects that may not have been observed in the full analysis? The authors point out a few loci with stronger effects in Asian ancestry samples and that the 3p21.31 haplotype is most common in Bangladeshi populations but what about variants that are exclusive to ancestry groups? Would these be captured in the present analysis and is it possible that population-specific causal variants would be missed when analyzing tag SNPs across a multi-ethnic sample?

Answer: This is an important point and something that we would have liked to leverage more for the multi-ethnic nature of the manuscript. The non-European ancestry group sub

meta-analyses were run but these were mostly low in power due to the small sample size e.g. EAS was the only sub population where the sample size for case was above 1,000 (N=1,414). We definitely cannot rule out potential population-specific causal variants. We expect in future data freezes, with larger non-European ancestry sample sizes to be able to run some of these analyses. However, we have now provided ancestry-specific allele frequency and meta- analysis results for the now 13 lead variants in **Supplementary Table 12**.

4. I also find the analysis of the effect at 3p21.31 in the Bangladeshi population lacking. It appears that this analysis is simply testing for association at a single proxy variant to provide additional replication of the association. If the claim is that the causal variants at this locus should be more pronounced due to increased frequency in this population then wouldn't a more in-depth analysis of haplotype structure, presence of rare, coding variants including a conditional analysis be more informative?

Answer: We thank the reviewer for pointing out that the inclusion of this section did

not add much value to the manuscript. Given that other reviewers had comments about this section, we have removed it from the manuscript.

5. Although full functional characterization of the associated loci will likely require years of dedicated research and is beyond the scope of this report, I do feel some additional analyses could be done with the data on hand to help guide functional follow-up studies. Firstly, for loci with two or more protein altering variants linked to the lead SNP a conditional analysis could be performed to determine if the combination of functional variants statistically out-performs the lead SNP. This could also help to understand how many independent signals are expected at each associated locus. For the eQTL analysis including GTEx and the Lung eQTL Consortium, a full colocalization analysis (using eCAVIAR or something similar) is warranted. I would also be interested to know if the overlap of effects on lung expression and COVID-19 severity go in an expected direction? And if so, is there evidence that other expression-influencing variants also impact COVID-19 severity? This is very hard for me to parse from Table S11 and would be of interest to labs following up on these genes.

Answer: The reviewer makes a valid observation, however due to the diversity of the cohorts studies and interdependability of colocalization and finemapping, we are unable to leverage the multi-ethnic nature of this study to consistently define credible sets of causal loci. We discuss these challenges in depth in our responses to **reviewer 2, comment section “GWAS analysis” point 5**. Overall, we feel that these questions are important to follow up on, but our focus is to provide as robust as possible data to share with the scientific community, and that independent research groups can use to pursue these questions more in depth.

6. As I mentioned in my summary, I believe the authors and organizers are to be commended for this global effort. However, I do think that the lack of inclusion of any groups or cohorts from continental Africa should be commented on. Although I don't think any one study can solve the continued lack of diversity in genomics, it would be useful for the authors include a comment highlighting the relative benefit of prioritizing inclusion of African and other non-European ancestry populations in further efforts. Particularly given the large potential for ancestry-specific genetic effects and the apparent reduced severity of COVID-19 across Africa.

Answer: We thank the reviewer for these positive comments. We fully agree the lack of diversity in our analyses as well is a continued problem. The Initiative has been communicating with research groups from continental Africa, and we are happy to be expecting new datasets from some of these countries. We hope that in the long run we will acquire enough data to release good quality meta-analysis results for non-European ancestry groups. Further in-depth analyses utilizing the data released by the Initiative will be driven by researchers who are already part of the initiative and

anyone else who wishes to use the data for downstream analyses. We have attempted to incorporate a note about the expansion to non-European populations into the Discussion at page 14:

The COVID-19 Host Genetics Initiative continues to pursue expansion of the datasets included in the consortium's analyses to populations from underrepresented populations in upcoming data releases.

minor comments:

Line 58, what is meant by “variable ascertainment strategies” this is worth expanding on in Methods

Answer: By different ascertainment strategies we were referring to the fact that contributing studies are very different from each other in their recruitment approach: e.g. hospital-based recruitment, direct-to-consumer companies, biobank-based studies, etc. We have kept the main text as is for now because we feel that this information is available throughout the manuscript, but we are happy to expand if still needed.

Should Table 1 contain a V2G column? Seems so from the legend but didn't come through in the manuscript pdf

Answer: We had indeed decided to move this column to the expanded results table Supplementary Table 2, and have removed the mention of V2G in the main Table 1 legend.

Typos throughout, a few examples:

Line 26 typo “international collaboration **is** a blue-print”
Line 238 typo “Lastly, there are **six** remaining* loci”

Line 355 typo “to to”

Line 428 clarify “and with the enrollment of vaccines”

Answer: We thank the reviewer for pointing out these typos and incoherences and have fixed them in the manuscript.

References

1. Severe Covid-19 GWAS Group *et al.* Genomewide Association Study of Severe

- Covid-19 with Respiratory Failure. *N. Engl. J. Med.* **383**, 1522–1534 (2020).
2. Shelton, J. F. *et al.* Trans-ethnic analysis reveals genetic and non-genetic associations with COVID-19 susceptibility and severity. *bioRxiv* (2020) doi:10.1101/2020.09.04.20188318.
 3. Kosmicki, J. A. *et al.* Genetic association analysis of SARS-CoV-2 infection in 455,838 UK Biobank participants. *bioRxiv* (2020) doi:10.1101/2020.10.28.20221804.
 4. Pairo-Castineira, E. *et al.* Genetic mechanisms of critical illness in Covid-19. *Nature* (2020) doi:10.1038/s41586-020-03065-y.
 5. Roberts, G. H. L. *et al.* AncestryDNA COVID-19 host genetic study identifies three novel loci. *bioRxiv* (2020) doi:10.1101/2020.10.06.20205864.
 6. Li, M.-X., Yeung, J. M. Y., Cherny, S. S. & Sham, P. C. Evaluating the effective numbers of independent tests and significant p-value thresholds in commercial genotyping arrays and public imputation reference datasets. *Hum. Genet.* **131**, 747–756 (2012).
 7. Roberts, G. H. L. *et al.* Novel COVID-19 phenotype definitions reveal phenotypically distinct patterns of genetic association and protective effects. *bioRxiv* (2021) doi:10.1101/2021.01.24.21250324.
 8. Fingerlin, T. E. *et al.* Genome-wide association study identifies multiple susceptibility loci for pulmonary fibrosis. *Nat. Genet.* **45**, 613–620 (2013).
 9. Eyre, S. *et al.* High-density genetic mapping identifies new susceptibility loci for rheumatoid arthritis. *Nat. Genet.* **44**, 1336–1340 (2012).
 10. Tsoi, L. C. *et al.* Large scale meta-analysis characterizes genetic architecture for common psoriasis associated variants. *Nat. Commun.* **8**, 15382 (2017).
 11. Langefeld, C. D. *et al.* Transancestral mapping and genetic load in systemic

- lupus erythematosus. *Nat. Commun.* **8**, 16021 (2017).
12. Kichaev, G. *et al.* Leveraging Polygenic Functional Enrichment to Improve GWAS Power. *Am. J. Hum. Genet.* **104**, 65–75 (2019).
 13. Kwok, A. J., Mentzer, A. & Knight, J. C. Host genetics and infectious disease: new tools, insights and translational opportunities. *Nat. Rev. Genet.* **22**, 137–153 (2021).
 14. Ning, Z., Pawitan, Y. & Shen, X. High-definition likelihood inference of genetic correlations across human complex traits. *Nat. Genet.* 1–6 (2020).
 15. Bulik-Sullivan, B. K. *et al.* LD Score regression distinguishes confounding from polygenicity in genome-wide association studies. *Nat. Genet.* **47**, 291–295 (2015).
 16. Batty, G. D., Gale, C. R., Kivimäki, M., Deary, I. J. & Bell, S. Comparison of risk factor associations in UK Biobank against representative, general population based studies with conventional response rates: prospective cohort study and individual participant meta- analysis. *BMJ* **368**, m131 (2020).
 17. Niedzwiedz, C. L. *et al.* Ethnic and socioeconomic differences in SARS-CoV-2 infection: prospective cohort study using UK Biobank. *BMC Med.* **18**, 160 (2020).
 18. Papageorge, N. W. *et al.* Socio-demographic factors associated with self-protecting behavior during the Covid-19 pandemic. *J. Popul. Econ.* 1–48 (2021).
 19. Griffith, G. J. *et al.* Collider bias undermines our understanding of COVID-19 disease risk and severity. *Nat. Commun.* **11**, 5749 (2020).
 20. Dennis, J. M., McGovern, A. P., Vollmer, S. J. & Mateen, B. A. Improving Survival of Critical Care Patients With Coronavirus Disease 2019 in England: A National Cohort Study, March to June 2020. *Crit. Care Med.* **49**, 209–214

- (2021).
21. Weissbrod, O. *et al.* Functionally informed fine-mapping and polygenic localization of complex trait heritability. *Nat. Genet.* **52**, 1355–1363 (2020).
 22. Ulirsch, J. C. *et al.* Interrogation of human hematopoiesis at single-cell and single-variant resolution. *Nat. Genet.* **51**, 683–693 (2019).
 23. Zeberg, H. & Pääbo, S. The major genetic risk factor for severe COVID-19 is inherited from Neanderthals. *Nature* **587**, 610–612 (2020).
 24. Ghossaini, M. *et al.* Open Targets Genetics: systematic identification of trait-associated genes using large-scale genetics and functional genomics. *Nucleic Acids Res.* (2020) doi:10.1093/nar/gkaa840.
 25. Xiao, G. *et al.* CXCL16/CXCR6 chemokine signaling mediates breast cancer progression by pERK1/2-dependent mechanisms. *Oncotarget* **6**, 14165–14178 (2015).
 26. Wei, Q. *et al.* LZTFL1 suppresses lung tumorigenesis by maintaining differentiation of lung epithelial cells. *Oncogene* **35**, 2655–2663 (2016).
 27. Vuille-dit-Bille, R. N. *et al.* Human intestine luminal ACE2 and amino acid transporter expression increased by ACE-inhibitors. *Amino Acids* **47**, 693–705 (2015).

Reviewer Reports on the First Revision:

Referee #1 (Remarks to the Author):

I commend Ganna and Colleagues for an overall well performed study, they have been thorough and have addressed my comments to satisfaction. I have no further comments.

Referee #2 (Remarks to the Author):

I thank the authors for the thorough and careful review of their manuscript. This version is much improved. I have no additional major comments.

Minor comments:

1- please add the third GWAS plot to figure 2

2- Given that this version of the data is focused mostly on Europeans please add a sentence to the discussion mentioning that more non-European ancestry samples will be added in future releases [otherwise the claims "worldwide" or "multi-ethnic" sound exaggerated].

3- Please add a few sentences to the manuscript (main text) mentioning the heritability estimate for covid reported previously by Pairo-Castineira et al and briefly discuss why the author's estimate is largely different from this previous report.

Referee #3 (Remarks to the Author):

I thank the authors for their detailed response to my comments on the previous version of the manuscript. Addressing the host genetic factors that contribute to COVID-19 susceptibility remains a critical question in global health and I again commend the authors for organizing such a large collaborative network so quickly. Although any study with this design and scope will have limitations, I feel the current version of the manuscript adequately addresses them where possible. In specific, acknowledging the challenges in quantifying variable sociodemographic factors across contexts, fine mapping associated loci by meta-analysis in diverse populations and the necessity to increase population representation globally. I look forward to seeing how these issues evolve as this initiative continues forward. I am satisfied with the authors response to my review and have no additional major critiques.

Author Rebuttals to First Revision:

Referee #2 (Remarks to the Author):

I thank the authors for the thorough and careful review of their manuscript. This version is much improved. I have no additional major comments.

Minor comments:

1- please add the third GWAS plot to figure 2

We thank the reviewer for their final comments. In this case, we have agreed with the editor that we will keep the format of Figure 2 as it is, such that only the two analyses with larger sample sizes and thus better power are displayed. We have kept the third analysis as an Extended Data Figure 4.

2- Given that this version of the data is focused mostly on Europeans please add a sentence to the discussion mentioning that more non-European ancestry samples will be added in future releases [otherwise the claims "worldwide" or "multi-ethnic" sound exaggerated].

As requested, we have removed the word world-wide from our title and the main text.

We have also added a sentence to the discussion: "We plan to release ancestry-specific results in full once the sample sizes allow for a well-powered meta-analysis."

3- Please add a few sentences to the manuscript (main text) mentioning the heritability estimate for covid reported previously by Pairo-Castineira et al and briefly discuss why the author's estimate is largely different from this previous report.

We have added the following sentence to the results section describing heritability:

“The values are low compared to previously published studies (Pairo-Castineira et al., 2020) but may be explained by differences in reported estimate scale (observed vs. liability), the specific method used, disease prevalence estimates, phenotypic differences between patient cohorts or ascertainment of controls.”

Pairo-Castineira, E., Clohisey, S., Klaric, L., Bretherick, A. D., Rawlik, K., Pasko, D., Walker, S.,

Parkinson, N., Fourman, M. H., Russell, C. D., Furniss, J., Richmond, A., Gountouna, E.,

Wrobel, N., Harrison, D., Wang, B., Wu, Y., Meynert, A., Griffiths, F., ... Baillie, J. K. (2020).

Genetic mechanisms of critical illness in Covid-19. *Nature*. [https://doi.org/10.1038/s41586-020-](https://doi.org/10.1038/s41586-020-03065-y)

03065-y